# BENIGN OVERFITTING IN SINGLE-HEAD ATTENTION

## ABSTRACT

The phenomenon of *benign overfitting*, where a trained neural network perfectly fits noisy training data but still achieves near-optimal test performance, has been extensively studied in recent years for linear models and fully-connected/convolutional networks. In this work, we study benign overfitting in a single-head softmax attention model, which is the fundamental building block of Transformers. We prove that under appropriate conditions, the model exhibits benign overfitting in a classification setting already after two steps of gradient descent. Moreover, we show conditions where a minimum-norm/maximum-margin interpolator exhibits benign overfitting. We study how the overfitting behavior depends on the signal-to-noise ratio (SNR) of the data distribution, namely, the ratio between norms of signal and noise tokens, and prove that a sufficiently large SNR is both necessary and sufficient for benign overfitting.

## 1 INTRODUCTION

Neural networks often exhibit a remarkable phenomenon, known as *benign overfitting*, where they achieve a perfect fit to noisy training examples and still generalize well to unseen data (Zhang et al., 2021; Bartlett et al., 2020). This phenomenon contradicts classical wisdom in machine learning, and has become a central research question in the theory of deep learning. Existing works on benign overfitting study under what conditions the phenomenon occurs in different architectures. These works focus on linear models, and on shallow fully-connected and convolutional neural networks.

In recent years, Transformers (Vaswani, 2017) have emerged as a leading neural network architecture, with impactful applications across a wide range of domains such as natural language processing and computer vision. The fundamental building block of Transformers is the attention mechanism, which allows them to process sequences and focus different parts of the input. Despite the central role of the attention mechanism, we currently do not understand their overfitting behavior and the conditions under which they exhibit benign overfitting.

In this work, we show the first benign-overfitting results for the attention mechanism. We consider classification with a single-head softmax attention model, and study the conditions that allow for benign overfitting. In our results, the data distribution consists of two tokens: a *signal token*, which can be used for correctly classifying clean test examples, and a *noise token*, which is independent of the label but can be used for interpolating (i.e., perfectly fitting) noisy training examples. We study the singnal-to-noise ratio (SNR), namely, the expected ratio between the norms of signal and noise tokens, that allows for benign overfitting.

Below we summarize our main contributions:

- In Theorem 4 (Section 3) we show that under appropriate conditions, gradient descent with the logistic loss exhibits benign overfitting already after two iterations. This result holds when the SNR is $\Theta(1/\sqrt{n})$, where $n$ is the number of training samples.

- We then turn to consider other natural learning rules, which allow for benign overfitting under a weaker requirement on the SNR. In Theorems 6 and 8 (Section 4), we prove that minimum-norm (i.e., maximum-margin) interpolators exhibit benign overfitting when the SNR is $\Omega(1/\sqrt{n})$ without requiring an upper bound on the SNR.

- In Theorem 10 (Section 4), we prove that the above requirement on the SNR is tight. Namely, if the SNR is smaller than it, then the min-norm interpolator exhibits harmful overfitting, where it fits the training data but has poor generalization performance.

- In Section 6, we complement our theoretical results with an empirical study. We show that a sufficiently large SNR and input dimension are necessary to achieve benign overfitting.

The paper is structured as follows. In Section 2, we provide some preliminaries and define the data distribution and the single-head attention model. In Sections 3 and 4 we state our main results on benign overfitting with gradient descent and with min-norm interpolators. In Section 5 we discuss the main proof ideas, with all formal proofs deferred to the appendix. Finally, in Section 6 we show empirical results.

RELATED WORK

**Optimization in Transformers.** Li et al. (2023) provided a theoretical analysis of training a shallow Vision Transformer (ViT) for a classification task. They showed that the sample complexity required to achieve a zero generalization error is correlated with the inverse of the fraction of label-relevant tokens, the token noise level, and the initial model error. Ataee Tarzanagh et al. (2023a) showed that optimizing the attention layer via gradient descent leads to convergence to an SVM solution, where the implicit bias of the attention mechanism depends on whether the parameters are represented as a product of key-query matrices or directly as a combined matrix, with different norm-minimization objectives in each case. Ataee Tarzanagh et al. (2023b) provided a regularization path analysis and prove that the attention weights converge in a direction to a max-margin solution that separates locally optimal tokens from non-optimal. They also showed that running gradient descent, with a specific initialization direction and without optimizing the attention head, converges in a direction to the same max-margin solution. Vasudeva et al. (2024) expanded on their findings by identifying non-trivial data settings for which the convergence of GD is provably global, i.e., without requiring assumptions about the initialization direction. They also provided convergence rate bounds and analysis for optimizing both the attention weights and the attention head, although they did not consider the case of noisy data labels, as we do in our work. Another line of work looks at the learning dynamics of single-layer linear attention models trained on linear regression tasks (Zhang et al., 2024; Ahn et al., 2023; Wu et al., 2023). Additional works that consider optimization dynamics in Transformers include Jelassi et al. (2022); Oymak et al. (2023).

**Benign overfitting.** A significant body of research has explored why neural networks (NNs) that perfectly interpolate the training data can still generalize well (Zhang et al., 2021; Bartlett et al., 2020). This has sparked substantial interest in studying overfitting and generalization in NNs trained to fit datasets with noisy labels. The literature on benign overfitting is broad and cannot be reasonably covered here. We refer the reader to the surveys Bartlett et al. (2021); Belkin (2021). Most relevant to our work are Cao et al. (2022); Kou et al. (2023); Meng et al. (2023) that studied benign overfitting in convolutional neural networks. Their data distribution resembles ours, as we discuss in Section 2.1. Benign overffiting in fully-connected two-layer neural network classification was studied in Frei et al. (2022; 2023); Xu et al. (2023); Xu & Gu (2023); Kornowski et al. (2024); George et al. (2024); Karhadkar et al. (2024) for various activation functions, data distributions and loss functions (both the logistic and the hinge losses).

## 2 PRELIMINARIES

**Notations.** We use bold-face letters to denote vectors and matrices, and let $[m]$ be shorthand for $\{1, 2, \ldots, m\}$. Given a vector $\boldsymbol{x}$, we denote by $x_j$ its $j$-th coordinate. Let $\boldsymbol{I}_d$ be the $d \times d$ identity matrix, and let $\boldsymbol{0}_d$ (or just $\boldsymbol{0}$, if $d$ is clear from the context) denote the zero vector in $\mathbb{R}^d$. We let $\|\cdot\|$ denote the Euclidean norm. We denote a multivariate Gaussian distribution with mean vector $\boldsymbol{\mu}$ and covariance matrix $\boldsymbol{\Sigma}$ by $N(\boldsymbol{\mu}, \boldsymbol{\Sigma})$. We use standard big-Oh notation, with $\Theta(\cdot), \Omega(\cdot), O(\cdot)$ hiding universal constants and $\widetilde{\Theta}(\cdot), \widetilde{\Omega}(\cdot), \widetilde{O}(\cdot)$ hiding constants and factors that are polylogarithmic in the problem parameters. We use $\mathbb{I}(\cdot)$ to denote the indicator variable of an event. For a finite set $\mathcal{A}$, denote the uniform distribution over $\mathcal{A}$ by $\mathrm{Unif}(\mathcal{A})$ and let $|\mathcal{A}|$ be its cardinality.

### 2.1 DATA GENERATION SETTING

In this work we focus on the following data distribution:

**Definition 1.** *Let $\boldsymbol{\mu}_1, \boldsymbol{\mu}_2 \in \mathbb{R}^d$ such that $\|\boldsymbol{\mu}_1\| = \|\boldsymbol{\mu}_2\| = \rho$ for some $\rho > 0$ and $\langle \boldsymbol{\mu}_1, \boldsymbol{\mu}_2 \rangle = 0$, be two fixed orthogonal vectors representing the signal contained in each data point. Define $\mathcal{D}_{clean}$ as the distribution over $\mathbb{R}^{2 \times d} \times \{\pm 1\}$ of labelled data such that a data point $(\boldsymbol{X}, \widetilde{y})$ is generated by the following procedure:*

1. *Sample the label $\widetilde{y} \sim \mathsf{Unif}\{\pm 1\}$.*

2. *Generate a vector $\boldsymbol{u}$, which represents the signal, as follows: If $\widetilde{y} = +1$, set $\boldsymbol{u} = \boldsymbol{\mu}_1$; and if $\widetilde{y} = -1$, set $\boldsymbol{u} = \boldsymbol{\mu}_2$.*

3. *Generate a vector $\boldsymbol{\xi}$, which represents the noise, from the Gaussian distribution $\boldsymbol{\xi} \sim \mathcal{N}(\boldsymbol{0}, \boldsymbol{I}_d - \boldsymbol{\mu}_1 \boldsymbol{\mu}_1^\top / \rho^2 - \boldsymbol{\mu}_2 \boldsymbol{\mu}_2^\top / \rho^2)$.*

4. *Denote $\boldsymbol{X} = (\boldsymbol{x}^{(1)}, \boldsymbol{x}^{(2)})^\top$. Select $k \sim \mathsf{Unif}\{1, 2\}$ and set $\boldsymbol{x}^{(k)} = \boldsymbol{u}$. Set the other token $\boldsymbol{x}^{(3-k)} = \boldsymbol{\xi}$.*

To study the overfitting behavior we also need to introduce label-flipping noise:

**Definition 2.** *Let $\eta \in [0, 1/2)$ be the label flipping probability. We define $\mathcal{D}$ as the distribution over $\mathbb{R}^{2 \times d} \times \{\pm 1\}$ which is the $\eta$-label-flipped version of $\mathcal{D}_{clean}$. Namely, to generate $(\boldsymbol{X}, y) \sim \mathcal{D}$, first generate $(\boldsymbol{X}, \widetilde{y}) \sim \mathcal{D}_{clean}$, then let $y = \widetilde{y}$ with probability $1 - \eta$ and $y = -\widetilde{y}$ with probability $\eta$.*

Our data distribution resembles the distributions considered by Kou et al. (2023); Cao et al. (2022); Meng et al. (2023). They proved benign overfitting in two-layer convolutional neural networks, and in their setting each data point consists of two patches $\boldsymbol{x}^{(1)}, \boldsymbol{x}^{(2)}$ (rather than two tokens in our setting). Since our single-head attention model is invariant to the order of the tokens, we assume without loss of generality throughout this work that $\boldsymbol{x}^{(1)}$ is the signal token and $\boldsymbol{x}^{(2)}$ is the noise token in all data points. Note that the noise token $\boldsymbol{x}^{(2)} = \boldsymbol{\xi}$ is independent of the label, and that it is generated from $\mathcal{N}(\boldsymbol{0}, \boldsymbol{I}_d - \boldsymbol{\mu}_1 \boldsymbol{\mu}_1^\top / \rho^2 - \boldsymbol{\mu}_2 \boldsymbol{\mu}_2^\top / \rho^2)$, ensuring that it is orthogonal to the signal vector. Note that when the dimension $d$ is large, $\|\boldsymbol{\xi}\| \approx \sqrt{d-2} \approx \sqrt{d}$ by standard concentration bounds. Therefore, we denote the signal-to-noise ratio (SNR) as $\mathrm{SNR} = \|\boldsymbol{\mu}\| / \sqrt{d} = \rho / \sqrt{d}$.

We consider a training dataset $\{(\boldsymbol{X}_i, y_i)\}_{i=1}^n$ of $n$ samples generated i.i.d. from the distribution $\mathcal{D}$. Denote the index set of data whose labels are not flipped by $\mathcal{C} = \{i : \widetilde{y}_i = y_i\}$ ("clean examples"), and the index set of data whose labels are flipped by $\mathcal{N} = \{i : \widetilde{y}_i = -y_i\}$ ("noisy examples"). For indices in $\mathcal{C}$, we further denote $\mathcal{C}_1 := \mathcal{C} \cap \{i : \boldsymbol{x}_i^{(1)} = \boldsymbol{\mu}_1\}, \mathcal{C}_2 := \mathcal{C} \cap \{i : \boldsymbol{x}_i^{(1)} = \boldsymbol{\mu}_2\}$, and define the subsets $\mathcal{N}_1, \mathcal{N}_2$ of $\mathcal{N}$ analogously.

## 2.2 Single-Head Attention Model

We consider the following single-head attention model:

$$f(\boldsymbol{X}; \boldsymbol{W}, \boldsymbol{p}) = \boldsymbol{v}^\top \boldsymbol{X}^\top \mathbb{S}(\boldsymbol{X} \boldsymbol{W} \boldsymbol{q}),$$

where $\mathbb{S} : \mathbb{R}^d \to \mathbb{R}^d$ is the softmax function, the key-query matrix $\boldsymbol{W} \in \mathbb{R}^{d \times d}$ and the linear head vector $\boldsymbol{v} \in \mathbb{R}^d$ are the trainable parameters, and the query vector $\boldsymbol{q} \in \mathbb{R}^d$ is an arbitrary fixed unit vector. We follow Ataee Tarzanagh et al. (2023b) and assume that $\boldsymbol{q} = (1, 0, \ldots, 0)^\top$, obtaining the following model:

$$f(\boldsymbol{X}; \boldsymbol{p}, \boldsymbol{v}) = \boldsymbol{v}^\top \boldsymbol{X}^\top \mathbb{S}(\boldsymbol{X} \boldsymbol{p}), \tag{1}$$

Here the trained parameters are $\boldsymbol{p}, \boldsymbol{v} \in \mathbb{R}^d$. Thus, instead of the key-query matrix $\boldsymbol{W}$ we have a vector $\boldsymbol{p}$ that controls the attention. Throughout this paper we will use the model (1). We denote the output of the softmax layer $\mathbb{S}(\boldsymbol{X}_i \boldsymbol{p})$ by $\boldsymbol{s}_i = (s_{i,1}, s_{i,2})^\top$, and denote the output of the attention layer $\boldsymbol{X}_i^\top \boldsymbol{s}_i$ by $\boldsymbol{r}_i = s_{i,1} \boldsymbol{\mu}_i + s_{i,2} \boldsymbol{\xi}_i$, where $0 \leq s_{i,1}, s_{i,2} \leq 1$, $s_{i,1} + s_{i,2} = 1$ are the attention on two tokens of the $i$-th sample.

## 3 Benign Overfitting with Gradient Descent

In this section, we study the joint optimization of the head $\boldsymbol{v}$ and attention weights $\boldsymbol{p}$ using the logistic loss function. We show that the model exhibits benign overfitting after just two iterations of gradient descent (GD).

Formally, for a training dataset $\{(\boldsymbol{X}_i, y_i)\}_{i=1}^n$ we define the empirical risk as

$$\mathcal{L}(\boldsymbol{v}, \boldsymbol{p}) = \frac{1}{n} \sum_{i=1}^n \ell(y_i \cdot f(\boldsymbol{X}_i; \boldsymbol{p}, \boldsymbol{v})),$$

where $\ell(z) = \log(1 + \exp(-z))$ is the logistic loss function, and $f$ is the model from Eq. (1). We consider GD optimization. Starting from $\boldsymbol{p}_0 = \boldsymbol{0}$ and $\boldsymbol{v}_0 = \boldsymbol{0}$, we have

$$\boldsymbol{v}_{t+1} = \boldsymbol{v}_t - \beta \nabla_{\boldsymbol{v}} \mathcal{L}(\boldsymbol{v}_t, \boldsymbol{p}_t) \qquad \text{and} \qquad \boldsymbol{p}_{t+1} = \boldsymbol{p}_t - \beta \nabla_{\boldsymbol{p}} \mathcal{L}(\boldsymbol{v}_t, \boldsymbol{p}_t),$$

where $\beta$ is the step size. When we discuss some fixed $t$, we sometimes write in the subscript "$t = \cdot$", e.g., $\boldsymbol{p}_{t=2}$ instead of $\boldsymbol{p}_2$. We make the following assumptions:

**Assumption 3** (Assumptions for GD with SNR $= \Theta(1/\sqrt{n})$). *Let $\delta \in (0, 0.5)$ be a desired probability of failure. For universal constants $C_\rho \geq 6, C_\beta \geq 16$, as well as a sufficiently large universal constant $C$ that may depend on $C_\rho$ and $C_\beta$, the following conditions hold:*

1. *Number of samples $n$ is sufficiently large: $n \geq C \log(1/\delta)$.*

2. *Dimension $d$ is sufficiently large: $d \geq C n^2 \log(n/\delta)$.*

3. *Signal strength satisfies $\rho = C_\rho \cdot \sqrt{d/n}$*

4. *Label flipping rate satisfies $\eta \leq 1/C$.*

5. *Step size satisfies $\beta = C_\beta \cdot (n/d)$.*

6. *Initialization at zero: $\|\boldsymbol{v}_0\| = \|\boldsymbol{p}_0\| = 0$.*

Item 1 is required to estimate the number of clean examples compared to noisy examples. The assumption of high dimensionality (Item 2) is important for enabling benign overfitting (see empirical results in Section 6), and implies that noise tokens from different training samples are nearly-orthogonal. This assumption appears in many prior works on benign overfitting in neural network classification (e.g., Cao et al. (2022); Kou et al. (2023); Meng et al. (2023); Frei et al. (2022; 2023); Xu et al. (2023); Kornowski et al. (2024); Xu & Gu (2023)). Item 3 states that the signal-to-noise ratio (SNR) is $\frac{\rho}{\sqrt{d}} = \Theta(1/\sqrt{n})$. In Section 5 we will discuss how the SNR affects the dynamics of GD. Interestingly, SNR of $\Theta(1/\sqrt{n})$ matches the lower bound of the required SNR that allows for benign overfitting with the min-norm (i.e. max-margin) learning rule that we will study in Section 4. Item 4 ensures the flipping rate is small enough to allow the model to learn the signal token. Item 5, namely, using a step size of $\Theta(n/d)$, is required to achieve benign overfitting after two iterations; with a smaller step size, the model will need more iterations to fit the noisy samples, which we will demonstrate empirically in Section 6.

We now state our main result on benign overfitting with GD:

**Theorem 4.** *Suppose that Assumption 3 holds. Then, with probability at least $1 - \delta$ over the training dataset, after two iterations of GD we have:*

- *Higher softmax probability for optimal tokens:*

$$s_{i,1}^{t=2} > 1/2, \; \forall i \in \mathcal{C} \qquad \text{and} \qquad s_{i,2}^{t=2} \geq 1 - 1/c_\rho^2, \; \forall i \in \mathcal{N}$$

  *where $s_{i,j}^t$ is the softmax probability of the $j^{th}$ token in the $i^{th}$ sample at time $t$.*

- *The classifier $\boldsymbol{X} \mapsto \text{sign}(f(\boldsymbol{X}; \boldsymbol{v}_{t=2}, \boldsymbol{p}_{t=2}))$ correctly classifies all training data points:*

$$y_i = \text{sign}(f(\boldsymbol{X}_i; \boldsymbol{v}_{t=2}, \boldsymbol{p}_{t=2})), \; \forall i \in [n].$$

- *The classifier $\boldsymbol{X} \mapsto \text{sign}(f(\boldsymbol{X}; \boldsymbol{v}_{t=2}, \boldsymbol{p}_{t=2})$ generalizes well:*

$$\mathbb{P}_{(\boldsymbol{X}, y) \sim \mathcal{D}}(y \neq \text{sign}(f(\boldsymbol{X}; \boldsymbol{v}_{t=2}, \boldsymbol{p}_{t=2}))) \leq \eta + \exp(-d/C_1 n^2),$$

  *where $C_1 := C_1(c_\rho, c_\beta)$ is a constant.*

We can also conclude that for the clean-labeled distribution $\mathcal{D}_{\text{clean}}$ we have

$$\mathbb{P}_{(\boldsymbol{X},y)\sim\mathcal{D}_{\text{clean}}}(y \neq \text{sign}(f(\boldsymbol{X};\boldsymbol{v}_{t=2},\boldsymbol{p}_{t=2}))) \leq \exp(-d/C_1 n^2),$$

which approaches zero as $d$ grows (see Assumption 3, item 2).

Theorem 4 shows that after two iterations of GD, the attention focuses on the signal tokens for clean examples, and on the noise tokens for noisy examples. The model uses the noise tokens for interpolating noisy training examples, while still achieving good generalization performance using the signal token.

## 4 BENIGN OVERFITTING OF MAX-MARGIN SOLUTION

In the previous section we showed that GD exhibits benign overfitting in a setting where the SNR is $\Theta(1/\sqrt{n})$. We now turn to study the overfitting behavior of single-head attention models, when using another learning rule, which returns solutions that interpolate the training data with large margin while keeping the parameters norms small. As we will show, such a learning rule allows us to obtain benign overfitting under a weaker requirement on the SNR, namely, the SNR is $\Omega(1/\sqrt{n})$ without requiring an upper bound on it.

We note that learning rules that return min-norm (or max-margin) solutions are considered natural, and hence understanding properties of min-norm interpolators has attracted much interest in recent years, even in settings where the implicit bias of GD does not necessarily lead to a min-norm solution (see, e.g., Savarese et al. (2019); Ongie et al. (2019); Ergen & Pilanci (2021); Hanin (2021); Debarre et al. (2022); Boursier & Flammarion (2023)). More directly related to our work, min-norm interpolation with Transformers has been studied in Ataee Tarzanagh et al. (2023b;a), and benign/tempered overfitting in min-norm univariate neural network interpolators has been studied in Joshi et al. (2023).

We first consider the following learning rule:

$$(\boldsymbol{v}_{(r,R)}, \boldsymbol{p}_{(r,R)}) = \underset{\|\boldsymbol{v}\|\leq r, \|\boldsymbol{p}\|\leq R}{\text{argmax}} \min_{i\in[n]} y_i \cdot f(\boldsymbol{X}_i;\boldsymbol{p},\boldsymbol{v}), \tag{2}$$

where $f$ is the model from (1). The learning rule returns a solution that maximizes the margin $\min_{i\in[n]} y_i \cdot f(\boldsymbol{X}_i;\boldsymbol{p},\boldsymbol{v})$ under a restriction on the parameter norms. We make the following assumption:

**Assumption 5** (Assumptions for max-margin with SNR $= \Omega(1/\sqrt{n})$). *Let $\delta \in (0, 0.5)$ be a desired probability of failure. There exists a sufficiently large constant $C$ such that the following hold:*

1. *Dimension $d$ is sufficiently large: $d \geq Cn^2 \log(n/\delta)$.*

2. *Number of samples $n$ is sufficiently large: $n \geq C \log(1/\delta)$.*

3. *Signal strength: $\rho \geq C\sqrt{d/n}$.*

4. *Label flipping rate: $0 \leq \eta \leq 1/C$.*

5. *Norm constraint of $\boldsymbol{p}$ satisfies: $R \geq C\sqrt{\eta n/d + 1/\rho^2} \log(\rho n)$.*

Items 1, 2 and 4 are similar to Assumption 3. Item 3 requires SNR $\geq \Omega(1/\sqrt{n})$, which is a weaker requirement than the $\Theta(1/\sqrt{n})$ requirement in Assumption 3. We will show later a lower bound on the required SNR for benign overfitting, implying that the $\Omega(1/\sqrt{n})$ bound is tight. Item 5 provides the lower bound for the norm constraint of $\boldsymbol{p}$ so that the model can allocate enough attention on signal token to achieve benign overfitting. Note that the norm constraint $r$ for $\boldsymbol{v}$ can take any positive value. Intuitively, since the model is linear in $\boldsymbol{v}$, once $\boldsymbol{p}$ is properly learned, $\boldsymbol{v}$ can achieve accurate classification even with a small norm.

With these assumptions in place, we give our result on benign overfitting with the learning rule (2).

**Theorem 6.** *Suppose that Assumption 5 holds, and consider the classifier $\boldsymbol{X} \rightarrow \text{sign}(f(\boldsymbol{X};\boldsymbol{p}_{(r,R)},\boldsymbol{v}_{(r,R)}))$, where $(\boldsymbol{v}_{(r,R)},\boldsymbol{p}_{(r,R)})$ is the solution to Problem (2). Then, with probability at least $1 - \delta$ over the training dataset, we have:*

- *The classifier $\operatorname{sign}(f(\boldsymbol{X}; \boldsymbol{p}_{(r,R)}, \boldsymbol{v}_{(r,R)}))$ correctly classifies all training data points:*

$$y_i = \operatorname{sign}(f(\boldsymbol{X}_i; \boldsymbol{p}_{(r,R)}, \boldsymbol{v}_{(r,R)})), \ \forall i \in [n].$$

- *The classifier $\operatorname{sign}(f(\boldsymbol{X}; \boldsymbol{p}_{(r,R)}, \boldsymbol{v}_{(r,R)}))$ generalizes well on test data:*

$$\mathbb{P}_{(\boldsymbol{X},y)\sim\mathcal{D}}(y \neq \operatorname{sign}(f(\boldsymbol{X}; \boldsymbol{p}_{(r,R)}, \boldsymbol{v}_{(r,R)})))$$
$$\leq \eta + \exp(-\Omega(d/n^2)) + \exp\Big(-\Omega\big(\frac{(1-\zeta)}{\sqrt{\eta n/d + 1/\rho^2}} - \frac{\log(d)}{R}\big)^2\Big),$$

*where $\zeta = \Theta(\sqrt{\eta n/d + 1/\rho^2}\log(\rho n)/R)$.*

**Remark 7.** *To see why Theorem 6 implies benign overfitting, consider the limit $R \to \infty$. Then, the upper bound for test error becomes $\eta + \exp(-\Omega(d/n^2)) + \exp(-\Theta((1/\rho^2 + \eta n/d)^{-1}))$, which can be arbitrarily close to $\eta$ if $d$ is large (see Assumption 5, item 1).*

Next, we consider the following learning rule, which explicitly requires to minimize the parameters norms while allowing interpolation with margin at least $\gamma$:

$$(\boldsymbol{v}_\gamma, \boldsymbol{p}_\gamma) = \underset{\|\boldsymbol{p}\|^2 + \|\boldsymbol{v}\|^2}{\operatorname{argmin}} \ \text{s.t.} \ \min_{i\in[n]} y_i f(\boldsymbol{X}_i; \boldsymbol{p}, \boldsymbol{v}) \geq \gamma, \tag{3}$$

where $f$ is the model from Eq. (1). We show that under Assumption 5, the solution $(\boldsymbol{v}_\gamma, \boldsymbol{p}_\gamma)$ exhibits benign overfitting for large enough $\gamma$ and $d$:

**Theorem 8.** *Suppose that Assumption 5 (items 1 through 4) holds, and consider the classifier $\boldsymbol{X} \to \operatorname{sign}(f(\boldsymbol{X}; \boldsymbol{p}_\gamma, \boldsymbol{v}_\gamma))$, where $(\boldsymbol{v}_\gamma, \boldsymbol{p}_\gamma)$ is a solution of Problem (3). Then there exists $\gamma_0$ such that for any $\gamma \geq \gamma_0$, with probability at least $1 - \delta$ over the training dataset, we have:*

- *The classifier $\operatorname{sign}(f(\boldsymbol{X}; \boldsymbol{p}_\gamma, \boldsymbol{v}_\gamma))$ correctly classifies all training data points:*

$$y_i = \operatorname{sign}(f(\boldsymbol{X}_i; \boldsymbol{p}_\gamma, \boldsymbol{v}_\gamma)), \ \forall i \in [n].$$

- *The classifier $\operatorname{sign}(f(\boldsymbol{X}; \boldsymbol{p}_\gamma, \boldsymbol{v}_\gamma))$ generalizes well on test data:*

$$\mathbb{P}_{(\boldsymbol{X},y)\sim\mathcal{D}}(y \neq \operatorname{sign}(f(\boldsymbol{X}; \boldsymbol{p}_\gamma, \boldsymbol{v}_\gamma))) \leq \eta + \exp(-\Omega(d/n^2)) + \exp(-\Theta((1/\rho^2 + \eta n/d)^{-1})).$$

Thus, for large enough $\gamma$, the theorem implies that the trained model interpolates the training data, and the test error approaches $\eta$ as $d \to \infty$.

Note that Theorems 6 and 8 hold only when $\text{SNR} = \Omega(1/\sqrt{n})$. This raises the question: what is the overfitting behavior of min-norm interpolators when the SNR is smaller? We now consider the case where $\rho \leq \sqrt{1/Cn}$ for some sufficiently large universal constant $C$. We will show that in this case, although the model can correctly classify all training samples, the test error of learning rule (2) is at least a universal constant, indicating that benign overfitting does not happen. Formally, we make the following assumptions:

**Assumption 9** (Assumptions for max-margin with $\text{SNR} \leq O(1/\sqrt{n})$). *Let $\delta \in (0, 0.5)$ be a desired probability of failure. There exists a sufficiently large constant $C$ such that the following hold:*

1. *Dimension $d$ is sufficiently large: $d \geq Cn^2 \log(n/\delta)$*

2. *Number of samples $n$ is sufficiently large: $n \geq C\log(1/\delta)$.*

3. *Signal strength: $\rho \leq \sqrt{d/Cn}$.*

4. *Label flipping rate is a constant $\eta \in (0, 1/2)$.*

5. *The norm of $\boldsymbol{p}$ should be sufficiently large: $R \geq C\sqrt{\frac{n}{d}}\log\big(\frac{n\rho}{d}\big)$.*

Compared with Assumption 5, the main difference is in the second item that $\text{SNR} \leq O(1/\sqrt{n})$. Additionally, the condition on $\eta$ is relaxed, as in our analysis clean and noisy samples can be treated equivalently when the norm of the signal token is sufficiently small. With these assumptions in place, we can state the following theorem which characterizes the training error and test error of the single-head attention model when the SNR is small:

**Theorem 10.** *Suppose that Assumption 9 holds, and consider the classifier $\boldsymbol{X} \rightarrow \operatorname{sign}(f(\boldsymbol{X}; \boldsymbol{p}_{(r,R)}, \boldsymbol{v}_{(r,R)}))$, where $(\boldsymbol{v}_{(r,R)}, \boldsymbol{p}_{(r,R)})$ is a solution of Problem (2). Then, with probability at least $1 - \delta$ over the training data, we have:*

- *The classifier $\operatorname{sign}(f(\boldsymbol{X}; \boldsymbol{p}_{(r,R)}, \boldsymbol{v}_{(r,R)}))$ correctly classifies all training data points:*

$$y_i = \operatorname{sign}(f(\boldsymbol{X}_i; \boldsymbol{p}_{(r,R)}, \boldsymbol{v}_{(r,R)})), \ \forall i \in [n].$$

- *The classifier $\operatorname{sign}(f(\boldsymbol{X}; \boldsymbol{p}_{(r,R)}, \boldsymbol{v}_{(r,R)}))$ does not generalize well on test data:*

$$\mathbb{P}_{(\boldsymbol{X},y) \sim \mathcal{D}_{clean}}(y \neq \operatorname{sign}(f(\boldsymbol{X}; \boldsymbol{p}_{(r,R)}, \boldsymbol{v}_{(r,R)}))) \geq \frac{1}{16}.$$

## 5 PROOF IDEAS

In this section we briefly discuss the main proof ideas. The formal proofs are deferred to the appendix.

### 5.1 PROOF IDEAS FOR SECTION 3

In this subsection we discuss the main proof idea of Theorem 4. Since the initialization is at zero, $\boldsymbol{v}_t$ is a linear combination of the training data tokens. Therefore, we can express $\boldsymbol{v}_{t=1}$ as $\lambda_1^{t=1}\boldsymbol{\mu}_1 + \lambda_2^{t=1}\boldsymbol{\mu}_2 + \sum_{i=1}^n y_i \theta_i^{t=1}\boldsymbol{\xi}_i$, where $\lambda_1^{t=1} > 0, \lambda_2^{t=1} < 0$. Note that $\lambda_1^t > 0, \lambda_2^t < 0$ holds since $|\mathcal{C}| > |\mathcal{N}|$. We begin by analyzing the first step of GD. Specifically, we show that after one step, the coefficients of $\boldsymbol{v}_{t=1}$ can be estimated as $|\lambda_k^{t=1}| \approx \frac{\beta}{8}(1 - 2\eta), k \in [2]$ and $\theta_i^{t=1} = \frac{\beta}{4n}, i \in [n]$. Moreover, we have $\boldsymbol{p}_{t=1} = 0$, and hence for a training sample $(\boldsymbol{X}_j = (\boldsymbol{\mu}_k, \boldsymbol{\xi}_j), y_j)$, the margin is:

$$y_j f(\boldsymbol{X}_j; \boldsymbol{v}_{t=1}, \boldsymbol{p}_{t=1}) = \frac{1}{2}y_j \boldsymbol{v}_{t=1}^\top (\boldsymbol{x}_j^{(1)} + \boldsymbol{x}_j^{(2)}) \approx \frac{1}{2}y_j \lambda_1^{t=1} \|\boldsymbol{\mu}_k\|^2 + \frac{1}{2}\theta_j^{t=1} \|\boldsymbol{\xi}_j\|^2,$$

where in the last approximate equality we use the high dimensional setting (i.e. by item 2 in our assumption $d \gg n^2 \log(n)$) to neglect the $\sum_{i \in [n]: i \neq j} y_i y_j \theta_j^{t=1} \boldsymbol{\xi}_i^\top \boldsymbol{\xi}_j$ term, since it is much smaller (in absolute value) than the other terms. Indeed, we have w.h.p. that $|\boldsymbol{\xi}_i^\top \boldsymbol{\xi}_j| \leq \sqrt{d}\log(n), \|\boldsymbol{\xi}_j\|^2 \approx d$ and recall that $\|\boldsymbol{\mu}_k\|^2 = C_\rho^2(d/n)$ (item 3 in our assumption). Therefore, for a clean sample $j \in \mathcal{C}$, the margin is $y_j f(\boldsymbol{X}_j; \boldsymbol{v}_{t=1}, \boldsymbol{p}_{t=1}) \approx \frac{\beta(1-2\eta)}{16}\frac{dC_\rho^2}{n} + \frac{\beta}{8n}d > 0$, for large enough $C_\rho$. On the other hand, for a noisy sample $j \in \mathcal{N}$, we have $y_j f(\boldsymbol{X}_j; \boldsymbol{v}_{t=1}, \boldsymbol{p}_{t=1}) \approx -\frac{\beta(1-2\eta)}{16}\frac{dC_\rho^2}{n} + \frac{\beta}{8n}d < 0$. This implies that the classifier $\operatorname{sign}(f(\boldsymbol{X}; \boldsymbol{v}_{t=1}, \boldsymbol{p}_{t=1})$ does not correctly classify noisy training samples, but still correctly classifies clean training samples. Together with $\boldsymbol{p}_{t=1} = \boldsymbol{0}$, the classifier $\operatorname{sign}(f(\boldsymbol{X}; \boldsymbol{v}_{t=1}, \boldsymbol{p}_{t=1})$ will also correctly classify, with high probability, a clean test sample.

Moreover, since the loss function $\ell$ is decreasing, the loss of noisy samples, denoted $\ell_{t=1,j}, j \in \mathcal{N}$, dominates the loss of clean samples $\ell_{t=1,i}, i \in \mathcal{C}$. This implies that after two iterations, the coefficients $|\theta_j^{t=2}|, j \in \mathcal{N}$, of the second (noise) tokens in $\boldsymbol{v}_{t=2}$, corresponding to noisy samples, grow faster than the coefficients $|\lambda_i^{t=2}|$ of the first (signal) tokens. This property is important to allow for interpolation of noisy examples. We also show that $\boldsymbol{p}_{t=2}$ focuses on optimal tokens, namely, on the noise token for noisy samples (i.e. $s_{i,2}^{t=2} \geq 1 - 1/c_\rho^2, \forall i \in \mathcal{N}$), and on the signal token for clean training and test samples. Using this property we conclude that the model parameterized by $(\boldsymbol{v}_{t=2}, \boldsymbol{p}_{t=2})$ exhibits benign overfitting.

**Remark 11.** *Note that our proof implies the following behavior of GD. After the first iteration, the model correctly classifies only the clean training samples, resulting in an expected training accuracy of $1 - \eta$. Additionally, the model successfully classifies a clean test sample w.h.p., leading to the same expected test accuracy. After the second iteration, the model interpolates the training data, achieving a training accuracy of $1$. This is shown empirically in Figure 1. When using a smaller step size, we empirically observe a similar trend: after the first iteration, the model learns the signal tokens, and with more iterations, it captures the noisy tokens of the noisy samples and fits the entire dataset. This behavior is shown in Figure 2.*

### 5.2 PROOF IDEAS FOR SECTION 4

Here we provide the proof sketch for Theorem 6. There are mainly two parts in our proof:

- First we determine the convergence behavior of $\boldsymbol{p}$ and $\boldsymbol{v}$ when the norm constraint $R$ is sufficiently large.
- Using properties derived from this convergence, we can analyze the training and test errors.

The first part of the proof builds upon techniques from Ataee Tarzanagh et al. (2023b), which shows that jointly solving for $\boldsymbol{v}$ and $\boldsymbol{p}$ leads to convergence to their respective max-margin solutions. While their approach focuses on the asymptotic case where $R, r \to \infty$ under specific conditions on the training data, our work extends these techniques to the signal-noise data model and provides non-asymptotic results.

To begin, consider the output of the attention layer $\boldsymbol{r}_i = \boldsymbol{X}_i^\top \mathbb{S}(\boldsymbol{X}_i \boldsymbol{p})$ which is a combination of signal and noise tokens. This can be considered as a "token selection" based on softmax probabilities. Since $\{\boldsymbol{r}_i\}_{i \in [n]}$ determines the model's output, we prove that only by selecting signal tokens for clean samples and noise tokens for noisy samples can we reach the maximum margin when performing SVM on $(\boldsymbol{r}_i, y_i)_{i \in [n]}$ and we refer to this as *optimal tokens*.

**Definition 12** (Optimal Token). *We define the* optimal token *for sample* $(\boldsymbol{X}_i, y_i)$ *as*

$$\boldsymbol{r}_i^\star = \boldsymbol{x}_i^{(1)} = \boldsymbol{\mu}_k, \ i \in \mathcal{C}_k, k \in \{1, 2\} \qquad and \qquad \boldsymbol{r}_i^\star = \boldsymbol{x}_i^{(2)} = \boldsymbol{\xi}_i, \ i \in \mathcal{N} \qquad (4)$$

Based on this optimal token selection, we define the corresponding max-margin solution for $\boldsymbol{p}$ and $\boldsymbol{v}$, denoted by $\boldsymbol{p}_{mm}$ and $\boldsymbol{v}_{mm}$. We first define $\boldsymbol{p}_{mm}$ as follows:

**Definition 13** (p-SVM).

$$\boldsymbol{p}_{mm} = \operatorname*{argmin}_{\boldsymbol{p} \in \mathbb{R}^d} \|\boldsymbol{p}\| \quad subject \ to:$$

$$\boldsymbol{p}^\top(\boldsymbol{\mu}_k - \boldsymbol{\xi}_i) \geq 1, i \in \mathcal{C}_k \quad and \quad \boldsymbol{p}^\top(\boldsymbol{\xi}_i - \boldsymbol{\mu}_i) \geq 1, i \in \mathcal{N}$$

*for all* $k \in \{1, 2\}, i \in [n]$. *Let* $\Xi := 1/\|\boldsymbol{p}_{mm}\|$ *be the margin induced by* $\boldsymbol{p}_{mm}$.

Then for a given $\boldsymbol{p}$, we define $\boldsymbol{v}(\boldsymbol{p})$ as the standard max-margin classifier on $(\boldsymbol{r}_i, y_i)_{i \in [n]}$ and $\boldsymbol{v}_{mm}$ as the standard max-margin classifier on $(\boldsymbol{r}_i^\star, y_i)_{i \in [n]}$ which represents the limiting case when $\boldsymbol{p} = \boldsymbol{p}_{mm}$ and $R \to +\infty$.

**Definition 14** (v-SVM).

$$\boldsymbol{v}(\boldsymbol{p}) := \operatorname*{argmin}_{\boldsymbol{v} \in \mathbb{R}^d} \|\boldsymbol{v}\| \ s.t. \ y_i \cdot \boldsymbol{v}^\top \boldsymbol{r}_i \geq 1, \quad for \ all \ i \in [n]. \qquad (5)$$

$\Gamma(\boldsymbol{p}) := 1/\|\boldsymbol{v}(\boldsymbol{p})\|$ *is the **label margin** induced by* $\boldsymbol{v}(\boldsymbol{p})$. *When* $\boldsymbol{r}_i = \boldsymbol{r}_i^\star, i \in [n]$, *we define*

$$\boldsymbol{v}_{mm} := \operatorname*{argmin}_{\boldsymbol{v} \in \mathbb{R}^d} \|\boldsymbol{v}\| \ s.t. \ y_i \cdot \boldsymbol{v}^\top \boldsymbol{r}_i^\star \geq 1, \quad for \ all \ i \in [n]. \qquad (6)$$

$\Gamma := 1/\|\boldsymbol{v}_{mm}\|$ *is the label margin induced by* $\boldsymbol{v}_{mm}$.

To show the optimality of this token selection, we prove that any other token selection that incorporates other tokens in $\boldsymbol{r}_i$ will strictly reduce the label margin. This is formalized in the following proposition:

**Proposition 15** (optimal token condition). *Suppose that Assumption 5 holds, with probability at least* $1 - \delta$ *over the training dataset, for all* $\boldsymbol{p}$, *the token selection under* $\boldsymbol{p}$ *results in a label margin (Def. 14) of at most* $\Gamma - \frac{C}{\|\boldsymbol{v}_{mm}\|^3 n \rho^2} \cdot \max_{i \in [n]}(1 - s_{i\alpha_i})$ *where* $\alpha_i = \mathbb{I}(i \in \mathcal{C}) + 2\mathbb{I}(i \in \mathcal{N})$ *and* $C > 0$ *is some constant.*

Then, it is natural to make a conjecture that when jointly optimizing $\boldsymbol{p}$ and $\boldsymbol{v}$ for (2), they will converge to their respective max-margin solutions $\boldsymbol{p}_{mm}$ and $\boldsymbol{v}_{mm}$ as $R, r \to \infty$. We verify and formalize it in the following theorem.

**Theorem 16.** *Suppose that Assumption 5 holds, with probability at least* $1 - \delta$ *on the training dataset, we have*

- *The margin induced by* $\boldsymbol{p}_{(r,R)}/R$ *in p-SVM is at least* $(1 - \zeta)\Xi$, *where*

$$\zeta = \frac{\log(4\sqrt{\rho^2 + (1 + \kappa)d}\|\boldsymbol{v}_{mm}\|^3 d\rho^2)}{R\Xi}.$$

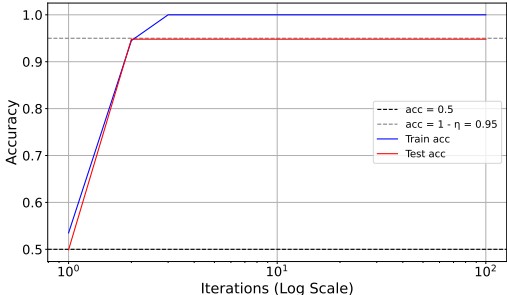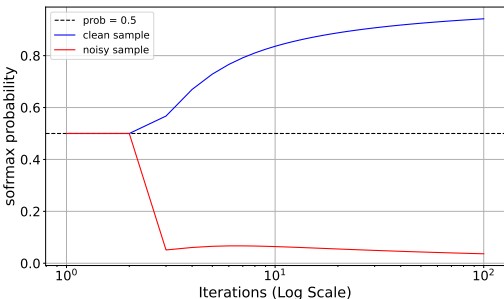

Figure 1: The left panel shows the train and test accuracies during training. It shows that benign overfitting occurs after 2 iterations. After the first iteration, the model correctly classifies the clean training examples, but not the noisy ones. In the right panel, we show the softmax probability of the signal token for clean and noisy samples (average of the softmax probabilities $s_{j,1}^t$ over $\mathcal{C}$ and $\mathcal{N}$ respectively). We see that after 2 iterations, the attention focuses on signal tokens for clean examples, and on noise tokens for noisy examples. This aligns with Theorem 4. Parameters: $n = 200, d = 40000, \beta = 0.025, \rho = 30, \eta = 0.05, \text{test sample size} = 2000$.

- *The label margin induced by $\boldsymbol{v}_{(r,R)}/r$ in v-SVM is at least $(1 - \gamma)\Gamma$, where $\gamma = \frac{2\sqrt{\rho^2+(1+\kappa)d}}{\Gamma \exp((1-\zeta)R\Xi)}$.*

Here, $(\zeta, \gamma)$ quantify the difference between $(\boldsymbol{p}_{(r,R)}, \boldsymbol{v}_{(r,R)})$ and $(\boldsymbol{p}_{mm}, \boldsymbol{v}_{mm})$. As $R \to \infty$, both $\zeta$ and $\gamma$ converge to 0. Thus, for sufficiently large $R$, we conclude that $\boldsymbol{p}_{(r,R)}^\top(\boldsymbol{\mu}_k - \boldsymbol{\xi}_i)$ becomes large for $i \in \mathcal{C}_k$. This ensures that $\boldsymbol{p}_{(r,R)}$ captures sufficient information about signal tokens, which enhances the accuracy of test sample predictions. Specifically, the attention weight on a signal token is lower bounded by $0.5(1 - \zeta)R\Xi \leq \langle \boldsymbol{p}_{(r,R)}, \boldsymbol{\mu}_j \rangle$. Since the signal token remains invariant between training and test data, we can estimate the attention layer's output for a new test sample $(\boldsymbol{X}, y)$.

**Lemma 17.** *Suppose that Assumption 5 holds, with probability at least $1 - \delta$ on the training dataset, for a given test sample $(\boldsymbol{X}, y)$ with $\boldsymbol{X} = (\boldsymbol{\mu}^\star, \boldsymbol{\xi}^\star)$, where the signal $\boldsymbol{\mu}^\star$ can be $\boldsymbol{\mu}_1$ or $\boldsymbol{\mu}_2$, we have with probability at least $1 - \exp\left(-\frac{1}{2}(\frac{1}{2}(1 - \zeta)\Xi - K/R)^2\right)$ that $\langle \boldsymbol{p}_{(r,R)}, \boldsymbol{\mu}^\star \rangle - \langle \boldsymbol{p}_{(r,R)}, \boldsymbol{\xi}^\star \rangle \geq K$, for $K \leq \frac{1}{2}(1 - \zeta)R\Xi$. Here $\zeta, \Xi$ follow the definitions in Theorem 16.*

Therefore, if $K$ is large, which is equivalent to $R$ is large, the attention weight on the signal token is much greater than the noise token. As a result, the signal token $\boldsymbol{\mu}^\star$ will dominate the attention layer's output, i.e. $\boldsymbol{r}^\star \to \boldsymbol{\mu}^\star$.

Finally, from Theorem 16, $\boldsymbol{v}_{(r,R)}$ converges to $\boldsymbol{v}_{mm}$, ensuring that it can make accurate predictions on $(\boldsymbol{\mu}_k, y)$ if $(\boldsymbol{\mu}_k, y)$ comes from the clean set. Thus, w.h.p. the learning of signal token $y \cdot \langle \boldsymbol{v}_{(r,R)}, \boldsymbol{\mu}^\star \rangle$ is large enough to eliminate the randomness introduced by the noise token (denoted by $\Delta(\boldsymbol{\xi}^\star)$ here) and the model will make accurate prediction with high probability: $y \cdot f(\boldsymbol{p}_{(r,R)}, \boldsymbol{v}_{(r,R)}; \boldsymbol{X}) \approx y \cdot \boldsymbol{v}_{(r,R)}^\top \boldsymbol{\mu}^\star - \Delta(\boldsymbol{\xi}^\star) \geq 0$.

## 6 EXPERIMENTS

We complement our theoretical results with an empirical study on benign overfitting in single-head softmax attention. We trained single-head softmax attention models (Eq. (1)) on data generated as specified in Section 2.1 using GD with a fixed step size and the logistic loss function. In all figures, the x-axis corresponds to the time and has a log scale. We add 1 to the time so that the initialization $t = 0$ can be shown in the log scale (i.e. iteration $10^0$ is the initialization).

In Figure 1, we consider a setting similar to Theorem 4, and demonstrate that benign overfitting occurs after two iterations, and that the behavior of GD aligns with our discussion in Remark 11. We also plot how the softmax probabilities evolve during training, and see after two iterations a behavior similar to the first item of Theorem 4. In Figure 2, we consider a similar setting, but with a smaller step size. Here, benign overfitting occurs after about 150 iterations.

In Figure 3a, we explore the behavior of GD with different SNR levels. When the SNR is too small the model exhibits catastrophic overfitting, namely, it fits the training data but has trivial generalization performance. When the SNR is sufficiently large we observe benign overfitting. In Figure 3b, we investigate the overfitting behavior with different dimensions $d$. If $d$ is sufficiently large we observe benign overfitting. If it is very small we are not able to overfit, namely, the training accuracy does not reach $1$. For intermediate values of $d$ we observe harmful overfitting. Thus, we see that high dimensionality is crucial for benign overfitting. Interestingly, we can see that achieving benign overfitting is possible even when $d \ll n^2$, suggesting that our assumption on $d$ in the theoretical results might not be tight.

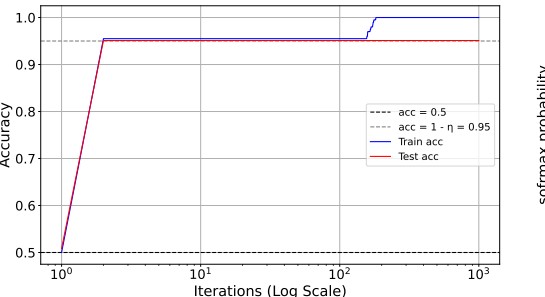 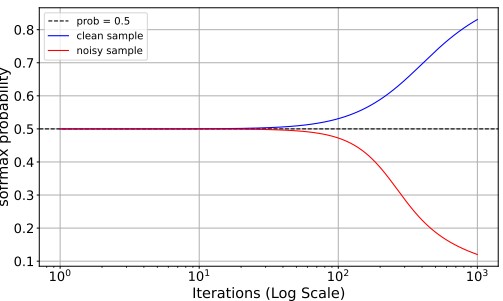

Figure 2: The left panel shows train and test accuracies during training with a small step size. The clean training samples are correctly classified already after one iteration, but in contrast to Theorem 4 and Figure 1, benign overfitting occurs after about $150$ iterations. In the right panel we see that the attention starts separating signal and noise tokens shortly before benign overfitting occurs. Parameters: $n = 200, d = 40000, \beta = 0.0001, \rho = 30, \eta = 0.05$, test sample size $= 2000$.

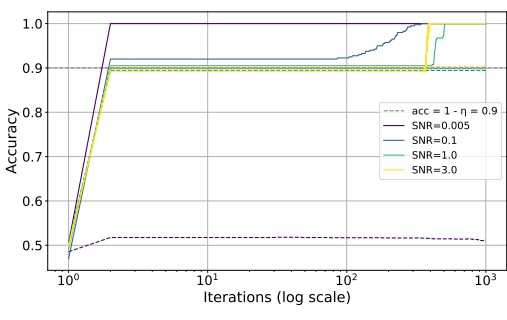

(a) Accuracy - different SNR's

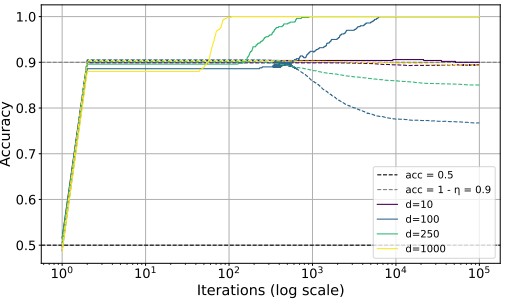

(b) Accuracy - different dimensions $d$

Figure 3: Comparing train (solid lines) and test (dashed lines) accuracies, with different SNR (left panel) and different dimensions (right panel). In the left panel, we observe that for small SNR (purple line), the model exhibits catastrophic overfitting, similar to Theorem 10. For larger SNR values, the model demonstrates benign overfitting. In the right panel, we see that for small $d$ (purple line), the model is unable to fit the data (at least in the first $10^5$ first iterations), and both the train and test accuracies are at the noise-rate level. For intermediate values of $d$ (green and blue lines), the model exhibits harmful overfitting, and for larger $d$ (yellow line) the model exhibits benign overfitting. We note that benign overfitting occurs here for $d = 2n \ll n^2$, which suggests that the assumptions on $d$ in our theorems are loose. Parameters (left panel): $n = 400, d = 40000, \beta = 0.00015, \eta = 0.1$, test sample size $= 2000$. Parameters (right panel): $n = 500, \beta = 0.02, \rho = 30, \eta = 0.1$, test sample size $= 10000$.

# 7 CONCLUSION

This paper took an initial step in establishing the benign overfitting phenomenon in a single-head softmax attention model. Our results open up several future directions, including analyzing gradient descent for more than 2 steps, more complex data distributions containing more than 2 tokens and varying sequence length, and the self-attention architecture.

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

# A  APPENDIX

**Remark 18.** *Throughout our proofs, we assume without loss of generality that $\boldsymbol{\mu}_1 = (\rho, 0, 0, ..., 0)^\top$, $\boldsymbol{\mu}_2 = (0, \rho, 0, ..., 0)^\top$ and $\boldsymbol{\xi}_i = (0, 0, \boldsymbol{\xi}^\top)$ for $\boldsymbol{\xi} \sim \mathcal{N}(\mathbf{0}, \boldsymbol{I}_{d-2})$. Indeed, since $\boldsymbol{\mu}_1$ and $\boldsymbol{\mu}_2$ are orthogonal, we can find orthogonal matrix $\boldsymbol{A} \in \mathbb{R}^{d \times d}$ such that $\boldsymbol{A}\boldsymbol{\mu}_1 = (\rho, 0, 0, ..., 0)^\top, \boldsymbol{A}\boldsymbol{\mu}_2 = (0, \rho, 0, ..., 0)^\top$ and $\boldsymbol{A}\boldsymbol{\xi}_i \sim \mathcal{N}(\boldsymbol{A}\mathbf{0}, \boldsymbol{A}(\boldsymbol{I}_d - \boldsymbol{\mu}_1\boldsymbol{\mu}_1^\top/\rho^2 - \boldsymbol{\mu}_2\boldsymbol{\mu}_2^\top/\rho^2)\boldsymbol{A}^\top)$, which mean that $\boldsymbol{A}\boldsymbol{\xi}_i = (0, 0, \boldsymbol{\xi}^\top)$ for $\boldsymbol{\xi} \sim \mathcal{N}(\mathbf{0}, \boldsymbol{I}_{d-2})$. We emphasize that an orthogonal transformation does not affect our results.*

## A.1  PROOFS FOR SEC. 3

### A.1.1  NOTATIONS FOR SEC. 3.

Given $a, b, c \in \mathbb{R}$, we denote by $c(a \pm b)$ the close segment $[c(a - b), c(a + b)]$. Given vector $\boldsymbol{x}$, we denote by $\boldsymbol{x}[i]$ the $i^{\text{th}}$ coordinate of $\boldsymbol{x}$, and $\boldsymbol{x}[i : j]$ denotes the subvector containing the elements from the $i^{\text{th}}$ to the $j^{\text{th}}$, inclusive. We also list some key notations used in this section for convenience.

Table 1: Usefull notation.

| | |
|---|---|
| $\boldsymbol{x}_{i,j}$ | $j^{\text{th}}$ token in the $i^{\text{th}}$ sample |
| $\boldsymbol{\gamma}_{i,j}^t$ | $y_i \boldsymbol{v}_t^\top \boldsymbol{x}_{i,j}$ i.e. $j^{\text{th}}$ token score in time $t$ |
| $\alpha_{i,j}^t$ | softmax probability of the $j^{\text{th}}$ token in the $i^{\text{th}}$ sample in time $t$ |
| $\ell_{t,i}$ | $\ell(\boldsymbol{X}_i; \boldsymbol{v}_t, \boldsymbol{p}_t)$ |

We remind that $\mathcal{C}, \mathcal{N} \subseteq [n]$ denotes the indices of clean and noisy training examples, and $\mathcal{C}_k, \mathcal{N}_k$ denotes the clean and noisy examples from cluster $k \in \{1, 2\}$. For example if $i \in \mathcal{C}_1$, then $x_{i,1} = \mu_1$ and $y_1 = 1$, and for $j \in \mathcal{N}_1$ we have that $x_{j,1} = \mu_1$ and $y_1 = -1$. Let $\mathbb{S}'(\boldsymbol{v}) := \nabla\mathbb{S}(\boldsymbol{v}) = \text{diag}(\mathbb{S}(\boldsymbol{v})) - \mathbb{S}(\boldsymbol{v})\mathbb{S}(\boldsymbol{v})^\top$ denote the Jacobian of the softmax function $\mathbb{S}(\boldsymbol{v})$ at $\boldsymbol{v} \in \mathbb{R}^d$.

### A.1.2 ADDITIONAL LEMMAS & DEFINITIONS FOR SEC 3.

The following equations will be useful throughout the proof:

$$\nabla_{\boldsymbol{v}}\mathcal{L}(\boldsymbol{v},\boldsymbol{p}) = \frac{1}{n}\sum_{i=1}^{n}\ell'_i \cdot y_i \boldsymbol{X}_i^{\top}\mathbb{S}(X_i\boldsymbol{p}) \tag{7}$$

$$\nabla_{\boldsymbol{p}}\mathcal{L}(\boldsymbol{v},\boldsymbol{p}) = \frac{1}{n}\sum_{i=1}^{n}\ell'_i \cdot \boldsymbol{X}_i^{\top}\mathbb{S}'(X_i\boldsymbol{p})\boldsymbol{\gamma}_i, \quad \text{where } \boldsymbol{\gamma}_i = y_i\boldsymbol{v}^{\top}\boldsymbol{X}_i \tag{8}$$

$$\ell'(x) = -1/(1+\exp(x)) \tag{9}$$

$$\mathbb{S}'(\boldsymbol{v}) = \text{diag}(\mathbb{S}(\boldsymbol{v})) - \mathbb{S}(\boldsymbol{v})\mathbb{S}(\boldsymbol{v})^{\top} \tag{10}$$

**Definition 19** (Good Training Set). *We say that a training set* $(\boldsymbol{X}_1,\dots,\boldsymbol{X}_n)$ *is* good *if*

- $\|\boldsymbol{\xi}_i\|_2^2 \in (1 \pm o_n(1))d$, *for all* $i \in [n]$.

- $|\langle \boldsymbol{\xi}_i, \boldsymbol{\xi}_j \rangle| \leq \sqrt{d\log(12n^2/\delta)}$, *for any* $i,j \in [n]$.

- $|\mathcal{N}_k| \in \frac{n}{2}(\eta \pm o_n(1))$ *and* $|\mathcal{C}_k| = \frac{n}{2}(1 - \eta \pm o_n(1))$, *for* $k \in \{1,2\}$.

**Definition 20** (Good Test Sample). *We say that a test sample* $(\boldsymbol{X} = (\boldsymbol{x}_1,\boldsymbol{x}_2), y)$ *is* good *w.r.t. a training set* $(\boldsymbol{X}_1,\dots,\boldsymbol{X}_n)$ *and constant* $C_1$ *if*

$$|\langle \boldsymbol{x}_{i,2}, \boldsymbol{x}_2 \rangle| \leq \frac{d}{C_1 n}, \quad \forall i \in [n]$$

Next we write Lemma 59 slightly different, and also add a formal proof for completeness:

**Lemma 21.** *Let* $\delta > 0$ *and* $C > 0$. *Suppose that Assumption 23 (item 1) holds with constant* $C$, *then with probability at least* $1 - \delta/2$ *we have that*

$$|\mathcal{C}_k| \in \frac{n}{2}(1 - \eta \pm \sqrt{2/C}), \quad |\mathcal{N}_k| \in \frac{n}{2}(\eta \pm \sqrt{2/C}), \quad \forall k \in \{1,2\}.$$

*Moreover, we have*

$$|\mathcal{C}_k| \in \frac{n}{2}(1 - \eta \pm o_n(1)), \quad |\mathcal{N}_k| \in \frac{n}{2}(\eta \pm o_n(1)), \quad \forall k \in \{1,2\}.$$

*Proof.* By Hoeffding's inequality,

$$\mathbb{P}\left(\left||\mathcal{C}_j| - \frac{n}{2}(1-\eta)\right| \geq \sqrt{n\log(16/\delta)/2}\right) \leq \delta/8,$$

which means that with probability at least $1 - \delta/8$ we have that $|\mathcal{C}_j| \in \frac{n}{2}(1 - \eta \pm c_n)$, where $c_n = \sqrt{2n\log(16/\delta)}/n$. Hence, if $n \geq C\log(16/\delta)$, then $c_n = \sqrt{2\log(16/\delta)}/\sqrt{n} \leq \sqrt{2/C}$. Similarly, we can estimate $|\mathcal{N}_k|$ for $k \in \{1,2\}$, and by union bound, the result follows. $\square$

**Lemma 22.** *Let* $\boldsymbol{z},\boldsymbol{\gamma},\boldsymbol{p} \in \mathbb{R}^2$ *and let* $\boldsymbol{\alpha} = \mathbb{S}(\boldsymbol{p})$, *then*

$$\boldsymbol{z}^T\mathbb{S}'(\boldsymbol{p})\boldsymbol{\gamma} = (\gamma_1 - \gamma_2)(1 - \alpha_1)\alpha_1(z_1 - z_2)$$

*Proof.* Observe that $\alpha_1 + \alpha_2 = 1$. Therefore,

$$\boldsymbol{z}^T\mathbb{S}'(\boldsymbol{p})\boldsymbol{\gamma} = \boldsymbol{z}^T\text{diag}(\boldsymbol{\alpha})\boldsymbol{\gamma} - \boldsymbol{z}^T\boldsymbol{\alpha}\boldsymbol{\alpha}^{\top}\boldsymbol{\gamma} = \sum_{i=1}^{2}z_i\alpha_i\gamma_i - \sum_{i=1}^{2}\alpha_i z_i \sum_{i=1}^{2}\alpha_i\gamma_i$$

$$= z_1\alpha_1\gamma_1 + z_2\alpha_2\gamma_2 - (\alpha_1 z_1 + \alpha_2 z_2)(\alpha_1\gamma_1 + \alpha_2\gamma_2)$$
$$= (\gamma_2 - (\alpha_1\gamma_1 + \alpha_2\gamma_2))\alpha_2 z_2 + (\gamma_1 - (\alpha_1\gamma_1 + \alpha_2\gamma_2))\alpha_1 z_1$$
$$= (\alpha_1\gamma_2 - \alpha_1\gamma_1)\alpha_2 z_2 + (\alpha_2\gamma_1 - \alpha_2\gamma_2)\alpha_1 z_1$$
$$= -\alpha_1(\gamma_1 - \gamma_2)\alpha_2 z_2 + \alpha_2(\gamma_1 - \gamma_2)\alpha_1 z_1$$
$$= \alpha_1(\gamma_1 - \gamma_2)\alpha_2(z_1 - z_2)$$

$\square$

Lemma 22 allows us to analyze $\nabla_{\boldsymbol{p}}\mathcal{L}$ as a function of the score gap.

A.1.3 PROOF OF THM. 4

*Proof of Thm. 4.* To simplify the proof, we will use the following assumption, which is slightly weaker than Assumption 3:

**Assumption 23** (Assumptions for GD with SNR = $\Theta(1/\sqrt{n})$). *Let $\delta > 0$ be a desired probability of failure. For constants $c_\rho \geq 6, c_\beta \geq 16c_\rho \log(c_\rho^2)$, there exists some large enough constant $C = C(c_\beta)$, such that the following hold:*

1. *Number of samples $n$ should be sufficiently large: $n \geq C \log(16/\delta)$*

2. *Dimension $d$ should be sufficiently large: $d \geq Cn^2 \log(12n^2/\delta)$.*

3. *Signal strength is: $\rho = c_\rho \sqrt{d/n}$*

4. *Label flipping rate $\eta$: $\eta \leq 1/C$.*

5. *The step size $\beta$ satisfies: $\beta = (c_\beta \cdot n)/(c_\rho^2 \cdot d)$.*

6. *Initialization at zero: $\|\boldsymbol{v}_0\| = \|\boldsymbol{p}_0\| = 0$.*

Apart from slight adjustments to the constants within the logarithm at items 1 and 2 (which can be absorbed into $C$), the only changes are $c_\beta \geq 16c_\rho \log(c_\rho^2)$ (instead of $C_\beta \geq 16$) and $\beta = (c_\beta \cdot n)/(c_\rho^2 \cdot d)$ (instead of $\beta = C_\beta \cdot (n/d)$). Indeed, given $C_\beta \geq 16, C_\rho \geq 6$ and $\beta = C_\beta \cdot (n/d)$ which satisfy Assumption 3, define $c_\rho := C_\rho \geq 6, c_\beta := C_\beta c_\rho^2 \geq 16c_\rho \log(c_\rho^2)$, which holds for any $c_\rho \geq 6$. We also have that $\beta = C_\beta \cdot (n/d) = (c_\beta/c_\rho^2) \cdot (n/d)$, i.e., $\beta, c_\rho, c_\beta$ satisfy Assumption 23.

Next, under Assumption 23, we argue that with probability at least $1 - \delta$ the training set is good (Def. 19) i.e.:

- $|\mathcal{C}_k| \in \frac{n}{2}(\eta \pm o_n(1))$ and $\mathcal{N}_k \in \frac{n}{2}(1 - \eta \pm o_n(1))$, for $k \in \{1, 2\}$.

- $\|\boldsymbol{\xi}_i\|_2^2 \in (1 \pm o_n(1))d$, for any $i \in [n]$.

- $|\langle \boldsymbol{\xi}_i, \boldsymbol{\xi}_j \rangle| \leq \sqrt{d \log(12n^2/\delta)}$, for any $i, j \in [n]$.

Indeed, this holds by Lemma 57, Lemma 21, and the union bound. We emphasize that the notation $o_n(1)$ represents a term that becomes arbitrarily small as $n$ increases, and thus it can be bounded by a small constant if $C$ from Assumption 1 is large enough.

Next, we show that under a good training set, the model exhibits benign overfitting, already after two iterations. See Remark 18 for the data setting used throughout the proof.

**GD after 1 iteration.** We start by analyzing the first coordinate of $\boldsymbol{v}_1$ (i.e. $\boldsymbol{v}$ after one iteration of GD). By assumption 23 ( item6), we have that $\boldsymbol{p}_0 = \boldsymbol{v}_0 = \boldsymbol{0}$, which implies that $\ell'_{0,i} = -1/2$, for any $i \in [n]$. Hence

$$-\beta \nabla_{\boldsymbol{v}} \mathcal{L}(\boldsymbol{v}_0, \boldsymbol{p}_0)[1] = -\frac{\beta}{2n} \sum_{i=1}^n \ell'_{0,i} \cdot y_i \boldsymbol{x}_{i,1}[1] = \frac{\beta}{4n} \sum_{i \in \mathcal{C}_1} y_i \rho + \frac{\beta}{4n} \sum_{i \in \mathcal{N}_1} y_i \rho$$

$$= \frac{\beta}{4n}(|\mathcal{C}_1| - |\mathcal{N}_1|)\rho$$

$$\in \frac{\beta}{8}(1 - 2\eta \pm o_n(1))\rho \qquad \text{"good" training set}$$

In the same way, we can estimate the second coordinate of $\boldsymbol{v}_{t=1}$:

$$\boldsymbol{v}_{t=1}[2] = \frac{\beta}{4n} \sum_{i \in \mathcal{C}_2} y_i \rho + \frac{\beta}{4n} \sum_{i \in \mathcal{N}_2} y_i \rho \in -\frac{\beta}{8}(1 - 2\eta \pm o_n(1))\rho,$$

where we remind that $y_i = -1$, when $i \in \mathcal{C}_2$, hence $\boldsymbol{v}_{t=1}[2]$ has the same bounds as $\boldsymbol{v}_{t=1}[1]$, just with opposite sign. We move to analyze the rest of the coordinates of $\boldsymbol{v}_{t=1}$:

$$\boldsymbol{v}_t[3:d] = \frac{\beta}{4n} \sum_{i=1}^{n} y_i \boldsymbol{\xi}_i.$$

Overall, we can write $\boldsymbol{v}_{t=1}$ as $\lambda_1^{t=1} \boldsymbol{\mu}_1 + \lambda_2^{t=1} \boldsymbol{\mu}_2 + \sum_{i=1}^{n} y_i \theta_i^{t=1} \boldsymbol{\xi}_i$ with

$$\lambda_1^{t=1} \in \frac{\beta}{8}(1 - 2\eta \pm o_n(1)), \ \ \lambda_2^{t=1} \in -\frac{\beta}{8}(1 - 2\eta \pm o_n(1)), \ \ \theta_i^{t=1} = \frac{\beta}{4n}. \tag{11}$$

Moreover, since $\boldsymbol{\gamma}_i^{t=0} = \boldsymbol{0}$ for every $i \in [n]$, we have that $\boldsymbol{p}_1 = \boldsymbol{0}$ (see Eq. 8).

**Preparation for next iteration.** To estimate $(\boldsymbol{v}_{t=2}, \boldsymbol{p}_{t=2})$, we first need to estimate the loss for clean/noisy samples and the score difference, i.e. $\gamma_{i,1}^1 - \gamma_{i,2}^1, i \in \mathcal{C}$ and $\gamma_{j,2}^1 - \gamma_{j,1}^1, j \in \mathcal{N}$.

We remind that $\|\boldsymbol{\mu}_j\|^2 = \rho^2 = c_\rho^2 d/n$ (Assumption 23 (item 3)). For $j \in \mathcal{C}_k$, where $k \in \{1, 2\}$ we have that

$$
\begin{aligned}
y_j f(\boldsymbol{X}_j; \boldsymbol{v}_{t=1}, \boldsymbol{p}_{t=1}) &= \frac{1}{2} y_j \boldsymbol{v}_{t=1}^\top (\boldsymbol{x}_{j,1} + \boldsymbol{x}_{j,2}) && \text{since } \boldsymbol{p}_1 = \boldsymbol{0} \\
&= \frac{1}{2} |\lambda_k^{t=1}| \|\boldsymbol{\mu}_k\|^2 + \frac{1}{2} \theta_j^{t=1} \|\boldsymbol{\xi}_j\|^2 + \frac{1}{2} \sum_{i \in [n]: i \neq j} y_i y_j \theta_j^{t=1} \boldsymbol{\xi}_i^\top \boldsymbol{\xi}_j && y_j \lambda_k^{t=1} > 0
\end{aligned}
$$
$$\tag{12}$$

Since the training set is "good" then by Eq. 11, we can bound $y_j f(\boldsymbol{X}_j; \boldsymbol{v}_{t=1}, \boldsymbol{p}_{t=1})$ as follows:

$$
\begin{aligned}
y_j f(\boldsymbol{X}_j; \boldsymbol{v}_{t=1}, \boldsymbol{p}_{t=1}) &\leq \frac{\beta}{16}(1 - 2\eta + o_n(1)) \cdot c_\rho^2 \cdot \frac{d}{n} + \frac{\beta}{8n} d(1 + o_n(1)) + \frac{\beta}{8n} n\sqrt{d \log(12n^2/\delta)} \\
&\leq \left( \frac{c_\rho^2(1 - 2\eta) + 2 + o_n(1)}{16} \right) \cdot \frac{\beta d}{n} && \text{Assumption 23 (item 2)} \\
&= c_\beta \cdot \left( \frac{(1 - 2\eta) + 2/c_\rho^2 + o_n(1)}{16} \right) && \text{Assumption 23 (item 5)} \\
&\leq \frac{1.1 c_\beta}{16}, && \tag{13}
\end{aligned}
$$

where the last inequality holds since $c_\rho \geq 5$, which implies that $2/c_\rho^2 + o_n(1) \leq 0.1$. Similarly, we have that

$$
\begin{aligned}
y_j f(\boldsymbol{X}_j; \boldsymbol{v}_{t=1}, \boldsymbol{p}_{t=1}) &\geq \frac{\beta}{16}(1 - 2\eta - o_n(1)) \cdot c_\rho^2 \cdot \frac{d}{n} + \frac{\beta}{8n} d(1 - o_n(1)) - \frac{\beta}{8n} n\sqrt{d \log(12n^2/\delta)} \\
&\geq \left( \frac{c_\rho^2(1 - 2\eta) + 2 - o_n(1)}{16} \right) \cdot \frac{\beta d}{n} \\
&= c_\beta \cdot \left( \frac{(1 - 2\eta) + 2/c_\rho^2 - o_n(1)}{16} \right) \\
&\geq \frac{0.9 c_\beta}{16} && \tag{14}
\end{aligned}
$$

For $j \in \mathcal{N}_k$ we have:

$$
\begin{aligned}
y_j f(\boldsymbol{X}_j; \boldsymbol{v}_{t=1}, \boldsymbol{p}_{t=1}) &= \frac{1}{2} y_j \boldsymbol{v}_{t=1}^\top (\boldsymbol{x}_{j,1} + \boldsymbol{x}_{j,2}) && \text{since } \boldsymbol{p}_1 = \boldsymbol{0} \\
&= -\frac{1}{2} |\lambda_k^{t=1}| \|\boldsymbol{\mu}_k\|^2 + \frac{1}{2} \theta_j^{t=1} \|\boldsymbol{\xi}_j\|^2 + \frac{\beta}{8n} \sum_{i \in [n]: i \neq j} y_i y_j \boldsymbol{\xi}_i^\top \boldsymbol{\xi}_j && y_j \lambda_k^{t=1} < 0
\end{aligned}
$$

Since the training set is good then by Eq. 11, we can bound $y_j f(\boldsymbol{X}_j; \boldsymbol{v}_{t=1}, \boldsymbol{p}_{t=1})$ as follows:

$$
\begin{aligned}
y_j f(\boldsymbol{X}_j; \boldsymbol{v}_{t=1}, \boldsymbol{p}_{t=1}) &\leq -\frac{\beta}{16}(1 - 2\eta - o_n(1)) \cdot c_\rho^2 \cdot \frac{d}{n} + \frac{\beta}{8n}d(1 + o_n(1)) + \frac{\beta}{8n}n\sqrt{d\log(12n^2/\delta)} \\
&\leq \left(\frac{-c_\rho^2(1 - 2\eta) + 2 + o_n(1)}{16}\right) \cdot \frac{\beta d}{n} \\
&= c_\beta \cdot \left(\frac{-(1 - 2\eta) + 2/c_\rho^2 + o_n(1)}{16}\right) \\
&\leq \frac{-0.9 c_\beta}{16},
\end{aligned}
\tag{15}
$$

where the last inequality holds for small enough $\eta$ and since $c_\rho \geq 5$, which implies that $2/c_\rho^2 + 2\eta + o_n(1) \leq 0.1$. Similarly, we have that

$$
\begin{aligned}
y_j f(\boldsymbol{X}_j; \boldsymbol{v}_{t=1}, \boldsymbol{p}_{t=1}) &\geq -\frac{\beta}{16}(1 - 2\eta + o_n(1)) \cdot c_\rho^2 \cdot \frac{d}{n} + \frac{\beta}{8n}d(1 - o_n(1)) - \frac{\beta}{8n}n\sqrt{d\log(12n^2/\delta)} \\
&\geq \left(\frac{-c_\rho^2(1 - 2\eta) + 2 - o_n(1)}{16}\right) \cdot \frac{\beta d}{n} \\
&\geq c_\beta \left(\frac{-(1 - 2\eta) + 2/c_\rho^2 - o_n(1)}{16}\right) \\
&\geq \frac{-1.1 c_\beta}{16}.
\end{aligned}
\tag{16}
$$

We remind that $-\ell'_{1,j} = 1/(1 + \exp(y_i f(\boldsymbol{X}_i; \boldsymbol{v}_{t=1}, \boldsymbol{p}_{t=1})))$ and that $\beta = c_\beta \cdot n/(dc_\rho^2)$ for some constant $c_\beta \geq 16c_\rho$. Combine with Eqs. 13 and 14, we have that

$$
i \in \mathcal{C}, \quad -\ell'_{t=1,i} \geq 1/(1 + \exp(1.1 c_\beta/16)) := m_{\mathcal{C}}^{t=1} > 0
\tag{17}
$$

$$
i \in \mathcal{C}, \quad -\ell'_{t=1,i} \leq 1/(1 + \exp(0.9 c_\beta/16)) := M_{\mathcal{C}}^{t=1} \leq 1/(4c_\rho^2),
\tag{18}
$$

where the last inequality holds since $c_\beta \geq 16c_\rho$ and since $1 + \exp(0.9 c_\rho) \geq 4c_\rho^2$ for any $c_\rho \geq 6$.

Moreover, by Eqs. 15 and 16, we have that

$$
j \in \mathcal{N}, \quad -\ell'_{t=1,j} \geq 1/(1 + \exp(-0.9 c_\beta/16)) := m_{\mathcal{N}}^{t=1} \geq 0.99
\tag{19}
$$

$$
j \in \mathcal{N}, \quad -\ell'_{t=1,j} \leq 1/(1 + \exp(-1.1 c_\beta/16)) := M_{\mathcal{N}}^{t=1} \leq 1
\tag{20}
$$

The notations $M_{\mathcal{C}}^t$ and $m_{\mathcal{C}}^t$ ($M_{\mathcal{N}}^t$ and $m_{\mathcal{N}}^t$) denote the upper and lower bounds, respectively, on the derivative of the loss for clean (noisy) samples at time $t$, and we use them throughout the proof. We remind that $\gamma_{i,j}^t = y_i \boldsymbol{v}_t^\top \boldsymbol{x}_{i,j}$. Then by Eq. 11, for $i \in \mathcal{C}_k$ we have that

$$
\begin{aligned}
\gamma_{i,1}^{t=1} &\in \frac{\beta}{8}(1 - 2\eta \pm o_n(1))\rho^2 = \frac{c_\beta}{8}(1 - 2\eta \pm o_n(1)) \\
\gamma_{i,2}^{t=1} &\in \frac{\beta d}{4n}(1 \pm o_n(1)) = \frac{c_\beta}{4}(1/c_\rho^2 \pm o_n(1)) \\
\gamma_{i,1}^{t=1} - \gamma_{i,2}^{t=2} &\in \frac{c_\beta}{8}(1 - 2/c_\rho^2 - 2\eta \pm o_n(1)).
\end{aligned}
\tag{21}
$$

where in the calculation of $\gamma_{i,2}^{t=1}$ we use $\sum_{i \in [n]: i \neq j} y_i y_j \theta_j^{t=1} \boldsymbol{\xi}_i^\top \boldsymbol{\xi}_j = o_n(1) \cdot d$, since the training set is good. For $i \in \mathcal{N}_k$, we have that

$$
\begin{aligned}
\gamma_{i,1}^{t=1} &\in -\frac{\beta}{8}(1 - 2\eta \pm o_n(1))\rho^2 = -\frac{c_\beta}{8}(1 - 2\eta \pm o_n(1)) \\
\gamma_{i,2}^{t=1} &\in \frac{\beta d}{4n}(1 \pm o_n(1)) = \frac{c_\beta}{4}(1/c_\rho^2 \pm o_n(1)) \\
\gamma_{i,2}^{t=1} - \gamma_{i,1}^{t=2} &\in \frac{c_\beta}{8}(1 + 2/c_\rho^2 - 2\eta \pm o_n(1)).
\end{aligned}
\tag{22}
$$

**GD after 2 iterations.**

**Analysis of $v_{t=2}$.**

Observe that

$$-\beta\nabla_{\boldsymbol{v}}\mathcal{L}(\boldsymbol{v}_1,\boldsymbol{p}_1) = -\frac{\beta}{n}\sum_{i=1}^n \ell'_{1,i}\cdot y_i\boldsymbol{X}_i^\top\mathbb{S}(X_i\boldsymbol{p}_1) = -\frac{\beta}{2n}\sum_{i=1}^n \ell'_{1,i}\cdot y_i(\boldsymbol{x}_{i,1}+\boldsymbol{x}_{i,2}).$$

We start by analyzing the first coordinate of $\nabla_{\boldsymbol{v}}\mathcal{L}(\boldsymbol{v}_1,\boldsymbol{p}_1)$.

$$-\beta\nabla_{\boldsymbol{v}}\mathcal{L}(\boldsymbol{v}_1,\boldsymbol{p}_1)[1] = \frac{\beta}{2n}\sum_{i\in\mathcal{C}_1}-\ell'_{1,i}\cdot y_i\boldsymbol{x}_{i,1}[1] + \frac{\beta}{2n}\sum_{i\in\mathcal{N}_1}-\ell'_{1,i}\cdot y_i\boldsymbol{x}_{i,1}[1]$$

$$= \frac{\beta}{2n}\sum_{i\in\mathcal{C}_1}-\ell'_{1,i}\cdot\rho - \frac{\beta}{2n}\sum_{i\in\mathcal{N}_1}-\ell'_{1,i}\cdot\rho$$

$$= \frac{\beta}{2n}\left(\sum_{i\in\mathcal{C}_1}-\ell'_{1,i} - \sum_{j\in\mathcal{N}_1}-\ell'_{1,j}\right)\cdot\rho. \tag{23}$$

Observe that

$$\sum_{i\in\mathcal{C}_k}-\ell'_{1,i} - \sum_{j\in\mathcal{N}_k}-\ell'_{1,j} \geq \frac{n}{2}(1-\eta-o_n(1))\cdot m_{\mathcal{C}} - \frac{n}{2}(\eta+o_n(1))\cdot M_{\mathcal{N}} \quad\text{good training set}$$

$$> 0 \qquad\qquad\qquad\qquad\qquad\qquad\qquad\text{Eqs. 17 and 20,}$$

where the last inequality holds for small enough $\eta\leq 1/C$, where $C:=C(c_\rho,c_\beta)$ (see Assumption 4). Substituting it into Eq. 23, we obtain that

$$-\beta\nabla_{\boldsymbol{v}}\mathcal{L}(\boldsymbol{v}_1,\boldsymbol{p}_1)[1] > 0.$$

On the other hand, by Eq. 23, we can upper bound the first coordinate of the gradient of $\boldsymbol{v}$ by

$$-\beta\nabla_{\boldsymbol{v}}\mathcal{L}(\boldsymbol{v}_1,\boldsymbol{p}_1)[1] \leq \frac{\beta}{2n}\left(\sum_{i\in\mathcal{C}_1}-\ell'_{1,i}\right)\cdot\rho$$

$$\leq \frac{\beta}{17}\cdot\rho \qquad\qquad\qquad -\ell'_{1,i}<1/16,\text{Eq. 18.}$$

Similarly, we can estimate the second coordinate of $\nabla_{\boldsymbol{v}}\mathcal{L}(\boldsymbol{v}_1,\boldsymbol{p}_1)$:

$$0 \geq -\beta\nabla_{\boldsymbol{v}}\mathcal{L}(\boldsymbol{v}_1,\boldsymbol{p}_1)[2] \geq -\frac{\beta}{17}\cdot\rho.$$

Write $\boldsymbol{v}_{t=2} = \lambda_1^{t=2}\boldsymbol{\mu}_1 + \lambda_2^{t=2}\boldsymbol{\mu}_2 + \sum_{i=1}^n y_i\theta_i^{t=2}\boldsymbol{\xi}_i$. Together with Eq. 11, we get that

$$\lambda_1^{t=2} = \lambda_1^{t=1} - \beta\nabla_{\boldsymbol{v}}\mathcal{L}(\boldsymbol{v}_1,\boldsymbol{p}_1)[1]/\rho \leq \frac{\beta}{8}(1+o_n(1)) + \frac{\beta}{17} \leq \frac{3\beta}{16} \tag{24}$$

$$\lambda_1^{t=2} \geq \lambda_1^{t=1} \geq \frac{\beta}{8}(1-2\eta-o_n(1)) \tag{25}$$

$$\lambda_2^{t=2} = \lambda_2^{t=1} - \beta\nabla_{\boldsymbol{v}}\mathcal{L}(\boldsymbol{v}_1,\boldsymbol{p}_1)[2] \geq -\frac{\beta}{8}(1+o_n(1)) - \frac{\beta}{17} \geq -\frac{3\beta}{16} \tag{26}$$

$$\lambda_2^{t=2} \leq \lambda_2^{t=1} \leq -\frac{\beta}{8}(1-2\eta-o_n(1)). \tag{27}$$

Next, we analyze the rest of the coordinates of $\nabla_{\boldsymbol{v}}\mathcal{L}(\boldsymbol{v}_1,\boldsymbol{p}_1)$.

$$-\beta\nabla_{\boldsymbol{v}}\mathcal{L}(\boldsymbol{v}_1,\boldsymbol{p}_1)[3:d] = \frac{\beta}{2n}\sum_{i\in\mathcal{C}}-\ell'_{1,i}\cdot y_i\boldsymbol{\xi}_i + \frac{\beta}{2n}\sum_{j\in\mathcal{N}}-\ell'_{1,j}\cdot y_j\boldsymbol{\xi}_j,$$

and use it to analyze the coefficients of the noise (second) tokens in $\boldsymbol{v}_{t=2}$, i.e., $\theta_i^{t=2}$. Indeed, for $i\in\mathcal{C}$ we have that

$$\theta_i^{t=2} = \theta_i^{t=1} - \frac{\beta}{2n}\ell'_{1,i} = \frac{\beta}{2n}(-\ell'_{1,i}+0.5) \qquad\text{Eq. 11}$$

$$\in \left[\frac{\beta}{2n}(m_{\mathcal{C}}+0.5), \frac{\beta}{2n}(M_{\mathcal{C}}+0.5)\right]. \tag{28}$$

For $j \in \mathcal{N}$ we have that

$$\theta_j^{t=2} = \theta_j^{t=1} - \frac{\beta}{2n}\ell'_{1,j} = \frac{\beta}{2n}(-\ell'_{1,j} + 0.5) \qquad \text{Eq. 11}$$

$$\in \left[\frac{\beta}{2n}(m_\mathcal{N} + 0.5), \frac{\beta}{2n}(M_\mathcal{N} + 0.5)\right]. \tag{29}$$

Now we move to analyze $\boldsymbol{p}_{t=2}$ and show that $\boldsymbol{p}_{t=2}$ focuses on optimal tokens for training samples.

$\boldsymbol{p}_{t=2}$ **focuses on optimal tokens**. Observe that $\boldsymbol{p}_2 = -\beta\nabla_{\boldsymbol{p}}\mathcal{L}(\boldsymbol{v}_1, \boldsymbol{p}_1)$. Therefore, for $j \in \mathcal{C}_k$

$$\boldsymbol{p}_2^\top(\boldsymbol{x}_{j,1} - \boldsymbol{x}_{j,2})$$

$$= -(\boldsymbol{x}_{j,1} - \boldsymbol{x}_{j,2})^\top\beta\nabla_{\boldsymbol{p}}\mathcal{L}(\boldsymbol{v}_t, \boldsymbol{p}_t) = (\boldsymbol{x}_{j,1} - \boldsymbol{x}_{j,2})^\top\frac{\beta}{n}\sum_{i=1}^n -\ell'_{1,i} \cdot \boldsymbol{X}_i^\top\mathbb{S}'(X_i\boldsymbol{p}_t)\boldsymbol{\gamma}_i^{t=1}$$

$$= \frac{\beta}{n}\sum_{i=1}^n -\ell'_{1,i} \cdot \boldsymbol{x}_{j,1}^\top\boldsymbol{X}_i^\top\mathbb{S}'(X_i\boldsymbol{p}_t)\boldsymbol{\gamma}_i^{t=1} - \frac{\beta}{n}\sum_{i=1}^n -\ell'_{1,i} \cdot \boldsymbol{x}_{j,2}^\top\boldsymbol{X}_i^\top\mathbb{S}'(X_i\boldsymbol{p}_t)\boldsymbol{\gamma}_i^{t=1}$$

$$= \frac{\beta}{n}\sum_{i\in[n]} -\ell'_{1,i} \cdot (\gamma_{i,1}^{t=1} - \gamma_{i,2}^{t=1})(1 - \alpha_{i,1}^{t=1})\alpha_{i,1}^{t=1}(\boldsymbol{x}_{j,1}^\top\boldsymbol{x}_{i,1} + \boldsymbol{x}_{j,2}^\top\boldsymbol{x}_{i,2}) \qquad \text{Lemma 22}$$

$$= \frac{\beta}{n}(-\ell'_{1,j})(\gamma_{j,1}^{t=1} - \gamma_{j,2}^{t=1})(1 - \alpha_{j,1})\alpha_{j,1}(\|\boldsymbol{x}_{j,1}\|^2 + \|\boldsymbol{x}_{j,2}\|^2)$$

$$+ \frac{\beta}{n}\sum_{i\in\mathcal{C}_k:i\neq j} -\ell'_{1,i} \cdot (\gamma_{i,1}^{t=1} - \gamma_{i,2}^{t=1})(1 - \alpha_{i,1}^{t=1})\alpha_{i,1}^{t=1}(\boldsymbol{x}_{j,1}^\top\boldsymbol{x}_{i,1})$$

$$- \frac{\beta}{n}\sum_{i\in\mathcal{N}_k:i\neq j} -\ell'_{1,i} \cdot (\gamma_{i,2}^{t=1} - \gamma_{i,1}^{t=1})(1 - \alpha_{i,1}^{t=1})\alpha_{i,1}^{t=1}(\boldsymbol{x}_{j,1}^\top\boldsymbol{x}_{i,1})$$

$$+ \frac{\beta}{n}\sum_{i\in[n]:i\neq j} -\ell'_{1,i} \cdot (\gamma_{i,1}^{t=1} - \gamma_{i,2}^{t=1})(1 - \alpha_{i,1}^{t=1})\alpha_{i,1}^{t=1}(\boldsymbol{x}_{j,2}^\top\boldsymbol{x}_{i,2}).$$

Observe that $\alpha_{i,1}^{t=1} = \alpha_{i,2}^{t=1} = 1/2$. In Eqs. 21 and 22 we calculate the score differences (e.g. $\gamma_{i,1}^{t=1} - \gamma_{i,2}^{t=1}$). Overall, we can lower bound the above equation by:

$$\geq \frac{\beta}{4n}\left(m_\mathcal{C} \cdot \frac{c_\beta}{8}(1 - 2/c_\rho^2 - 2\eta - o_n(1)) \cdot d(1 - o_n(1))\right)$$

$$+ \frac{\beta}{4n}\left((1 - \eta - o_n(1)) \cdot \frac{n}{2} \cdot m_\mathcal{C}\frac{c_\beta}{8}(1 - 2/c_\rho^2 - 2\eta - o_n(1))\frac{d}{n}c_\rho^2\right)$$

$$- \frac{\beta}{4n}\left((\eta + o_n(1)) \cdot \frac{n}{2} \cdot M_\mathcal{N}\frac{c_\beta}{8}(1 + 2/c_\rho^2 - 2\eta + o_n(1))\frac{d}{n}c_\rho^2\right)$$

$$- \frac{\beta}{4n}\left(n \cdot M_\mathcal{N}\frac{c_\beta}{8}(1 + 2/c_\rho^2 - 2\eta + o_n(1))\sqrt{d\log(12n^2/\delta)}\right).$$

The first term dominates the last term since $d \gg n\sqrt{d\log(12n^2/\delta)}$ (see Assumption 23 (item 2)). The second term dominates the third term for small enough $\eta$ (see Assumption 4). Overall, we obtain that

$$\boldsymbol{p}_2^\top(\boldsymbol{x}_{j,1} - \boldsymbol{x}_{j,2}) > 0, \tag{30}$$

which means that for any $i \in \mathcal{C}$ we have:

$$\alpha_{i,1}^{t=2} = \frac{1}{1 + \exp(-\boldsymbol{p}_2^\top(\boldsymbol{x}_{j,1} - \boldsymbol{x}_{j,2}))} > \frac{1}{2}. \tag{31}$$

For $j \in \mathcal{N}_k$,

$$\boldsymbol{p}_2^\top (\boldsymbol{x}_{j,2} - \boldsymbol{x}_{j,1}) = \frac{\beta}{n}(-\ell'_{1,j})(\gamma_{j,2}^{t=1} - \gamma_{j,1}^{t=1})(1 - \alpha_{j,1})\alpha_{j,1}(\|\boldsymbol{x}_{j,1}\|^2 + \|\boldsymbol{x}_{j,2}\|^2)$$

$$- \frac{\beta}{n} \sum_{i \in \mathcal{C}_k : i \neq j} -\ell'_{1,i} \cdot (\gamma_{i,1}^{t=1} - \gamma_{i,2}^{t=1})(1 - \alpha_{i,1}^{t=1})\alpha_{i,1}^{t=1}(\boldsymbol{x}_{j,1}^\top \boldsymbol{x}_{i,1})$$

$$+ \frac{\beta}{n} \sum_{i \in \mathcal{N}_k : i \neq j} -\ell'_{1,i} \cdot (\gamma_{i,2}^{t=1} - \gamma_{i,1}^{t=1})(1 - \alpha_{i,1}^{t=1})\alpha_{i,1}^{t=1}(\boldsymbol{x}_{j,1}^\top \boldsymbol{x}_{i,1})$$

$$- \frac{\beta}{n} \sum_{i \in [n] : i \neq j} -\ell'_{1,i} \cdot (\gamma_{i,1}^{t=1} - \gamma_{i,2}^{t=1})(1 - \alpha_{i,1}^{t=1})\alpha_{i,1}^{t=1}(\boldsymbol{x}_{j,2}^\top \boldsymbol{x}_{i,2}) \quad \text{Lemma 22}$$

Observe that $\alpha_{i,1}^{t=1} = \alpha_{i,2}^{t=1} = 1/2$. In Eq. 21 and Eq. 22 we calculate the score differences (e.g. $\gamma_{i,1}^{t=1} - \gamma_{i,2}^{t=1}$). Overall, we can lower bound the above equation by:

$$\geq \frac{\beta}{4n} \left( m_\mathcal{N} \cdot \frac{c_\beta}{8}(1 + 2/c_\rho^2 - 2\eta - o_n(1)) \cdot d(1 - o_n(1)) \right)$$

$$- \frac{\beta}{4n} \left( (1 - \eta + o_n(1)) \cdot \frac{n}{2} \cdot M_\mathcal{C} \frac{c_\beta}{8}(1 - 2/c_\rho^2 - 2\eta + o_n(1)) \frac{d}{n} c_\rho^2 \right)$$

$$+ \frac{\beta}{4n} \left( |\mathcal{N}_k| \cdot m_\mathcal{N} \frac{c_\beta}{8}(1 + 2/c_\rho^2 - 2\eta - o_n(1)) \frac{d}{n} c_\rho^2 \right)$$

$$- \frac{\beta}{4n} \left( n \cdot M_\mathcal{N} \frac{c_\beta}{8}(1 + 2/c_\rho^2 - 2\eta + o_n(1)) \sqrt{d \log(12n^2/\delta)} \right).$$

Observe that the third term is non-negative. Moreover, we argue that the first term is at least twice the sum of the second and last terms. Indeed, enough to show that

$$\left( m_\mathcal{N} \cdot (1 + 2/c_\rho^2 - 2\eta - o_n(1)) \cdot d(1 - o_n(1)) \right) \geq$$

$$2 \left( (1 + o_n(1)) \cdot \frac{1}{2} \cdot M_\mathcal{C} \cdot dc_\rho^2 \right) + 2 \left( n \cdot M_\mathcal{N}(1 + 2/c_\rho^2 + o_n(1)) \sqrt{d \log(12n^2/\delta)} \right) ,$$

which indeed holds since $n\sqrt{d \log(12n^2/\delta)} = d \cdot o_n(1)$, and $M_\mathcal{C} \cdot c_\rho^2 \leq 0.25$ while $m_\mathcal{N} \geq 0.99$ (see Eqs. 19 and 18). Overall, for any $i \in \mathcal{N}$ we have that:

$$\boldsymbol{p}_2^\top (\boldsymbol{x}_{j,2} - \boldsymbol{x}_{j,1}) \geq \frac{\beta}{8n} \left( m_\mathcal{N} \cdot \frac{c_\beta}{8}(1 + 2/c_\rho^2 - 2\eta - o_n(1)) \cdot d(1 - o_n(1)) \right)$$

$$= \frac{c_\beta}{8c_\rho^2} \left( m_\mathcal{N} \cdot \frac{c_\beta}{8}(1 + 2/c_\rho^2 - 2\eta - o_n(1)) \cdot (1 - o_n(1)) \right)$$

$$\geq 2 \log(c_\rho),$$

where that last inequality holds since $c_\beta \geq 16c_\rho \log(c_\rho)$, which implies that $0.9c_\beta^2/64c_\rho^2 \geq 2 \log(c_\rho) = \log(c_\rho^2)$. We conclude that,

$$\alpha_{i,2}^{t=2} = \frac{1}{1 + \exp(-\boldsymbol{p}_2^\top(\boldsymbol{x}_{j,2} - \boldsymbol{x}_{j,1}))} \geq \frac{1}{1 + \exp(-\log(c_\rho^2))} = \frac{1}{1 + 1/c_\rho^2}$$

$$= \frac{c_\rho^2}{c_\rho^2 + 1} \geq \frac{c_\rho^2 - 1}{c_\rho^2} = 1 - 1/c_\rho^2. \tag{32}$$

We conclude that for any $j \in \mathcal{N}$ we have that

$$\alpha_{j,2}^{t=2} \geq 1 - 1/c_\rho^2, \quad \alpha_{j,1}^{t=2} \leq 1/c_\rho^2. \tag{33}$$

Together with Eq. 31, this proves the third part of the Thm.

**The classifier** $\text{sign}(f(\boldsymbol{X}; \boldsymbol{v}_{t=2}, \boldsymbol{p}_{t=2}))$ **classifies correctly clean training samples.** Let $(\boldsymbol{X}_j = (\boldsymbol{x}_{j,1}, \boldsymbol{x}_{j,2}), y_j)$ for $j \in \mathcal{C}$. We remind that $\boldsymbol{x}_{j,1} = \boldsymbol{\mu}_k$ for $k \in \{1, 2\}$ and $\boldsymbol{x}_{2,j} = \boldsymbol{\xi}_j$. we have that,

$$f(\boldsymbol{X}_j; \boldsymbol{v}_{t=2}, \boldsymbol{p}_{t=2}) = \alpha_{j,1}^{t=2} \boldsymbol{v}_2^\top \boldsymbol{x}_{j,1} + \alpha_{j,2}^{t=2} \boldsymbol{v}_2^\top \boldsymbol{x}_{j,2},$$

and it suffices to prove that

$$y_j(f(\boldsymbol{X}_j; \boldsymbol{v}_2, \boldsymbol{p}_2)) > 0.$$

Indeed,

$$
\begin{aligned}
y_j f(\boldsymbol{X}_j; \boldsymbol{v}, \boldsymbol{p}) &= \alpha_{j,1}^{t=2} y_j \boldsymbol{v}_2^\top \boldsymbol{x}_{j,1} + \alpha_{j,2}^{t=2} y_j \boldsymbol{v}_2^\top \boldsymbol{x}_{j,2} \\
&= \alpha_{j,1}^{t=2} |\lambda_k| \, \|\boldsymbol{\mu}_k\|^2 + \alpha_{j,2}^{t=2} \theta_j^2 \, \|\boldsymbol{\xi}_j\|^2 + \alpha_{j,2}^{t=2} y_j \sum_{i \in [n]: i \neq j} y_i \theta_i^{t=2} \boldsymbol{\xi}_i^\top \boldsymbol{\xi}_j \quad y_j \lambda_k > 0 \\
&\geq \alpha_{j,1}^{t=2} |\lambda_k| \, \|\boldsymbol{\mu}_k\|^2 - \alpha_{j,2}^{t=2} n \max_i |\theta_i| \sqrt{d \log(12n^2/\delta)} \\
&\geq \alpha_{j,1}^{t=2} \left(\frac{\beta}{9}\right) \frac{d}{n} c_\rho^2 - \alpha_{j,2}^{t=2} n \frac{\beta}{2n} (M_\mathcal{N} + 1) \sqrt{d \log(12n^2/\delta)} \qquad \text{Eqs. 29, 25 and 27} \\
&\geq \frac{1}{2} \left(\frac{\beta}{9}\right) \frac{d}{n} c_\rho^2 - \frac{1}{2} n \frac{\beta}{2n} (M_\mathcal{N} + 1) \sqrt{d \log(12n^2/\delta)} \qquad \text{Eq. 31} \\
&> 0, \qquad\qquad\qquad\qquad\qquad\qquad\qquad\qquad\qquad\quad d \gg n \sqrt{d \log(12n^2/\delta)}
\end{aligned}
$$

as required.

**The classifier** $\mathrm{sign}(f(\boldsymbol{X}; \boldsymbol{v}_{t=2}, \boldsymbol{p}_{t=2}))$ **classifies correctly noisy training samples.** Let $(\boldsymbol{X}_j = (\boldsymbol{x}_{j,1}, \boldsymbol{x}_{j,2}), y_j)$ for $j \in \mathcal{N}$. We remind that $\boldsymbol{x}_{j,1} = \boldsymbol{\mu}_k$ for $k \in \{1, 2\}$ and $\boldsymbol{x}_{2,j} = \boldsymbol{\xi}_j$. we have that,

$$f(\boldsymbol{X}_j; \boldsymbol{v}_{t=2}, \boldsymbol{p}_{t=2}) = \alpha_{j,1}^{t=2} \boldsymbol{v}_2^\top \boldsymbol{x}_{j,1} + \alpha_{j,2}^{t=2} \boldsymbol{v}_2^\top \boldsymbol{x}_{j,2},$$

and it suffices to prove that

$$y_j(f(\boldsymbol{X}_j; \boldsymbol{v}_2, \boldsymbol{p}_2)) > 0.$$

Indeed,

$$
\begin{aligned}
y_j f(\boldsymbol{X}_j; \boldsymbol{v}, \boldsymbol{p}) &= \alpha_{j,1}^{t=2} y_j \boldsymbol{v}_2^\top \boldsymbol{x}_{j,1} + \alpha_{j,2}^{t=2} y_j \boldsymbol{v}_2^\top \boldsymbol{x}_{j,2} \\
&= -\alpha_{j,1}^{t=2} |\lambda_k| \, \|\boldsymbol{\mu}_k\|^2 + \alpha_{j,2}^{t=2} \theta_j^2 \, \|\boldsymbol{\xi}_j\|^2 + \alpha_{j,2}^{t=2} y_j \sum_{i \in [n]: i \neq j} y_i \theta_i^{t=2} \boldsymbol{\xi}_i^\top \boldsymbol{\xi}_j \quad y_j \lambda_k < 0 \\
&\geq -\alpha_{j,1}^{t=2} \left(\frac{3\beta}{16}\right) \frac{d}{n} c_\rho^2 + \alpha_{j,1}^{t=2} \frac{\beta}{2n} m_\mathcal{N} d(1 - o_n(1)) - \alpha_{j,2}^{t=2} \frac{\beta}{n} d \cdot o_n(1) \quad \text{Eqs. 29, 24 and 26} \\
&\geq -\frac{1}{c_\rho^2} \left(\frac{3\beta}{16}\right) \frac{d}{n} c_\rho^2 + \left(1 - \frac{1}{c_\rho^2}\right) \frac{\beta}{2n} 0.99 d(1 - o_n(1)) - \frac{\beta}{n} d \cdot o_n(1) \quad \text{Eqs. 33 and 19} \\
&> 0,
\end{aligned}
$$

as required.

**The classifier** $\mathrm{sign}(f(\boldsymbol{X}; \boldsymbol{v}_{t=2}, \boldsymbol{p}_{t=2}))$ **classifies correctly clean test samples.**

Let $(\boldsymbol{X} = (\boldsymbol{x}_1, \boldsymbol{x}_2), y)$ be a fresh clean sample i.e. $(\boldsymbol{X}, y) \sim \mathcal{D}_{\text{clean}}$. Observe that $\boldsymbol{x}_1 = \boldsymbol{\mu}_k$ for some $k \in \{1, 2\}$ and $y = 1$ iff $k = 1$. By Remark 58, with probability at least $1 - 6n \exp(-d/4C_1 n^2)$ for some constant $C_1 = C_1(c_\rho, c_\beta)$ that will be chosen later, we have that $(\boldsymbol{X} = (\boldsymbol{x}_1, \boldsymbol{x}_2), y)$ is a good test sample w.r.t. $C_1$ (Def. 20). We work under the event that $(\boldsymbol{X} = (\boldsymbol{x}_1, \boldsymbol{x}_2), y)$ is a good test

sample and show that $y = \text{sign}(f(\boldsymbol{X}; \boldsymbol{v}_{t=2}, \boldsymbol{p}_{t=2}))$. Recall that $\boldsymbol{p}_2 = -\beta \nabla_{\boldsymbol{p}} \mathcal{L}(\boldsymbol{v}_1, \boldsymbol{p}_1)$ and therefore:

$$\boldsymbol{p}_2^\top (\boldsymbol{x}_1 - \boldsymbol{x}_2)$$

$$= -(\boldsymbol{x}_1 - \boldsymbol{x}_2)^\top \beta \nabla_{\boldsymbol{p}} \mathcal{L}(\boldsymbol{v}_{t=1}, \boldsymbol{p}_{t=1}) = (\boldsymbol{x}_1 - \boldsymbol{x}_2)^\top \frac{\beta}{n} \sum_{i=1}^n -\ell'_{1,i} \cdot \boldsymbol{X}_i^\top \mathbb{S}'(X_i \boldsymbol{p}_t) \boldsymbol{\gamma}_i^{t=1}$$

$$= \frac{\beta}{n} \sum_{i=1}^n -\ell'_{1,i} \cdot \boldsymbol{x}_1^\top \boldsymbol{X}_i^\top \mathbb{S}'(X_i \boldsymbol{p}_t) \boldsymbol{\gamma}_i^{t=1} - \frac{\beta}{n} \sum_{i=1}^n -\ell'_{1,i} \cdot \boldsymbol{x}_2^\top \boldsymbol{X}_i^\top \mathbb{S}'(X_i \boldsymbol{p}_t) \boldsymbol{\gamma}_i^{t=1}$$

$$= \frac{\beta}{n} \sum_{i \in [n]} -\ell'_{1,i} \cdot (\gamma_{i,1}^{t=1} - \gamma_{i,2}^{t=1})(1 - \alpha_{i,1}^{t=1})\alpha_{i,1}^{t=1}(\boldsymbol{x}_1^\top \boldsymbol{x}_{i,1} + \boldsymbol{x}_2^\top \boldsymbol{x}_{i,2}) \qquad \text{Lemma 22}$$

$$= \frac{\beta}{n} \sum_{i \in \mathcal{C}_k} -\ell'_{1,i} \cdot (\gamma_{i,1}^{t=1} - \gamma_{i,2}^{t=1})(1 - \alpha_{i,1}^{t=1})\alpha_{i,1}^{t=1}(\boldsymbol{x}_1^\top \boldsymbol{x}_{i,1})$$

$$- \frac{\beta}{n} \sum_{i \in \mathcal{N}_k} -\ell'_{1,i} \cdot (\gamma_{i,2}^{t=1} - \gamma_{i,1}^{t=1})(1 - \alpha_{i,1}^{t=1})\alpha_{i,1}^{t=1}(\boldsymbol{x}_1^\top \boldsymbol{x}_{i,1})$$

$$+ \frac{\beta}{n} \sum_{i \in [n]: i \neq j} -\ell'_{1,i} \cdot (\gamma_{i,1}^{t=1} - \gamma_{i,2}^{t=1})(1 - \alpha_{i,1}^{t=1})\alpha_{i,1}^{t=1}(\boldsymbol{x}_2^\top \boldsymbol{x}_{i,2})$$

Observe that $\alpha_{i,1}^{t=1} = \alpha_{i,2}^{t=1} = 1/2$. In Eq. 21 and Eq. 22 we calculate the score $\boldsymbol{\gamma}_i^{t=1}$. Overall, we can lower bound the above equation by:

$$+ \frac{\beta}{4n} \left( (1 - \eta - o_n(1)) \cdot \frac{n}{2} \cdot m_\mathcal{C} \frac{c_\beta}{8} (1 - 2/c_\rho^2 - 2\eta - o_n(1)) \frac{d}{n} c_\rho^2 \right)$$

$$- \frac{\beta}{4n} \left( (\eta + o_n(1)) \cdot \frac{n}{2} \cdot M_\mathcal{N} \frac{c_\beta}{8} (1 + 2/c_\rho^2 - 2\eta + o_n(1)) \frac{d}{n} c_\rho^2 \right)$$

$$- \frac{\beta}{4n} \left( n \cdot M_\mathcal{N} \frac{c_\beta}{8} (1 + 2/c_\rho^2 - 2\eta + o_n(1)) \frac{d}{C_1 n} \right).$$

Once again, the first term dominates the last two terms when $C_1$ is large enough and when $\eta$ is small enough (Assumption 4). This means that the softmax probability of the first token is:

$$\frac{1}{1 + \exp(-\boldsymbol{p}_2^\top (\boldsymbol{x}_1 - \boldsymbol{x}_2))} > \frac{1}{2}. \tag{34}$$

Let $\boldsymbol{x}_1 = \boldsymbol{\mu}_k$ for $k \in \{1, 2\}$ and $\boldsymbol{x}_2 = \boldsymbol{\xi}$. Then,

$$f(\boldsymbol{X}; \boldsymbol{v}, \boldsymbol{p}) = \alpha_1 \boldsymbol{v}_2^\top \boldsymbol{x}_1 + \alpha_2 \boldsymbol{v}_2^\top \boldsymbol{x}_2,$$

where $\alpha_1, \alpha_2$ are the softmax probabilities of $\boldsymbol{p}_2$ for $\boldsymbol{X}$. It suffices to prove that

$$y(f(\boldsymbol{X}; \boldsymbol{v}_2, \boldsymbol{p}_2)) > 0.$$

Since the test sample is "good", we have that $\forall i : \boldsymbol{\xi}_i^\top \boldsymbol{\xi} \leq \frac{d}{C_1 n}$, which implies that

$$yf(\boldsymbol{X}; \boldsymbol{v}_2, \boldsymbol{p}_2) = \alpha_1 y \boldsymbol{v}_2^\top \boldsymbol{x}_1 + \alpha_2 y \boldsymbol{v}_2^\top \boldsymbol{x}_2$$

$$= \alpha_1 |\lambda_k| \|\boldsymbol{\mu}_k\|^2 + \alpha_2 y \sum_{i=1}^n y_i \theta_i \boldsymbol{\xi}_i^\top \boldsymbol{\xi} \qquad y\lambda_k > 0$$

$$\geq \alpha_1 |\lambda_k| \|\boldsymbol{\mu}_k\|^2 - \alpha_2 n \max_i |\theta_i| \frac{d}{C_1 n}$$

$$\geq \alpha_1 \left( \frac{\beta}{9} \right) \frac{d}{n} c_\rho^2 - \alpha_2 n \frac{\beta}{2n} (M_\mathcal{N} + 1) \frac{d}{C_1 n} \qquad \text{Eqs. 29, 25 and 27}$$

$$\geq \frac{1}{2} \left( \frac{\beta}{9} \right) \frac{d}{n} c_\rho^2 - \frac{1}{2} n \frac{\beta}{2n} (M_\mathcal{N} + 1) \frac{d}{C_1 n} \qquad \text{Eq. 34}$$

$$> 0,$$

where the last inequality holds for large enough $C_1$. Overall,

$$\mathbb{P}_{(\boldsymbol{X},y)\sim\mathcal{D}}(y \neq \text{sign}(f(\boldsymbol{X};\boldsymbol{v}_{t=2},\boldsymbol{p}_{t=2})))$$
$$\leq \eta + \mathbb{P}_{(\boldsymbol{X},y)\sim\mathcal{D}_{\text{clean}}}(y \neq \text{sign}(f(\boldsymbol{X};\boldsymbol{v}_{t=2},\boldsymbol{p}_{t=2})))$$
$$\leq \eta + 6n^2 \exp(-d/4C_1 n^2).$$

By Assumption 23 (item 2), we can also upper bound the above term by $\eta + \exp(-d/C_1 n^2)$, for a slightly larger $C_1$. This proves the last part of the theorem. $\qquad\square$

## A.2 Proofs for Sec. 4

### A.2.1 Additional Notation

We first introduce some additional notations. Denote

$$n_1 = |\mathcal{C}|, \quad n_2 = |\mathcal{N}|; \quad n_{1i} = |\mathcal{C}_i|, \quad n_{2i} = |\mathcal{N}_i| \text{ for } i = 1, 2.$$

Denote the output of the softmax layer $\mathbb{S}(\boldsymbol{X}_i\boldsymbol{p})$ by

$$\boldsymbol{s}_i = (1 - \beta_i, \beta_i)^\top.$$

Denote the output of the attention layer $\boldsymbol{X}_i^\top \boldsymbol{s}_i$ by $\boldsymbol{r}_i = (1-\beta_i)\boldsymbol{\mu}_i + \beta_i\boldsymbol{\xi}_i$, where $0 \leq \beta_i \leq 1$ is the attention on the noise token of each sample. Then $f(\boldsymbol{X}_i;\boldsymbol{p},\boldsymbol{v}) = \langle \boldsymbol{v}, \boldsymbol{r}_i \rangle$ can be treated as a linear classifier on $(y_i, \boldsymbol{r}_i)_{i\in[n]}$. Additionally, from the property of log function, item 1 in Assumption 5 can be understood as $d \geq Cn^2 \log(\text{poly}(n)/\delta)$ and the same is for item 5.

### A.2.2 Proof of Thm. 6

**Proof Sketch**

There are two main parts in our proof. In the first part, we prove that only by selecting signal tokens for clean samples and noise tokens for non-clean samples can we reach the maximum margin when doing SVM on $(y_i, \boldsymbol{r}_i)_{i\in[n]}$.

**Definition 24** (Optimal Token). *We define the "optimal token" for sample $(\boldsymbol{X}_i, y_i)$ as*

$$\boldsymbol{r}_i^\star = \boldsymbol{\mu}_i, \; i \in \mathcal{C}$$
$$\boldsymbol{r}_i^\star = \boldsymbol{\xi}_i, \; i \in \mathcal{N} \tag{35}$$

Next we define the respective max-margin solution for $\boldsymbol{p}$ and $\boldsymbol{v}$. We will show that when jointly optimizing parameters $\boldsymbol{p}$ and $\boldsymbol{v}$ for (2), they will converge to their respective max-margin solutions as $R, r \to \infty$, which are $\boldsymbol{p}_{mm}$ and $\boldsymbol{v}_{mm}$ defined as follows.

**Definition 25.** *(p-SVM)*

$$\boldsymbol{p}_{mm} = \underset{\boldsymbol{p}}{\arg\min} \|\boldsymbol{p}\|$$

*subjected to*

$$\boldsymbol{p}^\top(\boldsymbol{\mu}_i - \boldsymbol{\xi}_i) \geq 1, i \in \mathcal{C}$$
$$\boldsymbol{p}^\top(\boldsymbol{\xi}_i - \boldsymbol{\mu}_i) \geq 1, i \in \mathcal{N} \tag{36}$$

*for all $i \in [n]$. $\Xi = 1/\|\boldsymbol{p}_{mm}\|$ is the margin induced by $\boldsymbol{p}_{mm}$.*

Then for a given $\boldsymbol{p}$, we define $\boldsymbol{v}(\boldsymbol{p})$ as the standard max-margin classifier on $(y_i, \boldsymbol{r}_i)_{i\in[n]}$ and $\boldsymbol{v}_{mm}$ as the standard max-margin classifier on $(y_i, \boldsymbol{r}_i^\star)_{i\in[n]}$ which can be understood as the limit scenario when $\boldsymbol{p} = \boldsymbol{p}_{mm}$ and $R \to +\infty$ .

**Definition 26.** *(v-SVM)*

$$\boldsymbol{v}(\boldsymbol{p}) = \underset{\boldsymbol{v}\in\mathbb{R}^d}{\arg\min} \|\boldsymbol{v}\| \text{ s.t. } y_i \cdot \boldsymbol{v}^\top \boldsymbol{r}_i \geq 1, \quad \text{for all } i \in [n]. \tag{37}$$

$\Gamma(\boldsymbol{p}) = 1/\|\boldsymbol{v}(\boldsymbol{p})\|$ *is the **label margin** induced by $\boldsymbol{v}$ and $\boldsymbol{p}$. When $\boldsymbol{r}_i = \boldsymbol{r}_i^\star, i \in [n]$,*

$$\boldsymbol{v}_{mm} = \underset{\boldsymbol{v}\in\mathbb{R}^d}{\arg\min} \|\boldsymbol{v}\| \text{ s.t. } y_i \cdot \boldsymbol{v}^\top \boldsymbol{r}_i^\star \geq 1, \quad \text{for all } i \in [n]. \tag{38}$$

$\Gamma = 1/\|\boldsymbol{v}_{mm}\|$ *is the label margin induced by $\boldsymbol{v}_{mm}$.*

After proving the converfnece direction of $\boldsymbol{p}_R$ and $\boldsymbol{v}_r$, we can utilize their properties similar to $\boldsymbol{p}_{mm}$ and $\boldsymbol{v}_{mm}$ to proceed the training and test error analysis. Therefore proving that the model exhibits benign-overfitting.

It is worth noting that in the first part, we show the optimality of the token selection in (35) is strict in the sense that mixing other tokens in $\boldsymbol{r}_i$ will shrink the label margin. We formalize this into the following proposition:

**Proposition 15** (optimal token condition). *Suppose that Assumption 5 holds, with probability at least $1 - \delta$ over the training dataset, for all $\boldsymbol{p}$, the token selection under $\boldsymbol{p}$ results in a label margin (Def. 14) of at most $\Gamma - \frac{C}{\|\boldsymbol{v}_{mm}\|^3 n \rho^2} \cdot \max\limits_{i \in [n]} (1 - s_{i\alpha_i})$ where $\alpha_i = \mathbb{I}(i \in \mathcal{C}) + 2\mathbb{I}(i \in \mathcal{N})$ and $C > 0$ is some constant.*

We will give detailed proof in the following.

**Optimal Token Condition**

Since $\boldsymbol{v}_{mm}$ satisfies the KKT conditions of the max-margin problem (37), by the stationarity condition, we can represent $\boldsymbol{v}_{mm}$ as

$$\boldsymbol{v}_{mm} = \lambda_1 \boldsymbol{\mu}_1 + \lambda_2 \boldsymbol{\mu}_2 + \sum_{i \in [n]} y_i \theta_i \boldsymbol{\xi}_i. \tag{39}$$

Note that the conditions in (37) can be written as:

**Condition 1** (Optimal tokens).

$$\begin{cases} \boldsymbol{v}^\top \boldsymbol{\mu}_1 \geq 1 \\ -\boldsymbol{v}^\top \boldsymbol{\mu}_2 \geq 1 \\ y_i \boldsymbol{v}^\top \boldsymbol{\xi}_i \geq 1, i \in \mathcal{N} \end{cases}$$

Plugging (39) in the condition 1, we can rewrite these conditions as:

$$\begin{cases} \lambda_1 \cdot \|\boldsymbol{\mu}_1\|^2 \geq 1 \\ -\lambda_2 \cdot \|\boldsymbol{\mu}_2\|^2 \geq 1 \\ \theta_i \cdot \|\boldsymbol{\xi}_i\|^2 + y_i y_{i'} \sum\limits_{i' \neq i} \theta_{i'} \langle \boldsymbol{\xi}_i, \boldsymbol{\xi}_{i'} \rangle \geq 1, i \in \mathcal{N} \end{cases}$$

Then we introduce a lemma to estimate the coefficients $\theta_i$ of $\boldsymbol{v}_{mm}$ under this condition:

**Lemma 27** (balanced noise factor for KKT points). *Suppose that Assumption 5 holds, under Condition 1, we have that for $\boldsymbol{v}_{mm}$,*

$$\theta_i = 0, \quad i \in \mathcal{C}; \tag{40}$$

$$\theta_i \in \left[ \frac{(1 - \kappa)d - 4n_2\sqrt{d\log(6n^2/\delta)}}{(1 + \kappa)d((1 - \kappa)d - 2n_2\sqrt{d\log(6n^2/\delta)})}, \frac{1}{(1 - \kappa)d - 2n_2\sqrt{d\log(6n^2/\delta)}} \right], \quad i \in \mathcal{N}. \tag{41}$$

*Proof of Lemma 27.* Note that Condition 1 does not have any constraint for samples with $i \in \mathcal{C}$. Thus we have $\theta_i = 0$ for any $i \in \mathcal{C}$ in the representation (39). For $\theta_i$ with $i \in \mathcal{N}$, we first prove the upper bound by contradiction. Denote $j = \operatorname{argmax}\limits_{i \in \mathcal{N}} \theta_i$. Then we have

$$y_j \boldsymbol{v}^\top \boldsymbol{\xi}_j = \sum_{i \in \mathcal{N}} y_i y_j \theta_i \langle \boldsymbol{\xi}_i, \boldsymbol{\xi}_j \rangle = \theta_j \|\boldsymbol{\xi}_j\|_2^2 + \sum_{i \neq j, i \in \mathcal{N}} y_i y_j \theta_i \langle \boldsymbol{\xi}_i, \boldsymbol{\xi}_j \rangle$$

$$\geq \theta_j \cdot (1 - \kappa)d - n_2 \theta_j \cdot 2\sqrt{d\log(6n^2/\delta)},$$

where the inequality is from Lemma 57 and the definition of $j$. Consider the contrary case when $\theta_j > \frac{1}{(1-\kappa)d - 2n_2\sqrt{d\log(6n^2/\delta)}}$, we have

$$y_j \boldsymbol{v}^\top \boldsymbol{\xi}_j > \frac{1}{(1 - \kappa)d - 2n_2\sqrt{d\log(6n^2/\delta)}} \cdot \left((1 - \kappa)d - n_2 \cdot 2\sqrt{d\log(6n^2/\delta)}\right) = 1.$$

By the complementary slackness, if $y_j \boldsymbol{v}^\top \boldsymbol{\xi}_j > 1$, then we must have $\theta_j = 0$, and thus we reach a contradiction.

Then we prove for the lower bound. For $\forall j \in \mathcal{N}$ we have

$$1 \leq \theta_j \|\boldsymbol{\xi}_j\|_2^2 + \sum_{i \neq j, i \in \mathcal{N}} y_i y_j \theta_i \langle \boldsymbol{\xi}_i, \boldsymbol{\xi}_j \rangle$$

$$\leq \theta_j \cdot (1 + \kappa)d + n_2 \max_{i \in \mathcal{N}} \theta_i \cdot 2\sqrt{d \log(6n^2/\delta)}$$

$$\leq \theta_j \cdot (1 + \kappa)d + \frac{n_2}{(1 - \kappa)d - 2n_2\sqrt{d \log(6n^2/\delta)}} \cdot 2\sqrt{d \log(6n^2/\delta)}.$$

The second inequality is due to Lemma 57 and the last inequality is from the upper bound we just get. Therefore, we have

$$\theta_j \geq \frac{(1 - \kappa)d - 4n_2\sqrt{d \log(6n^2/\delta)}}{(1 + \kappa)d((1 - \kappa)d - 2n_2\sqrt{d \log(6n^2/\delta)})}.$$

This completes the proof. □

Then we introduce a lemma to estimate $\|\boldsymbol{v}_{mm}\|$:

**Lemma 28** (Norm of $\boldsymbol{v}_{mm}$). *Suppose that Assumption 5 holds, for the solution $\boldsymbol{v}_{mm}$ of (37) under the token selection (35), we have*

$$\frac{2}{\rho^2} + \frac{\eta n}{2d} \leq \|\boldsymbol{v}_{mm}\|^2 \leq \frac{2}{\rho^2} + \frac{5\eta n}{d}.$$

*This implies*

$$\|\boldsymbol{v}_{mm}\| = \Theta\left(\sqrt{\frac{1}{\rho^2} + \frac{\eta n}{d}}\right).$$

*Proof of Lemma 28.* As $\boldsymbol{v}_{mm}$ is the max-margin solution and satisfies KKT condition, it can be represented as

$$\boldsymbol{v}_{mm} = \lambda_1 \boldsymbol{\mu}_1 + \lambda_2 \boldsymbol{\mu}_2 + \sum_{i \in \mathcal{C}} y_i \theta_i \boldsymbol{\xi}_i + \sum_{i \in \mathcal{N}} y_i \theta_i \boldsymbol{\xi}_i. \tag{42}$$

As $\boldsymbol{v}_{mm}$ satisfies Condition 1, we have $\lambda_1 \geq 1/\rho^2$ and $\lambda_2 \leq -1/\rho^2$. So we could lower bound $\|\boldsymbol{v}_{mm}\|$ as

$$\|\boldsymbol{v}_{mm}\|^2 \geq \lambda_1^2 \|\boldsymbol{\mu}_1\|^2 + \lambda_2^2 \|\boldsymbol{\mu}_2\|^2 + \sum_{i \in \mathcal{N}} \theta_i^2 \|\boldsymbol{\xi}_i\|^2 + \sum_{i \in \mathcal{N}} \sum_{j \in \mathcal{N}} y_i y_j \theta_i \theta_j \langle \boldsymbol{\xi}_i, \boldsymbol{\xi}_j \rangle$$

$$\geq \frac{2}{\rho^2} + \frac{n_2(1 - \kappa)}{d} + O\left(\frac{\eta^2 n^2}{d^{3/2}}\right) \geq \frac{2}{\rho^2} + \frac{\eta n}{2d}.$$

The second inequality is from Lemma 27 that $\theta_i = \Theta(1/d)$ for $i \in \mathcal{N}$ and the last inequality is from Assumption 5.

Then to upper bound $\|\boldsymbol{v}_{mm}\|$, consider the following possible solution $\widetilde{\boldsymbol{v}}$

$$\widetilde{\boldsymbol{v}} = \rho^{-2}\boldsymbol{\mu}_1 - \rho^{-2}\boldsymbol{\mu}_2 + \sum_{i \in \mathcal{N}} 2y_i\boldsymbol{\xi}_i/d.$$

For $i \in \mathcal{C}$, we have

$$y_i \widetilde{\boldsymbol{v}}^\top \boldsymbol{r}_i = y_i \widetilde{\boldsymbol{v}}^\top \boldsymbol{\mu}_i \geq 1.$$

And for $i \in \mathcal{N}$, we have

$$y_i \widetilde{\boldsymbol{v}}^\top \boldsymbol{r}_i = y_i \widetilde{\boldsymbol{v}}^\top \boldsymbol{\xi}_i = 2\|\boldsymbol{\xi}_i\|^2/d + \sum_{j \in \mathcal{N}, j \neq i} 2y_i y_j \langle \boldsymbol{\xi}_i, \boldsymbol{\xi}_j \rangle/d$$

$$\geq 2(1 - \kappa) - 2n_2\sqrt{\log(6n^2/\delta)/d} \geq 1.$$

The first inequality is from Lemma 57 and the second inequality is from Assumption 5. Therefore, $\widetilde{v}$ is a possible solution of SVM problem 26 when $p$ converges to $p_{mm}$. So we have

$$\|\boldsymbol{v}_{mm}\|^2 \leq \|\widetilde{\boldsymbol{v}}\|^2 = 2/\rho^2 + \sum_{i \in \mathcal{N}} 4\|\boldsymbol{\xi}_i\|^2/d^2 + \sum_{i \in \mathcal{N}} \sum_{j \in \mathcal{N}} 4y_i y_j \langle \boldsymbol{\xi}_i, \boldsymbol{\xi}_j \rangle/d^2 \leq \frac{2}{\rho^2} + \frac{5\eta n}{d}.$$

The last inequality is from Lemma 57, Lemma 59 and Assumption 5. Combine the results above, we have $\|\boldsymbol{v}_{mm}\|^2 = \Theta(\frac{1}{\rho^2} + \frac{\eta n}{d})$. $\qquad\square$

Based on the lemmas above, we introduce our main proposition in this section:

**Proposition 15** (optimal token condition)**.** *Suppose that Assumption 5 holds, with probability at least $1 - \delta$ over the training dataset, for all $p$, the token selection under $p$ results in a label margin (Def. 14) of at most $\Gamma - \frac{C}{\|\boldsymbol{v}_{mm}\|^3 n \rho^2} \cdot \max_{i \in [n]}(1 - s_{i\alpha_i})$ where $\alpha_i = \mathbb{I}(i \in \mathcal{C}) + 2\mathbb{I}(i \in \mathcal{N})$ and $C > 0$ is some constant.*

*Proof of Proposition 15.* The main idea is to show the optimality of the token selection rule in the sense that mixing any other tokens will shrink the label margin. For a given $p$, we say a sample $\boldsymbol{x}_i$ is a "mixed sample" if $\boldsymbol{r}_i \neq \boldsymbol{r}_i^\star$. We say $\boldsymbol{r}_i$ is a mixture of optimal token and non-optimal token in this case. Note that for any $p$ with finite norm, $\boldsymbol{r}_i \neq \boldsymbol{r}_i^\star$. This notation is introduced for the clearness of the proof.

We use contradiction to prove Proposition 15 by showing that any token selection different from (35) can only result in a strictly smaller label margin than that for the max-margin problem (37). Since $\boldsymbol{v}$ satisfies the KKT conditions of the max-margin problem, we can write $\boldsymbol{v}$ as

$$\boldsymbol{v} = \lambda_1 \boldsymbol{\mu}_1 + \lambda_2 \boldsymbol{\mu}_2 + \sum_{i \in \mathcal{C}} y_i \theta_i \boldsymbol{\xi}_i + \sum_{i \in \mathcal{N}} y_i \theta_i \boldsymbol{\xi}_i. \tag{43}$$

For a given $p$, denote $\boldsymbol{v}'$ as the max-margin solution in (37), and $\Gamma' = 1/\|\boldsymbol{v}'\|$ as the new label margin. According to Lemma 28, we have

$$\|\boldsymbol{v}_{mm}\|^2 = \Theta\left(\frac{1}{\rho^2} + \frac{\eta n}{d}\right) = \Omega(1/\rho^2).$$

Then we have

$$\Gamma - \frac{C}{\|\boldsymbol{v}_{mm}\|^3 n \rho^2} \cdot \max_{i \in [n]}(1 - s_{i\alpha_i}) \geq \Gamma - \frac{C}{\|\boldsymbol{v}_{mm}\|^3 n \rho^2} \geq \frac{\Gamma}{2}$$

for sufficiently large $d$. Here the last inequality uses $\|\boldsymbol{v}_{mm}\|^2 = \Omega(1/\rho^2)$. Thus we only need consider the case when the new label margin $\Gamma' \geq \Gamma/2$, or equivalently,

$$\|\boldsymbol{v}'\| \leq 2\|\boldsymbol{v}_{mm}\|. \tag{44}$$

Assume that there are $k$ samples ($0 < k \leq n$) that violate the token selection rule (35) and among them, $p$ samples are from clean set $\mathcal{C}$ and $k - p$ samples are from label-flipped set $\mathcal{N}$. Denote the indices of the $k$ samples as $I_v$. Then we consider the following three scenarios:

1. $p \neq 0, k - p = 0$. (All mixed samples come from $\mathcal{C}$)

2. $p = 0, k - p \neq 0$. (All mixed samples come from $\mathcal{N}$)

3. $p \neq 0, k - p \neq 0$. (Mixed samples are from both sets)

We will separately discuss each scenario and show that Proposition 15 holds in all cases.
**Case 1:** $p \neq 0, k - p = 0$

Under this scenario, we have:
$$I_v \cap \mathcal{C} = I_v; \quad I_v \cap \mathcal{N} = \varnothing.$$
We proceed to analyze this scenario by dividing it into three distinct subcases.

- $p < n_1, I_v \cap \mathcal{C}_1 \neq \varnothing, I_v \cap \mathcal{C}_2 \neq \varnothing$

- $p < n_1$, $I_v \cap \mathcal{C}_i \neq \varnothing$, $I_v \cap \mathcal{C}_{i'} = \varnothing$, $(i, i' \in [2], i \neq i')$

- $p = n_1$

***Case 1.1*** $p < n_1$, $I_v \cap \mathcal{C}_1 \neq \varnothing$, $I_v \cap \mathcal{C}_2 \neq \varnothing$

In this case, both clusters exist clean samples that are not mixed. Denote the index of mixed samples $I_v$ as $\{k_1, k_2, ..., k_p\}$. For every mixed sample $k_i$, we have $\boldsymbol{r}_{k_i} = \beta_{k_i}\boldsymbol{\mu}_{k_i} + (1 - \beta_{k_i})\boldsymbol{\xi}_{k_i}$. Then the conditions under *Case 1.1* become

**Condition 2** ($p$ clean samples violating optimal token selection)**.**

$$\begin{cases} \boldsymbol{v}^\top \boldsymbol{\mu}_1 \geq 1 \\ -\boldsymbol{v}^\top \boldsymbol{\mu}_2 \geq 1 \\ y_i \boldsymbol{v}^\top \boldsymbol{\xi}_i \geq 1, i \in \mathcal{N} \\ y_i \boldsymbol{v}^\top \boldsymbol{r}_i \geq 1, i \in I_v \end{cases}$$

From the condition above, we could see that in this case, mixing one more clean sample is equal to adding one more constraint. Therefore, mixing $p$ samples will not result in a better solution than only mixing one sample, i.e. larger max-margin in our setting. So we can reduce this case to mixing only one clean sample with index $k^\star = \underset{i \in I_v}{\operatorname{argmin}} \beta_i$. Denote $r_{k^\star} = \beta\boldsymbol{\mu}_{k^\star} + (1 - \beta)\boldsymbol{\xi}_{k^\star}$ for some $\beta \in [0, 1)$.

Without loss of generality, we assume $\boldsymbol{\mu}_{k^\star} = \boldsymbol{\mu}_1$, $y_{k^\star} = +1$. Then the conditions become:

**Condition 3** (one clean sample violating optimal token selection)**.**

$$\begin{cases} \boldsymbol{v}^\top \boldsymbol{\mu}_1 \geq 1 \\ -\boldsymbol{v}^\top \boldsymbol{\mu}_2 \geq 1 \\ y_i \boldsymbol{v}^\top \boldsymbol{\xi}_i \geq 1, i \in \mathcal{N} \\ y_{k^\star} \boldsymbol{v}^\top \boldsymbol{r}_{k^\star} \geq 1 \end{cases}$$

Denote $\boldsymbol{v}'$ as the optimal solution under this condition. $\boldsymbol{v}'$ can also be written in the form of (43) with coefficients denoted as $\lambda_1'$, $\lambda_2'$ and $\theta_i'$, $i \in [n]$. Plugging this representation into the condition 3, we have:

$$\begin{cases} \lambda_1' \cdot \|\boldsymbol{\mu}_1\|^2 \geq 1 \\ -\lambda_2' \cdot \|\boldsymbol{\mu}_2\|^2 \geq 1 \\ \theta_i' \cdot \|\boldsymbol{\xi}_i\|^2 + \sum_{i' \neq i} y_i y_{i'} \theta_{i'}' \langle \boldsymbol{\xi}_i, \boldsymbol{\xi}_{i'} \rangle \geq 1, i \in \mathcal{N} \\ \beta\lambda_1' \cdot \|\boldsymbol{\mu}_1\|^2 + (1 - \beta)(\theta_{k^\star}' \|\boldsymbol{\xi}_{k^\star}\|^2 + \sum_{i \neq k^\star} y_{k^\star} y_i \theta_i' \langle \boldsymbol{\xi}_i, \boldsymbol{\xi}_{k^\star} \rangle) \geq 1 \end{cases}$$

First, we introduce another lemma similar to Lemma 27 to characterize the scale of $\theta_i'$, $i \in [n]$ in this case.

**Lemma 29.** *Suppose that Assumption 5 holds, under Condition 3, we have*

$$\theta_i' = 0, \quad i \in \mathcal{C}\backslash\{k^\star\};$$

$$\theta_i \in \left[ \frac{(1 - \kappa)d - 4n_2\sqrt{d\log(6n^2/\delta)}}{(1 + \kappa)d((1 - \kappa)d - 2n_2\sqrt{d\log(6n^2/\delta)})}, \frac{1}{(1 - \kappa)d - 2n_2\sqrt{d\log(6n^2/\delta)}} \right], \quad i \in \mathcal{N}.$$

*Proof of Lemma 29.* Same as Condition 1, Condition 3 does not have any constraint for samples with $i \in \mathcal{C}\backslash\{k^\star\}$. Thus we have $\theta_i' = 0$ for any $i \in \mathcal{C}\backslash\{k^\star\}$.

Meanwhile, Condition 3 introduces an additional constraint compared to Condition 1. Consequently, the feasible region for $\{\theta_i'\}_{i \in \mathcal{N}}$ under Condition 3 is a subset of the feasible region for $\{\theta_i\}_{i \in \mathcal{N}}$ under Condition 1. Therefore, the bounds established in Lemma 27 remain applicable to $\{\theta_i'\}_{i \in \mathcal{N}}$. $\square$

From this lemma, We can see that $\theta_i' = \Theta(1/d)$ for $i \in \mathcal{N}$. To proceed, we introduce a crucial lemma:

**Lemma 30.** *Suppose that Assumption 5 holds, denote $v$ and $v'$ as the optimal solutions under condition 1 and condition 3 respectively. We have*

$$\|v'\|_2^2 - \|v_{mm}\|_2^2 \geq \frac{C_1(1 - \beta\lambda_1'\rho^2)^2}{(1-\beta)^2(1+\kappa)d} + \widetilde{O}\Big(\frac{\eta n}{d^{3/2}}\Big).$$

*where $0 < C_1 \leq 1$ is a constant.*

*Proof of Lemma 30.* We consider two cases under this scenario:

- $\theta_k' = 0$ in $v'$

  In this case, from Lemma 29 we have $\beta\lambda_1' \geq (1 + o(1))/\rho^2$ and all other conditions are the same as the optimal selection. In order to get $\min\|v\|$, we have $\lambda_1' = (1 + o(1))/\beta\rho^2$. Consider another solution $v_0$ which has parameters $\lambda_{01} = 1/\rho^2$, $\lambda_{02} = \lambda_2'$, $\theta_{0i} = \theta_i'(i \in [n])$. As $v_0$ satisfies all the inequities under Condition 1, we have $\Gamma_0 \leq \Gamma$ So we have

  $$\Gamma^2 - \Gamma'^2 \geq \Gamma_0^2 - \Gamma'^2 = \frac{1}{\|v_0\|^2} - \frac{1}{\|v'\|^2} = \frac{(\lambda_{01}^2 - \lambda_1'^2) \cdot \|\mu_1\|^2}{\|v_0\|^2 \cdot \|v'\|^2}$$

  $$= \frac{(1 + o(1))/\beta^2 - 1}{\|v_0\|^2 \cdot \|v'\|^2} = \frac{(1+\beta)(1-\beta) + o(1)}{\beta^2\|v_0\|^2 \cdot \|v'\|^2} \geq \frac{1-\beta}{\|v_0\|^2 \cdot \|v'\|^2}.$$

  Therefore,

  $$\Gamma - \Gamma' \geq \frac{1-\beta}{(\Gamma_0 + \Gamma')\|v_0\|^2 \cdot \|v'\|^2} \geq \frac{1-\beta}{2\Gamma_0\|v_0\|^2 \cdot \|v'\|^2}.$$

  Set $c = \frac{1}{2\Gamma_0\|v_0\|^2 \cdot \|v'\|^2} = \frac{1}{2\|v_0\|\|v'\|^2}$. we have $\Gamma' \leq \Gamma - c(1 - \beta)$. Moreover, we could upper bound $c$ as

  $$c = \frac{1}{2\|v_0\|\|v'\|^2} \leq \frac{1}{2r_{mm}^3}.$$

  The last inequality is from $\|v'\| \geq \|v_0\| \geq r_{mm}$.

- $\theta_k' \neq 0$ in $v'$

  From KKT condition, we have

  $$\theta_{k^*}' \cdot \Big[\beta\lambda_1' \cdot \|\mu_1\|^2 + (1-\beta)(\theta_{k^*}'\|\xi_{k^*}\|^2 + \sum_{i \neq k^*} y_{k^*}y_i\theta_i'\langle\xi_i, \xi_{k^*}\rangle) - 1\Big] = 0.$$

  As $\theta_{k^*}' > 0$, we have

  $$\beta\lambda_1' \cdot \|\mu_1\|^2 + (1-\beta)(\theta_{k^*}'\|\xi_{k^*}\|^2 + \sum_{i \in \mathcal{N}} y_{k^*}y_i\theta_i'\theta_i'\langle\xi_i, \xi_{k^*}\rangle) = 1.$$

  So we can estimate $\theta_{k^*}'$ as

  $$\theta_{k^*}'\|\xi_{k^*}\|^2 = \frac{1 - \beta\lambda_1'\rho^2}{1-\beta} - \sum_{i \in \mathcal{N}} y_{k^*}y_i\theta_i'\theta_i'\langle\xi_i, \xi_{k^*}\rangle \leq \frac{1 - \beta\lambda_1'\rho^2}{1-\beta} + 2n_2 \max_{i \in \mathcal{N}} \theta_i'\sqrt{d\log(6n^2/\delta)}$$

  $$= \frac{1 - \beta\lambda_1'\rho^2}{1-\beta} + \frac{2n_2\sqrt{d\log(6n^2/\delta)}}{(1-\kappa)d - 2n_2\sqrt{d\log(6n^2/\delta)}}. \tag{45}$$

  The first inequality is from Lemma 57 and the last equality is from Lemma 29. We can also lower bound it as

  $$\theta_{k^*}'\|\xi_{k^*}\|^2 = \frac{1 - \beta\lambda_1'\rho^2}{1-\beta} - \sum_{i \in \mathcal{N}} y_{k^*}y_i\theta_i'\theta_i'\langle\xi_i, \xi_{k^*}\rangle \geq \frac{1 - \beta\lambda_1'\rho^2}{1-\beta} - 2n_2 \max_{i \in \mathcal{N}} \theta_i'\sqrt{d\log(6n^2/\delta)}$$

  $$= \frac{1 - \beta\lambda_1'\rho^2}{1-\beta} - \frac{2n_2\sqrt{d\log(6n^2/\delta)}}{(1-\kappa)d - 2n_2\sqrt{d\log(6n^2/\delta)}}. \tag{46}$$

The first inequality is from Lemma 57 and the last equality is from Lemma 29. Therefore, we have $\theta'_{k^*} = \Theta(\frac{1-\beta\lambda'_1\rho^2}{(1-\beta)d}) \pm O(\frac{\eta n}{d^{3/2}})$.

Then from the third inequality in Condition 3, we have

$$\theta'_i \cdot \|\boldsymbol{\xi}_i\|^2 + \sum_{i' \in \mathcal{N}, i' \neq i} y_i y_{i'} \theta'_{i'} \langle \boldsymbol{\xi}_i, \boldsymbol{\xi}_{i'} \rangle \geq 1 - y_i y_{k^*} \theta'_{k^*} \langle \boldsymbol{\xi}_i, \boldsymbol{\xi}_{k^*} \rangle$$

$$\geq 1 - \left[ \frac{1-\beta\lambda'_1\rho^2}{(1-\beta)(1+\kappa)d} + O\left(\frac{\eta n}{d^{3/2}}\right) \right] \cdot |\langle \boldsymbol{\xi}_i, \boldsymbol{\xi}_{k^*} \rangle|$$

$$\geq 1 - \frac{2(1-\beta\lambda'_1\rho^2)\sqrt{\log(6n^2/\delta)}}{(1-\beta)(1+\kappa)\sqrt{d}} - \widetilde{O}\left(\frac{\eta n}{d}\right)$$

$$\geq 1 - \frac{2\sqrt{\log(6n^2/\delta)}}{\sqrt{d}} - \widetilde{O}\left(\frac{\eta n}{d}\right)$$

$$= 1 - \frac{3\sqrt{\log(6n^2/\delta)}}{\sqrt{d}}. \tag{47}$$

The second inequality is from (45); The third inequality is from Lemma 57 and the last inequality is from the first inequality in Condition 3 that $\lambda'_1\rho^2 \geq 1$.

Consider $\widetilde{\boldsymbol{v}} = \widetilde{\lambda}_1 \boldsymbol{\mu}_1 + \widetilde{\lambda}_2 \boldsymbol{\mu}_2 + \sum_{i \in [n]} y_i \widetilde{\theta}_i \boldsymbol{\xi}_i$, which has $\widetilde{\lambda}_1 = \lambda'_1$, $\widetilde{\lambda}_2 = \lambda'_2$, $\widetilde{\theta}_i = \theta'_i/(1 - \frac{3\sqrt{\log(6n^2/\delta)}}{\sqrt{d}})$ for $i \in \mathcal{N}$ and $\widetilde{\theta}'_i = 0$ for $i \in \mathcal{C}$. We can verify that $\widetilde{\boldsymbol{v}}$ satisfies all conditions for $\boldsymbol{v}_{mm}$. For $\forall i \in \mathcal{N}$, we have

$$\widetilde{\theta}_i \cdot \|\boldsymbol{\xi}_i\|^2 + \sum_{i' \in \mathcal{N}, i' \neq i} y_i y_{i'} \widetilde{\theta}_{i'} \langle \boldsymbol{\xi}_i, \boldsymbol{\xi}_{i'} \rangle$$

$$= \left[ \theta'_i \cdot \|\boldsymbol{\xi}_i\|^2 + \sum_{i' \in \mathcal{N}, i' \neq i} y_i y_{i'} \theta'_{i'} \langle \boldsymbol{\xi}_i, \boldsymbol{\xi}_{i'} \rangle \right] / \left( 1 - \frac{3\sqrt{\log(6n^2/\delta)}}{\sqrt{d}} \right) \geq 1.$$

The last inequality is from (47). Meanwhile, we have $\widetilde{\lambda}_1 \|\boldsymbol{\mu}_1\|^2 = \lambda'_1 \|\boldsymbol{\mu}_1\|^2 \geq 1$, $-\widetilde{\lambda}_2 \|\boldsymbol{\mu}_2\|^2 = -\lambda'_2 \|\boldsymbol{\mu}_2\|^2 \geq 1$. So $\widetilde{\boldsymbol{v}}$ is a possible solution for Condition 3, which implies $\|\boldsymbol{v}_{mm}\| \leq \|\widetilde{\boldsymbol{v}}\|$.

Next we estimate the difference between $\|\boldsymbol{v}'\|^2$ and $\|\widetilde{\boldsymbol{v}}\|^2$. We write the expansion of $\|\widetilde{\boldsymbol{v}}\|^2$ and $\|\boldsymbol{v}'\|^2$:

$$\|\widetilde{\boldsymbol{v}}\|^2 = \widetilde{\lambda}_1^2 \|\boldsymbol{\mu}_1\|^2 + \widetilde{\lambda}_2^2 \|\boldsymbol{\mu}_2\|^2 + \sum_{i \in \mathcal{N}} \widetilde{\theta}_i^2 \|\boldsymbol{\xi}_i\|^2 + \sum_{i,j \in \mathcal{N}; i \neq j} y_i y_j \widetilde{\theta}_i \widetilde{\theta}_j \langle \boldsymbol{\xi}_i, \boldsymbol{\xi}_j \rangle,$$

$$\|\boldsymbol{v}'\|^2 = \lambda_1'^2 \|\boldsymbol{\mu}_1\|^2 + \lambda_2'^2 \|\boldsymbol{\mu}_2\|^2 + \sum_{i \in \mathcal{N} \cup \{k^*\}} \theta_i'^2 \|\boldsymbol{\xi}_i\|^2 + \sum_{i,j \in \mathcal{N} \cup \{k^*\}; i \neq j} y_i y_j \theta'_i \theta'_j \langle \boldsymbol{\xi}_i, \boldsymbol{\xi}_j \rangle.$$

From the construction of $\widetilde{\boldsymbol{v}}$, we have $\lambda'_1 = \lambda_1$, $\lambda'_2 = \lambda_2$. So we have

$$\|\boldsymbol{v}'\|^2 - \|\widetilde{\boldsymbol{v}}\|^2 \geq \theta_{k^*}'^2 \|\boldsymbol{\xi}_{k^*}\|^2 + \underbrace{\sum_{i \in \mathcal{N}} (\theta_i'^2 - \widetilde{\theta}_i^2) \|\boldsymbol{\xi}_i\|^2}_{I_1} + \underbrace{\sum_{i \in \mathcal{N} \cup \{k^*\}} \sum_{j \in \mathcal{N} \cup \{k^*\} \setminus \{i\}} y_i y_j \theta'_i \theta'_j \langle \boldsymbol{\xi}_i, \boldsymbol{\xi}_j \rangle}_{I_2}$$

$$- \underbrace{\sum_{i \in \mathcal{N}} \sum_{j \in \mathcal{N} \setminus \{i\}} y_i y_j \widetilde{\theta}_i \widetilde{\theta}_j \langle \boldsymbol{\xi}_i, \boldsymbol{\xi}_j \rangle}_{I_3}.$$

From (46), we have

$$\theta'_{k^\star} \|\boldsymbol{\xi}_{k^\star}\| \geq \frac{1-\beta\lambda'_1\rho^2}{(1-\beta)\sqrt{(1+\kappa)d}} - \widetilde{O}\left(\frac{\eta n}{d}\right).$$

We then bound the last three terms respectively. First we have

$$|I_1| = \sum_{i\in\mathcal{N}}(\widetilde{\theta}_i^2 - \theta_i'^2)\|\boldsymbol{\xi}_i\|^2 \leq \left(\frac{1}{(1-\widetilde{O}(1/\sqrt{d}))^2} - 1\right) \cdot \sum_{i\in\mathcal{N}}\theta_i'^2\|\boldsymbol{\xi}_i\|^2$$

$$\leq \frac{\widetilde{O}(1/\sqrt{d})}{(1-\widetilde{O}(1/\sqrt{d}))^2} \cdot \frac{n_2(1+\kappa)d}{\left((1-\kappa)d - 2n_2\sqrt{d\log(6n^2/\delta)}\right)^2}$$

$$= \widetilde{O}\left(\frac{\eta n}{d^{3/2}}\right).$$

The first inequality is from the definition of $\widetilde{\theta}_i$; The second inequality is from Lemma 27 and Lemma 57.

Then we bound $|I_2 - I_3|$ as:

$$|I_2 - I_3| = \sum_{i\in\mathcal{N}}\sum_{j\in\mathcal{N}\setminus\{i\}}(\widetilde{\theta}_i\widetilde{\theta}_j - \theta_i'\theta_j') \cdot |\langle\boldsymbol{\xi}_i,\boldsymbol{\xi}_j\rangle| + \theta_k'\sum_{i\in\mathcal{N}}\theta_i'|\langle\boldsymbol{\xi}_{k^*},\boldsymbol{\xi}_i\rangle|$$

$$\leq \left(\frac{1}{(1-\widetilde{O}(1/\sqrt{d}))^2} - 1\right)\sum_{i\in\mathcal{N}}\sum_{j\in\mathcal{N}\setminus\{i\}}\theta_i'\theta_j' \cdot |\langle\boldsymbol{\xi}_i,\boldsymbol{\xi}_j\rangle| + n_2\theta_{k^*}' \cdot \max_{i\in\mathcal{N}}\theta_i' \cdot |\langle\boldsymbol{\xi}_{k^*},\boldsymbol{\xi}_i\rangle|$$

$$\leq \frac{\widetilde{O}(1/\sqrt{d})}{(1-\widetilde{O}(1/\sqrt{d}))^2} \cdot \frac{(n_2)^2 2\sqrt{d\log(6n^2/\delta)}}{\left((1-\kappa)d - 2\eta n\sqrt{d\log(6n^2/\delta)}\right)^2} + \theta_{k^*}' \cdot \Theta\left(\frac{\eta n}{\sqrt{d}}\right)$$

$$= \widetilde{O}\left(\frac{\eta^2 n^2}{d^2}\right) + \Theta\left(\frac{\eta n}{d^{3/2}}\right)$$

$$= \widetilde{O}\left(\frac{\eta n}{d^{3/2}}\right).$$

The first inequality is from the definition of $\widetilde{\theta}_i$; The second inequality is from Lemma 27 and Lemma 57. Combining the above results, we finally have

$$\|\boldsymbol{v}'\|_2^2 - \|\boldsymbol{v}_{mm}\|_2^2 \geq \frac{C_1(1-\beta\lambda_1'\rho^2)^2}{(1-\beta)^2(1+\kappa)d} + \widetilde{O}\left(\frac{\eta n}{d^{3/2}}\right).$$

$\square$

Now we can prove the main proposition in this case.

*Proof of Proposition 15 in Case 1.1.* From Lemma 30 we have

$$\|\boldsymbol{v}'\|_2^2 - \|\boldsymbol{v}_{mm}\|_2^2 \geq \frac{C_1(1-\beta\lambda_1'\rho^2)^2}{(1-\beta)^2(1+\kappa)d} + o\left(\frac{1}{d}\right) \geq \frac{C_1(1-\beta\lambda_1'\rho^2)^2}{(1+\kappa)d}(1-\beta) = T(1-\beta).$$

In the last equation we substitute $T = \frac{C_1(1-\beta\lambda_1'\rho^2)^2}{(1+\kappa)d} \geq 0$. Then we have

$$\Gamma^2 - \Gamma'^2 = \frac{1}{\|\boldsymbol{v}_{mm}\|^2} - \frac{1}{\|\boldsymbol{v}'\|^2} = \frac{\|\boldsymbol{v}'\|^2 - \|\boldsymbol{v}_{mm}\|^2}{\|\boldsymbol{v}_{mm}\|^2 \cdot \|\boldsymbol{v}'\|^2} \geq \frac{T(1-\beta)}{\|\boldsymbol{v}_{mm}\|^2 \cdot \|\boldsymbol{v}'\|^2}.$$

Therefore,

$$\Gamma - \Gamma' \geq \frac{T(1-\beta)}{(\Gamma+\Gamma')\|\boldsymbol{v}_{mm}\|^2 \cdot \|\boldsymbol{v}'\|^2} \geq \frac{T(1-\beta)}{2\Gamma\|\boldsymbol{v}_{mm}\|^2 \cdot \|\boldsymbol{v}'\|^2} = \frac{T(1-\beta)}{2\|\boldsymbol{v}_{mm}\|\|\boldsymbol{v}'\|^2} \geq \frac{T(1-\beta)}{2\|\boldsymbol{v}'\|^3}.$$

The last inequality is from $\|\boldsymbol{v}'\| \geq \|\boldsymbol{v}_{mm}\|$. This implies

$$\Gamma' \leq \Gamma - \frac{T(1-\beta)}{2\|\boldsymbol{v}'\|^3} \leq \Gamma - \frac{C_1}{\|\boldsymbol{v}_{mm}\|^3 n\rho^2}(1-\beta).$$

The last inequality is from our assumption that $\|\boldsymbol{v}'\| \leq 2\|\boldsymbol{v}_{mm}\|$ and $\rho^2 = \Omega(d/n)$. $\square$

Next we consider the other case.

***Case 1.2*** $p = n_1$

Next we consider the case when all clean samples are mixed. In this case, all samples in clean set are mixed, so the first two inequalities in Condition 3 do not hold, which means that $\lambda'_1$ may be smaller than $\lambda_1$. But we could still prove that Lemma 30 holds. We first write down the condition in this case:

**Condition 4** (All clean samples violate optimal token selection rule)**.**

$$\begin{cases} y_i \boldsymbol{v}^\top \boldsymbol{\xi}_i \geq 1, i \in \mathcal{N} \\ y_i \boldsymbol{v}^\top \boldsymbol{r}_i \geq 1, i \in \mathcal{C} \end{cases}$$

Plugging the representation (43) into the condition, we have:

$$\begin{cases} \theta'_i \cdot \|\boldsymbol{\xi}_{i'}\|^2 + \sum\limits_{i' \neq i} y_i y_{i'} \theta'_{i'} \langle \boldsymbol{\xi}_i, \boldsymbol{\xi}_{i'} \rangle \geq 1, i \in \mathcal{N} \\ \beta_i \lambda'_i \cdot \|\boldsymbol{\mu}_i\|^2 + (1 - \beta_i)(\theta'_i \cdot \|\boldsymbol{\xi}_i\|^2 + \sum\limits_{j \neq i} y_i y_j \theta'_i \langle \boldsymbol{\xi}_i, \boldsymbol{\xi}_j \rangle) \geq 1, i \in \mathcal{C} \end{cases}$$

*Proof of Lemma 30.* First we assume that $\max\{\lambda'_1 \cdot \|\boldsymbol{\mu}_1\|^2, -\lambda'_2 \cdot \|\boldsymbol{\mu}_2\|^2\} = q$ in optimal $\boldsymbol{v}'$. If $q \geq 1$, this is the same as *Case 1.3*. So we assume that $q \leq 1$. Denote $k^\star = \underset{i \in \mathcal{C}}{\operatorname{argmin}} \frac{1 - \beta_i q}{1 - \beta_i}$ and $\beta = \beta_{k^\star}$, consider the following condition

**Condition 5** (Relaxed version of Condition 4)**.**

$$\begin{cases} \theta'_i \cdot \|\boldsymbol{\xi}_{i'}\|^2 + \sum\limits_{i' \neq i} y_i y_{i'} \theta'_{i'} \langle \boldsymbol{\xi}_i, \boldsymbol{\xi}_{i'} \rangle \geq 1, i \in \mathcal{N} \\ \theta'_i \cdot \|\boldsymbol{\xi}_{i'}\|^2 + \sum\limits_{i' \neq i} y_i y_{i'} \theta'_{i'} \langle \boldsymbol{\xi}_i, \boldsymbol{\xi}_{i'} \rangle \geq \frac{1 - \beta q}{1 - \beta}, i \in \mathcal{C} \end{cases}$$

Compared with Condition 4, the second inequality is relaxed for $i \in \mathcal{C}$. Therefore, denote the max-margin solution as $\widehat{\boldsymbol{v}}$ under Condition 5, we must have $\|\widehat{\boldsymbol{v}}\| \leq \|\boldsymbol{v}'\|$. Then we will prove that Lemma 30 still holds between $\|\boldsymbol{v}_{mm}\|$ and $\|\widehat{\boldsymbol{v}}\|$, which indicates $\|\boldsymbol{v}'\|_2^2 - \|\boldsymbol{v}_{mm}\|_2^2 \geq \|\widehat{\boldsymbol{v}}\|_2^2 - \|\boldsymbol{v}_{mm}\|_2^2 \geq \frac{C_1(1 - \beta \lambda'_1 \rho^2)^2}{(1 - \beta)^2(1 + \kappa)d} + o\left(\frac{1}{d}\right)$. Denote the parameters in $\widehat{\boldsymbol{v}}$ are $\widehat{\lambda_1}, \widehat{\lambda_2}$ and $\widehat{\theta_i}$, we first introduce the following lemma to estimate $\widehat{\theta_i}$. Here we denote $\alpha = \frac{1 - \beta q}{1 - \beta}$ for convenience.

**Lemma 31.** *Suppose that Assumption 5 holds, under Condition 5, we have*

$$\widehat{\theta_i} \in \left[ \frac{\alpha}{(1 + \kappa)d}\left(1 - \frac{2n\sqrt{d \log(6n^2/\delta)}}{(1 - \kappa)d - 2n\sqrt{d \log(6n^2/\delta)}}\right), \frac{\alpha}{((1 - \kappa)d - 2n\sqrt{d \log(6n^2/\delta)})} \right], i \in \mathcal{C},$$

$$\widehat{\theta_i} \in \left[ \frac{1}{(1 + \kappa)d}\left(1 - \frac{2\alpha n\sqrt{d \log(6n^2/\delta)}}{(1 - \kappa)d - 2n\sqrt{d \log(6n^2/\delta)}}\right), \frac{\alpha}{((1 - \kappa)d - 2n\sqrt{d \log(6n^2/\delta)})} \right], i \in \mathcal{N}.$$

*Proof of Lemma 31.* Denote $j = \underset{i \in [n]}{\operatorname{argmax}} \, \widehat{\theta_i}$, we have

$$\widehat{\theta_i} \cdot \|\boldsymbol{\xi}_i\|^2 + \sum_{j \neq i} y_i y_j \widehat{\theta_i} \langle \boldsymbol{\xi}_i, \boldsymbol{\xi}_j \rangle \geq \widehat{\theta_j} \|\boldsymbol{\xi}_j\|^2 - n\widehat{\theta_j}\sqrt{d \log(6n^2/\delta)}$$

$$\geq \widehat{\theta_j}((1 - \kappa)d - 2n\sqrt{d \log(6n^2/\delta)}).$$

The two inequalities are from Lemma 57 and our definition of j. Consider the contrary case when $\widehat{\theta_j} > \frac{\alpha}{((1 - \kappa)d - 2n\sqrt{d \log(6n^2/\delta)})}$, we have

$$y_j \widehat{\boldsymbol{v}}^\top \boldsymbol{\xi}_j > \alpha.$$

By the complementary slackness condition, if $y_j \widehat{\boldsymbol{v}}^\top \boldsymbol{\xi}_j > \alpha \geq 1$, then we must have $\widehat{\theta_j} = 0$, and thus we reach a contradiction.

Then we lower bound $\widehat{\theta}_i$, for $i \in \mathcal{C}$ we have

$$\alpha \leq \widehat{\theta}_i \cdot \|\boldsymbol{\xi}_i\|^2 + \sum_{j \neq i} y_i y_j \widehat{\theta}_i \langle \boldsymbol{\xi}_i, \boldsymbol{\xi}_j \rangle \leq \widehat{\theta}_i (1 + \kappa) d + 2n \max_{i \in [n]} \widehat{\theta}_i \sqrt{d \log(6n^2/\delta)}$$

$$\leq \widehat{\theta}_i (1 + \kappa) d + \frac{2\alpha n \sqrt{d \log(6n^2/\delta)}}{(1 - \kappa) d - 2n \sqrt{d \log(6n^2/\delta)}}.$$

The second inequality is from Lemma 57 and the last inequality is from the upper bound of $\widehat{\theta}_i$ we just derived. Therefore, we have

$$\widehat{\theta}_i \geq \frac{\alpha}{(1 + \kappa) d} \left( 1 - \frac{2n \sqrt{d \log(6n^2/\delta)}}{(1 - \kappa) d - 2n \sqrt{d \log(6n^2/\delta)}} \right).$$

Similarly, for $i \in \mathcal{N}$, we have

$$\widehat{\theta}_i \geq \frac{1}{(1 + \kappa) d} \left( 1 - \frac{2\alpha n \sqrt{d \log(6n^2/\delta)}}{(1 - \kappa) d - 2n \sqrt{d \log(6n^2/\delta)}} \right).$$

$\square$

Note that we only consider the case when $\|\widehat{\boldsymbol{v}}\| \leq \|\boldsymbol{v}'\| \leq 2\|\boldsymbol{v}_{mm}\|$. And from Lemma 31 we have $\widehat{\theta}_i = \Theta(\alpha/d)$ for $i \in \mathcal{C}$. So we must have $\alpha = O(\log n)$ is some constant. Otherwise, for $i \in \mathcal{C}$ we have

$$\widehat{\theta}_i \|\boldsymbol{\xi}_i\|^2 \geq \alpha - \sum_{i' \neq i} y_i y_{i'} \widehat{\theta}_i \langle \boldsymbol{\xi}_i, \boldsymbol{\xi}_{i'} \rangle = \Omega(\alpha).$$

It further yields that

$$\|\widehat{\boldsymbol{v}}\|^2 = \Omega(\frac{1}{\rho^2}) + \Omega(\frac{\eta n}{d}) + \sum_{i \in \mathcal{C}} \widehat{\theta}_i^2 \|\boldsymbol{\xi}_i\|^2 = \Omega(\frac{1}{\rho^2} + \frac{\eta n}{d} + \frac{n\alpha^2}{d}) = \Omega(\frac{n \log^2 n}{d}), \quad (48)$$

which contradicts with $\|\boldsymbol{v}''\| = \Theta(\sqrt{1/\rho^2 + \eta n/d})$.

Then the difference between $\|\boldsymbol{v}_{mm}\|_2^2$ and $\|\widehat{\boldsymbol{v}}\|_2^2$ becomes

$$\|\widehat{\boldsymbol{v}}\|^2 - \|\boldsymbol{v}_{mm}\|^2 \geq \sum_{i \in \mathcal{C}} \widehat{\theta}_i^2 \|\boldsymbol{\xi}_i\|^2 - 2/\rho^2 + \underbrace{\sum_{i \in \mathcal{N}} (\widehat{\theta}_i^2 - \theta_i^2) \|\boldsymbol{\xi}_i\|^2}_{I_1} + \underbrace{\sum_{i \in [n]} \sum_{j \in [n] \setminus \{i\}} y_i y_j \widehat{\theta}_i \widehat{\theta}_j \langle \boldsymbol{\xi}_i, \boldsymbol{\xi}_j \rangle}_{I_2}$$

$$- \underbrace{\sum_{i \in \mathcal{N}} \sum_{j \in \mathcal{N} \setminus \{i\}} y_i y_j \theta_i \theta_j \langle \boldsymbol{\xi}_i, \boldsymbol{\xi}_j \rangle}_{I_3}.$$

We will bound every term sequentially. For $i \in \mathcal{C}$, we have

$$\widehat{\theta}_i \|\boldsymbol{\xi}_i\|^2 \geq \alpha - \sum_{i' \in [n], i' \neq i} y_i \widehat{\theta}_{i'} \langle \boldsymbol{\xi}_i, \boldsymbol{\xi}_{i'} \rangle \geq \alpha - n \max_{i \in [n]} \widehat{\theta}_i \cdot 2\sqrt{d \log(6n^2/\delta)}$$

$$= \alpha - \frac{2\alpha n \sqrt{\log(6n^2/\delta)}}{(1 - \kappa) \sqrt{d} - 2n \sqrt{\log(6n^2/\delta)}} = \alpha - \widetilde{O}\left( \frac{n}{\sqrt{d}} \right).$$

The second inequality is from Lemma 57; The first equality is from Lemma 29 and the last equality is from Assumption 5. This implies

$$\sum_{i \in \mathcal{C}} \widehat{\theta}_i^2 \|\boldsymbol{\xi}_i\|^2 - 2/\rho^2 \geq \frac{n_1 \alpha^2}{(1 + \kappa) d} - \frac{2}{\rho^2} - \widetilde{O}\left( \frac{n}{d^{3/2}} \right) \geq \frac{C_2 n_1 \alpha^2}{(1 + \kappa) d} - \widetilde{O}\left( \frac{n}{d^{3/2}} \right).$$

The second inequality is due to the SNR condition $\rho/\sqrt{d} = \Omega(1/\sqrt{n})$ so there exists a constant $C_2$ that $\frac{2}{\rho^2} \leq \frac{(1 - C_2) n_1 \alpha^2}{(1 + \kappa) d}$.

Then for $|I_1|$ we have

$$|I_1| \leq (\max_{i \in \mathcal{N}} \theta_i^2 - \min_{i \in \mathcal{N}} \widehat{\theta}_i^2) \sum_{i \in \mathcal{N}} \|\boldsymbol{\xi}_i\|^2$$

$$\leq \left( \left( \frac{1}{(1-\kappa)d - 2\eta n\sqrt{d\log(6n^2/\delta)}} \right)^2 - \left( \frac{1}{(1+\kappa)d} \left( 1 - \frac{2n\sqrt{d\log(6n^2/\delta)}}{(1-\kappa)d - 2n\sqrt{d\log(6n^2/\delta)}} \right) \right)^2 \right) \cdot n_2(1+\kappa)d$$

$$\leq \left( \frac{\sqrt{(1+\kappa)d}}{(1-\kappa)d - 2\eta n\sqrt{d\log(6n^2/\delta)}} \right)^2 \left( 1 - \left( \frac{(1-\kappa)d - 4\eta n\sqrt{d\log(6n^2/\delta)}}{(1+\kappa)d} \right)^2 \right) \cdot n_2$$

$$= \Theta\left( \frac{1}{d} \right) \cdot \Theta\left( \frac{\eta n\sqrt{\log(6n^2/\delta)}}{\sqrt{d}} \right) \cdot n_2$$

$$= \widetilde{O}\left( \frac{\eta^2 n^2}{d^{3/2}} \right).$$

The second inequality is from Lemma 27 and Lemma 31; The third inequality is from the fact that $\eta < 1$.

As for the last two terms, we bound them respectively, for $I_2$ we have

$$|I_2| \leq \sum_{i \in [n]} \sum_{j \in [n] \setminus \{i\}} |y_i y_j \widehat{\theta}_i \widehat{\theta}_j \langle \boldsymbol{\xi}_i, \boldsymbol{\xi}_j \rangle| \leq n^2 \max_{i \in [n]} \widehat{\theta}_i^2 \cdot 2\sqrt{d\log(6n^2/\delta)}$$

$$\leq n^2 \frac{\alpha^2}{((1-\kappa)d - 2n\sqrt{d\log(6n^2/\delta)})^2} \cdot 2\sqrt{d\log(6n^2/\delta)}$$

$$= \widetilde{O}\left( \frac{n^2}{d^{3/2}} \right).$$

The first inequality is from triangle inequality; The second inequality is from Lemma 57; The third inequality is from Lemma 29. Last for $I_3$, we have

$$|I_3| \leq \sum_{i \in \mathcal{N}} \sum_{j \in \mathcal{N} \setminus \{i\}} |y_i y_j \theta_i \theta_j \langle \boldsymbol{\xi}_i, \boldsymbol{\xi}_j \rangle| \leq (n_2)^2 \max_{i \in \mathcal{N}} \theta_i^2 \cdot 2\sqrt{d\log(6n^2/\delta)}$$

$$\leq (n_2)^2 \frac{1}{((1-\kappa)d - 2\eta n\sqrt{d\log(6n^2/\delta)})^2} \cdot 2\sqrt{d\log(6n^2/\delta)}$$

$$= \widetilde{O}\left( \frac{\eta^2 n^2}{d^{3/2}} \right).$$

The first inequality is from triangle inequality; The second inequality is from Lemma 57; The third inequality is from Lemma 27. Combining the results above, we have

$$\|\boldsymbol{v}'\|^2 - \|\boldsymbol{v}_{mm}\|^2 \geq \frac{C_2 n_1 (1-\beta q)^2}{(1-\beta)^2 (1+\kappa)d} + \widetilde{O}\left( \frac{n^2}{d^{3/2}} \right) \geq \frac{C_1 (1-\beta q)^2}{(1-\beta)^2 (1+\kappa)d}.$$

Therefore, we could then use the same method as above to prove that Proposition 15 also holds in this case.

***Case 1.3*** $p < n_1$, $I_v \cap \mathcal{C}_i \neq \varnothing$, $I_v \cap \mathcal{C}_{i'} = \varnothing$

For the case when only one of the clusters in clean sets are all mixed, we can follow similar method in *Case 1.2* to prove that Lemma 30 still holds. Without losing generality, assume all clean samples with label $y_i = +1$ violate optimal token selection while only part of clean samples with label $y_i = -1$ violate. we have

**Condition 6** (One cluster and a clean sample in the opposite cluster violating optimal token selection).

$$\begin{cases} -\boldsymbol{v}^\top \boldsymbol{\mu}_2 \geq 1 \\ y_i \boldsymbol{v}^\top \boldsymbol{\xi}_i \geq 1, i \in \mathcal{N} \\ y_i \boldsymbol{v}^\top \boldsymbol{r}_i \geq 1, i \in \mathcal{C}_{+1} \\ y_i \boldsymbol{v}^\top \boldsymbol{r}_i \geq 1, i \in \mathcal{C}_{-1} \cap I_v \end{cases}$$

Similar to previous analysis, mixing multiple samples with label $-1$ will not result in a better solution than only mixing one sample with label $-1$. Thus we can reduce this case to mixing only one clean sample and denote this mixed sample as $k_{-1}$. Therefore, we have

$$
\begin{cases}
-\lambda_2' \cdot \|\boldsymbol{\mu}_2\|^2 \geq 1 \\
\theta_i' \cdot \|\boldsymbol{\xi}_{i'}\|^2 + \sum_{i' \neq i} y_i y_{i'} \theta_{i'}' \langle \boldsymbol{\xi}_i, \boldsymbol{\xi}_{i'} \rangle \geq 1, i \in \mathcal{N} \\
y_{k_{-1}} \beta \lambda_2' \cdot \|\boldsymbol{\mu}_2\|^2 + (1 - \beta)(\theta_{k_{-1}}' \cdot \|\boldsymbol{\xi}_{k_{-1}}\|^2 + \sum_{i \neq k_{-1}} y_{k_{-1}} y_i \theta_i' \langle \boldsymbol{\xi}_i, \boldsymbol{\xi}_{k_{-1}} \rangle) \geq 1 \\
\beta \lambda_1' \cdot \|\boldsymbol{\mu}_1\|^2 + (1 - \beta)(\theta_{k_i}' \cdot \|\boldsymbol{\xi}_{k_i}\|^2 + \sum_{i \neq k_i} y_{k_i} y_i \theta_i' \langle \boldsymbol{\xi}_i, \boldsymbol{\xi}_{k_i} \rangle) \geq 1, i \in \mathcal{C}_{+1}
\end{cases}
$$

Denote $q = \lambda_1' \cdot \|\boldsymbol{\mu}_1\|^2$ and $q \leq 1$. Denote $k^\star = \underset{i \in \mathcal{C}_{+1}}{\operatorname{argmin}} \frac{1 - \beta_i q}{1 - \beta_i}$ and $\beta = \beta_{k^\star}$, we can further reduce the condition to

**Condition 7** (Relaxed version of Condition 6).

$$
\begin{cases}
\theta_i' \cdot \|\boldsymbol{\xi}_{i'}\|^2 + \sum_{i' \neq i} y_i y_{i'} \theta_{i'}' \langle \boldsymbol{\xi}_i, \boldsymbol{\xi}_{i'} \rangle \geq 1, i \in \mathcal{N} \\
\theta_i' \cdot \|\boldsymbol{\xi}_{i'}\|^2 + \sum_{i' \neq i} y_i y_{i'} \theta_{i'}' \langle \boldsymbol{\xi}_i, \boldsymbol{\xi}_{i'} \rangle \geq \frac{1 - \beta q}{1 - \beta}, i \in \mathcal{C}_{+1}
\end{cases}
$$

Condition 7 relax the constraints in Condition 6. Meanwhile, it differs from Condition 4 only in that the last inequality holds for clean samples with label $+1$. Therefore, we can follow the proof above to show that Lemma 30 still holds in this case.

$\square$

Then we consider the second scenario.

**Case 2:** $p = 0, k - p \neq 0$

Similar to the previous part, there are two cases we need to consider under this scenario:

1. $k - p < n_2$.

2. $k - p = n_2$.

We will go over every case sequentially.

*Case 2.1* $k - p < n_2$

In this case, part of noisy samples are mixed. Denote the mixed samples as $k_1, k_2, ..., k_{k-p}$. And for every mixed sample $k_i$, we have $\boldsymbol{r}_i = \beta_i \boldsymbol{\xi}_{k_i} + (1 - \beta_i) \boldsymbol{\mu}_{k_i}$. Then the conditions under *Case 2.1* become:

**Condition 8** ($k - p$ noisy samples violating optimal token selection rule).

$$
\begin{cases}
\boldsymbol{v}^\top \boldsymbol{\mu}_1 \geq 1 \\
-\boldsymbol{v}^\top \boldsymbol{\mu}_2 \geq 1 \\
y_i \boldsymbol{v}^\top \boldsymbol{\xi}_i \geq 1, i \in \mathcal{N}, i \notin [k - p] \\
y_{k_i} \boldsymbol{v}^\top \boldsymbol{r}_{k_i} \geq 1, i \in [k - p]
\end{cases}
$$

We could also write the last inequality as

$$
y_{k_i} \beta_i \boldsymbol{v}^\top \boldsymbol{\xi}_{k_i} + y_{k_i} (1 - \beta_i) \boldsymbol{v}^\top \boldsymbol{\mu}_{k_i} \geq 1, i \in [k - p].
$$

Therefore,

$$
y_{k_i} \boldsymbol{v}^\top \boldsymbol{\xi}_{k_i} \geq (1 - y_{k_i} (1 - \beta_i) \boldsymbol{v}^\top \boldsymbol{\mu}_{k_i}) / \beta_i, i \in [k - p].
$$

For noisy samples, we have $y_i = -1$ when $\boldsymbol{\mu}_i = \boldsymbol{\mu}_1$ and $y_i = 1$ when $\boldsymbol{\mu}_i = \boldsymbol{\mu}_2$, so $y_{k_i} \boldsymbol{v}^\top \boldsymbol{\mu}_{k_i} \leq 0$ and thus $(1 - y_{k_i} (1 - \beta_i) \boldsymbol{v}^\top \boldsymbol{\mu}_{k_i}) / \beta_i \geq 1$. Compared to the constraint in Condition 1 that $y_{k_i} \boldsymbol{v}^\top \boldsymbol{\mu}_{k_i} \geq 1, i \in \mathcal{N}$, the new condition is strengthened. So mixing 1 more noisy samples is equal to strengthening

1 constraint in the original setting. Therefore, mixing $k - p$ samples will not result in a better solution than only mixing 1 noisy sample. Similarly, we can simplify this case to mixing only 1 noisy sample and denote this sample as $k_*$. We have $\boldsymbol{r}_{k^*} = \beta \boldsymbol{\xi}_{k^*} + (1 - \beta) \boldsymbol{\mu}_{k^*}$ and assume that $\boldsymbol{\xi}_{k^*} = \boldsymbol{\mu}_1$.

Denote $\boldsymbol{v}''$ is the optimal solution under this condition, and the parameters in $\boldsymbol{v}''$ are $\lambda_1''$, $\lambda_2''$ and $\theta_i''$. Then the conditions become:

**Condition 9** (1 noisy sample violating optimal token selection rule)**.**

$$
\begin{cases}
\boldsymbol{v}^\top \boldsymbol{\mu}_1 \geq 1 \\
-\boldsymbol{v}^\top \boldsymbol{\mu}_2 \geq 1 \\
y_i \boldsymbol{v}^\top \boldsymbol{\xi}_i \geq 1, i \in \mathcal{N}, i \neq k^\star \\
y_{k^\star} \boldsymbol{v}^\top \boldsymbol{r}_{k^\star} \geq 1
\end{cases}
$$

Plugging the representation (43) into the condition, we have:

$$
\begin{cases}
\lambda_1'' \cdot \|\boldsymbol{\mu}_1\|^2 \geq 1 \\
-\lambda_2'' \cdot \|\boldsymbol{\mu}_2\|^2 \geq 1 \\
\theta_i'' \cdot \|\boldsymbol{\xi}_i\|^2 + \sum_{i' \neq i} y_i y_{i'} \theta_{i'}'' \langle \boldsymbol{\xi}_i, \boldsymbol{\xi}_{i'} \rangle \geq 1, i \in \mathcal{N}, i \neq k^\star \\
-(1 - \beta) \lambda_1'' \cdot \|\boldsymbol{\mu}_1\|^2 + \beta (\theta_{k^\star}'' \cdot \|\boldsymbol{\xi}_{k^\star}\|^2 + \sum_{i \neq k^\star} y_{k^\star} y_i \theta_i'' \langle \boldsymbol{\xi}_i, \boldsymbol{\xi}_{k^\star} \rangle) \geq 1
\end{cases}
$$

We first introduce the following lemma which estimates the parameters of the noises. We define

$$
\alpha = \frac{1 + (1 - \beta) \lambda_1'' \|\boldsymbol{\mu}_1\|^2}{\beta}
$$

for the convenience of the following proof.

**Lemma 32.** *Suppose that Assumption 5 holds, under Condition 9, we have*

$$
\theta_{k^\star}'' \leq \frac{\alpha}{(1 - \kappa)d - 2n_2 \sqrt{d \log(6n^2/\delta)}}
$$

$$
\theta_{k^\star}'' \geq \frac{\alpha}{(1 + \kappa)d} \left( 1 - \frac{2n_2 \sqrt{d \log(6n^2/\delta)}}{(1 - \kappa)d - 2n_2 \sqrt{d \log(6n^2/\delta)}} \right)
$$

$$
\max_{i \in \mathcal{N}, i \neq k^\star} \theta_i'' \leq \frac{(1 - \kappa)d + 2(\alpha - n_2) \sqrt{d \log(6n^2/\delta)}}{((1 - \kappa)d - 2n_2 \sqrt{d \log(6n^2/\delta)})^2}
$$

$$
\min_{i \in \mathcal{N}, i \neq k^\star} \theta_i'' \geq \frac{1}{(1 + \kappa)d} \cdot \left( 1 - \frac{2\alpha n_2 \sqrt{d \log(6n^2/\delta)}}{(1 - \kappa)d - 2n_2 \sqrt{d \log(6n^2/\delta)}} \right).
$$

*Proof of Lemma 31.* From the last inequality in Condition 9 we have

$$
\theta_{k_*}'' \|\boldsymbol{\xi}_{k_*}\|^2 + \sum_{i \in \mathcal{N}, i \neq k_*} y_i y_{k_*} \theta_i'' \langle \boldsymbol{\xi}_i, \boldsymbol{\xi}_{k_*} \rangle \geq \alpha > 1.
$$

The last inequality is because $\lambda_1'' \|\boldsymbol{\mu}_1\|^2 \geq 1$ and $0 < \beta < 1$. Denote $j = \operatorname*{argmax}_{i \in [n]} \theta_i''$, we have

$$
y_j \boldsymbol{v}''^\top \boldsymbol{\xi}_j = \theta_j'' \|\boldsymbol{\xi}_j\|^2 + \sum_{i \in \mathcal{N}, i \neq j} y_i y_j \theta_i'' \langle \boldsymbol{\xi}_i, \boldsymbol{\xi}_j \rangle
$$

$$
\geq \theta_j'' (1 - \kappa)d - n_2 \max_{i \in [n]} \theta_i'' \cdot 2 \sqrt{d \log(6n^2/\delta)}
$$

$$
= \theta_j'' ((1 - \kappa)d - n_2 \cdot 2 \sqrt{d \log(6n^2/\delta)})
$$

The first inequality is due to Lemma 57 and the last equation is from our definition of j. Consider the contrary case when $\theta_j'' > \frac{\alpha}{(1-\kappa)d - 2n_2 \sqrt{d \log(6n^2/\delta)}}$, we have

$$
y_j \boldsymbol{v}''^\top \boldsymbol{\xi}_j > \alpha.
$$

By the complementary slackness condition, if $y_j \boldsymbol{v}''^\top \boldsymbol{\xi}_j > \frac{1 + \lambda_1''(1-\beta)\|\boldsymbol{\mu}_1\|^2}{\beta}$ then we must have $\theta_j'' = 0$, and thus we reach a contradiction. Therefore, we have $\theta_{k^\star}'' \le \theta_j'' \le \frac{\alpha}{(1-\kappa)d - 2n_2\sqrt{d\log(6n^2/\delta)}}$.

Then denote $j' = \underset{i \in [n], i \ne k^\star}{\operatorname{argmax}} \theta_i''$, we have

$$
\begin{aligned}
y_{j'} \boldsymbol{v}''^\top \boldsymbol{\xi}_{j'} &= \theta_{j'}'' \|\boldsymbol{\xi}_{j'}\|^2 + \sum_{i \in \mathcal{N}, i \ne j'} y_i y_{j'} \theta_i'' \langle \boldsymbol{\xi}_i, \boldsymbol{\xi}_{j'} \rangle \\
&\ge \theta_{j'}''(1-\kappa)d - n_2 \max_{i \in [n], i \ne j'} \theta_i'' \cdot 2\sqrt{d\log(6n^2/\delta)} - \theta_{k^\star}'' \sqrt{d\log(6n^2/\delta)} \\
&\ge \theta_{j'}''\left((1-\kappa)d - n_2 \cdot 2\sqrt{d\log(6n^2/\delta)}\right) - \frac{2\alpha\sqrt{d\log(6n^2/\delta)}}{(1-\kappa)d - 2n_2\sqrt{d\log(6n^2/\delta)}}.
\end{aligned}
$$

The first inequality is from Lemma 57 and the second inequality is from the upper bound of $\theta_{k^\star}''$ we just get. Consider the case when $\theta_{j'}'' > \frac{(1-\kappa)d + 2(\alpha - n_2)\sqrt{d\log(6n^2/\delta)}}{((1-\kappa)d - 2n_2\sqrt{d\log(6n^2/\delta)})^2}$, we have

$$
y_{j'} \boldsymbol{v}''^\top \boldsymbol{\xi}_{j'} > 1.
$$

By the complementary slackness condition, if $y_{j'} \boldsymbol{v}''^\top \boldsymbol{\xi}_{j'} > 1$ then we must have $\theta_{j'}'' = 0$, and thus we reach a contradiction.

Then we estimate the lower bound of $\theta_j''$ when $j \ne k_*$. We have

$$
1 \le y_j \boldsymbol{v}''^\top \boldsymbol{\xi}_j = \theta_j'' \|\boldsymbol{\xi}_j\|^2 + \sum_{i \in [n], i \ne j} y_i y_j \theta_i'' \langle \boldsymbol{\xi}_i, \boldsymbol{\xi}_j \rangle \le \theta_j''(1+\kappa)d + n_2 \max_{i \in [n]} \theta_i'' \cdot 2\sqrt{d\log(6n^2/\delta)}
$$

$$
\le \theta_j''(1+\kappa)d + \frac{1 + \lambda_1''(1-\beta)\|\boldsymbol{\mu}_1\|^2}{\beta((1-\kappa)d - 2n_2\sqrt{d\log(6n^2/\delta)})} \cdot 2n_2\sqrt{d\log(6n^2/\delta)},
$$

where the last inequality is from the upper bound we just get. Therefore, we have

$$
\theta_j'' \ge \frac{1}{(1+\kappa)d} \cdot \left(1 - \frac{2n_2\sqrt{d\log(6n^2/\delta)}}{(1-\kappa)d - 2n_2\sqrt{d\log(6n^2/\delta)}} \cdot \frac{1 + \lambda_1''(1-\beta)\|\boldsymbol{\mu}_1\|^2}{\beta}\right)
$$

for all $j \in \mathcal{N}$ and $j \ne k_*$.
Lastly we lower bound $\theta_{k_*}''$. We have

$$
\frac{1 + (1-\beta)\lambda_1''\|\boldsymbol{\mu}_1\|^2}{\beta} \le y_{k_*} \boldsymbol{v}''^\top \boldsymbol{\xi}_{k_*} = \theta_{k_*}''(1+\kappa)d + n_2 \max_{i \in [n]} \theta_i'' \cdot 2\sqrt{d\log(6n^2/\delta)}.
$$

Similarly, we have

$$
\theta_{k_*}'' \ge \frac{1}{(1+\kappa)d} \cdot \frac{1 + (1-\beta)\lambda_1''\|\boldsymbol{\mu}_1\|^2}{\beta}\left(1 - \frac{2n_2\sqrt{d\log(6n^2/\delta)}}{(1-\kappa)d - 2n_2\sqrt{d\log(6n^2/\delta)}}\right).
$$

$\square$

After getting the bound of parameters, we could derive the norm difference as above

**Lemma 33.** *Suppose that Assumption 5 holds, denote $\boldsymbol{v}$ and $\boldsymbol{v}''$ as the optimal solutions under condition 1 and condition 9 respectively. We have*

$$
\|\boldsymbol{v}''\|_2^2 - \|\boldsymbol{v}_{mm}\|_2^2 \ge \frac{C_3(1-\beta)}{d},
$$

*where $C_3 = \Theta(1)$.*

*Proof of Lemma 33.* From the third inequality in Condition 9, for $i \in \mathcal{N}, i \ne k^\star$ we have

$$
\theta_i'' \cdot \|\boldsymbol{\xi}_i\|^2 + \sum_{i' \ne i, k^\star} y_i y_{i'} \theta_{i'}'' \langle \boldsymbol{\xi}_i, \boldsymbol{\xi}_{i'} \rangle \ge 1 - y_i y_{k^\star} \theta_{k^\star}'' \langle \boldsymbol{\xi}_i, \boldsymbol{\xi}_{k^\star} \rangle.
$$

Then we add $y_i y_{k^\star} w \langle \boldsymbol{\xi}_i, \boldsymbol{\xi}_{k^\star} \rangle$ on both sides, where we set $w = \theta''_{k^\star} - \frac{\alpha - 1}{(1+\kappa)d - 2\sqrt{d\log(6n^2/\delta)}} \leq \theta''_{k^\star}$. Then we have

$$\theta''_i \cdot \|\boldsymbol{\xi}_{i'}\|^2 + \sum_{i' \neq i, k^\star} y_i y_{i'} \theta''_{i'} \langle \boldsymbol{\xi}_i, \boldsymbol{\xi}_{i'} \rangle + y_i y_{k^\star} w \langle \boldsymbol{\xi}_i, \boldsymbol{\xi}_{k^\star} \rangle \geq 1 - y_i y_{k^\star} (\theta''_{k^\star} - w) \langle \boldsymbol{\xi}_i, \boldsymbol{\xi}_{k^\star} \rangle$$

$$\geq 1 - 2(\theta''_{k^\star} - w)\sqrt{d\log(6n^2/\delta)}$$

$$= \frac{(1+\kappa)d - 2\alpha\sqrt{d\log(6n^2/\delta)}}{(1+\kappa)d - 2\sqrt{d\log(6n^2/\delta)}}. \quad (49)$$

The second inequality is from Lemma 57. Now consider a new $\overline{\boldsymbol{v}} = \overline{\lambda}_1 \boldsymbol{\mu}_1 + \overline{\lambda}_2 \boldsymbol{\mu}_2 + \sum_{i \in [n]} y_i \overline{\theta}_i \boldsymbol{\xi}_i$ with

$$\overline{\lambda}_1 = \lambda''_1; \quad \overline{\lambda}_2 = \lambda''_2;$$

$$\overline{\theta}_i = \theta''_i / (1 - 2(\theta''_{k^\star} - w)\sqrt{d\log(6n^2/\delta)}) \text{ for } i \in [n], i \neq k^\star$$

and

$$\overline{\theta}_{k^\star} = \frac{w}{1 - 2(\theta''_{k^\star} - w)\sqrt{d\log(6n^2/\delta)}}.$$

We can prove that $\overline{\boldsymbol{v}}$ satisfies all constraints for $\boldsymbol{v}_{mm}$.

From the first two inequalities in Condition 9, we have $\overline{\lambda}_1 \|\boldsymbol{\mu}_1\|^2 = \lambda''_1 \|\boldsymbol{\mu}_1\|^2 \geq 1$, $-\overline{\lambda}_2 \|\boldsymbol{\mu}_2\|^2 = -\lambda''_2 \|\boldsymbol{\mu}_2\|^2 \geq 1$. Then by dividing $1 - 2(\theta''_{k^\star} - w)\sqrt{d\log(6n^2/\delta)}$ on both sides of (49), for $\forall i \in \mathcal{N}, i \neq k^\star$ we have

$$\overline{\theta}_i \cdot \|\boldsymbol{\xi}_i\|^2 + \sum_{i' \neq i} y_i y_{i'} \overline{\theta}_i \langle \boldsymbol{\xi}_i, \boldsymbol{\xi}_{i'} \rangle \geq 1.$$

Lastly we prove that $\overline{\theta}_{k^\star} \|\boldsymbol{\xi}_{k^\star}\|^2 + \sum_{i \neq k^\star} y_i y_{k^\star} \overline{\theta}_i \langle \boldsymbol{\xi}_i, \boldsymbol{\xi}_{k^\star} \rangle \geq 1$. From the last inequality in Condition 9 we have

$$\theta''_{k^\star} \cdot \|\boldsymbol{\xi}_{k^\star}\|^2 + \sum_{i \neq k^\star} y_{k^\star} y_i \theta''_i \langle \boldsymbol{\xi}_i, \boldsymbol{\xi}_{k^\star} \rangle \geq \alpha.$$

Dividing $1 - 2(\theta''_{k^\star} - w)\sqrt{d\log(6n^2/\delta)}$ on both sides, we get

$$\frac{\theta''_{k^\star} \|\boldsymbol{\xi}_{k^\star}\|^2}{1 - 2(\theta''_{k^\star} - w)\sqrt{d\log(6n^2/\delta)}} + \sum_{i \neq k^\star} y_i y_{k^\star} \overline{\theta}_i \langle \boldsymbol{\xi}_i, \boldsymbol{\xi}_{k^\star} \rangle \geq \frac{\alpha}{1 - 2(\theta''_{k^\star} - w)\sqrt{d\log(6n^2/\delta)}}.$$

Therefore we have

$$\overline{\theta}_{k^\star} \|\boldsymbol{\xi}_{k^\star}\|^2 + \sum_{i \neq k^\star} y_i y_{k^\star} \overline{\theta}_i \langle \boldsymbol{\xi}_i, \boldsymbol{\xi}_{k^\star} \rangle \geq \frac{\alpha - (\theta''_{k^\star} - w)\|\boldsymbol{\xi}_{k^\star}\|^2}{1 - 2(\theta''_{k^\star} - w)\sqrt{d\log(6n^2/\delta)}} \geq \frac{\alpha - (\theta''_{k^\star} - w)(1+\kappa)d}{1 - 2(\theta''_{k^\star} - w)\sqrt{d\log(6n^2/\delta)}} = 1.$$

The second inequality is from Lemma 57 and the last equality is by our definition $\theta''_{k^\star} - w = \frac{\alpha - 1}{(1+\kappa)d - 2\sqrt{d\log(6n^2/\delta)}}$. Thus, $\overline{\boldsymbol{v}}$ is a possible solution under Condition 1 and $\|\overline{\boldsymbol{v}}\| \geq \|\boldsymbol{v}_{mm}\|$.

Next we estimate the difference between $\|\boldsymbol{v}''\|^2$ and $\|\overline{\boldsymbol{v}}\|^2$. The expansion of $\|\boldsymbol{v}''\|^2$ and $\|\overline{\boldsymbol{v}}\|^2$ are:

$$\|\boldsymbol{v}''\|^2 = \lambda''^2_1 \|\boldsymbol{\mu}_1\|^2 + \lambda''^2_2 \|\boldsymbol{\mu}_2\|^2 + \sum_{i \in \mathcal{N}} \theta''^2_i \|\boldsymbol{\xi}_i\|^2 + \sum_{i \in \mathcal{N}} \sum_{j \in \mathcal{N}} y_i y_j \theta''_i \theta''_j \langle \boldsymbol{\xi}_i, \boldsymbol{\xi}_j \rangle,$$

$$\|\overline{\boldsymbol{v}}\|^2 = \overline{\lambda}^2_1 \|\boldsymbol{\mu}_1\|^2 + \overline{\lambda}^2_2 \|\boldsymbol{\mu}_2\|^2 + \sum_{i \in \mathcal{N}} \overline{\theta}^2_i \|\boldsymbol{\xi}_i\|^2 + \sum_{i \in \mathcal{N}} \sum_{j \in \mathcal{N}} y_i y_j \overline{\theta}_i \overline{\theta}_j \langle \boldsymbol{\xi}_i, \boldsymbol{\xi}_j \rangle.$$

According to the condition (44), we have $\|\boldsymbol{v}''\| \leq 2\|\boldsymbol{v}_{mm}\| = \Theta(\sqrt{1/\rho^2 + \eta n/d})$, which implies that $\alpha = O(\sqrt{n}\log n)$. Otherwise, we have

$$\theta''_{k^\star} \|\boldsymbol{\xi}_{k^\star}\|^2 \geq \alpha - \sum_{i \neq k^\star} y_{k^\star} y_i \theta''_i \langle \boldsymbol{\xi}_i, \boldsymbol{\xi}_{k^\star} \rangle = \Omega(\alpha).$$

It further yields that

$$\|\boldsymbol{v}''\|^2 = \Omega(\frac{1}{\rho^2}) + \Omega(\frac{\eta n}{d}) + \theta_{k^\star}''^2 \|\boldsymbol{\xi}_{k^\star}\|^2 = \Omega(\frac{1}{\rho^2} + \frac{\eta n}{d} + \frac{\alpha^2}{d}) = \Omega(\frac{n\log^2 n}{d}),$$

which contradicts with $\|\boldsymbol{v}''\| = \Theta(\sqrt{1/\rho^2 + \eta n/d})$. We decompose the difference between $\|\boldsymbol{v}''\|^2$ and $\|\overline{\boldsymbol{v}}\|^2$ into four terms:

$$\|\boldsymbol{v}''\|^2 - \|\overline{\boldsymbol{v}}\|^2 = \underbrace{(\theta_{k^\star}''^2 - \overline{\theta}_{k^\star}^2)\|\boldsymbol{\xi}_{k^\star}\|^2}_{I_1} + \underbrace{\sum_{i\in\mathcal{N},i\neq k^\star}(\theta_i''^2 - \overline{\theta}_i^2)\|\boldsymbol{\xi}_i\|^2}_{I_2} - \underbrace{\sum_{i\in\mathcal{N}}\sum_{j\in\mathcal{N}} y_i y_j \overline{\theta}_i \overline{\theta}_j \langle \boldsymbol{\xi}_i, \boldsymbol{\xi}_j \rangle}_{I_3}$$

$$+ \underbrace{\sum_{i\in\mathcal{N}}\sum_{j\in\mathcal{N}} y_i y_j \theta_i'' \theta_j'' \langle \boldsymbol{\xi}_i, \boldsymbol{\xi}_j \rangle}_{I_4}.$$

We now estimate $I_1$ to $I_4$ sequentially. For the first term,

$$I_1 \geq (\theta_{k^\star}''^2 - \overline{\theta}_{k^\star}^2)(1-\kappa)d = (\theta_{k^\star}'' - \overline{\theta}_{k^\star})(\theta_{k^\star}'' + \overline{\theta}_{k^\star})(1-\kappa)d$$

$$= \frac{(\alpha-1)(1 - 2\theta_{k^\star}''\sqrt{d\log(6n^2/\delta)})}{(1+\kappa)d - 2\sqrt{d\log(6n^2/\delta)}} \cdot \Omega\left(\frac{1}{d}\right) \cdot (1-\kappa)d$$

$$= \Omega\left(\frac{\alpha-1}{d}\right),$$

where the first inequality is from Lemma 57; the second equality is from Lemma 31; and the last equality uses the fact that $\alpha = O(\sqrt{n}\log n)$. Then we can further upper bound $\max_{i\in\mathcal{N},i\neq k^\star} \theta_i''$ as

$$\max_{i\in\mathcal{N},i\neq k^\star} \theta_i'' \leq \frac{(1-\kappa)d + 2(\alpha - n_2)\sqrt{d\log(6n^2/\delta)}}{((1-\kappa)d - 2n_2\sqrt{d\log(6n^2/\delta)})^2} = O(\frac{1}{d}). \tag{50}$$

For the second term $I_2$, we have

$$|I_2| \leq \sum_{i\in\mathcal{N},i\neq k^\star}(\overline{\theta}_i^2 - \theta_i''^2)(1+\kappa)d$$

$$\leq \left(\frac{1}{(1 - (\theta_{k^\star}'' - w)\sqrt{d\log(6n^2/\delta)})^2} - 1\right)\max_{i\in\mathcal{N},i\neq k^\star} \theta_i''^2 \cdot \eta n(1+\kappa)d$$

$$= \frac{(\alpha-1)\sqrt{d\log(6n^2/\delta)}}{(1+\kappa)d - \sqrt{d\log(6n^2/\delta)}} \cdot O(\frac{\eta n}{d}) = \widetilde{O}\left(\frac{(\alpha-1)\eta n}{d^{3/2}}\right).$$

The second inequality is from Lemma 31. The first equality is from (50) and the last equality is from Assumption 5.

Then we bound $|-I_3 + I_4|$ as:

$$|-I_3 + I_4| \leq \sum_{i\in\mathcal{N}}\sum_{j\in\mathcal{N}\setminus\{i\}} |\overline{\theta}_i\overline{\theta}_j - \theta_i''\theta_j''| \cdot |\langle \boldsymbol{\xi}_i, \boldsymbol{\xi}_j \rangle|$$

$$\leq \sum_{i\in\mathcal{N}\setminus\{k^\star\}}\sum_{j\in\mathcal{N}\setminus\{k^\star,i\}} |\overline{\theta}_i\overline{\theta}_j - \theta_i''\theta_j''| \cdot |\langle \boldsymbol{\xi}_i, \boldsymbol{\xi}_j \rangle| + 2\sum_{t\in\mathcal{N}\setminus\{k^\star\}} |\overline{\theta}_{k^\star}\overline{\theta}_t - \theta_{k^\star}''\theta_t''| \cdot |\langle \boldsymbol{\xi}_{k^\star}, \boldsymbol{\xi}_t \rangle|$$

$$\leq (\eta n)^2\left(\frac{1}{(1 - (\theta_{k^\star}'' - w)\sqrt{d\log(6n^2/\delta)})^2} - 1\right)\max_{i\in\mathcal{N},i\neq k^\star} \theta_i''^2 \cdot 2\sqrt{d\log(6n^2/\delta)}$$

$$+ \eta n\left(\theta_{k^\star}'' - \frac{\overline{\theta}_{k^\star}}{1 - 2(\theta_{k^\star}'' - w)\sqrt{d\log(6n^2/\delta)}}\right)\max_{i\in\mathcal{N},i\neq k^\star} \theta_i'' 4\sqrt{d\log(6n^2/\delta)}$$

$$\leq \frac{(\alpha-1)\sqrt{d\log(6n^2/\delta)}}{(1+\kappa)d - \sqrt{d\log(6n^2/\delta)}} \cdot O(\frac{(\eta n)^2(1+\kappa)}{d^{3/2}}) + \frac{\alpha-1}{d}\cdot O(\eta n\frac{c_1}{d})\cdot 2\sqrt{d\log(6n^2/\delta)}$$

$$= O\left(\frac{(\alpha-1)\eta^2 n^2}{d^2} + \frac{(\alpha-1)\eta n}{d^{3/2}}\right).$$

The third inequality is from Lemma 27 and Lemma 31; The fourth inequality is from the fact that

$$\theta_{k^\star}'' - \frac{\overline{\theta}_{k^\star}}{1 - 2(\theta_{k^\star}'' - w)\sqrt{d\log(6n^2/\delta)}} = \frac{\theta_{k^\star}'' - \overline{\theta}_{k^\star} - 2\theta_{k^\star}''(\theta_{k^\star}'' - w)\sqrt{d\log(6n^2/\delta)}}{1 - 2(\theta_{k^\star}'' - w)\sqrt{d\log(6n^2/\delta)}}$$

$$= \frac{\Omega(\frac{\alpha-1}{d}) - O(\frac{\alpha(\alpha-1)}{d^{3/2}})}{1 - 2(\theta_{k^\star}'' - w)\sqrt{d\log(6n^2/\delta)}} > 0$$

So we have $\theta_{k^\star}'' - \frac{\overline{\theta}_{k^\star}}{1 - 2(\theta_{k^\star}'' - w)\sqrt{d\log(6n^2/\delta)}} \leq \theta_{k^\star}'' - \overline{\theta}_{k^\star}$; The last equality is from Assumption 5.

Combining the above results, we have

$$\|\boldsymbol{v}''\|_2^2 - \|\boldsymbol{v}_{mm}\|_2^2 \geq \Theta\left(\frac{\alpha-1}{d}\right) + O\left(\frac{(\alpha-1)\eta n}{d^{3/2}}\right) \geq \frac{C_3(1-\beta)}{d}.$$

Here $C_3 = \Theta(1)$ is a constant. □

Now we can prove the main proposition in this case.

*Proof of Proposition 15 under Case 2.1.* From Lemma 33 we have

$$\|\boldsymbol{v}''\|_2^2 - \|\boldsymbol{v}_{mm}\|_2^2 \geq \frac{C_3(1-\beta)}{d} = T'(1-\beta).$$

Here we substitute $T' = \frac{C_3}{d} \geq 0$. Then we have

$$\Gamma^2 - \Gamma''^2 = \frac{1}{\|\boldsymbol{v}_{mm}\|^2} - \frac{1}{\|\boldsymbol{v}''\|^2} = \frac{\|\boldsymbol{v}''\|^2 - \|\boldsymbol{v}_{mm}\|^2}{\|\boldsymbol{v}''\|^2 \cdot \|\boldsymbol{v}_{mm}\|^2} \geq \frac{T'(1-\beta)}{\|\boldsymbol{v}''\|^2 \cdot \|\boldsymbol{v}_{mm}\|^2}.$$

Therefore,

$$\Gamma - \Gamma'' \geq \frac{T'(1-\beta)}{(\Gamma + \Gamma'')\|\boldsymbol{v}_{mm}\|^2 \cdot \|\boldsymbol{v}'\|^2} \geq \frac{T'(1-\beta)}{2\Gamma\|\boldsymbol{v}_{mm}\|^2 \cdot \|\boldsymbol{v}''\|^2} = \frac{T'(1-\beta)}{2\|\boldsymbol{v}_{mm}\|\|\boldsymbol{v}''\|^2} \geq \frac{T'(1-\beta)}{2\|\boldsymbol{v}''\|^3}.$$

The last inequality is from $\|\boldsymbol{v}''\| \geq \|\boldsymbol{v}_{mm}\|$. This implies

$$\Gamma'' \leq \Gamma - \frac{T'(1-\beta)}{2\|\boldsymbol{v}''\|^3} \leq \Gamma - \frac{C_1}{\|\boldsymbol{v}_{mm}\|^3 n\rho^2}(1-\beta).$$

The last inequality is from our assumption that $\|\boldsymbol{v}''\| \leq 2\|\boldsymbol{v}_{mm}\|$ and $\rho^2 = \Omega(d/n)$. □

Then we consider the other case.

***Case 2.2*** $k - p = n_2$

In this case, all noisy samples are mixed. From previous analysis, this is equivalent to strengthening all conditions $y_i \boldsymbol{v}^\top \boldsymbol{\xi}_i \geq 1$ while other conditions remain the same. As mixing $k - p$ samples will not result in a better solution than only mixing 1 noisy sample, the proof is the same as *Case 2.1* and we omit it for convenience.

Finally, we consider the last scenario.

**Case 3:** $p \neq 0, k - p \neq 0$

This scenario is more complex as both clean and noisy sets are mixed. There are four cases to consider

1. $p < n_1, k - p < n_2$. (Both clean and noisy sets are partially mixed)

2. $p < n_1, k - p = n_2$ (Clean set is partially mixed, noisy set is all mixed)

3. $p = n_1, k - p < n_2$ (Clean set is all mixed, noisy set is partially mixed)

4. $p = n_1, k - p = n_2$ (Both clean and noisy sets are all mixed)

We will go over every case to prove Proposition 15 holds.

**Case 3.1** $p < n_1, k - p < n_2$

This case is simple because from the analysis above, mixing 1 more clean sample is equivalent to adding 1 more constraint and mixing 1 more noisy sample is equivalent to strengthening 1 original constraint. So mixing both sets will not result in a better solution than only mixing 1 clean sample. Therefore, the proof is the same as *Case 1.1* and we omit is for convenience.

**Case 3.2** $p < n_1, k - p = n_2$

In this case, all noisy samples and part of clean samples are mixed. We can consider this case as an extension of *Case 2.2* by mixing some clean samples. From previous analysis, mixing 1 more clean sample is equivalent to adding 1 more constraint. So this case will not result in a better solution than *Case 2.2*. The following proof is the same as *Case 2.2* and we omit it for convenience.

**Case 3.3** $p = n_1, k - p < n_2$

In this case, all clean samples and part of noisy samples are mixed. We can consider this case as an extension of *Case 1.2* by mixing some noisy samples. From previous analysis, mixing 1 more noisy sample is equivalent to strengthening 1 original constraint. So this case will not result in a better solution than *Case 1.2*. The following proof is the same as *Case 1.2* and we omit it for convenience.

**Case 3.4** $p = n_1, k - p = n_2$

This case is more complex. We cannot simply consider it as an extension of *Case 2.2* because the analysis of *Case 2.2* is based on the condition that there exist clean samples that follow optimal token selection rule. Denote $\boldsymbol{r}_i = \beta_i \boldsymbol{\mu}_i + (1 - \beta_i)\boldsymbol{\xi}_i$ for $i \in \mathcal{C}$ and $\boldsymbol{r}_i = (1 - \beta_i)\boldsymbol{\mu}_i + \beta_i \boldsymbol{\xi}_i$ for $i \in \mathcal{N}$. The condition in this case becomes

**Condition 10** (All samples are mixed).

$$y_i \boldsymbol{v}''^\top \boldsymbol{r}_i \geq 1.$$

*This indicates*

$$\begin{cases} \beta_i y_i \lambda_i'' \|\boldsymbol{\mu}_i\|^2 + (1 - \beta_i)(\theta_i''\|\boldsymbol{\xi}_i\|^2 + \sum_{j \neq i} y_i y_j \theta_j'' \langle \boldsymbol{\xi}_i, \boldsymbol{\xi}_j \rangle) \geq 1, i \in \mathcal{C}, \\ (1 - \beta_i)y_i \lambda_i'' \|\boldsymbol{\mu}_i\|^2 + \beta_i(\theta_i''\|\boldsymbol{\xi}_i\|^2 + \sum_{j \neq i} y_i y_j \theta_j'' \langle \boldsymbol{\xi}_i, \boldsymbol{\xi}_j \rangle) \geq 1, i \in \mathcal{N}. \end{cases}$$

Assume that $\min\{\lambda_1'' \cdot \|\boldsymbol{\mu}_1\|^2, -\lambda_2'' \cdot \|\boldsymbol{\mu}_2\|^2\} = q$ in optimal $\boldsymbol{v}''$. If $q \geq 1$, we can directly follow the proof in *Case 2.2*. Otherwise, denote $\alpha = \frac{1 - \beta_i q}{1 - \beta_i}$. We have $\alpha > 1$ due to $q < 1$ and $0 \leq \beta_i < 1$. Without losing generality, we assume $\lambda_1'' \cdot \|\boldsymbol{\mu}_1\|^2 = q < 1$. Then consider the following relaxed condition

**Condition 11** (Relaxed version of constraints in Condition 10).

$$\theta_i''\|\boldsymbol{\xi}_i\|^2 + \sum_{j \neq i} y_i y_j \theta_j'' \langle \boldsymbol{\xi}_i, \boldsymbol{\xi}_j \rangle \geq \alpha, i \in \mathcal{C}_1.$$

Denote the optimal solution under Condition 11 as $\breve{\boldsymbol{v}}$ and the corresponding coefficients in $\breve{\boldsymbol{v}}$ as $\breve{\lambda}_1, \breve{\lambda}_2$ and $\breve{\theta}_i$, i.e.

$$\breve{\boldsymbol{v}} = \breve{\lambda}_1 \boldsymbol{\mu}_1 + \breve{\lambda}_2 \boldsymbol{\mu}_2 + \sum_{i \in [n]} \breve{\theta}_i \boldsymbol{\xi}_i.$$

Since the constraints in Condition 11 is a subset of the constraints in Condition 10, we have $\|\breve{\boldsymbol{v}}\| \leq \|\boldsymbol{v}''\|$. Meanwhile, we have the following lemma to estimate $\breve{\theta}_i$:

**Lemma 34.** *Suppose that Assumption 5 holds, under Condition 11, we have*

$$\breve{\theta}_i = 0, i \in [n] \backslash \mathcal{C}_1;$$

$$\breve{\theta}_i \in \left[ \frac{\alpha}{(1 + \kappa)d}\left(1 - \frac{n\sqrt{d\log(6n^2/\delta)}}{(1 - \kappa)d - n\sqrt{d\log(6n^2/\delta)}}\right), \frac{\alpha}{((1 - \kappa)d - 2n_{11}\sqrt{d\log(6n^2/\delta)}} \right], i \in \mathcal{C}_1.$$

*Proof of Lemma 34.* Note that Condition 11 does not have any constraint for samples with $i \in [n] \backslash \mathcal{C}_1$. Thus we have $\breve{\theta}_i = 0$ for any $i \in [n] \backslash \mathcal{C}_1$ in the representation (39). Denote $j = \underset{i \in \mathcal{C}_1}{\arg\max} \, \breve{\theta}_i$, then we have

$$\breve{\theta}_j \cdot \|\boldsymbol{\xi}_j\|^2 + \sum_{k \neq j} y_k y_j \breve{\theta}_k \langle \boldsymbol{\xi}_i, \boldsymbol{\xi}_j \rangle \geq \breve{\theta}_j \|\boldsymbol{\xi}_j\|^2 - 2 \breve{\theta}_j n_{11} \sqrt{d \log(6n^2/\delta)} \geq \breve{\theta}_j ((1 - \kappa)d - 2n_{11} \sqrt{d \log(6n^2/\delta)}).$$

The two inequalities are from Lemma 57 and our definition of $j$. Consider the contrary case when $\breve{\theta}_j > \frac{\alpha}{((1-\kappa)d - 2n_{11}\sqrt{d\log(6n^2/\delta)})}$, we have

$$y_j \breve{\boldsymbol{v}}^\top \boldsymbol{\xi}_j > \alpha.$$

By the complementary slackness condition, if $y_j \breve{\boldsymbol{v}}^\top \boldsymbol{\xi}_j > \alpha$, then we must have $\breve{\theta}_j = 0$, and thus we reach a contradiction.

Then we lower bound $\breve{\theta}_i$. For $\forall i \in \mathcal{C}_1$ we have

$$\alpha \leq \breve{\theta}_i \cdot \|\boldsymbol{\xi}_i\|^2 + \sum_{j \neq i} y_i y_j \breve{\theta}_i \langle \boldsymbol{\xi}_i, \boldsymbol{\xi}_j \rangle \leq \breve{\theta}_i (1 + \kappa)d + 2n_{11} \max_{i \in [n]} \breve{\theta}_i \sqrt{d \log(6n^2/\delta)}$$

$$\leq \breve{\theta}_i (1 + \kappa)d + \frac{2\alpha n_{11} \sqrt{d \log(6n^2/\delta)}}{(1 - \kappa)d - 2n_{11}\sqrt{d \log(6n^2/\delta)}}.$$

The second inequality is from Lemma 57 and the last inequality is from the upper bound of $\breve{\theta}_i$ we just derived. Therefore, we have

$$\breve{\theta}_i \geq \frac{\alpha}{(1 + \kappa)d} \left( 1 - \frac{2n_{11}\sqrt{d \log(6n^2/\delta)}}{(1 - \kappa)d - 2n_{11}\sqrt{d \log(6n^2/\delta)}} \right).$$

$\square$

From this Lemma we have $\breve{\theta}_i = \Theta(\alpha/d)$ for $i \in \mathcal{C}_1$. Similar as (48), under our assumption $\|\breve{\boldsymbol{v}}\| \leq 2\|\boldsymbol{v}_{mm}\|$, we have $\alpha = O(\log(n))$. Next we estimate the difference between $\|\breve{\boldsymbol{v}}\|^2$ and $\|\boldsymbol{v}_{mm}\|^2$. We can prove that Lemma 33 still holds in this case.

*Proof of Lemma 33.* Under this case, the difference between $\|\breve{\boldsymbol{v}}\|_2^2$ and $\|\boldsymbol{v}_{mm}\|_2^2$ becomes

$$\|\breve{\boldsymbol{v}}\|^2 - \|\boldsymbol{v}_{mm}\|^2 \geq \underbrace{\sum_{i \in [n]} (\breve{\theta}_i^2 - \theta_i^2)\|\boldsymbol{\xi}_i\|^2 - (\lambda_1^2 - \breve{\lambda}_1^2)\|\boldsymbol{\mu}_1\|^2 - (\lambda_2^2 - \breve{\lambda}_2^2)\|\boldsymbol{\mu}_2\|^2}_{I_1}$$

$$\underbrace{- \sum_{i \in \mathcal{N}} \sum_{j \in \mathcal{N} \backslash \{i\}} y_i y_j \theta_i \theta_j \langle \boldsymbol{\xi}_i, \boldsymbol{\xi}_j \rangle}_{I_2} + \underbrace{\sum_{i \in \mathcal{C}_1} \sum_{j \in \mathcal{C}_1 \backslash \{i\}} y_i y_j \breve{\theta}_i \breve{\theta}_j \langle \boldsymbol{\xi}_i, \boldsymbol{\xi}_j \rangle}_{I_3}$$

We then bound $I_1 \sim I_3$ respectively. For $I_1$ we have

$$|I_1| \geq \sum_{i \in \mathcal{C}_1} \breve{\theta}_i^2 \|\boldsymbol{\xi}_i\|^2 - \sum_{i \in \mathcal{N}} \theta_i^2 \|\boldsymbol{\xi}_i\|^2 - 2/\rho^2 \geq n_{11} \min_{i \in [n]} \breve{\theta}_i^2 (1 - \kappa)d - n_2 \max_{i \in \mathcal{N}} \theta_i^2 (1 + \kappa)d - 2/\rho^2$$

$$\geq \frac{\alpha^2 n_{11}(1 - \kappa)}{(1 + \kappa)^2 d} \left( 1 - \frac{2\sqrt{d \log(6n^2/\delta)}}{(1 - \kappa)d - 2n_{11}\sqrt{d \log(6n^2/\delta)}} \right) - \frac{n_2(1 + \kappa)d}{((1 - \kappa)d - 2n_2\sqrt{d \log(6n^2/\delta)})^2} - \frac{2}{\rho^2}$$

$$= \Omega\left( \frac{n}{d} \right).$$

The second inequality is from Lemma 57; The third inequality is from Lemma 27 and 34; The last equality is due to the SNR condition $\rho/\sqrt{d} = \Omega(1/\sqrt{n})$ so that $\frac{1}{\rho^2} \leq \frac{n}{4d}$. For $I_2$, we have

$$|I_2| \leq \sum_{i \in \mathcal{N}} \max_{i \in \mathcal{N}} \theta_i^2 \cdot 2\sqrt{d \log(6n^2/\delta)} \leq \frac{2n_2\sqrt{d \log(6n^2/\delta)}}{((1 - \kappa)d - 2n_2\sqrt{d \log(6n^2/\delta)})^2} = \widetilde{O}\left( \frac{n}{d^{3/2}} \right).$$

The first inequality is from Lemma 57; The second inequality is from Lemma 27. Similarly, for $|I_3|$ we have

$$|I_3| \leq \sum_{i \in \mathcal{C}_1} \max_{i \in \mathcal{C}_1} \breve{\theta}_i^2 \cdot 2\sqrt{d \log(6n^2/\delta)} \leq \frac{2n_{11}\alpha^2\sqrt{d \log(6n^2/\delta)}}{((1-\kappa)d - 2n_{11}\sqrt{d \log(6n^2/\delta)})^2} = \widetilde{O}\left(\frac{n}{d^{3/2}}\right).$$

The second inequality is from Lemma 34. Combining the above results, we have

$$\|\boldsymbol{v}''\|_2^2 - \|\boldsymbol{v}\|_2^2 \geq \Theta\left(\frac{n_{11}}{d}\right) - \widetilde{O}\left(\frac{n}{d^{3/2}}\right) \geq \frac{C_3 n(1-\beta)}{d}.$$

The remaining proof is the same as *Case 2.1* and we omit it for convenience. $\qquad\square$

Therefore, we complete the proof for all possible scenarios. $\qquad\square$

**Training and Test Error Analysis**

From Proposition 15 we can analyze the properties of both parameters to estimate the training and test error.

In this section, we first get the convergence direction of parameters $\boldsymbol{p}$ and $\boldsymbol{v}$. The main difference between our setting with Ataee Tarzanagh et al. (2023b) is that they only consider the infinite case and their results hold only when $R, r \to \infty$. We extend their results to the finite case. Specifically, given fixed upper bound $R$ and $r$ for $\|\boldsymbol{p}\|$ and $\|\boldsymbol{v}\|$ respectively, we denote the solution of the constrained optimization (2) as $(\boldsymbol{v}_r, \boldsymbol{p}_R)$ in this section for brevity.

Our main theorem in this section estimates the corresponding deviation of $\boldsymbol{p}_R/R$ and $\boldsymbol{v}_r/r$ from their convergence direction $\boldsymbol{p}_{mm}/\|\boldsymbol{p}_{mm}\|$ and $\boldsymbol{v}_{mm}/\|\boldsymbol{v}_{mm}\|$. For a given $\boldsymbol{p}$, it is elementary that the margin induced by $\boldsymbol{p}$ is $\min_{i,t_i \neq \alpha_i}(\boldsymbol{x}_{i\alpha_i} - \boldsymbol{x}_{it_i})^\top \boldsymbol{p}/\|\boldsymbol{p}\|$, thus when $\|\boldsymbol{p}\| = 1$, the margin becomes $\min_{i,t_i \neq \alpha_i}(\boldsymbol{x}_{i\alpha_i} - \boldsymbol{x}_{it_i})^\top \boldsymbol{p}$. And for a given $\boldsymbol{v}$, the label margin induced by $\boldsymbol{v}$ is $\min_i y_i \boldsymbol{v}^\top \boldsymbol{r}_i/\|\boldsymbol{v}\|$. Recall that the label margin induced by $\boldsymbol{v}_{mm}$ is $\Gamma$ and the margin of $p$-SVM induced by $\boldsymbol{p}_{mm}$ is $\Xi$.

First we introduce a lemma to estimate the norm of $\|\boldsymbol{p}_{mm}\|$. This will benefit our proof of the main theorem.

**Lemma 35** (Norm of $\boldsymbol{p}_{mm}$)**.** *Suppose that Assumption 5 holds, recall that the solution of (p-SVM) is $\boldsymbol{p}_{mm}$. With probability at least $1 - \delta$ on the training dataset we have*

$$\frac{1}{\rho^2} + \frac{\eta n}{d} \leq \|\boldsymbol{p}_{mm}\|^2 \leq \frac{8}{\rho^2} + \frac{17\eta n}{d}.$$

*This implies*

$$\|\boldsymbol{p}_{mm}\| = \Theta\left(\sqrt{\frac{1}{\rho^2} + \frac{\eta n}{d}}\right).$$

*Proof of Lemma 35.* First we prove the upper bound. Consider the following possible solution $\widetilde{\boldsymbol{p}}$:

$$\widetilde{\boldsymbol{p}} = \frac{2(\boldsymbol{\mu}_1 + \boldsymbol{\mu}_2)}{\rho^2} + \sum_{i \in \mathcal{N}} 4\frac{\boldsymbol{\xi}_i}{d}. \tag{51}$$

We then proved that $\widetilde{\boldsymbol{p}}$ satisfies (36). For $k \in \mathcal{C}$ we have

$$\widetilde{\boldsymbol{p}}^\top(\boldsymbol{\mu}_k - \boldsymbol{\xi}_k) = 2 - \sum_{i \in \mathcal{N}} 4\frac{\langle \boldsymbol{\xi}_i, \boldsymbol{\xi}_k \rangle}{d} \geq 2 - \frac{4n_2\sqrt{d\log(6n^2/\delta)}}{d} \geq 1.$$

The first inequality is from the definition of $d$ in Lemma 57 and the second inequality is from Assumption 5. And for $k \in \mathcal{N}$, we have

$$\widetilde{\boldsymbol{p}}^\top(\boldsymbol{\xi}_k - \boldsymbol{\mu}_k) = -2 + \sum_{i \in \mathcal{N}} 4\frac{\langle \boldsymbol{\xi}_i, \boldsymbol{\xi}_k \rangle}{d} \geq -2 + 4(1-\kappa) + \sum_{i \in \mathcal{N}, i \neq k} 4\frac{\langle \boldsymbol{\xi}_i, \boldsymbol{\xi}_k \rangle}{d}$$

$$\geq -2 + 4(1-\kappa) + \frac{4n_2\sqrt{d\log(6n^2/\delta)}}{d} \geq 1.$$

The first and second inequalities are from Lemma 57; The last inequality is from Assumption 5.

Therefore, the max-margin solution $\boldsymbol{p}_{mm}$ must have no greater norm than $\widetilde{\boldsymbol{p}}$. So we can upper bound $\boldsymbol{p}_{mm}$ as

$$\|\boldsymbol{p}_{mm}\|^2 \leq \|\widetilde{\boldsymbol{p}}\|^2 = \frac{8}{\rho^2} + \frac{16}{d^2}\Big(\sum_{i\in\mathcal{N}}\|\boldsymbol{\xi}_i\|^2 + \sum_{i,j\in\mathcal{N}, i\neq j}\langle\boldsymbol{\xi}_i, \boldsymbol{\xi}_j\rangle\Big)$$

$$\leq \frac{8}{\rho^2} + \frac{16}{d^2}\big((1+\kappa)n_2 d + 2n_2^2\sqrt{d\log(6n^2/\delta)}\big) \leq \frac{8}{\rho^2} + \frac{17\eta n}{d}.$$

The second inequality is from Lemma 57; The last inequality is from the definition of $d$ in Assumption 5.

Then we prove for the lower bound. As $\boldsymbol{p}_{mm}$ is the max-margin solution and satisfies KKT condition, it can be expressed as the sum of signal and noise tokens. Then we decompose $\boldsymbol{p}_{mm} = \boldsymbol{p}_{\boldsymbol{\mu}}^{mm} + \boldsymbol{p}_{\boldsymbol{\xi}}^{mm}$ where $\boldsymbol{p}_{\boldsymbol{\mu}}^{mm} = f_1^{mm}\boldsymbol{\mu}_1 + f_2^{mm}\boldsymbol{\mu}_2$ and $\boldsymbol{p}_{\boldsymbol{\xi}}^{mm} = \sum_{i\in[n]}g_i^{mm}\boldsymbol{\xi}_i$. Note that $\boldsymbol{\mu}_j \perp \boldsymbol{\xi}_i$ for all $j \in \{\pm1\}, i \in [n]$. From Lemma 39, we have $f_j^{mm} \geq 0.9/\rho^2$, so we can lower bound $\|\boldsymbol{p}_{\boldsymbol{\mu}}^{mm}\|_2^2$ as

$$\|\boldsymbol{p}_{\boldsymbol{\mu}}^{mm}\|_2^2 = f_1^{mm2}\|\boldsymbol{\mu}_1\|^2 + f_2^{mm2}\|\boldsymbol{\mu}_2\|^2 \geq \frac{2 \cdot 0.9^2}{\rho^2} \geq \frac{1}{\rho^2}.$$

As for $\|\boldsymbol{p}_{\boldsymbol{\xi}}^{mm}\|_2$, from p-SVM condition, for every noisy sample we have

$$\boldsymbol{p}_{mm}^\top(\boldsymbol{\xi}_i - \boldsymbol{\mu}_i) \geq 1,$$

which indicates

$$\boldsymbol{p}_{\boldsymbol{\xi}}^{mm\top}\boldsymbol{\xi}_i = \boldsymbol{p}_{mm}^\top\boldsymbol{\xi}_i \geq 1 + \boldsymbol{p}_{mm}^\top\boldsymbol{\mu}_i \geq 1.9.$$

The last inequality is from Lemma 39. Sum up the inequality for all noisy sample, we have

$$\sum_{i\in\mathcal{N}}\boldsymbol{p}_{\boldsymbol{\xi}}^{mm\top}\boldsymbol{\xi}_i \geq 1.9n_2.$$

Thus,

$$\|\boldsymbol{p}_{\boldsymbol{\xi}}^{mm}\| \geq \frac{1.9n_2}{\|\sum_{i\in\mathcal{N}}\boldsymbol{\xi}_i\|} = \frac{1.9n_2}{\sqrt{\sum_{i\in\mathcal{N}}\|\boldsymbol{\xi}_i\|^2 + \sum_{i,j\in\mathcal{N}}\langle\boldsymbol{\xi}_i, \boldsymbol{\xi}_j\rangle}} \geq \frac{1.9n_2}{\sqrt{2\cdot n_2\cdot(1+\kappa)d}} \geq \sqrt{\frac{\eta n}{d}}.$$

The second inequality is from Lemma 57 and the last inequality is from Assumption 5. Therefore,

$$\|\boldsymbol{p}_{mm}\|^2 = \|\boldsymbol{p}_{\boldsymbol{\mu}}^{mm}\|_2^2 + \|\boldsymbol{p}_{\boldsymbol{\xi}}^{mm}\|_2^2 \geq \frac{1}{\rho^2} + \frac{\eta n}{d}.$$

Combining the results above, we have

$$\|\boldsymbol{p}_{mm}\|^2 = \Theta\Big(\frac{1}{\rho^2} + \frac{\eta n}{d}\Big).$$

$\square$

**Definition 36.** *Let $f : \mathbb{R}^2 \to \mathbb{R}^d$. We say that*

$$\lim_{x,y\to\infty} f(x,y) = L$$

*iff $\forall\epsilon > 0 \exists M$ such that $\forall x, y > M$ we have that $\|f(x,y) - L\| < \epsilon$.*

**Remark 37.** *Let $g : \mathbb{R} \to \mathbb{R}$ be a function with $\lim_{x\to\infty} g(x) = \infty$. Assume that $\lim_{x,y\to\infty} f(x,y) = L$, then $\lim_{x\to\infty} f(x, g(x)) = L$ and $\lim_{x\to\infty} f(g(x), x) = L$*

Now we introduce our key theorem:

**Theorem 16.** *Suppose that Assumption 5 holds, with probability at least $1 - \delta$ on the training dataset, we have*

- *The margin induced by $\boldsymbol{p}_{(r,R)}/R$ in p-SVM is at least $(1-\zeta)\Xi$, where*

$$\zeta = \frac{\log(4\sqrt{\rho^2 + (1+\kappa)d}\|\boldsymbol{v}_{mm}\|^3 d\rho^2)}{R\Xi}.$$

- *The label margin induced by $\boldsymbol{v}_{(r,R)}/r$ in v-SVM is at least $(1-\gamma)\Gamma$, where $\gamma = \frac{2\sqrt{\rho^2 + (1+\kappa)d}}{\Gamma \exp((1-\zeta)R\Xi)}$.*

*Proof of Theorem 16.* From Proposition 15, we have that for any $\|\boldsymbol{p}\|$, the label margin $1/\|\boldsymbol{v}(\boldsymbol{p})\|$ is at most

$$\Gamma - \frac{C\max_{i\in[n]}(1-s_{i\alpha_i})}{\|\boldsymbol{v}_{mm}\|^3 n\rho^2},$$

where $\alpha_i = 1$ for $i \in \mathcal{C}$ and $\alpha_i = 2$ for $i \in \mathcal{N}$. Recall that $\boldsymbol{s}_i = \mathbb{S}(\boldsymbol{X}_i\boldsymbol{p})$ is the softmax probability vector. We define $q_i^{\boldsymbol{p}} = 1 - s_{i\alpha_i}$ to measure the amount of non-optimality (attention on non-optimal token).

We first consider the convergence of $\boldsymbol{p}_R$ and use contradiction to prove the first statement. Denote $\boldsymbol{p}_R^{mm} = R\boldsymbol{p}_{mm}/\|\boldsymbol{p}_{mm}\|$ which has the same norm as $\boldsymbol{p}_R$ and the direction of $\boldsymbol{p}_{mm}$. Suppose the margin induced by $\boldsymbol{p}_R/R$ is at most $(1-\zeta)\Xi$, i.e. $\min_{i,t_i\neq\alpha_i}(\boldsymbol{x}_{i\alpha_i}-\boldsymbol{x}_{it_i})^\top\boldsymbol{p}_R \leq (1-\zeta)R\Xi, \forall i \in [n]$. Note that here each sequence only has two tokens, thus $t_i, \alpha_i \in [2]$, and $t_i = 3 - \alpha_i$.

According to Lemma 35, we have

$$\Xi = \|\boldsymbol{p}_{mm}\|_2^{-1} = \Theta((\eta n/d + 1/\rho^2)^{-1/2}).$$

Following the definition of $q_i^{\boldsymbol{p}}$ above, we set $\widehat{q}_{max} = \sup_{i\in[n]} q_i^{\boldsymbol{p}_R}$ and $q_{max}^* = \sup_{i\in[n]} q_i^{\boldsymbol{p}_R^{mm}}$ to be the worst non-optimality in $\boldsymbol{p}_R$ and $\boldsymbol{p}_R^{mm}$. Then we have

$$q_i^{\boldsymbol{p}_R^{mm}} = \frac{\exp(\boldsymbol{x}_{it_i}^\top\boldsymbol{p}_R^{mm})}{\sum_{t\in[2]}\exp(\boldsymbol{x}_{it}^\top\boldsymbol{p}_R^{mm})} \leq \frac{\exp(\boldsymbol{x}_{it_i}^\top\boldsymbol{p}_R^{mm})}{\exp(\boldsymbol{x}_{i\alpha_i}^\top\boldsymbol{p}_R^{mm})} \leq \exp(-R\Xi).$$

The last inequality is from the definition of $\boldsymbol{p}_{mm}$ that $\boldsymbol{p}_{mm}^\top(\boldsymbol{x}_{i\alpha_i}-\boldsymbol{x}_{it}) \geq 1$, so $\boldsymbol{p}_R^{mm\top}(\boldsymbol{x}_{i\alpha_i}-\boldsymbol{x}_{it}) \geq R/\|\boldsymbol{p}_{mm}\| = R\Xi$. Thus, $q_{max}^* = \sup_{i\in[n]} q_i^{\boldsymbol{p}_{mm}} \leq \exp(-R\Xi)$. Then denote the output of attention layer $\boldsymbol{r}_i = \boldsymbol{X}_i^\top\mathbb{S}(\boldsymbol{X}_i\boldsymbol{p}_R^{mm})$. Define $\epsilon_i = \|\boldsymbol{r}_i - \boldsymbol{x}_{i\alpha_i}\|$, we have $y_i \cdot \boldsymbol{r}_i^\top\boldsymbol{v}_{mm} \geq y_i \cdot \boldsymbol{x}_{i\alpha_i}^\top\boldsymbol{v}_{mm} - \|\boldsymbol{r}_i - \boldsymbol{x}_{i\alpha_i}\| \cdot \|\boldsymbol{v}_{mm}\| \geq 1 - \epsilon_i/\Gamma$. So if we set $\epsilon_{max} = \sup_{i\in[n]} \epsilon_i$, $\boldsymbol{v}_{mm}$ achieves a label margin of at least $\Gamma - \epsilon_{max}$ on $(y_i, \boldsymbol{r}_i)_{i\in[n]}$. To better estimate $\epsilon_{max}$, we define $M = \sup_{i\in[n]} \|\boldsymbol{\mu}_i - \boldsymbol{\xi}_i\| \leq \sqrt{\rho^2 + (1+\kappa)d}$, then we have

$$\epsilon_{max} = M \cdot q_{max}^* \leq M\exp(-R\Xi). \tag{52}$$

This implies the max-margin achieved by $(\boldsymbol{p}_R^{mm}, \boldsymbol{v}_r^{mm})$ is at least

$$y_i f(\boldsymbol{p}_R^{mm}, \boldsymbol{v}_r^{mm}; \boldsymbol{x}_i) = y_i \boldsymbol{v}_r^{mm\top}\boldsymbol{r}_i \geq r\Gamma - r\epsilon_{max} \geq r\Gamma - rM\exp(-R\Xi). \tag{53}$$

The first inequality is from $y_i \cdot \boldsymbol{r}_i^\top\boldsymbol{v}_r^{mm} \geq r(\Gamma - \epsilon_i)$ and the last inequality is from (52).

Then we consider the case when $\min_{i,t_i\neq\alpha_i}(x_{i\alpha_i} - x_{it_i})^\top\boldsymbol{p}_R \leq (1-\zeta)R\Xi$ the minimal margin constraint is $\zeta$-violated by $\boldsymbol{p}_R$. Without losing generality we assume that $1 = \underset{i\in[n]}{\operatorname{argmin}}[(\boldsymbol{x}_{i\alpha_i} - \boldsymbol{x}_{it})^\top\boldsymbol{p}_R]_{t\neq\alpha_i}$. Then we have

$$\widehat{q}_{max} \geq \frac{\exp(\boldsymbol{x}_{1t_1}^\top\boldsymbol{p}_R)}{\sum_{t\in[2]}\exp(\boldsymbol{x}_{1t}^\top\boldsymbol{p}_R)} \geq \frac{1}{2}\frac{\exp(\boldsymbol{x}_{1t_1}^\top\boldsymbol{p}_R)}{\exp(\boldsymbol{x}_{1\alpha_1}^\top\boldsymbol{p}_R)} \geq \frac{1}{2\exp((1-\zeta)R\Xi)}.$$

From Proposition 15, optimizing v-SVM on $(y_i, \widehat{\boldsymbol{r}}_i)_{i\in[n]}$ can achieve the max-margin at most

$$\min_{i\in[n]} y_i f(\boldsymbol{p}_R, \boldsymbol{v}_r; \boldsymbol{x}_i) \leq \Gamma - \frac{C}{2\|\boldsymbol{v}_{mm}\|^3 n\rho^2} \cdot e^{-(1-\zeta)R\Xi}. \tag{54}$$

And from the definition $\zeta = \frac{1}{R\Xi}\log(2M\|\boldsymbol{v}_{mm}\|^3 n\rho^2/C)$, we have

$$\frac{C}{2\|\boldsymbol{v}_{mm}\|^3 n\rho^2}\exp(-(1-\zeta)R\Xi) > M\exp(-R\Xi)$$

for sufficiently large $R$, which implies

$$\min_{i\in[n]} y_i \cdot f(\boldsymbol{p}_R, \boldsymbol{v}_r; \boldsymbol{x}_i) < \min_{i\in[n]} y_i \cdot f(\boldsymbol{p}_R^{mm}, \boldsymbol{v}_r^{mm}; \boldsymbol{x}_i).$$

This contradicts with the problem definition (2) to maximize the margin.

Then we prove for the second statement. When the margin induced by $\boldsymbol{p}_R/R$ in $p$-SVM is less than $(1-\zeta)\Xi$, we can use the proof above to derive a contradiction, so $(\boldsymbol{x}_{i\alpha_1} - \boldsymbol{x}_{it})^\top \boldsymbol{p}_R \geq (1-\zeta)R\Xi$ must hold. Then set $\widehat{\boldsymbol{r}}_i = \boldsymbol{X}_i^\top \mathcal{S}(\boldsymbol{X}_i \boldsymbol{p}_R)$, we have that

$$\min_{i\in[n]} y_i \boldsymbol{v}_r^\top \widehat{\boldsymbol{r}}_i \leq \min_{i\in[n]} y_i \boldsymbol{v}_r^\top \boldsymbol{x}_{i\alpha_i} + \sup_{i\in[n]}|\boldsymbol{v}_r^\top(\widehat{\boldsymbol{r}}_i - \boldsymbol{x}_{i\alpha_i})|$$

$$\leq (1-\gamma)\Gamma r + M\exp(-(1-\zeta)R\Xi)r$$

$$\leq (1-\gamma/2)\Gamma r.$$

The second inequality is from previous analysis that $(\boldsymbol{x}_{i\alpha_i} - \boldsymbol{x}_{it})^\top \boldsymbol{p}_R \geq (1-\zeta)R\Xi$, so $|\widehat{\boldsymbol{r}}_i - \boldsymbol{x}_{i1}| \leq M\exp(-(1-\zeta)R\Xi)$; The last inequality is from our definition $\gamma = \frac{2M}{\Gamma\exp((1-\zeta)R\Xi)}$.

Therefore, combining with (53), we have

$$\gamma\Gamma r/2 > rM\exp(-R\Xi),$$

which implies

$$\min_{i\in[n]} y_i \cdot f(\boldsymbol{p}_R, \boldsymbol{v}_r; \boldsymbol{x}_i) < \min_{i\in[n]} y_i \cdot f(\boldsymbol{p}_R^{mm}, \boldsymbol{v}_r^{mm}; \boldsymbol{x}_i).$$

Again this contradicts with the problem definition (2). $\qquad\square$

Then we have the following lemma to bound the derivation $\zeta$ and $\gamma$:

**Lemma 38.** *Suppose that Assumption 5 holds, consider the same setting in Theorem 16, we have $\zeta < 0.2$ and $\gamma < 1$.*

*Proof of Lemma 38.* From the definition of $\zeta$ in Theorem 16, we have

$$\zeta = \frac{\log(2M\|\boldsymbol{v}_{mm}\|^3 n\rho^2/C)}{R\Xi} = C_1 \frac{1}{R\sqrt{\eta n/d + 1/\rho^2}}\log(M\|\boldsymbol{v}_{mm}\|^3 n\rho^2)$$

$$\leq C_2 \frac{1}{R\sqrt{\eta n/d + 1/\rho^2}}\log\left(\frac{n^2(\rho^2+d)(\rho^2\eta n+d)^3}{\rho^2 d^3}\right) = \frac{C_3}{R\sqrt{\eta n/d + 1/\rho^2}}\log(\rho n) < 0.2.$$

Here $C_1, C_2, C_3 = \Theta(1)$. The first inequality is from the upper bound of $\|\boldsymbol{v}_{mm}\|$ in Lemma 28 and the last inequality is from the definition of $R$ in Assumption 5. And for $\gamma$, we have

$$\gamma = \frac{2M}{\Gamma\exp((1-\zeta)R\Xi)} = C_1' \frac{M\|\boldsymbol{v}_{mm}\|}{\exp(R/\|\boldsymbol{v}_{mm}\|)} \leq C_2' \frac{\sqrt{(\rho^2+d)(\eta n/d + 1/\rho^2)}}{\exp(R/\sqrt{\eta n/d + 1/\rho^2})} < 1.$$

Here $C_1', C_2' = \Theta(1)$. The first inequality is from the lower and upper bound of $\|\boldsymbol{v}_{mm}\|$ in Lemma 28 and the last inequality is from the definition of $R$ in Assumption 5. $\qquad\square$

Then we can estimate $\langle \boldsymbol{p}_R, \boldsymbol{\mu}\rangle$ with the following lemma:

**Lemma 39.** *Suppose that Assumption 5 holds, with probability at least $1-\delta$ on the training dataset, $\boldsymbol{p}_R$ should satisfy*

$$0.5(1-\zeta)R\Xi \leq \langle \boldsymbol{p}_R, \boldsymbol{\mu}_j\rangle \leq R\rho$$

*for $j \in \{1, 2\}$.*

*Proof of Lemma 39.* The upper bound is given by

$$\langle \boldsymbol{p}_R, \boldsymbol{\mu}_j \rangle \leq \|\boldsymbol{p}_R\|\|\boldsymbol{\mu}_j\| = R\rho.$$

Then we use contradiction to prove for the lower bound. From Theorem 16, $\boldsymbol{p}_R$ satisfies

$$\boldsymbol{p}_R^\top (\boldsymbol{\mu}_i - \boldsymbol{\xi}_i) \geq (1-\zeta)R\Xi, i \in \mathcal{C}$$
$$\boldsymbol{p}_R^\top (\boldsymbol{\xi}_i - \boldsymbol{\mu}_i) \geq (1-\zeta)R\Xi, i \in \mathcal{N} \tag{55}$$

If $\langle \boldsymbol{p}_R, \boldsymbol{\mu}_j \rangle \leq 0.5(1-\zeta)R\Xi$, then for every clean sample from cluster j we must have $\langle \boldsymbol{p}_R, \boldsymbol{\xi}_i \rangle \leq -0.5(1-\zeta)R\Xi$ and thus

$$\langle \boldsymbol{p}_R, \sum_{i \in \mathcal{C}_j} \boldsymbol{\xi}_i \rangle = \sum_{i \in \mathcal{C}_j} \langle \boldsymbol{p}_R, \boldsymbol{\xi}_i \rangle \leq -0.5(1-\zeta)R\Xi n_{1j}.$$

So we could estimate $\|\boldsymbol{p}_R\|$ as follows

$$\|\boldsymbol{p}_R\| \geq 0.5(1-\zeta)R\Xi \cdot n_{1j} \frac{1}{\|\sum_{i \in \mathcal{C}_j} \boldsymbol{\xi}_i\|} = 0.5(1-\zeta)R\Xi \cdot n_{1j} \frac{1}{\sqrt{\sum_{i \in \mathcal{C}_j} \|\boldsymbol{\xi}_i\|^2 + \sum_{i,j \in \mathcal{C}_j} \langle \boldsymbol{\xi}_i, \boldsymbol{\xi}_j \rangle}}$$

$$\geq 0.5(1-\zeta)R\Xi \cdot n_{1j} \frac{1}{\sqrt{2 \cdot n_{1j} \cdot (1+\kappa)d}} \geq 0.4R\Xi \cdot \frac{\sqrt{n_{1j}}}{\sqrt{2(1+\kappa)d}}.$$

The first inequality is from the property of innerproduct; The second inequality is from Lemma 57 and the definition of d in Assumption 5; The last inequality is from Lemma 38. Meanwhile, from Lemma 35 we have $\|\boldsymbol{p}_{mm}\| \leq \sqrt{8/\rho^2 + 17\eta n/d}$. Recall that $\Xi = \|\boldsymbol{p}_{mm}\|^{-1}$. Therefore, we further have

$$\|\boldsymbol{p}_R\| \geq 0.4R\Xi \cdot \frac{\sqrt{n_{1j}}}{\sqrt{2(1+\kappa)d}} \geq \sqrt{\frac{0.4^2 n_{1j}}{(8/\rho^2 + 17\eta n/d) \cdot 2(1+\kappa)d}} \cdot R$$

$$\geq \sqrt{\frac{0.04(n - \eta n - O(\sqrt{n}))}{(8/\rho^2 + 17\eta n/d) \cdot (1+\kappa)d}} \cdot R > R.$$

The second inequality is from Lemma 35; The third inequality is from Lemma 59 and the last inequality is from Assumption 5 about SNR and $\eta$. This leads to a contradiction.

$\square$

Now we can estimate the output of attention layer for some test sample $(\boldsymbol{X}, y)$.

**Lemma 40.** *Suppose that Assumption 5 holds, with probability at least $1 - \delta$ on the training dataset, for a given a test sample $\boldsymbol{X}, y$, where $\boldsymbol{X} = (\boldsymbol{\mu}^\star, \boldsymbol{\xi}^\star)$, $\boldsymbol{\mu}^\star$ can be $\boldsymbol{\mu}_1$ or $\boldsymbol{\mu}_2$, we have with probability at least $1 - \exp\left(-\frac{1}{2}(\frac{1}{2}(1-\zeta)\Xi - K/R)^2\right)$ that*

$$\langle \boldsymbol{p}_R, \boldsymbol{\mu}^\star \rangle - \langle \boldsymbol{p}_R, \boldsymbol{\xi}^\star \rangle \geq K,$$

*where $K \leq \frac{1}{2}(1-\zeta)R\Xi$ and $\zeta, \Xi$ are defined in Theorem 16.*

*Proof of Lemma 40.* Note that $\boldsymbol{p}^\top \boldsymbol{\xi}^\star$ follows Gaussian distribution $\mathcal{N}(0, R^2)$, we have

$$\mathbb{P}(\langle \boldsymbol{p}_R, \boldsymbol{\mu}^\star \rangle - \langle \boldsymbol{p}_R, \boldsymbol{\xi}^\star \rangle < K) = \mathbb{P}(\langle \boldsymbol{p}_R, \boldsymbol{\xi}^\star \rangle > \langle \boldsymbol{p}_R, \boldsymbol{\mu}^\star \rangle - K) \leq \mathbb{P}(\boldsymbol{p}_R^\top \boldsymbol{\xi}^\star > \frac{1}{2}(1-\zeta)R\Xi - K)$$

$$\leq \exp\left(-\frac{1}{2}(\frac{1}{2}(1-\zeta)\Xi - K/R)^2\right).$$

The first inequality is from Lemma 39 and the second inequality comes from the property of Gaussian tail probability.
$\square$

We also have the following lemma to estimate $\boldsymbol{v}_r$. We first prove that $\boldsymbol{v}_r$ can be expressed as the sum of signal and noise tokens.

**Lemma 41.** *The solution of constrained optimization problem (2) $\boldsymbol{v}_r$ can be expressed in the form that*

$$\boldsymbol{v}_r = \lambda_1 \boldsymbol{\mu}_1 + \lambda_2 \boldsymbol{\mu}_2 + \sum_{i=1}^{n} \theta_i \boldsymbol{\xi}_i.$$

*Proof of Lemma 41.* Similar to Theorem 16, define $\widehat{\boldsymbol{r}}_i = \boldsymbol{X}_i^\top \mathcal{S}(\boldsymbol{X}_i \boldsymbol{p}_R)$ as the output of attention layer, we have

$$\boldsymbol{v}_r = \operatorname*{argmax}_{\|\boldsymbol{v}\| \leq r} \min_{i \in [n]} y_i \boldsymbol{v}^\top \boldsymbol{r}_i. \tag{56}$$

Then denote $s = \min_{i \in [n]} y_i \boldsymbol{v}^\top \boldsymbol{r}_i$ and $s_r = \min_{i \in [n]} y_i \boldsymbol{v}_r^\top \boldsymbol{r}_i$. Then (56) can be written as

$$(\boldsymbol{v}_r, s_r) = \operatorname*{argmax}_{\boldsymbol{v}, s} s, \text{ s.t. } y_i \boldsymbol{v}^\top \boldsymbol{r}_i \geq s, \quad 1 \leq i \leq n$$
$$\|\boldsymbol{v}\| \leq r.$$

The corresponding Lagrangian function is

$$L(s, \psi) = -s + \sum_{i=1}^{n} \psi_i y_i (s - y_i \boldsymbol{v}^\top \boldsymbol{r}_i) + \psi_0 (\|\boldsymbol{v}\|^2 - r^2).$$

Take derivative of this function on $(s, \boldsymbol{v})$, we have

$$-\sum_{i=1}^{n} \psi_i y_i \boldsymbol{r}_i + 2\psi_0 \boldsymbol{v} = 0.$$

Therefore from the last equation we can get

$$\boldsymbol{v} = \frac{1}{2\psi_0} \sum_{i=1}^{n} \psi_i y_i \boldsymbol{r}_i.$$

As $\boldsymbol{r}_i = \beta_i \boldsymbol{\mu}_i + (1 - \beta_i) \boldsymbol{\xi}_i$ for every $i \in [n]$, $\boldsymbol{v}$ can be expressed as the combination of signal and noise token of every sample:

$$\boldsymbol{v}_r = \lambda_1 \boldsymbol{\mu}_1 + \lambda_2 \boldsymbol{\mu}_2 + \sum_{i=1}^{n} \theta_i \boldsymbol{\xi}_i.$$

$\square$

Based on this representation, we can then bound the parameters in $\boldsymbol{v}_r$:

**Lemma 42.** *Suppose that Assumption 5 holds, denote $\boldsymbol{v}_r = \lambda_1 \boldsymbol{\mu}_1 + \lambda_2 \boldsymbol{\mu}_2 + \sum_{i \in [n]} \theta_i \boldsymbol{\xi}_i$. Then with probability at least $1 - \delta$ on the training dataset, we have*

$$\lambda_1 \geq (1 - \gamma) \Gamma r / \rho^2,$$
$$\lambda_2 \leq -(1 - \gamma) \Gamma r / \rho^2,$$
$$|\theta_i| \leq 2\sqrt{1/\rho^2 + 5\eta n/d} \cdot \Gamma r / \sqrt{d}.$$

*Proof of Lemma 42.* The first two statements are obvious because from Theorem 16 we have

$$y_i \boldsymbol{v}_r^\top \boldsymbol{\mu}_i \geq (1 - \gamma) \Gamma r,$$

for $\forall i \in \mathcal{C}$. This implies $|\lambda_j| \geq (1 - \gamma) \Gamma r / \rho^2$ for $j \in \{1, 2\}$. Meanwhile, we decompose $\boldsymbol{v}_r = \boldsymbol{v}_{\boldsymbol{\mu}} + \boldsymbol{v}_{\boldsymbol{\xi}}$ where $\boldsymbol{v}_{\boldsymbol{\mu}} = \lambda_1 \boldsymbol{\mu}_1 + \lambda_2 \boldsymbol{\mu}_2$ and $\boldsymbol{v}_{\boldsymbol{\xi}} = \sum_{i \in [n]} \theta_i \boldsymbol{\xi}_i$. And we can upper bound $\|\boldsymbol{v}_{\boldsymbol{\xi}}\|$ as

$$\|\boldsymbol{v}_{\boldsymbol{\xi}}\|^2 = \|\boldsymbol{v}_r\|^2 - \|\boldsymbol{v}_{\boldsymbol{\mu}}\|^2 \leq r^2 - \lambda_1^2 \rho^2 - \lambda_2^2 \rho^2 \leq r^2 (1 - 2(1 - \gamma)^2 \Gamma^2 / \rho^2).$$

The first inequality is from $\|\boldsymbol{v}\| \leq r$ and the second inequality is from the first two statements we just proved. Therefore, denote $j = \underset{i \in [n]}{\operatorname{argmax}}\ \theta_i$, we have

$$\theta_j^2 \|\boldsymbol{\xi}_j\|^2 \leq \|\boldsymbol{v}_{\boldsymbol{\xi}}\|^2 \leq r^2 (1 - 2(1-\gamma)^2 \Gamma^2/\rho^2).$$

Then we can upper bound $|\theta_j|$ as

$$\begin{aligned}
\theta_j^2 &\leq r^2(1 - 2(1-\gamma)^2 \Gamma^2/\rho^2)/\|\boldsymbol{\xi}_j\|^2 \leq r^2(1 - 2(1-\gamma)^2 \Gamma^2/\rho^2)/(1-\kappa)d \\
&= r^2\left(1 - \frac{2(1-\gamma)^2}{\|\boldsymbol{v}_{mm}\|^2 \rho^2}\right)/(1-\kappa)d \leq r^2\left(1 - \frac{1}{(2/\rho^2 + 5\eta n/d)\rho^2}\right)/(1-\kappa)d \\
&= \frac{1 + 5\eta n\rho^2/d}{2 + 5\eta n\rho^2/d} \cdot \frac{r^2}{(1-\kappa)d} \leq \left(\frac{1}{\rho^2} + \frac{5\eta n}{d}\right) \cdot \frac{\Gamma^2 r^2}{2d}.
\end{aligned}$$

The second inequality is from Lemma 57; The third inequality is from Lemma 28 that $\|\boldsymbol{v}_{mm}\| \leq \sqrt{2/\rho^2 + 5\eta n/d}$ and our definition of $\gamma = \frac{2\sqrt{\rho^2 + (1+\kappa)d}}{\Gamma \exp((1-\zeta)R\Xi)}$; The last inequality is from $\Gamma = \|\boldsymbol{v}_{mm}\|^{-1} \geq (2/\rho^2 + 5\eta n/d)^{-1}$. Thus, we can bound $|\theta_j|$ as

$$|\theta_j| \leq 2\sqrt{1/\rho^2 + 5\eta n/d} \cdot \Gamma r/\sqrt{d}.$$

$\square$

Therefore, we can prove the main theorem.

*Proof of Theorem 6.* First we show that the model can perfectly classify all training samples. From Theorem 16, we have

$$y_i \boldsymbol{v}_r^\top \boldsymbol{r}_i \geq (1-\gamma)\Gamma r > 0$$

for $\forall i \in [n]$. The last inequality is from Lemma 38. Thus $y_i = \operatorname{sign}(f(\boldsymbol{X}_i; \boldsymbol{p}_R, \boldsymbol{v}_r))$ for all $i \in [n]$. Then we bound the test error. Given a test sample $\boldsymbol{X}, y$, where $\boldsymbol{X} = (\boldsymbol{\mu}^\star, \boldsymbol{\xi}^\star)$, $\boldsymbol{\mu}^\star$ can be $\boldsymbol{\mu}_1$ or $\boldsymbol{\mu}_2$. From Remark58, with probability at least $1 - 6n \exp(-d/4C_1 n^2)$,

$$|\langle \boldsymbol{\xi}^\star, \boldsymbol{\xi}_i \rangle| \leq \frac{d}{C_1 n}. \tag{57}$$

According to Lemma 40, with probability at least $1 - \exp\left(-\frac{1}{2}(\frac{1}{2}(1-\zeta)\Xi - K/R)^2\right)$, we have

$$y \cdot f(\boldsymbol{p}_R, \boldsymbol{v}_r; \boldsymbol{X}) \geq \frac{\langle y\boldsymbol{v}_r, e^K \boldsymbol{\mu}^\star + \boldsymbol{\xi}^\star \rangle}{e^K + 1} \geq \frac{e^K(1-\gamma)\Gamma r \|\boldsymbol{\mu}^\star\|^2}{\rho^2(e^K + 1)} - \frac{1}{e^K + 1}\sum_{i \in [n]} |\theta_i| \cdot |\langle \boldsymbol{\xi}_i, \boldsymbol{\xi}^\star \rangle|. \tag{58}$$

Let $K = \log(\sqrt{d}\sqrt{1/\rho^2 + \eta n/d}) + C < \frac{1}{2}(1-\zeta)R\Xi$. By uniform bound, we have that with probability at least $1 - 6n\exp(-d/4C_1 n^2) - \exp\left(-\frac{1}{2}(\frac{1}{2}(1-\zeta)\Xi - K/R)^2\right)$,

$$\begin{aligned}
y \cdot f(\boldsymbol{p}_R, \boldsymbol{v}_r; \boldsymbol{X}) &\geq \frac{e^K(1-\gamma)\Gamma r - n \cdot d/(C_1 n) \cdot 2\sqrt{1/\rho^2 + \eta n/d} \cdot \Gamma r/\sqrt{d}}{1 + e^K} \\
&\geq \frac{0.8 e^K \Gamma r - \sqrt{d}/C_1 \cdot 2\sqrt{1/\rho^2 + \eta n/d} \cdot \Gamma r}{1 + e^K} \\
&> 0,
\end{aligned}$$

where the first inequality uses (57), (58) and Lemma 42; The second inequality is from Lemma 38 and the last inequality is from Assumption 5 and our selection of $K$. Therefore,

$$\mathbb{P}(y \neq f(\boldsymbol{p}_R, \boldsymbol{v}_r; \boldsymbol{X})) \leq \exp\left(-\frac{1}{2}(\frac{1}{2}(1-\zeta)\Xi - \frac{K}{R})^2\right) + \delta,$$

where $\zeta = \frac{\log(2M\|\boldsymbol{v}_{mm}\|^3 n\rho^2)}{R\Xi} = \Theta\Big(\frac{\sqrt{\eta n/d + 1/\rho^2}}{R}\log(\rho n)\Big)$, $K = \log(\sqrt{d}\sqrt{1/\rho^2 + \eta n/d}) + C = \Theta(\log(\sqrt{d/\rho^2 + \eta n})$ and $\Xi = \|\boldsymbol{p}_{mm}\|_2^{-1} = \Theta((\eta n/d + 1/\rho^2)^{-1/2})$. Plugging in the order of $\Xi$ and $K$, we have

$$
\begin{aligned}
\mathbb{P}_{(\boldsymbol{X},y)\sim\mathcal{D}}&(y \neq \text{sign}(f(\boldsymbol{X};\boldsymbol{p}_R,\boldsymbol{v}_r)))\\
&= \mathbb{P}_{(\boldsymbol{X},y)\sim\mathcal{D}}(y \neq \text{sign}(f(\boldsymbol{X};\boldsymbol{p}_R,\boldsymbol{v}_r)), y = -\widetilde{y})\\
&\quad + \mathbb{P}_{(\boldsymbol{X},y)\sim\mathcal{D}}(y \neq \text{sign}(f(\boldsymbol{X};\boldsymbol{p}_R,\boldsymbol{v}_r)), y = \widetilde{y})\\
&= \eta + \mathbb{P}_{(\boldsymbol{X},y)\sim\mathcal{D}}(y \neq \text{sign}(f(\boldsymbol{X};\boldsymbol{p}_R,\boldsymbol{v}_r)), y = \widetilde{y})\\
&\leq \eta + \exp(-d/C_1 n^2) + \exp\Big(-\Theta\big(\frac{(1-\zeta)}{\sqrt{\eta n/d + 1/\rho^2}} - \frac{\log(nd\sqrt{1/\rho^2 + \eta n/d})}{R}\big)^2\Big)\\
&= \eta + \exp(-\Omega(\frac{d}{n^2})) + \exp\Big(-\Omega\big(\frac{(1-\zeta)}{\sqrt{\eta n/d + 1/\rho^2}} - \frac{\log(d)}{R}\big)^2\Big),
\end{aligned}
$$

where $\zeta = \Theta\Big(\frac{\sqrt{\eta n/d + 1/\rho^2}}{R}\log(\rho n)\Big)$. This completes the proof. $\qquad\square$

### A.2.3 PROOF OF THM. 8

**Lemma 43.** *Consider the next joint-constrained max margin solution:*

$$
(\boldsymbol{v}_t, \boldsymbol{p}_t) = \underset{\|\boldsymbol{v}\|^2 + \|\boldsymbol{p}\|^2 \leq t}{\text{argmax}} \ \min_i y_i f(\boldsymbol{X}_i; \boldsymbol{p}, \boldsymbol{v}). \tag{59}
$$

*Let $r_t := \|\boldsymbol{v}_t\|$ and $R_t := \|\boldsymbol{v}_t\|$, then $(\boldsymbol{v}_t, \boldsymbol{p}_t) = \big(\boldsymbol{v}_{(r_t, R_t)}, \boldsymbol{p}_{(r_t, R_t)}\big)$, where $\big(\boldsymbol{v}_{(r_t, R_t)}, \boldsymbol{p}_{(r_t, R_t)}\big)$ is a solution to Problem 2. Moreover, under Assumption 5 (items 1-3), with probability at least $1 - \delta$ over the random data generation, we have that $r_t \to \infty, R_t \to \infty$ as $t \to \infty$.*

*Proof.* By Proposition 15, with probability at least $1 - \delta$, for all $\boldsymbol{p} \in \mathbb{R}^d$, the token selection under $\boldsymbol{p}$ results in a label margin of at most $\Gamma - c \cdot \max_{i\in[n]}(1 - s_{i\alpha_i}^{\boldsymbol{p}})$ in 26 (with $\boldsymbol{r}_i = \boldsymbol{X}_i^\top \boldsymbol{S}(\boldsymbol{X}_i\boldsymbol{p})$), where $\alpha_i = \mathbb{I}(i \in \mathcal{C}) + 2\mathbb{I}(i \in \mathcal{N})$, $\boldsymbol{s}_i^{\boldsymbol{p}} = \mathbb{S}(\boldsymbol{X}_i\boldsymbol{p})$ is the softmax probabilities, and $c := C/|\boldsymbol{v}_{mm}\|^3 n\rho^2$ is some constant (which may depends on $n$ and $d$, but not in $t$).

Observe that as the norm of $\boldsymbol{v}$ increases, the margin increases; thus, it's easy to verify that $\|\boldsymbol{v}_t\| \to \infty$ as $t \to \infty$. We argue that also $\|\boldsymbol{p}_t\| \to \infty$ as $t \to \infty$. To see that, assume by contradiction that $\|\boldsymbol{p}_t\| \leq R_0$ for some arbitrary large $t$ that will be determined later. Set $\Gamma = 1/\|\boldsymbol{v}_{mm}\|$, $\|\boldsymbol{v}_t\| = r_t$, $\widetilde{\boldsymbol{v}}_{mm} = (r_t - 1)\Gamma\boldsymbol{v}_{mm}$. Hence $t = r_t^2 + R_0^2$ and $\|\widetilde{\boldsymbol{v}}_{mm}\|^2 = (r-1)^2$. The idea is that by decreasing $\|\boldsymbol{v}_t\|$ by 1, we can choose $\boldsymbol{p}$ with $\|\boldsymbol{p}\|^2 + (r_t - 1)^2 = t = r_t^2 + R_0^2$, i.e., $\|\boldsymbol{p}\|^2 = 2r_t - 1 + R_0^2$, which can be arbitrary large for large enough $t$. Set $\Pi := 1/\|\boldsymbol{p}_{mm}\|$ and $\widetilde{\boldsymbol{p}}_{mm} := \sqrt{2r_t - 1 + R_0^2}\Pi\boldsymbol{p}_{mm}$. The proof strategy is obtaining a contradiction by proving that $(\widetilde{\boldsymbol{v}}_{mm}, \widetilde{\boldsymbol{p}}_{mm})$ is a strictly better solution compared to $(\boldsymbol{v}_t, \boldsymbol{p}_t)$. Define $q_i^{\boldsymbol{p}} = 1 - \boldsymbol{s}_{i\alpha_i}^{\boldsymbol{p}}$ to be the amount of non-optimality sopftmax probability where $\boldsymbol{s}_i^{\boldsymbol{p}} = \mathbb{S}(\boldsymbol{X}_i\boldsymbol{p})$ is the softmax probabilities and $\alpha_i = 1$ iff $i \in \mathcal{C}$ and 2 otherwise. Then we have that

$$
\max_i q_i^{\boldsymbol{p}_t} \geq \kappa
$$

where $\kappa > 0$ is a constant that depends just on $R_0$ and data parameters (e.g. $n, d, \rho, \delta$). On the other hand, for every $\epsilon > 0$, we have that

$$
q^* = \max_i q_i^{\widetilde{\boldsymbol{p}}_{mm}} \leq \epsilon,
$$

for large enough $r_t$ i.e. large enough $t$. Therefore, By Proposition 15 (see the first paragraph in the proof), we can upper bound the margin induced by $\boldsymbol{v}_t$ on $(Y_i, \boldsymbol{r}_i)$ for $\boldsymbol{r}_i = \boldsymbol{X}_i^\top\mathbb{S}(\boldsymbol{X}_i\boldsymbol{p}_t)$ by

$$
\min_{i\in[n]} y_i \boldsymbol{v}_t^\top \boldsymbol{r}_i \leq r_t(\Gamma - c\kappa),
$$

for some constant $c > 0$. On the other hand, the margin induced by $\widetilde{\boldsymbol{v}}_{mm}$ on $(Y_i, \boldsymbol{r}_i)$ for $\boldsymbol{r}_i = \boldsymbol{x}_{i\alpha_i}$ is $(r_t - 1)\Gamma$. This means that we margin induced by $\widetilde{\boldsymbol{v}}_{mm}$ on $(y_i, \boldsymbol{r}_i)$ for $\boldsymbol{r}_i = \boldsymbol{X}_i^\top \mathbb{S}(\boldsymbol{X}_i \widetilde{\boldsymbol{p}}_{mm})$ is at least

$$\min_i y_i \boldsymbol{r}_i^\top \widetilde{\boldsymbol{v}}_{mm} \geq \min_i y_i x_{i\alpha_i}^\top \widetilde{\boldsymbol{v}}_{mm} - q^* \left\| \boldsymbol{x}_i^{(1)} - \boldsymbol{x}_i^{(2)} \right\| \|\widetilde{\boldsymbol{v}}_{mm}\|$$
$$\geq (\boldsymbol{r}_t - 1)(\Gamma - M\epsilon),$$

where $M = \sup_{i \in n} \left\| \boldsymbol{x}_i^{(1)} - \boldsymbol{x}_i^{(2)} \right\|$. Observe that this lower bound is bigger than the previous upper bound when

$$(r_t - 1)(\Gamma - M\epsilon) > r_t(\Gamma - c\kappa)$$
$$M\epsilon < -(\Gamma - M\epsilon)/r_t + c\kappa.$$

Choose large enough $t$ such that $(\Gamma - M\epsilon)/r_t < c\kappa/2$ and $M\epsilon < c\kappa/2$, gives us the desired contradiction. Recall that $R_t := \|p_t\|$ and $r_t := \|v_t\|$. Since $r_t^2 + R_t^2 \leq t$, we have that $(\boldsymbol{v}_t, \boldsymbol{p}_t$ is a solution to Problem 2 with $r = r_t, R = R_t$, and $(\boldsymbol{v}_{(r_t, R_t)}, \boldsymbol{p}_{(r_t, R_t)})$ is a solution to Problem 59.

$\square$

*Proof of Thm. 8.* By Thm. 6, with probability at least $1 - \delta$, the training set is feasible, i.e. exists $(\boldsymbol{v}, \boldsymbol{p})$ such that $\min_{i \in [n]} y_i f(\boldsymbol{X}_i; \boldsymbol{v}, \boldsymbol{p}) > 0$. Therefore, for any $\gamma > 0$, with probability at least $1 - \delta$, we have that $\min_{i \in [n]} y_i f(\boldsymbol{X}_i; \boldsymbol{v}_\gamma, \boldsymbol{p}_\gamma) \geq \gamma$, which proves the first part of the Thm. Next, we show that the classifier $\text{sign}(f(\boldsymbol{X}; \boldsymbol{p}_\gamma, \boldsymbol{v}_\gamma))$ generalizes well, for large enough $\gamma$. Recall the next joint-constrained max margin solution:

$$(v_t, p_t) = \underset{\|\boldsymbol{v}\|^2 + \|\boldsymbol{p}\|^2 \leq t}{\text{argmax}} \min_i y_i f(\boldsymbol{X}_i; \boldsymbol{p}, \boldsymbol{v}), \tag{60}$$

which was introduced in Lemma 43. Fix $\gamma > 0$, and let $(\boldsymbol{v}_\gamma, \boldsymbol{p}_\gamma)$ be the solution of Problem 3. Define $t(\gamma) := \|\boldsymbol{v}_\gamma\|^2 + \|\boldsymbol{p}_\gamma\|^2$. We argue that $(\boldsymbol{v}_\gamma, \boldsymbol{p}_\gamma)$ is a solution to Problem 60 for $t = t(\gamma)$. Indeed, let

$$m := \max_{\|\boldsymbol{v}\|^2 + \|\boldsymbol{p}\|^2 \leq t(\gamma)} \min_{i \in [n]} y_i f(\boldsymbol{X}_i; \boldsymbol{p}, \boldsymbol{v})$$

be the maximum margin for Problem 60 with $t = t(\gamma)$. Assume by contradiction that

$$\min_{i \in [n]} y_i f(\boldsymbol{X}_i; \boldsymbol{p}_\gamma, \boldsymbol{v}_\gamma) < m,$$

which implies that

$$\gamma \leq \min_{i \in [n]} y_i f(\boldsymbol{X}_i; \boldsymbol{p}_\gamma, \boldsymbol{v}_\gamma) < m.$$

Let $(\boldsymbol{v}^*, \boldsymbol{p}^*)$ be a solution to Problem 60 with $t = t(\gamma)$ i.e. $\|\boldsymbol{v}^*\|^2 + \|\boldsymbol{p}^*\|^2 = t(\gamma)$ and $\min_{i \in [n]} y_i f(\boldsymbol{X}_i; \boldsymbol{p}^*, \boldsymbol{v}^*) = m > \gamma$. Write $\boldsymbol{v}' := (\gamma/m) \cdot \boldsymbol{v}^*$. We remind that $f(\boldsymbol{X}; \boldsymbol{p}, \boldsymbol{v}) = \boldsymbol{v}^\top \boldsymbol{X}^\top \mathbb{S}(\boldsymbol{X}\boldsymbol{p})$ and overall we get that

- $\|\boldsymbol{v}'\|^2 + \|\boldsymbol{p}^*\|^2 = (\gamma/m)^2 \|\boldsymbol{v}^*\|^2 + \|\boldsymbol{p}^*\|^2 < \|\boldsymbol{v}^*\|^2 + \|\boldsymbol{p}^*\|^2 = t(\gamma)$

- $\min_{i \in [n]} y_i f(\boldsymbol{X}_i; \boldsymbol{p}^*, \boldsymbol{v}') = \frac{\gamma}{m} \min_{i \in [n]} y_i f(\boldsymbol{X}_i; \boldsymbol{p}^*, \boldsymbol{v}^*) = \frac{\gamma}{m} \cdot m = \gamma,$

which contradicts the optimality of $(\boldsymbol{v}_\gamma, \boldsymbol{p}_\gamma)$ to Problem 3. We conclude that $(\boldsymbol{v}_\gamma, \boldsymbol{p}_\gamma)$ is a solution to Problem 60 for $t = t(\gamma)$, i.e. $(\boldsymbol{v}_\gamma, \boldsymbol{p}_\gamma) = (\boldsymbol{v}_{t(\gamma)}, \boldsymbol{p}_{t(\gamma)})$, where $(\boldsymbol{v}_{t(\gamma)}, \boldsymbol{p}_{t(\gamma)})$ is a solution for Problem 60 with $t = t(\gamma)$. Let $r_{t(\gamma)} := \|\boldsymbol{v}_{t(\gamma)}\|$ and $R_{t(\gamma)} := \|\boldsymbol{p}_{t(\gamma)}\|$. By Lemma 43 we have

$$(\boldsymbol{v}_\gamma, \boldsymbol{p}_\gamma) = (\boldsymbol{v}_{t(\gamma)}, \boldsymbol{p}_{t(\gamma)}) = \left( \boldsymbol{v}_{(r_{t(\gamma)}, R_{t(\gamma)})}, \boldsymbol{p}_{(r_{t(\gamma)}, R_{t(\gamma)})} \right), \tag{61}$$

and that $r_{t(\gamma)} \to \infty, R_{t(\gamma)} \to \infty$ as $t(\gamma) \to \infty$. Clearly $t(\gamma) \to \infty$ as $\gamma \to \infty$. By Thm. 6, The classifier $\text{sign}(f(\boldsymbol{X}; \boldsymbol{p}_R, \boldsymbol{v}_r))$ generalizes well on test data:

$$\mathbb{P}_{(\boldsymbol{X}, y) \sim \mathcal{D}}(y \neq \text{sign}(f(\boldsymbol{X}; \boldsymbol{p}_{(r,R)}, \boldsymbol{v}_{(r,R)})))$$
$$= \eta + \exp(-\Omega(d/n^2)) + \exp\left( -\Theta\left( \frac{(1 - \zeta)}{\sqrt{\frac{\eta n}{d} + \frac{1}{\rho^2}}} - \frac{\log(d)}{R} \right)^2 \right)$$

In particular, there exists $r_0, R_0$ such that for any $r \geq r_0, R \geq R_0$, the above probability can be upper bound by $\eta + \exp(-\Omega(d/n^2)) + \exp(-\Theta((1/\rho^2 + \eta n/d)^{-1}))$ (see Remark 7). Choose large enough $\gamma_0$ such that for any $\gamma \geq \gamma_0$ we have that $r_{t(\gamma)} \geq r_0$ and $R_{t(\gamma)} \geq R_0$. Then we conclude

$$\mathbb{P}_{(\boldsymbol{X},y)\sim\mathcal{D}}\left(y \neq \text{sign}(f(\boldsymbol{X};\boldsymbol{p}_\gamma,\boldsymbol{v}_\gamma))\right)$$
$$= \mathbb{P}_{(\boldsymbol{X},y)\sim\mathcal{D}}\left(y \neq \text{sign}\left(f(\boldsymbol{X};\boldsymbol{p}_{(r_{t(\gamma)},R_{t(\gamma)})},\boldsymbol{v}_{(r_{t(\gamma)},R_{t(\gamma)})})\right)\right)$$
$$\leq \eta + \exp(-\Omega(d/n^2)) + \exp(-\Theta((1/\rho^2 + \eta n/d)^{-1})),$$

where the first equality is from Eq. 61, as required. $\qquad\square$

### A.2.4 PROOF OF THM. 10

**Proof Sketch**

First we prove that in this case, only by selecting the noise token for every sample can we achieve the largest margin in the downstream task,

$$\boldsymbol{r}_i^* = \boldsymbol{\xi}_i, \forall i \in [n] \tag{62}$$

Similarly, we define the respective max-margin solution for $\boldsymbol{p}$ and $\boldsymbol{v}$ in this case.

**Definition 44** (p-SVM, negative case). *$\boldsymbol{p}$ should satisfy*

$$\boldsymbol{p}_{mm}(\alpha) = \underset{\boldsymbol{p}}{\arg\min} \|\boldsymbol{p}\|$$

*subjected to*

$$\boldsymbol{p}^\top(\boldsymbol{\xi}_i - \boldsymbol{\mu}_i) \geq 1, \tag{63}$$

*for all $1 \leq i \leq n$. $\Xi = 1/\|\boldsymbol{p}_{mm}\|$ is the margin induced by $\boldsymbol{p}_{mm}$.*

**Definition 45** (v-SVM, negative case).

$$\boldsymbol{v}(\boldsymbol{p}) = \underset{\boldsymbol{v}\in\mathbb{R}^d}{\arg\min} \|\boldsymbol{v}\| \text{ s.t. } y_i \cdot \boldsymbol{v}^\top\boldsymbol{r}_i \geq 1, \quad \text{for all } i \in [n]. \tag{64}$$

*$\Gamma(\boldsymbol{p}) = 1/\|\boldsymbol{v}(\boldsymbol{p})\|$ is the **label margin** induced by $\boldsymbol{v}$ and $\boldsymbol{p}$. When $\boldsymbol{r}_i = \boldsymbol{\xi}_i, i \in [n]$,*

$$\boldsymbol{v}_{mm} = \underset{\boldsymbol{v}\in\mathbb{R}^d}{\arg\min} \|\boldsymbol{v}\| \text{ s.t. } y_i \cdot \boldsymbol{v}^\top\boldsymbol{\xi}_i \geq 1, \quad \text{for all } i \in [n]. \tag{65}$$

*$\Gamma = 1/\|\boldsymbol{v}_{mm}\|$ is the label margin induced by $\boldsymbol{v}_{mm}$.*

To prove this token selection is optimal, we need to explain that the optimality of the token choice is strict in the sense that mixing other tokens will shrink the label margin. We formalize this into the following proposition:

**Proposition 46** (Optimal Token Condition). *Suppose that Assumption 9 holds, with probability at least $1 - \delta$ on the training dataset, for all $\boldsymbol{p}$, the token selection under $\boldsymbol{p}$ results in a label margin of at most $\Gamma - c \cdot \max_{i\in[n]}(1 - s_{i2})$.*

Then we derive the convergence direction of $\boldsymbol{p}$ and $\boldsymbol{v}$ by Theorem 16. Note that as $\|\boldsymbol{p}\| \to \infty$, the attention is more focused on the noise token for every training sample. Therefore, the output of signal token is upper bounded by a small value.

Consider a test sample $(\boldsymbol{X}, y), \boldsymbol{X} = (\boldsymbol{\mu}', \boldsymbol{\xi}')$. As $\|\boldsymbol{p}\|$ increasing, the noise token $\boldsymbol{\xi}'$ will will dominate the overall output if $\boldsymbol{p}_R^\top\boldsymbol{\xi}' \geq 0$, which indicates the output of attention layer will close to the noise token, $\boldsymbol{r}' \to \boldsymbol{\xi}'$. Meanwhile, we can prove that $\boldsymbol{p}_R$ and $\boldsymbol{v}_r$ are near orthogonal, so $\boldsymbol{p}_R^\top\boldsymbol{\xi}'$ and $\boldsymbol{v}_r^\top\boldsymbol{\xi}'$ are nearly independent variables subjected to Gaussian distribution. Therefore, the probability that $y_i\boldsymbol{v}_r^\top\boldsymbol{\xi}' < 0$ is at least constant order.

**Optimal Token Condition**

First we find the optimal token selection in this case.

**Proposition 46** (Optimal Token Condition). *Suppose that Assumption 9 holds, with probability at least $1 - \delta$ on the training dataset, for all $\boldsymbol{p}$, the token selection under $\boldsymbol{p}$ results in a label margin of at most $\Gamma - c \cdot \max_{i \in [n]}(1 - s_{i2})$.*

*Proof of Proposition 46.* Similar as above, we consider the following three situations:

1. $p \neq 0, k - p = 0$. (All wrong token selections come from clean set)

2. $p = 0, k - p \neq 0$. (All wrong token selections come from noisy set)

3. $p \neq 0, k - p \neq 0$. (Wrong token selections are from both sets)

We will discuss each situation specifically and prove that Proposition 15 holds in every possible case.

**Situation 1:** $p \neq 0, k - p = 0$

First, let's see the condition under the optimal choice of tokens:

**Condition 12** (Original Condition).

$$y_i \boldsymbol{v}^\top \boldsymbol{\xi}_i \geq 1, i \in [n]$$

Similarly, $\boldsymbol{v}_{mm}$ also satisfies the KKT conditions of the max-margin problem (37) in this case, so we could write $\boldsymbol{v}$ as

$$\boldsymbol{v} = \lambda_1 \boldsymbol{\mu}_1 + \lambda_2 \boldsymbol{\mu}_2 + \sum_{i \in [n]} y_i \theta_i \boldsymbol{\xi}_i. \tag{66}$$

Plugging (66) in the condition 12, we can rewrite these conditions as:

$$\theta_i \cdot \|\boldsymbol{\xi}_i\|^2 + \sum_{i' \neq i} y_i y_{i'} \theta_{i'} \langle \boldsymbol{\xi}_i, \boldsymbol{\xi}_{i'} \rangle \geq 1, i \in [n].$$

Then we introduce a lemma to estimate the parameters of optimal solution under this condition:

**Lemma 47** (Balanceing noise factor for KKT point). *Suppose that Assumption 9 holds, under Condition 12, we have*

$$\max_{i \in [n]} \theta_i \leq \frac{1}{(1 - \kappa)d - 2n\sqrt{d \log(6n^2/\delta)}},$$

$$\min_{i \in [n]} \theta_i \geq \frac{(1 - \kappa)d - 4n\sqrt{d \log(6n^2/\delta)}}{(1 + \kappa)d((1 - \kappa)d - 2n\sqrt{d \log(6n^2/\delta)})}.$$

*Proof of Lemma 47.* First we prove the upper bound. Denote $j = \operatorname{argmax}_{i \in [n]} \theta_i$, we have

$$y_j \boldsymbol{v}^\top \boldsymbol{\xi}_j = \sum_{i \in [n]} y_i y_j \theta_i \langle \boldsymbol{\xi}_i, \boldsymbol{\xi}_j \rangle = \theta_j \|\boldsymbol{\xi}_j\|_2^2 + \sum_{i \neq j, i \in [n]} y_i y_j \theta_i \langle \boldsymbol{\xi}_i, \boldsymbol{\xi}_j \rangle$$

$$\geq \theta_j \cdot (1 - \kappa)d - n\theta_j \cdot 2\sqrt{d \log(6n^2/\delta)}$$

The last inequality is because Lemma 57 and the definition of j. Consider the contrary case when $\theta_j > \frac{1}{(1-\kappa)d-2n\sqrt{d \log(6n^2/\delta)}}$, we have

$$y_j \boldsymbol{v}^\top \boldsymbol{\xi}_j > \frac{1}{(1 - \kappa)d - 2n\sqrt{d \log(6n^2/\delta)}} \cdot ((1 - \kappa)d - n \cdot 2\sqrt{d \log(6n^2/\delta)}) = 1.$$

By the KKT conditions, if $y_j \boldsymbol{v}^\top \boldsymbol{\xi}_j > 1$ then we must have $\theta_j = 0$, and thus we reach a contradiction.

Then we prove the lower bound. For $\forall j \in [n]$ we have

$$1 \leq \theta_j \|\boldsymbol{\xi}_j\|_2^2 + \sum_{i \neq j, i \in [n]} y_i y_j \theta_i \langle \boldsymbol{\xi}_i, \boldsymbol{\xi}_j \rangle \leq \theta_j \cdot (1 + \kappa)d + n \max_{i \in [n]} \theta_i \cdot 2\sqrt{d \log(6n^2/\delta)}$$

$$\leq \theta_j \cdot (1 + \kappa)d + \frac{n}{(1 - \kappa)d - 2n\sqrt{d \log(6n^2/\delta)}} \cdot 2\sqrt{d \log(6n^2/\delta)}.$$

The second inequality is due to Lemma 57 and the last inequality is from the upper bound we just get. Therefore, we have

$$\theta_j \geq \frac{(1 - \kappa)d - 4n\sqrt{d \log(6n^2/\delta)}}{(1 + \kappa)d((1 - \kappa)d - 2n\sqrt{d \log(6n^2/\delta)})}$$

This completes the proof.

$\square$

As for the signal parameters $\lambda_1$ and $\lambda_2$, to achieve the minimal norm for $\boldsymbol{v}$, it is obvious that $\lambda_1 = \lambda_2 = 0$. Then we can estimate $\|\boldsymbol{v}_{mm}\|$ in this case:

**Lemma 48** (Norm of $\boldsymbol{v}_{mm}$). *Suppose that Assumption 9 holds, with probability at least $1 - \delta$ on the training dataset, for the solution $\boldsymbol{v}_{mm}$ of* (37) *under the token selection* (62), *we have*

$$\frac{n}{2d} \leq \|\boldsymbol{v}_{mm}\|^2 \leq \frac{5n}{d}.$$

*This implies*

$$\|\boldsymbol{v}_{mm}\| = \Theta\left(\sqrt{\frac{n}{d}}\right).$$

*Proof of Lemma 48.* As $\boldsymbol{v}_{mm}$ is the max-margin solution and satisfies KKT condition, it can be represented as

$$\boldsymbol{v}_{mm} = \lambda_1 \boldsymbol{\mu}_1 + \lambda_2 \boldsymbol{\mu}_2 + \sum_{i \in \mathcal{C}} y_i \theta_i \boldsymbol{\xi}_i + \sum_{i \in [n]} y_i \theta_i \boldsymbol{\xi}_i. \tag{67}$$

As there is no constraint on $\lambda_1, \lambda_2$, both of them can take 0 to achieve max-margin. So we could lower bound $\|\boldsymbol{v}_{mm}\|$ as

$$\|\boldsymbol{v}_{mm}\|^2 \geq \sum_{i \in [n]} \theta_i^2 \|\boldsymbol{\xi}_i\|^2 + \sum_{i \in [n]} \sum_{j \in [n]} y_i y_j \theta_i \theta_j \langle \boldsymbol{\xi}_i, \boldsymbol{\xi}_j \rangle \geq O\left(\frac{n^2}{d^{3/2}}\right) \geq \frac{n}{2d}.$$

The second inequality is from Lemma 47 that $\theta_i = \Theta(1/d)$ for $i \in [n]$ and the last inequality is from Assumption 9.

Then to upper bound $\|\boldsymbol{v}_{mm}\|$, consider the following possible solution $\widetilde{\boldsymbol{v}}$

$$\widetilde{\boldsymbol{v}} = \sum_{i \in [n]} 2y_i \boldsymbol{\xi}_i / d.$$

For $i \in [n]$, we have

$$y_i \widetilde{\boldsymbol{v}}^\top \boldsymbol{r}_i = y_i \widetilde{\boldsymbol{v}}^\top \boldsymbol{\xi}_i = 2\|\boldsymbol{\xi}_i\|^2 / d + \sum_{j \in [n], j \neq i} 2y_i y_j \langle \boldsymbol{\xi}_i, \boldsymbol{\xi}_j \rangle / d$$

$$\geq 2(1 - \kappa) - 2n\sqrt{\log(6n^2/\delta)/d} \geq 1.$$

The first inequality is from Lemma 57 and the second inequality is from Assumption 9. Therefore, $\widetilde{\boldsymbol{v}}$ is a possible solution of SVM problem 26 when $\boldsymbol{p}$ converges to $\boldsymbol{p}_{mm}$. So we have

$$\|\boldsymbol{v}_{mm}\|^2 \leq \|\widetilde{\boldsymbol{v}}\|^2 = \sum_{i \in [n]} 4\|\boldsymbol{\xi}_i\|^2 / d^2 + \sum_{i \in [n]} \sum_{j \in [n]} 4y_i y_j \langle \boldsymbol{\xi}_i, \boldsymbol{\xi}_j \rangle / d^2 \leq \frac{5n}{d}.$$

The last inequality is from Lemma 57, Lemma 59 and Assumption 9. Combine the results above, we have $\|\boldsymbol{v}_{mm}\|^2 = \Theta(\frac{n}{d})$.

$\square$

Denote the mixed samples as $k_1, k_2, ..., k_p$. And for every mixed sample $k_i$, we have $\boldsymbol{r}_{k_i} = (1 - \beta_i)\boldsymbol{\mu}_{k_i} + \beta_i\boldsymbol{\xi}_{k_i}$. Without losing generality, we assume that $y_{k_i} = +1$ for all $i \in [p]$. Then the conditions under *Situation 1* become

**Condition 13** ($p$ clean samples violating optimal token selection).

$$\begin{cases} y_i\boldsymbol{v}^\top\boldsymbol{\xi}_i \geq 1, i \in [n]\backslash[p] \\ \boldsymbol{v}^\top\boldsymbol{r}_{k_i} \geq 1, i \in [p] \end{cases}$$

Denote the max-margin solution under this condition as $\boldsymbol{v}'$ with parameters $\lambda'_1, \lambda'_2, \theta'_i$. Plugging this representation into the condition 13, we have:

$$\begin{cases} \theta'_i \cdot \|\boldsymbol{\xi}_{i'}\|^2 + \sum_{i' \neq i} y_iy_{i'}\theta'_{i'}\langle\boldsymbol{\xi}_i, \boldsymbol{\xi}_{i'}\rangle \geq 1, i \in [n]\backslash[p] \\ (1 - \beta_i)\lambda'_1 \cdot \|\boldsymbol{\mu}_1\|^2 + \beta_i(\theta'_{k_i} \cdot \|\boldsymbol{\xi}_{k_i}\|^2 + \sum_{i' \neq k_i} y_{i'}\theta'_{i'}\langle\boldsymbol{\xi}_{k_i}, \boldsymbol{\xi}_{i'}\rangle) \geq 1, i \in [p] \end{cases}$$

We consider two cases: $\lambda'_1\|\boldsymbol{\mu}_1\|^2 < 1$ and $\lambda'_1\|\boldsymbol{\mu}_1\|^2 \geq 1$. First when $\lambda'_1\|\boldsymbol{\mu}_1\|^2 < 1$, the condition for mixed clean sample becomes:

$$\theta'_{k_i} \cdot \|\boldsymbol{\xi}_{k_i}\|^2 + \sum_{i' \neq k_i} y_{i'}\theta'_{i'}\langle\boldsymbol{\xi}_{k_i}, \boldsymbol{\xi}_{i'}\rangle \geq \frac{1 - (1 - \beta_i)\lambda'_1\|\boldsymbol{\mu}_1\|^2}{\beta_i} > 1,$$

which indicates that the condition for $\theta'_{k_i}$ is strengthened. So mixing 1 more clean sample is equal to strengthening 1 constraint in the original setting. Therefore, mixing p samples will not result in a better solution than only mixing 1 clean sample. Then we can simplify this case to mixing only 1 clean sample and denote this sample as $k_*$, $r_{k_*} = (1 - \beta)\boldsymbol{\mu}_1 + \beta\boldsymbol{\xi}_{k_*}$. Now the condition becomes:

**Condition 14** (1 clean sample violating optimal token selection).

$$\begin{cases} \theta'_i \cdot \|\boldsymbol{\xi}_{i'}\|^2 + \sum_{i' \neq i} y_iy_{i'}\theta'_{i'}\langle\boldsymbol{\xi}_i, \boldsymbol{\xi}_{i'}\rangle \geq 1, i \in [n]\backslash\{k_*\} \\ (1 - \beta)\lambda'_1 \cdot \|\boldsymbol{\mu}_1\|^2 + \beta(\theta'_{k_*} \cdot \|\boldsymbol{\xi}_{k_i}\|^2 + \sum_{i' \neq k_*} y_{i'}\theta'_{i'}\langle\boldsymbol{\xi}_{k_*}, \boldsymbol{\xi}_{i'}\rangle) \geq 1 \end{cases}$$

Similarly, we introduce the following lemma which estimates the parameters in $\boldsymbol{v}'$. We define

$$\alpha = \frac{1 - (1 - \beta)\lambda'_1\|\boldsymbol{\mu}_1\|^2}{\beta}$$

for the convenience of the following proof.

**Lemma 49.** *Suppose that Assumption 9 holds, under condition 14, with probability at least $1 - \delta$ on the training dataset, we have*

$$\theta'_{k_*} \leq \frac{\alpha}{(1 - \kappa)d - 2n\sqrt{d\log(6n^2/\delta)}},$$

$$\theta'_{k_*} \geq \frac{\alpha}{(1 + \kappa)d}\left(1 - \frac{2n\sqrt{d\log(6n^2/\delta)}}{(1 - \kappa)d - 2n\sqrt{d\log(6n^2/\delta)}}\right),$$

$$\max_{i \in [n]\backslash\{k_*\}} \theta'_i \leq \frac{(1 - \kappa)d + 2(\alpha - n)\sqrt{d\log(6n^2/\delta)}}{((1 - \kappa)d - 2n\sqrt{d\log(6n^2/\delta)})^2},$$

$$\min_{i \in [n]\backslash\{k_*\}} \theta'_i \geq \frac{1}{(1 + \kappa)d} \cdot \left(1 - \frac{2n\alpha\sqrt{d\log(6n^2/\delta)}}{(1 - \kappa)d - 2n\sqrt{d\log(6n^2/\delta)}}\right).$$

*Proof of Lemma 49.* Denote $j = \underset{i \in [n]}{\operatorname{argmax}}\,\theta'_i$, we have

$$y_j\boldsymbol{v}'^\top\boldsymbol{\xi}_j = \theta'_j\|\boldsymbol{\xi}_j\|^2 + \sum_{i \in [n], i \neq j} y_iy_j\theta'_i\langle\boldsymbol{\xi}_i, \boldsymbol{\xi}_j\rangle$$

$$\geq \theta'_j(1 - \kappa)d - n\max_{i \in [n]} \theta'_i \cdot 2\sqrt{d\log(6n^2/\delta)}$$

$$= \theta'_j((1 - \kappa)d - n \cdot 2\sqrt{d\log(6n^2/\delta)}).$$

The first inequality is due to Lemma 57 and the last equation is from our definition of j. Consider the contrary case when $\theta'_j > \frac{\alpha}{(1-\kappa)d - 2n\sqrt{d\log(6n^2/\delta)}}$, we have

$$y_j \boldsymbol{v}'^\top \boldsymbol{\xi}_j > \alpha.$$

By the KKT conditions, if $y_j \boldsymbol{v}'^\top \boldsymbol{\xi}_j > \frac{1 + \lambda'_1(1-\beta)\|\boldsymbol{\mu}_1\|^2}{\beta}$ then we must have $\theta'_j = 0$, and thus we reach a contradiction. Therefore, $\theta'_{k_\star} \leq \theta'_j \leq \frac{\alpha}{(1-\kappa)d - 2n\sqrt{d\log(6n^2/\delta)}}$. Then denote $j' = \operatorname*{argmax}_{i \in [n], i \neq k_\star} \theta''_i$, we have

$$
\begin{aligned}
y_{j'} \boldsymbol{v}'^\top \boldsymbol{\xi}_{j'} &= \theta'_{j'} \|\boldsymbol{\xi}_{j'}\|^2 + \sum_{i \in [n], i \neq j'} y_i y_{j'} \theta'_i \langle \boldsymbol{\xi}_i, \boldsymbol{\xi}_{j'} \rangle \\
&\geq \theta'_{j'}(1-\kappa)d - n \max_{i \in [n], i \neq j'} \theta'_i \cdot 2\sqrt{d\log(6n^2/\delta)} - \theta'_{k_\star} \sqrt{d\log(6n^2/\delta)} \\
&\geq \theta'_{j'}((1-\kappa)d - n \cdot 2\sqrt{d\log(6n^2/\delta)}) - \frac{2\alpha\sqrt{d\log(6n^2/\delta)}}{(1-\kappa)d - 2n\sqrt{d\log(6n^2/\delta)}}.
\end{aligned}
$$

The first inequality is from Lemma 57 and the second inequality is from the upper bound of $\theta'_{k_\star}$ we just get. Consider the case when $\theta'_{j'} > \frac{(1-\kappa)d + 2(\alpha-n)\sqrt{d\log(6n^2/\delta)}}{((1-\kappa)d - 2n\sqrt{d\log(6n^2/\delta)})^2}$, we have

$$y_{j'} \boldsymbol{v}'^\top \boldsymbol{\xi}_{j'} > 1.$$

By the complementary slackness condition, if $y_{j'} \boldsymbol{v}''^\top \boldsymbol{\xi}_{j'} > 1$ then we must have $\theta'_{j'} = 0$, and thus we reach a contradiction.

Next we estimate the lower bound of $\theta'_j$ when $j \neq k_*$. We have

$$
\begin{aligned}
1 &\leq y_j \boldsymbol{v}'^\top \boldsymbol{\xi}_j \\
&= \theta'_j \|\boldsymbol{\xi}_j\|^2 + \sum_{i \in [n], i \neq j} y_i y_j \theta'_i \langle \boldsymbol{\xi}_i, \boldsymbol{\xi}_j \rangle \\
&\leq \theta'_j(1+\kappa)d + n \max_{i \in [n]} \theta'_i \cdot 2\sqrt{d\log(6n^2/\delta)} \\
&\leq \theta'_j(1+\kappa)d + \frac{\alpha}{(1-\kappa)d - 2n\sqrt{d\log(6n^2/\delta)}} \cdot 2n\sqrt{d\log(6n^2/\delta)}
\end{aligned}
$$

The last inequality is from the upper bound of $\theta'_{k_*}$ we just get. Therefore, we have

$$\theta'_j \geq \frac{1}{(1+\kappa)d} \cdot \left(1 - \frac{2n\alpha\sqrt{d\log(6n^2/\delta)}}{(1-\kappa)d - 2n\sqrt{d\log(6n^2/\delta)}}\right)$$

for all $j \in [n]$ and $j \neq k_*$.

Last we lower bound $\theta'_{k_*}$. We have

$$
\begin{aligned}
\alpha &\leq y_k \boldsymbol{v}''^\top \boldsymbol{\xi}_{k_*} \\
&= \theta'_{k_*}(1+\kappa)d + n \max_{i \in [n]} \theta'_i \cdot 2\sqrt{d\log(6n^2/\delta)}
\end{aligned}
$$

Similarly, we have

$$\theta'_{k_*} \geq \frac{\alpha}{(1+\kappa)d}\left(1 - \frac{2n\sqrt{d\log(6n^2/\delta)}}{(1-\kappa)d - 2n\sqrt{d\log(6n^2/\delta)}}\right).$$

$\square$

Therefore, we could estimate the difference between $\|\boldsymbol{v}'\|^2$ and $\|\boldsymbol{v}_{mm}\|^2$.

**Lemma 50.** *Suppose that Assumption 9 holds, with probability at least $1 - \delta$ on the training dataset, denote $\boldsymbol{v}$ and $\boldsymbol{v}'$ as the optimal solutions under condition 12 and condition 14 respectively. We have*

$$\|\boldsymbol{v}'\|_2^2 - \|\boldsymbol{v}_{mm}\|_2^2 \geq \frac{C_1(1-\beta)}{d}.$$

*where $C_1 = \Theta(1)$ is a constant.*

*Proof of Lemma 50.* From the first inequality in Condition 14, for $i[n], i \neq k_\star$ we have

$$\theta_i' \cdot \|\boldsymbol{\xi}_i\|^2 + \sum_{i' \neq i, k_\star} y_i y_{i'} \theta_{i'}' \langle \boldsymbol{\xi}_i, \boldsymbol{\xi}_{i'} \rangle \geq 1 - y_i y_{k_\star} \theta_{k_\star}' \langle \boldsymbol{\xi}_i, \boldsymbol{\xi}_{k_\star} \rangle.$$

Then we add $y_i y_{k_\star} w \langle \boldsymbol{\xi}_i, \boldsymbol{\xi}_{k_\star} \rangle$ on both sides, where we set $w = \theta_{k_\star}' - \frac{\alpha-1}{(1+\kappa)d - 2\sqrt{d\log(6n^2/\delta)}} \leq \theta_{k^\star}'$. Then we have

$$\theta_i' \cdot \|\boldsymbol{\xi}_{i'}\|^2 + \sum_{i' \neq i, k^\star} y_i y_{i'} \theta_{i'}' \langle \boldsymbol{\xi}_i, \boldsymbol{\xi}_{i'} \rangle + y_i y_{k_\star} w \langle \boldsymbol{\xi}_i, \boldsymbol{\xi}_{k_\star} \rangle \geq 1 - y_i y_{k_\star} (\theta_{k^\star}' - w) \langle \boldsymbol{\xi}_i, \boldsymbol{\xi}_{k_\star} \rangle$$

$$\geq 1 - 2(\theta_{k_\star}' - w)\sqrt{d\log(6n^2/\delta)}$$

$$= \frac{(1+\kappa)d - 2\alpha\sqrt{d\log(6n^2/\delta)}}{(1+\kappa)d - 2\sqrt{d\log(6n^2/\delta)}}. \quad (68)$$

The second inequality is from Lemma 57. Now consider a new $\underline{\boldsymbol{v}} = \underline{\lambda}_1 \boldsymbol{\mu}_1 + \underline{\lambda}_2 \boldsymbol{\mu}_2 + \sum_{i \in [n]} y_i \underline{\theta}_i \boldsymbol{\xi}_i$ with

$$\underline{\lambda}_1 = \lambda_1'; \quad \underline{\lambda}_2 = \lambda_2';$$

$$\underline{\theta}_i = \theta_i' / (1 - 2(\theta_{k_\star}' - w)\sqrt{d\log(6n^2/\delta)}) \text{ for } i \in [n], i \neq k_\star$$

and

$$\underline{\theta}_{k_\star} = \frac{w}{1 - 2(\theta_{k_\star}' - w)\sqrt{d\log(6n^2/\delta)}}.$$

We can prove that $\underline{\boldsymbol{v}}$ satisfies all constraints for $\boldsymbol{v}_{mm}$.

By dividing $1 - 2(\theta_{k_\star}' - w)\sqrt{d\log(6n^2/\delta)}$ on both sides of (68), for $\forall i \in [n], i \neq k_\star$ we have

$$\underline{\theta}_i \cdot \|\boldsymbol{\xi}_i\|^2 + \sum_{i' \neq i} y_i y_{i'} \underline{\theta}_i \langle \boldsymbol{\xi}_i, \boldsymbol{\xi}_{i'} \rangle \geq 1.$$

Then we prove that $\underline{\theta}_{k^\star} \|\boldsymbol{\xi}_{k^\star}\|^2 + \sum_{i \neq k^\star} y_i y_{k_\star} \underline{\theta}_i \langle \boldsymbol{\xi}_i, \boldsymbol{\xi}_{k_\star} \rangle \geq 1$. From the last inequality in Condition 14 we have

$$\theta_{k_\star}' \cdot \|\boldsymbol{\xi}_{k_\star}\|^2 + \sum_{i \neq k_\star} y_{k_\star} y_i \theta_i' \langle \boldsymbol{\xi}_i, \boldsymbol{\xi}_{k_\star} \rangle \geq \alpha.$$

Dividing $1 - 2(\theta_{k_\star}' - w)\sqrt{d\log(6n^2/\delta)}$ on both sides, we get

$$\frac{\theta_{k_\star}' \|\boldsymbol{\xi}_{k_\star}\|^2}{1 - 2(\theta_{k_\star}' - w)\sqrt{d\log(6n^2/\delta)}} + \sum_{i \neq k_\star} y_i y_{k_\star} \underline{\theta}_i \langle \boldsymbol{\xi}_i, \boldsymbol{\xi}_{k_\star} \rangle \geq \frac{\alpha}{1 - 2(\theta_{k_\star}' - w)\sqrt{d\log(6n^2/\delta)}}.$$

Therefore we have

$$\underline{\theta}_{k_\star} \|\boldsymbol{\xi}_{k_\star}\|^2 + \sum_{i \neq k_\star} y_i y_{k_\star} \underline{\theta}_i \langle \boldsymbol{\xi}_i, \boldsymbol{\xi}_{k_\star} \rangle \geq \frac{\alpha - (\theta_{k_\star}' - w)\|\boldsymbol{\xi}_{k_\star}\|^2}{1 - 2(\theta_{k_\star}' - w)\sqrt{d\log(6n^2/\delta)}} \geq \frac{\alpha - (\theta_{k_\star}' - w)(1+\kappa)d}{1 - 2(\theta_{k_\star}' - w)\sqrt{d\log(6n^2/\delta)}} = 1.$$

The second inequality is from Lemma 57 and the last equality is by our definition $\theta_{k_\star}' - w = \frac{\alpha-1}{(1+\kappa)d - 2\sqrt{d\log(6n^2/\delta)}}$. Thus, $\underline{\boldsymbol{v}}$ is a possible solution under Condition 1 and $\|\underline{\boldsymbol{v}}\| \geq \|\boldsymbol{v}_{mm}\|$.

Next we estimate the difference between $\|\boldsymbol{v}'\|^2$ and $\|\underline{\boldsymbol{v}}\|^2$. The expansion of $\|\boldsymbol{v}'\|^2$ and $\|\underline{\boldsymbol{v}}\|^2$ are:

$$\|\boldsymbol{v}'\|^2 = \lambda_1'^2 \|\boldsymbol{\mu}_1\|^2 + \lambda_2'^2 \|\boldsymbol{\mu}_2\|^2 + \sum_{i \in [n]} \theta_i'^2 \|\boldsymbol{\xi}_i\|^2 + \sum_{i \in [n]} \sum_{j \in [n]} y_i y_j \theta_i' \theta_j' \langle \boldsymbol{\xi}_i, \boldsymbol{\xi}_j \rangle,$$

$$\|\underline{\boldsymbol{v}}\|^2 = \underline{\lambda}_1^2 \|\boldsymbol{\mu}_1\|^2 + \underline{\lambda}_2^2 \|\boldsymbol{\mu}_2\|^2 + \sum_{i \in [n]} \underline{\theta}_i^2 \|\boldsymbol{\xi}_i\|^2 + \sum_{i \in [n]} \sum_{j \in [n]} y_i y_j \underline{\theta}_i \underline{\theta}_j \langle \boldsymbol{\xi}_i, \boldsymbol{\xi}_j \rangle.$$

Similar to the condition (44), we have $\|\boldsymbol{v}'\| \leq 2\|\boldsymbol{v}_{mm}\| = \Theta(\sqrt{n/d})$, which implies that $\alpha = O(\sqrt{n}\log n)$. Otherwise, we have

$$\theta_{k_\star}' \|\boldsymbol{\xi}_{k_\star}\|^2 \geq \alpha - \sum_{i \neq k_\star} y_{k_\star} y_i \theta_i' \langle \boldsymbol{\xi}_i, \boldsymbol{\xi}_{k_\star} \rangle = \Omega(\alpha).$$

It further yields that

$$\|\boldsymbol{v}'\|^2 = \Omega(\frac{n}{d}) + \theta_{k_\star}'^2 \|\boldsymbol{\xi}_{k_\star}\|^2 = \Omega(\frac{n}{d} + \frac{\alpha^2}{d}) = \Omega(\frac{n \log^2 n}{d}),$$

which contradicts with $\|\boldsymbol{v}'\| = \Theta(\sqrt{n/d})$.

We decompose the difference between $\|\boldsymbol{v}'\|^2$ and $\|\underline{\boldsymbol{v}}\|^2$ into four terms:

$$\|\boldsymbol{v}'\|^2 - \|\underline{\boldsymbol{v}}\|^2 = \underbrace{(\theta_{k_\star}'^2 - \underline{\theta}_{k_\star}^2)\|\boldsymbol{\xi}_{k_\star}\|^2}_{I_1} + \underbrace{\sum_{i\in[n],i\neq k_\star} (\theta_i'^2 - \underline{\theta}_i^2)\|\boldsymbol{\xi}_i\|^2}_{I_2} - \underbrace{\sum_{i\in[n]}\sum_{j\in[n]} y_i y_j \underline{\theta}_i \underline{\theta}_j \langle \boldsymbol{\xi}_i, \boldsymbol{\xi}_j \rangle}_{I_3}$$

$$+ \underbrace{\sum_{i\in[n]}\sum_{j\in[n]} y_i y_j \theta_i' \theta_j' \langle \boldsymbol{\xi}_i, \boldsymbol{\xi}_j \rangle}_{I_4} .$$

We now estimate $I_1$ to $I_4$ sequentially. For the first term,

$$I_1 \geq (\theta_{k_\star}'^2 - \underline{\theta}_{k_\star}^2)(1-\kappa)d = (\theta_{k_\star}' - \underline{\theta}_{k_\star})(\theta_{k_\star}' + \underline{\theta}_{k_\star})(1-\kappa)d$$

$$= \frac{(\alpha-1)(1 - 2\theta_{k_\star}'\sqrt{d\log(6n^2/\delta)})}{(1+\kappa)d - 2\sqrt{d\log(6n^2/\delta)}} \cdot \Omega\left(\frac{1}{d}\right) \cdot (1-\kappa)d$$

$$= \Omega\left(\frac{\alpha-1}{d}\right),$$

where the first inequality is from Lemma 57; the second equality is from Lemma 49; and the last equality uses the fact that $\alpha = O(\sqrt{n}\log n)$. Then we can further upper bound $\max_{i\in[n],i\neq k_\star} \theta_i'$ as

$$\max_{i\in[n],i\neq k_\star} \theta_i' \leq \frac{(1-\kappa)d + 2(\alpha-n)\sqrt{d\log(6n^2/\delta)}}{((1-\kappa)d - 2n\sqrt{d\log(6n^2/\delta)})^2} = O(\frac{1}{d}). \tag{69}$$

For the second term $I_2$, we have

$$|I_2| \leq \sum_{i\in[n],i\neq k_\star} (\underline{\theta}_i^2 - \theta_i'^2)(1+\kappa)d$$

$$\leq \left(\frac{1}{(1-(\theta_{k_\star}' - w)\sqrt{d\log(6n^2/\delta)})^2} - 1\right) \max_{i\in[n],i\neq k_\star} \theta_i'^2 \cdot n(1+\kappa)d$$

$$= \frac{(\alpha-1)\sqrt{d\log(6n^2/\delta)}}{(1+\kappa)d - \sqrt{d\log(6n^2/\delta)}} \cdot O(\frac{n}{d}) = \tilde{O}\left(\frac{(\alpha-1)n}{d^{3/2}}\right).$$

The second inequality is from Lemma 49. The first equality is from (69) and the last equality is from Assumption 9.

Then we bound $|-I_3 + I_4|$ as:

$$|-I_3 + I_4| \leq \sum_{i\in[n]}\sum_{j\in[n]\setminus\{i\}} |\underline{\theta}_i \underline{\theta}_j - \theta_i' \theta_j'| \cdot |\langle \boldsymbol{\xi}_i, \boldsymbol{\xi}_j \rangle|$$

$$\leq \sum_{i\in[n]\setminus\{k_\star\}}\sum_{j\in[n]\setminus\{k_\star,i\}} |\underline{\theta}_i \underline{\theta}_j - \theta_i' \theta_j'| \cdot |\langle \boldsymbol{\xi}_i, \boldsymbol{\xi}_j \rangle| + 2\sum_{t\in[n]\setminus\{k_\star\}} |\underline{\theta}_{k_\star} \underline{\theta}_t - \theta_{k_\star}' \theta_t'| \cdot |\langle \boldsymbol{\xi}_{k_\star}, \boldsymbol{\xi}_t \rangle|$$

$$\leq n^2 \left(\frac{1}{(1-(\theta_{k_\star}' - w)\sqrt{d\log(6n^2/\delta)})^2} - 1\right) \max_{i\in[n],i\neq k_\star} \theta_i'^2 \cdot 2\sqrt{d\log(6n^2/\delta)}$$

$$+ n\left(\theta_{k_\star}' - \frac{\underline{\theta}_{k_\star}}{1 - 2(\theta_{k_\star}' - w)\sqrt{d\log(6n^2/\delta)}}\right) \max_{i\in[n],i\neq k_\star} \theta_i' 4\sqrt{d\log(6n^2/\delta)}$$

$$\leq \frac{(\alpha-1)\sqrt{d\log(6n^2/\delta)}}{(1+\kappa)d - \sqrt{d\log(6n^2/\delta)}} \cdot O(\frac{n^2(1+\kappa)}{d^{3/2}}) + \frac{\alpha-1}{d} \cdot O(\frac{n}{d}) \cdot 2\sqrt{d\log(6n^2/\delta)}$$

$$= O\left(\frac{(\alpha-1)n^2}{d^2} + \frac{(\alpha-1)n}{d^{3/2}}\right).$$

The third inequality is from Lemma 47 and Lemma 49; The fourth inequality is from the fact that

$$\theta'_{k_\star} - \frac{\underline{\theta}_{k_\star}}{1 - 2(\theta'_{k_\star} - w)\sqrt{d\log(6n^2/\delta)}} = \frac{\theta'_{k_\star} - \underline{\theta}_{k_\star} - 2\theta'_{k_\star}(\theta'_{k_\star} - w)\sqrt{d\log(6n^2/\delta)}}{1 - 2(\theta'_{k_\star} - w)\sqrt{d\log(6n^2/\delta)}}$$

$$= \frac{\Omega(\frac{\alpha-1}{d}) - O(\frac{\alpha(\alpha-1)}{d^{3/2}})}{1 - 2(\theta'_{k_\star} - w)\sqrt{d\log(6n^2/\delta)}} > 0$$

So we have $\theta'_{k_\star} - \frac{\underline{\theta}_{k_\star}}{1-2(\theta'_{k_\star}-w)\sqrt{d\log(6n^2/\delta)}} \leq \theta'_{k_\star} - \underline{\theta}_{k_\star}$; The last equality is from Assumption 5.

Combining the above results, we have

$$\|\boldsymbol{v}'\|_2^2 - \|\boldsymbol{v}_{mm}\|_2^2 \geq \Theta\left(\frac{\alpha-1}{d}\right) + O\left(\frac{(\alpha-1)\eta n}{d^{3/2}}\right) \geq \frac{C_1(1-\beta)}{d}.$$

Here $C_1 = \Theta(1)$ is a constant. $\qquad\qquad\square$

Then we consider the case when $\lambda'_1\|\boldsymbol{\mu}_1\|^2 \geq 1$. In this case, the condition for mixed clean sample becomes:

$$\theta'_{k_i} \cdot \|\boldsymbol{\xi}_{k_i}\|^2 + \sum_{i' \neq k_i} y_{ki}y_{i'}\theta'_{i'}\langle\boldsymbol{\xi}_{k_i}, \boldsymbol{\xi}_{i'}\rangle \geq \frac{1 - (1-\beta_i)\lambda'_1\|\boldsymbol{\mu}_1\|^2}{\beta_i},$$

and $\frac{1-(1-\beta_i)\lambda'_1\|\boldsymbol{\mu}_1\|^2}{\beta_i} \leq 1$, which indicates that the condition for $\theta'_{k_i}$ is relaxed. So mixing 1 more clean sample is equal to relaxing 1 constraint in the original setting. Therefore, mixing all clean samples will achieve the best result. From the data generalization model, there are $(1-\eta)n/2 + o(n)$ clean samples with label $+1$ and denote $S_{+1}$ as their set. Now the condition becomes:

**Condition 15** (All clean samples violating optimal token selection).

$$\begin{cases} \theta'_i \cdot \|\boldsymbol{\xi}_{i'}\|^2 + \sum_{i' \neq i} y_iy_{i'}\theta'_{i'}\langle\boldsymbol{\xi}_i, \boldsymbol{\xi}_{i'}\rangle) \geq 1, i \in [n]\setminus S_{+1} \\ (1-\beta)\lambda'_1 \cdot \|\boldsymbol{\mu}_1\|^2 + \beta(\theta'_i \cdot \|\boldsymbol{\xi}_i\|^2 + \sum_{i' \neq i} y_iy_{i'}\theta'_{i'}\langle\boldsymbol{\xi}_i, \boldsymbol{\xi}_{i'}\rangle) \geq 1, i \in S_{+1} \end{cases}$$

We have another lemma to estimate the scale of parameters in the max-margin solution in this case. Here $\alpha = \frac{1-(1-\widetilde{\beta})\lambda'_1\|\boldsymbol{\mu}_1\|^2}{\widetilde{\beta}}$ and $\widetilde{\beta} = \min_{i\in[n]}\{\beta_i\}$.

**Lemma 51.** *Suppose that Assumption 9 holds, under Condition 15, we have*

$$\max_{i\in[n]} \theta'_i \leq \frac{1}{(1-\kappa)d - 2n\sqrt{d\log(6n^2/\delta)}},$$

$$\min_{i\in[n]} \theta'_i \geq \frac{(1-\kappa)d\alpha - 2n\sqrt{d\log(6n^2/\delta)}(\alpha+1)}{(1+\kappa)d((1-\kappa)d - 2n\sqrt{d\log(6n^2/\delta)})}.$$

*Proof of Lemma 51.* First we prove the upper bound. Denote $j = \underset{i\in[n]}{\operatorname{argmax}}\, \theta_i$, we have

$$y_j\boldsymbol{v}^\top\boldsymbol{\xi}_j = \sum_{i\in[n]} y_iy_j\theta_i\langle\boldsymbol{\xi}_i, \boldsymbol{\xi}_j\rangle$$

$$= \theta_j\|\boldsymbol{\xi}_j\|_2^2 + \sum_{i\neq j, i\in[n]} y_iy_j\theta_i\langle\boldsymbol{\xi}_i, \boldsymbol{\xi}_j\rangle$$

$$\geq \theta_j \cdot (1-\kappa)d - n\theta_j \cdot 2\sqrt{d\log(6n^2/\delta)}$$

The last inequality is because Lemma 57 and the definition of j. Consider the contrary case when $\theta_j > \frac{1}{(1-\kappa)d-2n\sqrt{d\log(6n^2/\delta)}}$, we have

$$y_j\boldsymbol{v}^\top\boldsymbol{\xi}_j > \frac{1}{(1-\kappa)d - 2n\sqrt{d\log(6n^2/\delta)}} \cdot ((1-\kappa)d - n\cdot 2\sqrt{d\log(6n^2/\delta)}) = 1.$$

By the KKT conditions, if $y_j \boldsymbol{v}^\top \boldsymbol{\xi}_j > 1$ then we must have $\theta_j = 0$, and thus we reach a contradiction.

Then we prove the lower bound. For $\forall j \in S_{+1}$ we have

$$\alpha \leq \theta_j \|\boldsymbol{\xi}_j\|_2^2 + \sum_{i \neq j, i \in [n]} y_i y_j \theta_i \langle \boldsymbol{\xi}_i, \boldsymbol{\xi}_j \rangle$$

$$\leq \theta_j \cdot (1 + \kappa)d + n \max_{i \in [n]} \theta_i \cdot 2\sqrt{d \log(6n^2/\delta)}$$

$$\leq \theta_j \cdot (1 + \kappa)d + \frac{n}{(1 - \kappa)d - 2n\sqrt{d \log(6n^2/\delta)}} \cdot 2\sqrt{d \log(6n^2/\delta)}.$$

The second inequality is due to Lemma 57 and the last inequality is from the upper bound we just get. Therefore, we have

$$\theta_j \geq \frac{(1 - \kappa)d\alpha - 2n\sqrt{d \log(6n^2/\delta)}(\alpha + 1)}{(1 + \kappa)d((1 - \kappa)d - 2n\sqrt{d \log(6n^2/\delta)})}.$$

This completes the proof

$\square$

Then we can estimate the difference between $\|\boldsymbol{v}'\|^2$ and $\|\boldsymbol{v}_{mm}\|^2$ with the following lemma:

**Lemma 52.** *Suppose that Assumption 9 holds, denote $\boldsymbol{v}$ and $\boldsymbol{v}'$ as the optimal solutions under condition 12 and condition 15 respectively. We have*

$$\|\boldsymbol{v}'\|_2^2 - \|\boldsymbol{v}_{mm}\|_2^2 \geq \frac{C_2(1 - \beta)}{\rho^2}.$$

*where $C_2 = \Theta(1)$ is a constant.*

*Proof of Lemma 52.* Recall the expansion of $\|\boldsymbol{v}_{mm}\|^2$ and $\|\boldsymbol{v}'\|^2$:

$$\|\boldsymbol{v}_{mm}\|^2 = \sum_{i \in [n]} \theta_i^2 \|\boldsymbol{\xi}_i\|^2 + \sum_{i \in [n]} \sum_{j \in [n]} y_i y_j \theta_i \theta_j \langle \boldsymbol{\xi}_i, \boldsymbol{\xi}_j \rangle,$$

$$\|\boldsymbol{v}'\|^2 = \lambda_1'^2 \|\boldsymbol{\mu}_1\|^2 + \sum_{i \in [n]} \theta_i'^2 \|\boldsymbol{\xi}_i\|^2 + \sum_{i \in [n]} \sum_{j \in [n]} y_i y_j \theta_i' \theta_j' \langle \boldsymbol{\xi}_i, \boldsymbol{\xi}_j \rangle.$$

Then we have

$$\|\boldsymbol{v}'\|^2 - \|\boldsymbol{v}_{mm}\|^2 = \underbrace{\lambda_1'^2 \|\boldsymbol{\mu}_1\|^2}_{I_1} + \underbrace{\sum_{i \in [n]} (\theta_i'^2 - \theta_i^2) \|\boldsymbol{\xi}_i\|^2}_{I_2} - \underbrace{\sum_{i \in [n]} \sum_{j \in [n]} y_i y_j \theta_i \theta_j \langle \boldsymbol{\xi}_i, \boldsymbol{\xi}_j \rangle}_{I_3}$$

$$+ \underbrace{\sum_{i \in [n]} \sum_{j \in [n]} y_i y_j \theta_i'' \theta_j'' \langle \boldsymbol{\xi}_i, \boldsymbol{\xi}_j \rangle}_{I_4}.$$

We now estimate $I_1$ to $I_4$ sequentially. Here we use the same notation $\alpha = \frac{1 - (1 - \widetilde{\beta})\lambda_1' \|\boldsymbol{\mu}_1\|^2}{\widetilde{\beta}}$ and $\widetilde{\beta} = \min_{i \in [n]} \{\beta_i\}$ as in Lemma 51. First from our assumption $\lambda_1' \|\boldsymbol{\mu}_1\|^2 \geq 1$ we have

$$I_1 = \lambda_1'^2 \|\boldsymbol{\mu}_1\|^2 \geq 1/\rho^2.$$

Then for $I_2$, we have

$$|I_2| \leq n(\max_{i \in [n]} \theta_i^2 - \min_{i \in [n]} \theta_i'^2) \cdot (1 + \kappa)d$$

$$\leq \left( \frac{1}{((1 - \kappa)d - 2n\sqrt{d \log(6n^2/\delta)})^2} - \frac{1}{(1 + \kappa)^2 d^2} \cdot \left( \alpha - \frac{2n\sqrt{d \log(6n^2/\delta)}}{(1 - \kappa)d - 2n\sqrt{d \log(6n^2/\delta)}} \right)^2 \right) \cdot (1 + \kappa)dn$$

$$= d(1 + \kappa)n \cdot \frac{1 - \frac{1}{(1 + \kappa)^2 d^2}((1 - \kappa)d\alpha - 2(\alpha + 1)n\sqrt{d \log(6n^2/\delta)})^2}{((1 - \kappa)d - 2n\sqrt{d \log(6n^2/\delta)})^2}$$

$$= O\left( \frac{n}{d} \right).$$

The second inequality is from Lemma 47 and Lemma 51.

Then we bound $|-I_3 + I_4|$ as:

$$|-I_3 + I_4| \leq \sum_{i \in [n]} \sum_{j \in [n] \setminus \{i\}} (\theta_i' \theta_j' - \theta_i \theta_j) \cdot |\langle \boldsymbol{\xi}_i, \boldsymbol{\xi}_j \rangle|$$

$$\leq (n)^2 (\max_{i \in [n]} \theta_i'^2 - \min_{i \in [n]} \theta_i^2) \cdot 2\sqrt{d \log(6n^2/\delta)}$$

$$\leq (n)^2 \left[ \left( \frac{1}{(1-\kappa)d - 2n\sqrt{d \log(6n^2/\delta)}} \right)^2 - \left( \frac{(1-\kappa)d - 4n\sqrt{d \log(6n^2/\delta)}}{(1+\kappa)d((1-\kappa)d - 2n\sqrt{d \log(6n^2/\delta)})} \right)^2 \right] \cdot 2\sqrt{d \log(6n^2/\delta)}$$

$$= \widetilde{O}\left( \frac{\kappa n^2}{d^{3/2}} \right) = O\left( \frac{n^2}{d^2} \right).$$

The third inequality is from Lemma 47 and 51; The last two equalities are from Assumption 9. Combining the above results, we have

$$\|\boldsymbol{v}'\|_2^2 - \|\boldsymbol{v}_{mm}\|_2^2 \geq \frac{C}{\rho^2} + O\left( \frac{n}{d} \right) \geq \frac{C_2(1-\beta)}{\rho^2}.$$

Here $C_2 = \Theta(1)$ is a constant. $\hfill \square$

Therefore, combining Lemma 50 and 52, we have the following statement for the difference between $\|\boldsymbol{v}'\|$ and $\|\boldsymbol{v}_{mm}\|$:

$$\|\boldsymbol{v}'\|_2^2 - \|\boldsymbol{v}_{mm}\|_2^2 \geq \frac{C_3(1-\beta)}{d}. \tag{70}$$

Here $C_3 = \Theta(1)$ is a constant. The inequality is from the SNR condition that $\rho = o(\sqrt{d/n})$.

Now we can prove the main proposition in this scenario.

*Proof of Proposition 46 in case 1.* From (70) we have

$$\|\boldsymbol{v}''\|_2^2 - \|\boldsymbol{v}\|_2^2 \geq \frac{C_3(1-\beta)}{d} = S(1-\beta)$$

Here we substitute $S = \frac{C_3}{d} \geq 0$ Then we have

$$\Gamma^2 - \Gamma'^2 = \frac{1}{\|\boldsymbol{v}\|^2} - \frac{1}{\|\boldsymbol{v}'\|^2} = \frac{\|\boldsymbol{v}'\|^2 - \|\boldsymbol{v}\|^2}{\|\boldsymbol{v}'\|^2 \cdot \|\boldsymbol{v}\|^2} \geq \frac{S(1-\beta)}{\|\boldsymbol{v}'\|^2 \cdot \|\boldsymbol{v}\|^2}.$$

Therefore,

$$\Gamma - \Gamma' \geq \frac{S(1-\beta)}{(\Gamma + \Gamma')\|\boldsymbol{v}\|^2 \cdot \|\boldsymbol{v}'\|^2} \geq \frac{S(1-\beta)}{2\Gamma\|\boldsymbol{v}\|^2 \cdot \|\boldsymbol{v}'\|^2}.$$

Set $c = \frac{S}{2\Gamma\|\boldsymbol{v}\|^2 \cdot \|\boldsymbol{v}'\|^2} = \frac{S}{2\|\boldsymbol{v}\|\|\boldsymbol{v}'\|^2}$, we have $\Gamma' \leq \Gamma - c(1-\beta)$. And we can upper bound $c$ as

$$c = \frac{S}{2\|\boldsymbol{v}\|\|\boldsymbol{v}'\|^2} \leq \frac{S}{r_{mm}^3} \leq \frac{C_3}{r_{mm}^3 d}.$$

The first inequality is from $\|\boldsymbol{v}'\| \geq \|\boldsymbol{v}\|$ and the second equality is from $S = \frac{C_2}{d}$.

$\hfill \square$

**Situation 2:** $p = 0, k - p \neq 0$

Then we consider the case when all wrong token selections come from noisy set. Same as above, denote the mixed samples as $k_1, k_2, ..., k_{k-p}$. And for every mixed sample $k_i$, we have $\boldsymbol{r}_{k_i} = (1 - \beta_i)\boldsymbol{\mu}_{k_i} + \beta_i \boldsymbol{\xi}_{k_i}$. Without losing generality, we assume that $y_{k_i} = +1$ for all $i \in [k - p]$, so the corresponding signal token is $\boldsymbol{\mu}_2$. Then the conditions under *Situation 2* become

**Condition 16** (Change k-p noisy samples)**.**

$$\begin{cases} y_i \boldsymbol{v}^\top \boldsymbol{\xi}_i \geq 1, i \in [n] \setminus [k - p] \\ \boldsymbol{v}^\top \boldsymbol{r}_{k_i} \geq 1, i \in [k - p] \end{cases}$$

Denote the max-margin solution under this condition as $\boldsymbol{v}'$ with parameters $\lambda_1', \lambda_2', \theta_i'$, we can interpret the condition for parameters:

$$
\begin{cases}
\theta_i' \cdot \|\boldsymbol{\xi}_{i'}\|^2 + \sum_{i' \neq i} y_i y_{i'} \theta_{i'}' \langle \boldsymbol{\xi}_i, \boldsymbol{\xi}_{i'} \rangle \geq 1, i \in [n] \backslash [k-p] \\
(1 - \beta_i) \lambda_2' \cdot \|\boldsymbol{\mu}_2\|^2 + \beta_i (\theta_{k_i}' \cdot \|\boldsymbol{\xi}_{k_i}\|^2 + \sum_{i' \neq k_i} y_{k_i} y_{i'} \theta_{i'}' \langle \boldsymbol{\xi}_{k_i}, \boldsymbol{\xi}_{i'} \rangle) \geq 1, i \in [k-p]
\end{cases}
$$

Compare with Codition 13, the only difference is that we substitute $\lambda_1' \|\boldsymbol{\mu}_1\|^2$ with $\lambda_2' \|\boldsymbol{\mu}_2\|^2$. From the symmetry, we can see that the two conditions are actually the same. Thereofre, we can follow the proof of Situation 1 to prove for Proposition 46 under this situation.

**Situation 3:** $p \neq 0, k - p \neq 0$

Last we consider the case when wrong tokens come from both clean and noisy sets. Denote the mixed clean samples as $k_1, k_2, ..., k_p$ and the mixed noisy samples as $q_1, q_2, ..., q_{k-p}$. Without losing generality, we assume that $y_{k_i} = +1$ for $i \in [p]$ and $y_{q_i} = -1$ for $i \in [k-p]$, which indicates that their signal tokens are all $\boldsymbol{\mu}_1$. Then the conditions under *Situation 2* become

**Condition 17** (p clean samples and k-p noisy samples violating optimal token selection)**.**

$$
\begin{cases}
y_i \boldsymbol{v}^\top \boldsymbol{\xi}_i \geq 1, i \in [n] \backslash [k] \\
\boldsymbol{v}^\top \boldsymbol{r}_{k_i} \geq 1, i \in [p] \\
-\boldsymbol{v}^\top \boldsymbol{r}_{q_i} \geq 1, i \in [k-p]
\end{cases}
$$

Denote the max-margin solution under this condition as $\boldsymbol{v}''$ with parameters $\lambda_1'', \lambda_2'', \theta_i''$, we can interpret the condition for parameters:

$$
\begin{cases}
\theta_i'' \cdot \|\boldsymbol{\xi}_{i'}\|^2 + \sum_{i' \neq i} y_i y_{i'} \theta_{i'}'' \langle \boldsymbol{\xi}_i, \boldsymbol{\xi}_{i'} \rangle) \geq 1, i \in [n] \backslash [k] \\
(1 - \beta_i) \lambda_1'' \cdot \|\boldsymbol{\mu}_1\|^2 + \beta_i (\theta_{k_i}'' \cdot \|\boldsymbol{\xi}_{k_i}\|^2 + \sum_{i' \neq k_i} y_{k_i} y_{i'} \theta_{i'}'' \langle \boldsymbol{\xi}_{k_i}, \boldsymbol{\xi}_{i'} \rangle) \geq 1, i \in [p] \\
-(1 - \beta_i) \lambda_1'' \cdot \|\boldsymbol{\mu}_1\|^2 - \beta_i (\theta_{q_i}'' \cdot \|\boldsymbol{\xi}_{q_i}\|^2 + \sum_{i' \neq q_i} y_{q_i} y_{i'} \theta_{i'}'' \langle \boldsymbol{\xi}_{q_i}, \boldsymbol{\xi}_{i'} \rangle) \geq 1, i \in [k-p]
\end{cases}
$$

We consider three cases: $\lambda_1'' \|\boldsymbol{\mu}_1\|^2 \geq 1, 1 > \lambda_1'' \|\boldsymbol{\mu}_1\|^2 \geq -1$ and $\lambda_1'' \|\boldsymbol{\mu}_1\|^2 < -1$.

- $\lambda_1'' \|\boldsymbol{\mu}_1\|^2 \geq 1$

  First when $\lambda_1'' \|\boldsymbol{\mu}_1\|^2 \geq 1$, we have $\frac{1 - (1 - \beta_i) \lambda_1' \|\boldsymbol{\mu}_1\|^2}{\beta_i} \leq 1$, which indicates that the condition for mixed clean samples' parameter $\theta_{k_i}'$ is relaxed. Meanwhile, for the mixed noisy samples we have

  $$
  -\theta_{q_i}'' \cdot \|\boldsymbol{\xi}_{q_i}\|^2 + \sum_{i' \neq q_i} y_{q_i} y_{i'} \theta_{i'}'' \langle \boldsymbol{\xi}_{q_i}, \boldsymbol{\xi}_{i'} \rangle \geq \frac{1 + (1 - \beta_i) \lambda_1'' \|\boldsymbol{\mu}_1\|^2}{\beta_i} \geq 1,
  $$

  which indicates that the condition is strengthened. Therefore, this case is an extension of the second case of Situation 1 with strengthening some constraints. These constraints will not result in a better solution than Situation 1. The following proof is the same as Situation 1 and we omit it for convenience.

- $1 > \lambda_1'' \|\boldsymbol{\mu}_1\|^2 \geq -1$

  In this case, the constraints for both mixed clean and noisy samples are strengthened. So this can be taken as an extension of the first case in Situation 1 with strengthening some constraints. The following proof is the same as Situation 1 and we omit it for convenience.

- $\lambda_1'' \|\boldsymbol{\mu}_1\|^2 < -1$

  In this case, the constraints are strengthened for mixed clean samples while relaxed for the mixed noisy samples. So we consider it as the extension of Situation 2 when $\lambda_1' \|\boldsymbol{\mu}_1\|^2 < -1$ with strengthening some constraints. The following proof is the same as Situation 2 and we omit it for convenience.

Therefore, we complete the proof for all possible situations. $\qquad\square$

**Training and Test error analysis**

From Proposition 46 we can derive the convergence direction of $\boldsymbol{p}$ and $\boldsymbol{v}$, i.e. $\boldsymbol{p}_{mm}$ and $\boldsymbol{v}_{mm}$. Note that Theorem 16 does not depend on the selection of optimal tokens, so it still holds in this case when optimal tokens are noise tokens for all samples. We restate it here for convenience:

**Theorem 53.** *Suppose that Assumption 9 holds, with probability at least $1 - \delta$ on the training dataset, we have*

- *the margin induced by $\boldsymbol{p}_R/R$ in p-SVM is at least $(1 - \zeta)\Xi$, where*

$$\zeta = \frac{\log(4\sqrt{(1+\kappa)d}\|\boldsymbol{v}_{mm}\|^3 d\rho^2)}{R\Xi}.$$

- *the label margin induced by $\boldsymbol{v}_r/r$ in v-SVM is at least $(1 - \gamma)\Gamma$, where $\gamma = \frac{2\sqrt{(1+\kappa)d}}{\Gamma\exp((1-\zeta)R\Xi)}$.*

Then we could estimate the test error in this case. From Theorem 53 we have

$$\boldsymbol{p}_R^\top(\boldsymbol{\xi}_i - \boldsymbol{\mu}_i) \geq (1 - \zeta)R\Xi, \forall i \in [n] \tag{71}$$

$$y_i \boldsymbol{v}_r^\top \boldsymbol{\xi}_i \geq (1 - \gamma)\Gamma r, \forall i \in [n]. \tag{72}$$

Here $\zeta, \gamma, \Xi, \Gamma$ are the same as the definition in Theorem 53. Similarly, we have the following lemma for $\zeta, \gamma$.

**Lemma 54.** *Suppose that Assumption 9 holds, with probability at least $1 - \delta$ on the training dataset, consider the same setting in Theorem 16, we have $\zeta < 0.2$ and $\gamma < 1$.*

*Proof of Lemma 54.* First we upper bound $\|\boldsymbol{p}_{mm}\|$. Consider the following possible solution $\widetilde{\boldsymbol{p}}$:

$$\widetilde{\boldsymbol{p}} = \sum_{i \in [n]} 2\frac{\boldsymbol{\xi}_i}{d}. \tag{73}$$

We then proved that $\widetilde{\boldsymbol{p}}$ satisfies (63). For $\forall k \in [n]$, we have

$$\widetilde{\boldsymbol{p}}^\top(\boldsymbol{\xi}_k - \boldsymbol{\mu}_k) = \sum_{i \in [n]} 2\frac{\langle\boldsymbol{\xi}_i, \boldsymbol{\xi}_k\rangle}{d} \geq 2(1 - \kappa) + \sum_{i \in [n], i \neq k} 2\frac{\langle\boldsymbol{\xi}_i, \boldsymbol{\xi}_k\rangle}{d}$$

$$\geq 2(1 - \kappa) + \frac{2n\sqrt{d\log(6n^2/\delta)}}{d} \geq 1.$$

The first and second inequalities are from Lemma 57; The last inequality is from Assumption 9.

Therefore, the max-margin solution $\boldsymbol{p}_{mm}$ must have no greater norm than $\widetilde{\boldsymbol{p}}$. So we can upper bound $\boldsymbol{p}_{mm}$ as

$$\|\boldsymbol{p}_{mm}\|^2 \leq \|\widetilde{\boldsymbol{p}}\|^2 = \frac{4}{d^2}\Big(\sum_{i \in [n]} \|\boldsymbol{\xi}_i\|^2 + \sum_{i,j \in [n], i \neq j} \langle\boldsymbol{\xi}_i, \boldsymbol{\xi}_j\rangle\Big)$$

$$\leq \frac{4}{d^2}\big((1 + \kappa)nd + 2n^2\sqrt{d\log(6n^2/\delta)}\big) \leq \frac{5n}{d}.$$

The second inequality is from Lemma 57; The last inequality is from the definition of $d$ in Assumption 9.

Then from the definition of $\zeta$ in Theorem 16, we have

$$\zeta = \frac{\log(4\sqrt{(1+\kappa)d}\|\boldsymbol{v}_{mm}\|^3 d\rho^2)}{R\Xi} \leq C_1 \frac{\sqrt{n/d}}{R}\log(4\sqrt{(1+\kappa)d}\|\boldsymbol{v}_{mm}\|^3 d\rho^2)$$

$$\leq C_2 \frac{\sqrt{n/d}}{R}\log\left(\frac{n^3}{d}\right) < 0.2.$$

Here $C_1, C_2 = \Theta(1)$. The first inequality is from $\Xi^{-1} = \|\boldsymbol{p}_{mm}\| \le \sqrt{5n/d}$; The second inequality is from the upper bound of $\|\boldsymbol{v}_{mm}\|$ in Lemma 48 and the last inequality is from the definition of $R$ in Assumption 9. And for $\gamma$, we have

$$\gamma = \frac{2M}{\Gamma \exp((1-\zeta)R\Xi)} = C_1' \frac{M\|\boldsymbol{v}_{mm}\|}{\exp(R/\|\boldsymbol{v}_{mm}\|)} \le C_2' \frac{\sqrt{d \cdot (n/d)}}{\exp(R/\sqrt{n/d})} < 1.$$

Here $C_1', C_2' = \Theta(1)$. The first inequality is from the lower and upper bound of $\|\boldsymbol{v}_{mm}\|$ in Lemma 28 and the last inequality is from the definition of $R$ in Assumption 5. $\qquad\square$

Then we have the following lemma to estimate the innerproduct of $\boldsymbol{p}_R$ and signal token:

**Lemma 55.** *Suppose that Assumption 9 holds, with probability at least $1 - \delta$ on the training dataset, we have*

$$|\langle \boldsymbol{p}_R, \boldsymbol{\mu}_j \rangle| \le 0.9(1 - \zeta)R\xi$$

*for $j \in \{1, 2\}$.*

*Proof of Lemma 55.* First we use contradiction to prove for the lower bound. Assume that $|\langle \boldsymbol{p}_R, \boldsymbol{\mu}_j \rangle| > 0.9(1 - \zeta)R\Xi$. We can estimate $\|\boldsymbol{p}_R\|$ as

$$\|\boldsymbol{p}_R\|^2 > (0.9(1-\zeta)R\Xi)^2/\rho^2 > (0.5\Xi^2/\rho^2) \cdot R^2 \ge (0.1d/n\rho^2) \cdot R^2 > R^2.$$

The second inequality is from Lemma 54 ; The third inequality is from $\Xi^2 = \|\boldsymbol{p}_{mm}\|^{-2} \ge d/(5n)$; The last inequality is from our SNR condition $\rho = o(\sqrt{d/n})$. This leads to a contradiction.

$\qquad\square$

From Lemma 41, we can denote $\boldsymbol{v}_r$ as

$$\boldsymbol{v}_r = \lambda_1 \boldsymbol{\mu}_1 + \lambda_2 \boldsymbol{\mu}_2 + \sum_{i \in [n]} y_i \theta_i \boldsymbol{\xi}_i.$$

Denote $\boldsymbol{v}_{\boldsymbol{\xi}} = \sum_{i \in [n]} y_i \theta_i \boldsymbol{\xi}_i$ as the noise part of $\boldsymbol{v}_r$. Then we prove that $\boldsymbol{p}_R, \boldsymbol{v}_{\boldsymbol{\xi}}$ are near orthogonal

**Lemma 56.** *Suppose that Assumption 9 holds, with probability at least $1 - \delta$ on the training dataset, we have*

$$|\langle \boldsymbol{p}_R, \boldsymbol{v}_{\boldsymbol{\xi}} \rangle| \le c$$

*for some constant $c \in (0, 1)$.*

*Proof of Lemma 56.* First plugging in the parameters in $\boldsymbol{v}_{\boldsymbol{\xi}}$ we have

$$\langle \boldsymbol{p}_R, \boldsymbol{v}_{\boldsymbol{\xi}} \rangle = \sum_{i \in [n]} y_i \theta_i \boldsymbol{p}_R^\top \boldsymbol{\xi}_i$$

$$= \sum_{y_i = +1} \theta_i \boldsymbol{p}_R^\top \boldsymbol{\xi}_i - \sum_{y_i = -1} \theta_i \boldsymbol{p}_R^\top \boldsymbol{\xi}_i$$

$$\le (n_{11} + n_{21})(\max_i \theta_i)(R\Xi + O(R\rho)) - (n_{12} + n_{22})(\min_i \theta_i)((1-\zeta)R\Xi - O(R\rho))$$

$$\le \underbrace{(n/2)(\max_i \theta_i - \min_i \theta_i)R\Xi}_{I_1} + \underbrace{O(\sqrt{n})(\max_i \theta_i)R\Xi}_{I_2} + \underbrace{n(\max_i \theta_i)(\zeta R\Xi + O(R\rho))}_{I_3}.$$

The first inequality is from Theorem 53 that $(1 - \zeta)R\Xi \le \boldsymbol{p}_R^\top(\boldsymbol{\xi}_i - \boldsymbol{\mu}_i) \le R\Xi$ and $\boldsymbol{p}_R^\top \boldsymbol{\mu}_i = O(R\rho)$ and the second inequality is from Lemma 59. Then we bound $I_1 \sim I_3$ respectively. For $I_1$, we need to first bound $\theta_i$. From Theorem 53 we have

$$(1 - \gamma)\Gamma r \le y_i \boldsymbol{v}_r^\top \boldsymbol{\xi}_i \le \Gamma r, \forall i \in [n].$$

Denote $j = \text{argmax}_i \theta_i$, we have

$$y_j \boldsymbol{v}_r^\top \boldsymbol{\xi}_j \ge \theta_j \|\boldsymbol{\xi}_j\|^2 + n\theta_j \sqrt{d \log(6n^2/\delta)} \ge \theta_j((1-\kappa)d + n\sqrt{d \log(6n^2/\delta)}).$$

Therefore, we can upper bound $\theta_j$ as

$$\theta_j \leq \frac{y_j \boldsymbol{v}_r^\top \boldsymbol{\xi}_i}{(1-\kappa)d + n\sqrt{d\log(6n^2/\delta)}} \leq \frac{\Gamma r}{(1-\kappa)d + n\sqrt{d\log(6n^2/\delta)}}. \tag{74}$$

Then we can lower bound $\theta_i$ as

$$y_i \boldsymbol{v}_r^\top \boldsymbol{\xi}_i \leq \theta_i \|\boldsymbol{\xi}_i\|^2 + n\theta_j \sqrt{d\log(6n^2/\delta)} \leq (1+\kappa)d\theta_i + \frac{\Gamma rn\sqrt{d\log(6n^2/\delta)}}{(1-\kappa)d + n\sqrt{d\log(6n^2/\delta)}}.$$

Therefore,

$$\theta_i \geq \frac{(1-\gamma)(1-\kappa)\Gamma rd - \gamma\Gamma rn\sqrt{d\log(6n^2/\delta)}}{(1+\kappa)d(1-\kappa)d + n\sqrt{d\log(6n^2/\delta)}}.$$

So we can estimate $I_1$ as

$$I_1 \leq (nR\Xi/2) \cdot \left( \frac{\Gamma r}{(1-\kappa)d + n\sqrt{d\log(6n^2/\delta)}} - \frac{(1-\gamma)(1-\kappa)\Gamma rd - \gamma\Gamma rn\sqrt{d\log(6n^2/\delta)}}{(1+\kappa)d(1-\kappa)d + n\sqrt{d\log(6n^2/\delta)}} \right)$$

$$\leq R\sqrt{nd}/2 \cdot \Gamma r \cdot \left( \frac{1 - \frac{(1-\gamma)(1-\kappa)}{1+\kappa} + \frac{\gamma n\log(6n^2/\delta)}{(1+\kappa)d}}{(1-\kappa)d + n\sqrt{d\log(6n^2/\delta)}} \right)$$

$$\leq Rr(\kappa + \gamma).$$

The second inequality is from $\Xi = \|\boldsymbol{p}_{mm}\| = \Theta(\sqrt{d/n})$ and the last inequality is from $\Gamma = \|\boldsymbol{v}_{mm}\|^{-1} = \Theta(\sqrt{d/n})$.

Then we bound $I_2$. From (74) we have $\max_i \theta_i = \Theta(\Gamma r/d)$. Therefore,

$$I_2 \leq O(\sqrt{n})\Theta(\Gamma r/d)R\Xi \leq Rr \cdot O(1/\sqrt{n}).$$

The last inequality is from $\Gamma, \Xi = \Theta(\sqrt{d/n})$.

Last we bound $I_3$ as

$$I_3 = n\Theta(\Gamma r/d)(\zeta R\Xi + O(R\rho))$$

$$\leq \Theta(r\sqrt{n/d})(\log(4\sqrt{(1+\kappa)d}\|\boldsymbol{v}_{mm}\|^3 d\rho^2) + O(R\rho))$$

$$\leq Rr \cdot O(\rho\sqrt{n/d}).$$

The first inequality is from $\Gamma, \Xi = \Theta(\sqrt{d/n})$ and the last inequality is from Assumption 9.

Combining the results above, we have

$$\langle \boldsymbol{p}_R, \boldsymbol{v}_\xi \rangle \leq I_1 + I_2 + I_3 \leq Rr \cdot O(\sqrt{1/n} + \rho\sqrt{n/d}) \leq c$$

for sufficiently large $d$ and $n$. Here the last inequality comes from Assumption 9. $\qquad\square$

With the lemmas above, we could prove for the main theorem

*Proof of Theorem 10.* First we show that the model can perfectly classify all training samples. From Theorem 16, we have

$$y_i \boldsymbol{v}_r^\top \boldsymbol{r}_i = y_i\beta_i \boldsymbol{v}_r^\top \boldsymbol{\xi}_i + y_i(1-\beta_i)\boldsymbol{v}_r^\top \boldsymbol{\mu}_i \geq \beta_i(1-\gamma)\Gamma r - 0.9(1-\beta_i)(1-\gamma)\Gamma r > 0,$$

for $\forall i \in [n]$. The last inequality is from Lemma 54. Thus $y_i = \text{sign}(f(\boldsymbol{X}_i; \boldsymbol{p}_R, \boldsymbol{v}_r))$ for all $i \in [n]$.

Then we bound the test error. This is equivalent to estimate $y \cdot f(\boldsymbol{p}_R, \boldsymbol{v}_r; \boldsymbol{X})$ and we could write it as

$$y \cdot f(\boldsymbol{p}_R, \boldsymbol{v}_r; \boldsymbol{X}) = y \cdot \frac{\exp(\langle \boldsymbol{p}_R, \boldsymbol{\mu}' \rangle)\boldsymbol{v}_r^\top \boldsymbol{\mu}' + \exp(\langle \boldsymbol{p}_R, \boldsymbol{\xi}' \rangle)\boldsymbol{v}_r^\top \boldsymbol{\xi}'}{\exp(\langle \boldsymbol{p}_R, \boldsymbol{\mu}' \rangle) + \exp(\langle \boldsymbol{p}_R, \boldsymbol{\xi}' \rangle)}.$$

We first upper bound the term $y \cdot \exp(\langle \boldsymbol{p}_R, \boldsymbol{\mu}' \rangle) \boldsymbol{v}_r^\top \boldsymbol{\mu}'$. From Theorem 53, the non-optimality of $i$-th sample is

$$1 - \beta_i = \frac{\exp(\langle \boldsymbol{p}_R, \boldsymbol{\mu}_i \rangle)}{\exp(\langle \boldsymbol{p}_R, \boldsymbol{\mu}_i \rangle) + \exp(\langle \boldsymbol{p}_R, \boldsymbol{\xi}_i \rangle)} \leq \frac{1}{1 + \exp((1-\zeta)\Xi R)} \text{ for all } i \in [n].$$

The last inequality is from the first statement in Theorem 53. Consider the sample that contains the same signal token as $\boldsymbol{\mu}'$, we have

$$(1 - \beta_i) \boldsymbol{v}_r^\top \boldsymbol{\mu}_i = \frac{\exp(\langle \boldsymbol{p}_R, \boldsymbol{\mu}_i \rangle) \boldsymbol{v}_r^\top \boldsymbol{\mu}_i}{\exp(\langle \boldsymbol{p}_R, \boldsymbol{\mu}_i \rangle) + \exp(\langle \boldsymbol{p}_R, \boldsymbol{\xi}_i \rangle)}.$$

Therefore,

$$y \cdot \exp(\langle \boldsymbol{p}_R, \boldsymbol{\mu}' \rangle) \boldsymbol{v}_r^\top \boldsymbol{\mu}' \leq \exp(\langle \boldsymbol{p}_R, \boldsymbol{\mu}_i \rangle) |\boldsymbol{v}_r^\top \boldsymbol{\mu}_i| \leq \frac{\exp(\langle \boldsymbol{p}_R, \boldsymbol{\mu}_i \rangle) + \exp(\langle \boldsymbol{p}_R, \boldsymbol{\xi}_i \rangle)}{1 + \exp((1-\zeta)\Xi R)} \cdot |\boldsymbol{v}_r^\top \boldsymbol{\mu}_i|$$

$$\leq \frac{2\exp(\langle \boldsymbol{p}_R, \boldsymbol{\xi}_i \rangle)}{\exp((1-\zeta)\Xi R)} \cdot |\boldsymbol{v}_r^\top \boldsymbol{\mu}_i| \leq \frac{2\exp(\Xi R)}{\exp((1-\zeta)\Xi R)} \cdot |\boldsymbol{v}_r^\top \boldsymbol{\mu}_i|$$

$$\leq 2\exp(\zeta \Xi R) \cdot \rho r = (4\sqrt{(1+\kappa)d}\|\boldsymbol{v}_{mm}\|^3 d\rho^2) \cdot \rho r \leq C n^{3/2} \rho^3 r \quad (75)$$

for some constant $C > 0$. Here the third inequality is from $\boldsymbol{p}_R^\top(\boldsymbol{\xi}_i - \boldsymbol{\mu}_i) \geq 0$; The fourth inequality is from the fact that $\langle \boldsymbol{p}_R, \boldsymbol{\xi}_i \rangle \leq \Xi R$ and the last inequality is from $\|\boldsymbol{v}_r\| \leq r, \|\boldsymbol{\mu}_i\| \leq \rho$. Then we can bound the test error as

$$\mathbb{P}(y \cdot f(\boldsymbol{p}_R, \boldsymbol{v}_r; \boldsymbol{X}) \leq 0) = \mathbb{P}(y \cdot \exp(\langle \boldsymbol{p}_R, \boldsymbol{\mu}' \rangle) \boldsymbol{v}_r^\top \boldsymbol{\mu}' + y \cdot \exp(\langle \boldsymbol{p}_R, \boldsymbol{\xi}' \rangle) \boldsymbol{v}_r^\top \boldsymbol{\xi}' \leq 0)$$

$$\geq \mathbb{P}(y \cdot \exp(\langle \boldsymbol{p}_R, \boldsymbol{\xi}' \rangle) \boldsymbol{v}_r^\top \boldsymbol{\xi}' \leq -C n^{3/2} \rho^3 r)$$

$$\geq \frac{1}{4} \mathbb{P}\left( y \boldsymbol{v}_{\boldsymbol{\xi}}^\top \boldsymbol{\xi}' \leq -e^{-R/C} \cdot C n^{3/2} \rho^3 r \mid \langle \boldsymbol{p}_R/R, \boldsymbol{\xi}' \rangle \in [1/C, C] \right)$$

$$\geq \frac{1}{4} \left( \frac{1}{2} - \frac{cC + C\exp(-R/C) n^{3/2} \rho^3}{\sqrt{2\pi(1-c^2)}} \right) \geq \frac{1}{16}.$$

The first inequality is from (75); the second inequality use the fact that there exists a constant $C > 0$ such that $\mathbb{P}(N(0,1) \in [1/C, C]) \geq 1/4$; the third inequality comes from Lemma 60 and the last inequality uses Assumption 9. $\qquad \square$

.

### A.3 SUPPLEMENT LEMMAS

Here we list some technical lemmas for the main proof.

**Lemma 57.** *(Properties of Training Data) Suppose that $\delta > 0$ and $\kappa = O(\sqrt{\log(6n/\delta)/d}) = \widetilde{O}(1/\sqrt{d})$. Then with probability at least $1 - \delta$, we have*

$$(1-\kappa)d \leq \|\boldsymbol{\xi}_i\|_2^2 \leq (1+\kappa)d$$

$$|\langle \boldsymbol{\xi}_i, \boldsymbol{\xi}_j \rangle| \leq 2\sqrt{d\log(6n^2/\delta)}$$

*for any $i, j \in [n]$.*

*Proof of Lemma 57.* By Bernstein's inequality (see Theorem 2.8.1 in Vershynin (2018)), with probability at least $1 - \delta/(3n)$ we have

$$|\|\boldsymbol{\xi}_i\|_2^2 - d| = O(\sqrt{d\log(6n/\delta)}).$$

Therefore, there exists $\kappa = O(\sqrt{\log(6n/\delta)/d})$ that

$$(1-\kappa)d \leq \|\boldsymbol{\xi}_i\|_2^2 \leq (1+\kappa)d.$$

Moreover, $\langle \boldsymbol{\xi}_i, \boldsymbol{\xi}_j \rangle$ has mean zero. For any $i, j \in [n]$ and $i \neq j$, by Bernstein's inequality, with probability at least $1 - \delta/(3n^2)$ we have

$$|\langle \boldsymbol{\xi}_i, \boldsymbol{\xi}_j \rangle| \leq 2\sqrt{d\log(6n^2/\delta)}.$$

Applying a union bound completes the proof. $\qquad \square$

Set $\delta = 6n \exp(-d/4C_1 n^2)$ for any constant $C_1 > 0$, we can follow the proof of Lemma 57 and conclude the next remark:

**Remark 58.** *(Properties of New Test Sample) Let $(\boldsymbol{X} = (\boldsymbol{\mu}_k, \boldsymbol{\xi}), y) \sim \mathcal{D}$. Then with probability at least $1 - 6n \exp(-d/4C_1 n^2)$, we have*

$$|\langle \boldsymbol{\xi}, \boldsymbol{\xi}_i \rangle| \leq \frac{d}{C_1 n}$$

*for any $i \in [n]$.*

**Lemma 59.** *With probability at least $1 - 6\delta$,*

$$\big| |\mathcal{C}| - n(1-\eta) \big| \leq \sqrt{n \log(\frac{1}{\delta})}; \quad \big| |\mathcal{N}| - n\eta \big| \leq \sqrt{n \log(\frac{1}{\delta})};$$

$$\big| |\mathcal{C}_i| - \frac{n(1-\eta)}{2} \big| \leq \sqrt{n \log(\frac{1}{\delta})}; \quad \big| |\mathcal{N}_i| - \frac{n\eta}{2} \big| \leq \sqrt{n \log(\frac{1}{\delta})}, \quad i = 1, 2.$$

*Proof.* Note that $|\mathcal{C}| \sim \text{Binom}(n, 1-\eta)$. Applying Hoeffding's inequality, we have

$$\mathbb{P}\big( \big| |\mathcal{C}| - (1-\eta)n \big| > t \big) \leq 2 \exp(-\frac{2t^2}{n}).$$

Let $t = \sqrt{n \log(1/\delta)}$. We have that with probability at least $1 - \delta$,

$$\big| |\mathcal{C}| - (1-\eta)n \big| \leq \sqrt{n \log(\frac{1}{\delta})}.$$

Similarly, note that $|\mathcal{N}| \sim \text{Binom}(n, \eta), |\mathcal{C}_1| \sim \text{Binom}(n, (1-\eta)/2), |\mathcal{C}_2| \sim \text{Binom}(n, (1-\eta)/2), |\mathcal{N}_1| \sim \text{Binom}(n, \eta/2)$ and $|\mathcal{N}_2| \sim \text{Binom}(n, \eta/2)$, we have that each of the following events holds with probability at least $1 - \delta$:

$$\big| |\mathcal{C}| - n(1-\eta) \big| \leq \sqrt{n \log(\frac{1}{\delta})}; \quad \big| |\mathcal{N}| - n\eta \big| \leq \sqrt{n \log(\frac{1}{\delta})};$$

$$\big| |\mathcal{C}_i| - n(1-\eta)/2 \big| \leq \sqrt{n \log(\frac{1}{\delta})}, \quad i = 1, 2;$$

$$\big| |\mathcal{N}_i| - n\eta/2 \big| \leq \sqrt{n \log(\frac{1}{\delta})}, \quad i = 1, 2.$$

$\square$

**Lemma 60.** *Suppose $X \sim N(0, \boldsymbol{I}_d)$, and $\boldsymbol{v}, \boldsymbol{p} \in \mathbb{R}^d$ are two vectors with $\|\boldsymbol{v}\| = \|\boldsymbol{p}\| = 1, \boldsymbol{v}^\top \boldsymbol{p} \leq c$ for some constant $c \in (0, 1)$. Given some constant $C > 1$, for $z < 0$,*

$$\mathbb{P}(\boldsymbol{v}^\top X < z | \boldsymbol{p}^\top X \in [1/C, C]) \geq \frac{1}{2} - \frac{1}{\sqrt{2\pi}} \frac{cC - z}{\sqrt{1 - c^2}}.$$

*Proof of Lemma 60.* Denote $x_v = v^\top X \sim N(0, 1), x_p = \boldsymbol{p}^\top X \sim N(0, 1)$. Then we have $x_v, x_p \sim \mathcal{N}(0, 1)$. Denote the covariance between $x_v, x_p$ by $c_0$, then we have

$$c_0 = \text{Cov}(x_v, x_p) = \boldsymbol{v}^\top \text{Cov}(X) \boldsymbol{p} = \boldsymbol{v}^\top \boldsymbol{p} \leq c.$$

Note that

$$x_v \stackrel{d}{=} c_0 x_p + \sqrt{1 - c_0^2} r,$$

where $r \sim N(0, 1)$ is independent of $x_p$. It follows that

$$\mathbb{P}(x_v < z | x_p \in [\frac{1}{C}, C]) = \mathbb{P}(r < \frac{z - c_0 x_p}{\sqrt{1 - c_0^2}} | x_p \in [\frac{1}{C}, C]) \geq \mathbb{P}(r < \frac{z - cC}{\sqrt{1 - c^2}}) \geq \frac{1}{2} - \frac{1}{\sqrt{2\pi}} \frac{cC - z}{\sqrt{1 - c^2}}.$$

$\square$

## A.4 Additional Experiments

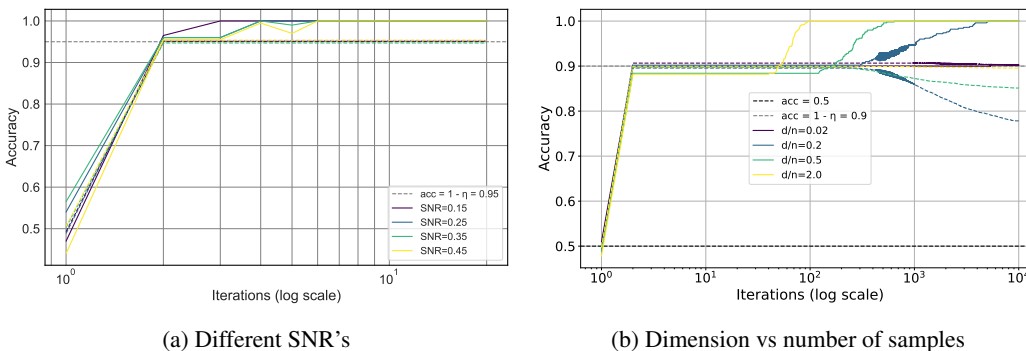

(a) Different SNR's

(b) Dimension vs number of samples

Figure 4: Comparing train (solid lines) and test (dashed lines) accuracies with different SNR (left panel) and different dimensions (right panel), as in Figure 3b. In the left panel, we see that for higher SNR more than two iterations are required to achieve benign overfitting. In the right panel, we see that for small $d$ (purple line), the model is unable to fit the data (at least in the first $10^5$ first iterations), and both the train and test accuracies are at the noise-rate level. For intermediate values of $d$ (green and blue lines), the model exhibits harmful overfitting, and for larger $d$ (yellow line) the model exhibits benign overfitting. We note that benign overfitting occurs here for $d = 2n \ll n^2$, which suggests that the assumptions on $d$ in our theorems are loose. Parameters: $n = 500, \beta = 0.02, \rho = 30, \eta = 0.1$, test sample size $= 10000$.

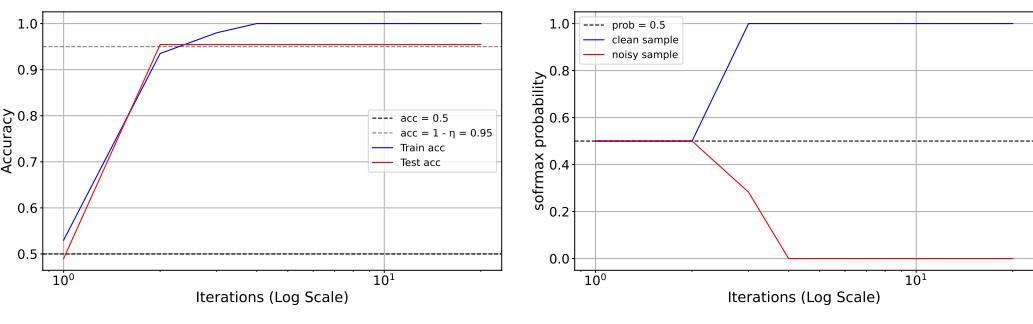

(a) train and test accuracy

(b) attention weights on signal token

Figure 5: Self-attention experiments. The left panel shows the train and test accuracies during training. It shows that benign overfitting also occurs after 2 iterations. In the right panel, we show the softmax probability of the signal token for clean and noisy samples (average of the softmax probabilities $s_{j,1}^t$ over $\mathcal{C}$ and $\mathcal{N}$ respectively). We see that after 2 iterations, the attention focuses on signal tokens for clean examples, and on noise tokens for noisy examples. This indicates that our results can be extended to self-attention mechanism. Parameters: $n = 200, d = 40000, \beta = 0.025, \rho = 20, \eta = 0.05$, test sample size $= 2000$.

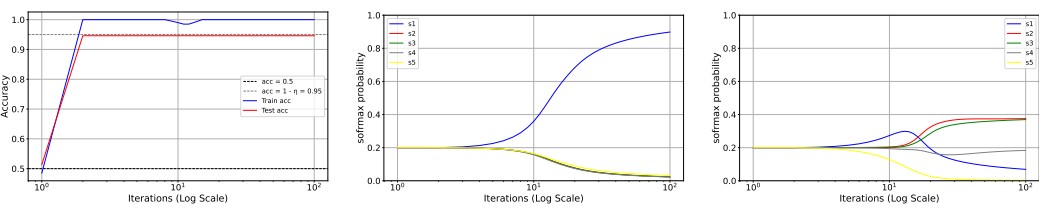

(a) train and test accuracy    (b) clean sample attention weights    (c) noisy sample attention weights

Figure 6: Multi-token experiments. The first panel shows the train and test accuracies during training. It shows that benign overfitting also occurs after 2 iterations. In the middle panel, we show that for clean samples the softmax probability of the signal token $s_{j,1}^t$ dominates the overall attention. While in the last panel, we show that for noisy samples the softmax probabilities of noise tokens are average. This indicates that our results can be extended to multi-token settings. Parameters: $T = 5, n = 200, d = 10000, \beta = 0.025, \rho = 15, \eta = 0.05$, test sample size $= 2000$.

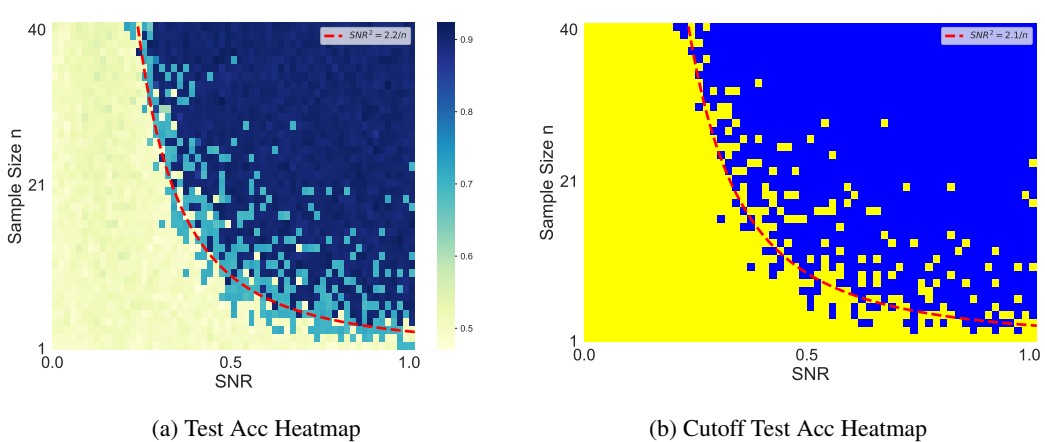

(a) Test Acc Heatmap            (b) Cutoff Test Acc Heatmap

Figure 7: The left panel presents a heatmap of the test acc, plotted across varying signal-to-noise ratios (SNR) and sample sizes (n). Yellow indicates small test acc, while blue represents high test loss. The right panel shows a heatmap with a cutoff value of 0.7, where values below 0.7 are categorized as 0 (blue) and values above 0.7 as 1 (green). In both panels, the red curves represent the expression $SNR^2 = 2.1/n$. This validates our tight bound of SNR $= \Theta(1/\sqrt{n})$ to achieve benign overfitting, and with a smaller SNR the model exhibits harmful overfitting. Parameters: $d = 900, \beta = 0.01, \eta = 0.1$, test sample size $= 2000$.

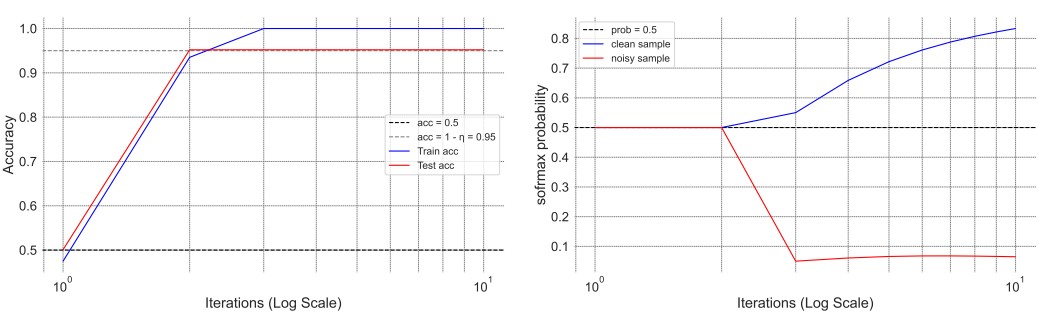

(a) train and test accuracy          (b) attention weights in signal token

Figure 8: The left panel shows the train and test accuracies during training (with Gaussian initialization, where each entry has variance 0.01). As in Figure 1, It shows that benign overfitting occurs after 2 iterations. After the first iteration, the model correctly classifies the clean training examples, but not the noisy ones. In the right panel, we show the softmax probability of the signal token for clean and noisy samples (average of the softmax probabilities $s_{j,1}^t$ over $\mathcal{C}$ and $\mathcal{N}$ respectively). We see that after 2 iterations, the attention focuses on signal tokens for clean examples, and on noise tokens for noisy examples. This aligns with Theorem 4. Parameters: $n = 200, d = 40000, \beta = 0.025, \rho = 30, \eta = 0.05$, test sample size $= 2000$.

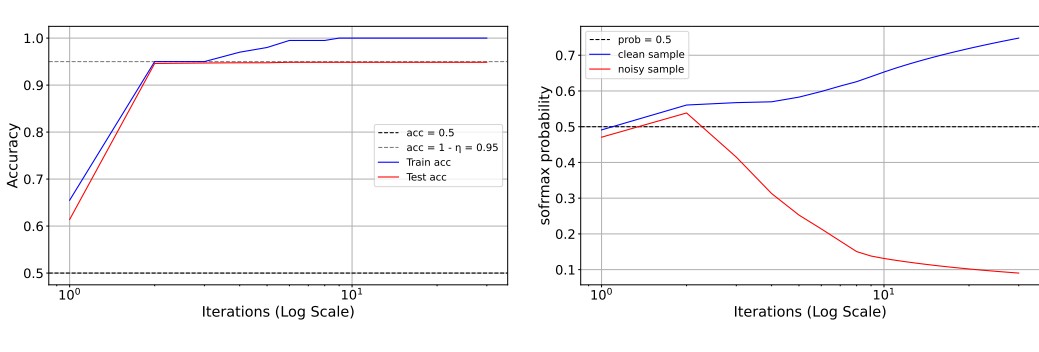

(a) train and test accuracy

(b) attention weights on signal token

Figure 9: Multi-layer experiments. The left panel shows the train and test accuracies during training in a 4-layer single-head attention model. It shows that benign overfitting also occurs after 2 iterations. After the first iteration, the model correctly classifies the clean training examples, but not the noisy ones. In the right panel, we show the softmax probability of the signal token for clean and noisy samples (average of the softmax probabilities $s_{j,1}^t$ over $\mathcal{C}$ and $\mathcal{N}$ respectively) in the first layer. We see that the attention focuses on signal tokens for clean examples, and on noise tokens for noisy examples. This indicates that our results can be extended to multi-layer models. Parameters: $n = 200, d = 10000, \beta = 0.025, \rho = 40, \eta = 0.05,$ test sample size $= 2000$.

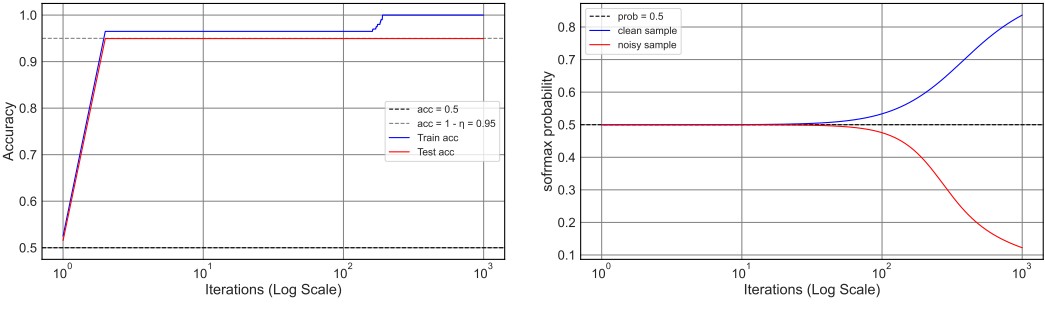

(a) Accuracy during iterations

(b) Softmax probability for signal (first) token

Figure 10: The left panel shows train and test accuracies during training with GD with weight decay, as in Figure 2. The clean training samples are correctly classified already after one iteration, but in contrast to Theorem 4 and Figure 1, benign overfitting occurs after about $150$ iterations. In the right panel we see that the attention starts separating signal and noise tokens shortly before benign overfitting occurs. Parameters: weight decay $= 0.01, n = 200, d = 40000, \beta = 0.0001, \rho = 30, \eta = 0.05,$ test sample size $= 2000$.

