3. *Generate noisy tokens $\boldsymbol{\xi}_\tau$ for $\tau \in \{2, 3, \ldots, T\}$, from the Gaussian distribution $\boldsymbol{\xi}_\tau \sim \mathcal{N}(\boldsymbol{0}, \boldsymbol{I}_d - \boldsymbol{\mu}_1 \boldsymbol{\mu}_1^\top / \rho^2 - \boldsymbol{\mu}_2 \boldsymbol{\mu}_2^\top / \rho^2)$ and let $\boldsymbol{x}_\tau = \boldsymbol{\xi}_\tau$.*

4. *Denote $\boldsymbol{X} = (\boldsymbol{x}_1, \boldsymbol{x}_2, \ldots, \boldsymbol{x}_T)^\top$.*

We make the following assumptions:

**Assumption 25** (Assumptions for GD with SNR $= \Theta(1/\sqrt{n})$)**.** *Let $\delta \in (0, 0.5)$ be a desired probability of failure. For any constants $C_T \geq 2, C_\rho \geq 6(C_T - 1), C_\beta \geq 16C_T^3$, as well as a sufficiently large universal constant $C$ that may depend on $C_\rho, C_T$ and $C_\beta$, the following conditions hold:*

1. *Number of samples $n$ is sufficiently large: $n \geq C \log(1/\delta)$.*

2. *Dimension $d$ is sufficiently large: $d \geq Cn^2 \log(n/\delta)$.*

3. *Signal strength satisfies $\rho = C_\rho \cdot \sqrt{d/n}$*

4. *Label flipping rate satisfies $\eta \leq 1/C$.*

5. *Step size satisfies $\beta = C_\beta \cdot (n/d)$.*

6. *Initialization at zero: $\|\boldsymbol{v}_0\| = \|\boldsymbol{p}_0\| = 0$.*

7. *Number of tokens satisfies: $2 \leq T \leq C_T$*

We now state our main result on benign overfitting with GD for the multiple token setting:

**Theorem 26.** *Suppose that Assumption 25 holds. Then, with probability at least $1 - \delta$ over the training dataset, after two iterations of GD we have:*

- *Higher softmax probability for optimal tokens:*

$$s_{i,1}^{t=2} > \frac{1}{T}, \ \forall i \in \mathcal{C} \qquad and \qquad s_{i,1}^{t=2} < \frac{1}{(T-1)c_\rho^2}, \ \forall i \in \mathcal{N}$$

  *where $s_{i,j}^t$ is the softmax probability of the $j^{th}$ token in the $i^{th}$ sample at time $t$.*

- *The classifier $\boldsymbol{X} \mapsto \mathrm{sign}(f(\boldsymbol{X}; \boldsymbol{v}_{t=2}, \boldsymbol{p}_{t=2}))$ correctly classifies all training data points:*

$$y_i = \mathrm{sign}(f(\boldsymbol{X}_i; \boldsymbol{v}_{t=2}, \boldsymbol{p}_{t=2})), \ \forall i \in [n].$$

- *The classifier $\boldsymbol{X} \mapsto \text{sign}(f(\boldsymbol{X}; \boldsymbol{v}_{t=2}, \boldsymbol{p}_{t=2}))$ generalizes well:*

$$\mathbb{P}_{(\boldsymbol{X},y) \sim \mathcal{D}}(y \neq \text{sign}(f(\boldsymbol{X}; \boldsymbol{v}_{t=2}, \boldsymbol{p}_{t=2}))) \leq \eta + \exp(-d/C_1 n^2),$$

  *where $C_1 := C_1(C_\rho, C_\beta)$ is a constant.*

We can also conclude that for the clean-labeled distribution $\mathcal{D}_{\text{clean}}$ we have

$$\mathbb{P}_{(\boldsymbol{X},y) \sim \mathcal{D}_{\text{clean}}}(y \neq \text{sign}(f(\boldsymbol{X}; \boldsymbol{v}_{t=2}, \boldsymbol{p}_{t=2}))) \leq \exp(-d/C_1 n^2),$$

which approaches zero as $d$ grows (see Assumption 25, item 2).

### A.1.5 ADDITIONAL DEFINITIONS & LEMMAS - MULTIPLE TOKENS.

**Definition 27** (Good Training Set). *We say that a training set $(\boldsymbol{X}_1, \ldots, \boldsymbol{X}_n)$ is* good *if*

- $\|\boldsymbol{\xi}_{i,\tau}\|_2^2 \in (1 \pm o_n(1))d$, *for all $i \in [n], \tau \in \{2, \ldots, T\}$.*

- $|\langle \boldsymbol{\xi}_{i,\tau}, \boldsymbol{\xi}_{j,\tau'} \rangle| \leq \sqrt{d \log(12n^2 T^2/\delta)}$, *for any $i, j \in [n], \tau, \tau' \in \{2, \ldots, T\}$ such that $(i, \tau) \neq (j, \tau')$.*

- $|\mathcal{N}_k| \in \frac{n}{2}(\eta \pm o_n(1))$ *and* $|\mathcal{C}_k| = \frac{n}{2}(1 - \eta \pm o_n(1))$, *for $k \in \{1, 2\}$.*

**Definition 28** (Good Test Sample). *We say that a test sample $(\boldsymbol{X} = (\boldsymbol{x}_1, \boldsymbol{x}_2, \ldots, \boldsymbol{x}_T), y)$ is* good *w.r.t. a training set $(\boldsymbol{X}_1, \ldots, \boldsymbol{X}_n)$ and constant $C_1$ if*

$$|\langle \boldsymbol{x}_{i,\tau}, \boldsymbol{x}_{\tau'} \rangle| \leq \frac{d}{C_1 n}, \quad \forall i \in [n], \tau, \tau' \in \{2, \ldots, T\}$$

Next we write Lemma 66 slightly different, and also add a formal proof for completeness:

Lemma 29 allows us to analyze $\nabla_{\boldsymbol{p}} \mathcal{L}$ as a function of the score gap.

**Lemma 29.** *Let $\boldsymbol{z}, \boldsymbol{\gamma}, \boldsymbol{p} \in \mathbb{R}^T$ and let $\boldsymbol{\alpha} = \mathbb{S}(\boldsymbol{p})$. Define $\gamma_{min} := \min_{\tau \geq 2} \gamma_\tau$, $\gamma_{max} := \max_{\tau \geq 2} \gamma_\tau$, $\gamma := (\gamma_{min} + \gamma_{max})/2$ and $\epsilon := (\gamma_{max} - \gamma_{min})/2$. Then*

$$\boldsymbol{z}^T \mathbb{S}'(\boldsymbol{p}) \boldsymbol{\gamma} \in (\gamma_1 - \gamma)(1 - \alpha_1)\alpha_1 \left( z_1 - \frac{\sum_{i=2}^T z_i \alpha_i}{1 - \alpha_1} \right) \pm \epsilon \left( 2 \sum_{i=2}^T z_i \alpha_i + \alpha_1 \sum_{i=2}^T z_i \alpha_i + (1 - \alpha_1)\alpha_1 z_1 \right)$$

*Proof.* Observe that $\sum_{i=1}^{T} \alpha_i = 1$. Therefore,

$$\boldsymbol{z}^T \mathbb{S}'(\boldsymbol{p})\boldsymbol{\gamma} = \boldsymbol{z}^T \text{diag}(\boldsymbol{\alpha})\boldsymbol{\gamma} - \boldsymbol{z}^T \boldsymbol{\alpha}\boldsymbol{\alpha}^\top \boldsymbol{\gamma} = \sum_{i=1}^{T} z_i \gamma_i \alpha_i - \sum_{i=1}^{T} z_i \alpha_i \sum_{i=1}^{T} \gamma_i \alpha_i$$

$$= z_1 \gamma_1 \alpha_1 + (\gamma \pm \epsilon) \sum_{i=2}^{T} z_i \alpha_i - \left( z_1 \alpha_1 + \sum_{i=2}^{T} z_i \alpha_i \right) \left( \gamma_1 \alpha_1 + (\gamma \pm \epsilon) \sum_{i=2}^{T} \alpha_i \right)$$

$$= \left( (\gamma \pm \epsilon) - \left( \alpha_1 \gamma_1 + (\gamma \pm \epsilon) \sum_{i=2}^{T} \alpha_i \right) \right) \sum_{i=2}^{T} z_i \alpha_i + \left( \gamma_1 - \left( \alpha_1 \gamma_1 + (\gamma \pm \epsilon) \sum_{i=2}^{T} \alpha_i \right) \right) \alpha_1 z_1$$

$$= ((\gamma \pm \epsilon) - (\alpha_1 \gamma_1 + (\gamma \pm \epsilon)(1 - \alpha_1))) \sum_{i=2}^{T} z_i \alpha_i + (\gamma_1 - (\alpha_1 \gamma_1 + (\gamma \pm \epsilon)(1 - \alpha_1))) \alpha_1 z_1$$

$$= (\alpha_1(\gamma \pm \epsilon) - \alpha_1 \gamma_1 \pm 2\epsilon) \sum_{i=2}^{T} z_i \alpha_i + (1 - \alpha_1)(\gamma_1 - \gamma \pm \epsilon)\alpha_1 z_1$$

$$= \alpha_1 (\gamma - \gamma_1) \sum_{i=2}^{T} z_i \alpha_i + (1 - \alpha_1)(\gamma_1 - \gamma)\alpha_1 z_1 \pm \epsilon \left( 2 \sum_{i=2}^{T} z_i \alpha_i + \alpha_1 \sum_{i=2}^{T} z_i \alpha_i + (1 - \alpha_1)\alpha_1 z_1 \right)$$

$$= (\alpha_1(\gamma \pm \epsilon) - \alpha_1 \gamma_1 \pm 2\epsilon) \sum_{i=2}^{T} z_i \alpha_i + (1 - \alpha_1)(\gamma_1 - \gamma \pm \epsilon)\alpha_1 z_1$$

$$= (\gamma_1 - \gamma)(1 - \alpha_1)\alpha_1 \left( z_1 - \frac{\sum_{i=2}^{T} z_i \alpha_i}{1 - \alpha_1} \right) \pm \epsilon \left( 2 \sum_{i=2}^{T} z_i \alpha_i + \alpha_1 \sum_{i=2}^{T} z_i \alpha_i + (1 - \alpha_1)\alpha_1 z_1 \right)$$

$\square$

We will show that in our setting the score difference between noisy tokens (i.e. $\epsilon$ from Lemma 29) is relatively small and thus the second term in Lemma 29 is negligible compare to the first term.

### A.1.6  PROOF OF THM. 26

*Proof of Thm. 26.* To simplify the proof, we will use the following assumption, which is slightly weaker than Assumption 25:

**Assumption 30** (Assumptions for GD with SNR $= \Theta(1/\sqrt{n})$)**.** *Let $\delta > 0$ be a desired probability of failure. For constants $c_T \geq 2, c_\rho \geq 6(T-1), c_\beta \geq 16T^3 c_\rho \log(c_\rho^2)$, there exists some large enough constant $C = C(c_\beta)$, such that the following hold:*

1. *Number of samples $n$ should be sufficiently large: $n \geq C \log(16/\delta)$*

2. *Dimension $d$ should be sufficiently large: $d \geq Cn^2 \log(12n^2 T^2/\delta)$.*

3. *Signal strength is: $\rho = c_\rho \sqrt{d/n}$*

4. *Label flipping rate $\eta$: $\eta \leq 1/C$.*

5. *The step size $\beta$ satisfies: $\beta = (c_\beta \cdot n)/(c_\rho^2 \cdot d)$.*

6. *Initialization at zero: $\|\boldsymbol{v}_0\| = \|\boldsymbol{p}_0\| = 0$.*

7. *The number of token $T$ satisfies: $2 \leq T \leq c_T$*

Apart from slight adjustments to the constants within the logarithm at items 1 and 2 (which can be absorbed into $C$), the only changes are $c_\beta \geq 16T^3 c_\rho \log(c_\rho^2)$ (instead of $C_\beta \geq 16T^3$) and $\beta = (c_\beta \cdot n)/(c_\rho^2 \cdot d)$ (instead of $\beta = C_\beta \cdot (n/d)$). Indeed, given $C_\beta \geq 16T^3, C_\rho \geq 6(T-1)$ and $\beta = C_\beta \cdot (n/d)$ which satisfy Assumption 25, define $c_\rho := C_\rho \geq 6(T-1), c_\beta := C_\beta c_\rho^2 \geq 16T^3 c_\rho \log(c_\rho^2)$,

which holds for any $c_\rho \geq 6(T-1)$. We also have that $\beta = C_\beta \cdot (n/d) = (c_\beta/c_\rho^2) \cdot (n/d)$ , i.e., $\beta, c_\rho, c_\beta$ satisfy Assumption 30.

Next, under Assumption 30, we argue that with probability at least $1 - \delta$ the training set is good (Def. 27) i.e.:

- $|\mathcal{C}_k| \in \frac{n}{2}(\eta \pm o_n(1))$ and $\mathcal{N}_k \in \frac{n}{2}(1 - \eta \pm o_n(1))$, for $k \in \{1, 2\}$.

- $\|\boldsymbol{\xi}_{i,\tau}\|_2^2 \in (1 \pm o_n(1))d$, for any $i \in [n], \tau \in \{2, \ldots, T\}$.

- $|\langle \boldsymbol{\xi}_{i,\tau}, \boldsymbol{\xi}_{j,\tau'} \rangle| \leq \sqrt{d \log(12n^2 T^2/\delta)}$, for any $i, j \in [n], \tau, \tau' \in \{2, \ldots, T\}$ such that $(i, \tau) \neq (j, \tau')$.

Indeed, this holds by Lemma 64, Lemma 21, and the union bound. We emphasize that the notation $o_n(1)$ represents a term that becomes arbitrarily small as $n$ increases, and thus it can be bounded by a small constant if $C$ from Assumption 1 is large enough.

Next, we show that under a good training set, the model exhibits benign overfitting, already after two iterations. See Remark 18 for the data setting used throughout the proof.

**GD after 1 iteration.** We start by analyzing the first coordinate of $\boldsymbol{v}_1$ (i.e. $\boldsymbol{v}$ after one iteration of GD). By assumption 30 (item 6), we have that $\boldsymbol{p}_0 = \boldsymbol{v}_0 = \boldsymbol{0}$, which implies that $\ell'_{0,i} = -1/2$, for any $i \in [n]$. Hence

$$-\beta \nabla_{\boldsymbol{v}} \mathcal{L}(\boldsymbol{v}_0, \boldsymbol{p}_0)[1] = -\frac{\beta}{Tn} \sum_{i=1}^n \ell'_{0,i} \cdot y_i \boldsymbol{x}_{i,1}[1] = \frac{\beta}{2Tn} \sum_{i \in \mathcal{C}_1} y_i \rho + \frac{\beta}{2Tn} \sum_{i \in \mathcal{N}_1} y_i \rho$$

$$= \frac{\beta}{2Tn}(|\mathcal{C}_1| - |\mathcal{N}_1|)\rho$$

$$\in \frac{\beta}{4T}(1 - 2\eta \pm o_n(1))\rho \qquad\qquad \text{"good" training set}$$

In the same way, we can estimate the second coordinate of $\boldsymbol{v}_{t=1}$:

$$\boldsymbol{v}_{t=1}[2] = \frac{\beta}{2Tn} \sum_{i \in \mathcal{C}_2} y_i \rho + \frac{\beta}{2Tn} \sum_{i \in \mathcal{N}_2} y_i \rho \in -\frac{\beta}{4T}(1 - 2\eta \pm o_n(1))\rho,$$

where we remind that $y_i = -1$, when $i \in \mathcal{C}_2$, hence $\boldsymbol{v}_{t=1}[2]$ has the same bounds as $\boldsymbol{v}_{t=1}[1]$, just with opposite sign. We move to analyze the rest of the coordinates of $\boldsymbol{v}_{t=1}$:

$$\boldsymbol{v}_t[3:d] = \frac{\beta}{2Tn} \sum_{i=1}^n y_i \sum_{\tau=2}^T \boldsymbol{\xi}_{i,\tau}.$$

Overall, we can write $\boldsymbol{v}_{t=1}$ as $\lambda_1^{t=1} \boldsymbol{\mu}_1 + \lambda_2^{t=1} \boldsymbol{\mu}_2 + \sum_{i=1}^n y_i \theta_i^{t=1} \sum_{\tau=2}^T \boldsymbol{\xi}_{i,\tau}$ with

$$\lambda_1^{t=1} \in \frac{\beta}{4T}(1 - 2\eta \pm o_n(1)), \quad \lambda_2^{t=1} \in -\frac{\beta}{4T}(1 - 2\eta \pm o_n(1)), \quad \theta_i^{t=1} = \frac{\beta}{2Tn}. \qquad (35)$$

Moreover, since $\boldsymbol{\gamma}_i^{t=0} = \boldsymbol{0}$ for every $i \in [n]$, we have that $\boldsymbol{p}_1 = \boldsymbol{0}$ (see Eq. 8).

**Preparation for next iteration.** To estimate $(\boldsymbol{v}_{t=2}, \boldsymbol{p}_{t=2})$, we first need to estimate the loss for clean/noisy samples and the score $\gamma_{i,\tau}$ (see Table 1).

We remind that $\|\boldsymbol{\mu}_j\|^2 = \rho^2 = c_\rho^2 d/n$ (Assumption 30 (item 3)). For $j \in \mathcal{C}_k$, where $k \in \{1, 2\}$ we have that

$$y_j f(\boldsymbol{X}_j; \boldsymbol{v}_{t=1}, \boldsymbol{p}_{t=1}) = \frac{1}{T} \cdot y_j \boldsymbol{v}_{t=1}^\top \sum_{\tau=1}^T \boldsymbol{x}_{j,\tau} \qquad\qquad \text{since } \boldsymbol{p}_1 = \boldsymbol{0}$$

$$\in \frac{1}{T}|\lambda_k^{t=1}| \|\boldsymbol{\mu}_k\|^2 + \frac{1}{T}\theta_j^{t=1} \sum_{\tau=1}^{T-1} \|\boldsymbol{\xi}_{j,\tau}\|^2 \pm \frac{\beta}{n}o(d) \qquad y_j \lambda_k^{t=1} > 0 \qquad (36)$$

where the last inequality holds since the training set is "good" and $T$ is a constant i.e.

$$\sum_{i,\tau,\tau':(i,\tau)\neq(j,\tau')} \boldsymbol{\xi}_{i,\tau}^{\top}\boldsymbol{\xi}_{j,\tau'} \in \pm o_n(1) \cdot d.$$

Since the training set is "good" then by Eq. 35, we can bound $y_j f(\boldsymbol{X}_j; \boldsymbol{v}_{t=1}, \boldsymbol{p}_{t=1})$ as follows:

$$
\begin{aligned}
y_j f(\boldsymbol{X}_j; \boldsymbol{v}_{t=1}, \boldsymbol{p}_{t=1}) &\leq \frac{\beta}{4T^2}(1 - 2\eta + o_n(1)) \cdot c_\rho^2 \cdot \frac{d}{n} + \frac{\beta(T-1)}{2T^2 n}d(1 + o_n(1)) + \frac{\beta}{n} \cdot o(d) \\
&\leq \left(\frac{c_\rho^2(1-2\eta) + 2(T-1) + o_n(1)}{4T^2}\right) \cdot \frac{\beta d}{n} \qquad \text{Assumption 30 (item 2)} \\
&= c_\beta \cdot \left(\frac{(1-2\eta) + 2(T-1)/c_\rho^2 + o_n(1)}{4T^2}\right) \qquad \text{Assumption 30 (item 5)} \\
&\leq \frac{1.1c_\beta}{4T^2},
\end{aligned}
\tag{37}
$$

where the last inequality holds since $c_\rho \geq 5(T-1)$, which implies that $2(T-1)/c_\rho^2 + o_n(1) \leq 0.1$. Similarly, we have that

$$
\begin{aligned}
y_j f(\boldsymbol{X}_j; \boldsymbol{v}_{t=1}, \boldsymbol{p}_{t=1}) &\geq \frac{\beta}{4T^2}(1 - 2\eta - o_n(1)) \cdot c_\rho^2 \cdot \frac{d}{n} + \frac{\beta}{2T^2 n}d(1 - o_n(1)) - \frac{\beta}{n}o(d) \\
&\geq \left(\frac{c_\rho^2(1-2\eta) + 2(T-1) - o_n(1)}{4T^2}\right) \cdot \frac{\beta d}{n} \\
&= c_\beta \cdot \left(\frac{(1-2\eta) + 2(T-1)/c_\rho^2 - o_n(1)}{4T^2}\right) \\
&\geq \frac{0.9c_\beta}{4T^2}
\end{aligned}
\tag{38}
$$

For $j \in \mathcal{N}_k$, where $k \in \{1, 2\}$ we have that

$$
\begin{aligned}
y_j f(\boldsymbol{X}_j; \boldsymbol{v}_{t=1}, \boldsymbol{p}_{t=1}) &= \frac{1}{T} \cdot y_j \boldsymbol{v}_{t=1}^{\top} \sum_{\tau=1}^{T} \boldsymbol{x}_{j,\tau} \qquad\qquad\qquad\qquad\qquad \text{since } \boldsymbol{p}_1 = \boldsymbol{0} \\
&\in -\frac{1}{T}|\lambda_k^{t=1}| \, \|\boldsymbol{\mu}_k\|^2 + \frac{1}{T}\theta_j^{t=1}\sum_{\tau=1}^{T-1} \|\boldsymbol{\xi}_{j,\tau}\|^2 \pm \frac{\beta}{n}o(d) \quad y_j \lambda_k^{t=1} > 0 \quad (39)
\end{aligned}
$$

where the last inequality holds since that the training set is "good" and $T$ is a constant. Since the training set is "good" then by Eq. 35, we can bound $y_j f(\boldsymbol{X}_j; \boldsymbol{v}_{t=1}, \boldsymbol{p}_{t=1})$ as follows:

$$
\begin{aligned}
y_j f(\boldsymbol{X}_j; \boldsymbol{v}_{t=1}, \boldsymbol{p}_{t=1}) &\leq -\frac{\beta}{4T^2}(1 - 2\eta - o_n(1)) \cdot c_\rho^2 \cdot \frac{d}{n} + \frac{\beta(T-1)}{2T^2 n}d(1 + o_n(1)) + \frac{\beta}{n} \cdot o(d) \\
&\leq \left(\frac{-c_\rho^2(1-2\eta) + 2(T-1) + o_n(1)}{4T^2}\right) \cdot \frac{\beta d}{n} \qquad \text{Assumption 30 (item 2)} \\
&= c_\beta \cdot \left(\frac{-(1-2\eta) + 2(T-1)/c_\rho^2 + o_n(1)}{4T^2}\right) \qquad \text{Assumption 30 (item 5)} \\
&\leq \frac{-0.9c_\beta}{4T^2},
\end{aligned}
\tag{40}
$$

where the last inequality holds since $c_\rho \geq 5(T-1)$, which implies that $2(T-1)/c_\rho^2 + 2\eta + o_n(1) \leq 0.1$. Similarly, we have that

$$
\begin{aligned}
y_j f(\boldsymbol{X}_j; \boldsymbol{v}_{t=1}, \boldsymbol{p}_{t=1}) &\geq -\frac{\beta}{4T^2}(1 - 2\eta + o_n(1)) \cdot c_\rho^2 \cdot \frac{d}{n} + \frac{\beta}{2T^2 n} d(1 - o_n(1)) - \frac{\beta}{n} o(d) \\
&\geq \left( \frac{-c_\rho^2(1 - 2\eta) + 2(T-1) - o_n(1)}{4T^2} \right) \cdot \frac{\beta d}{n} \\
&= c_\beta \cdot \left( \frac{-(1 - 2\eta) + 2(T-1)/c_\rho^2 - o_n(1)}{4T^2} \right) \\
&\geq \frac{-1.1 c_\beta}{4T^2}
\end{aligned} \tag{41}
$$

We remind that $-\ell'_{1,j} = 1/(1 + \exp(y_i f(\boldsymbol{X}_i; \boldsymbol{v}_{t=1}, \boldsymbol{p}_{t=1})))$ and that $\beta = c_\beta \cdot n/(dc_\rho^2)$ for some constant $c_\beta \geq 16 c_\rho$. Combine with Eqs. 37 and 38, we have that

$$
i \in \mathcal{C}, \quad -\ell'_{t=1,i} \geq 1/(1 + \exp(1.1 c_\beta/4T^2)) := m_{\mathcal{C}}^{t=1} > 0 \tag{42}
$$
$$
i \in \mathcal{C}, \quad -\ell'_{t=1,i} \leq 1/(1 + \exp(0.9 c_\beta/4T^2)) := M_{\mathcal{C}}^{t=1} \leq 1/(4(T-1)c_\rho^2), \tag{43}
$$

where the last inequality holds since $c_\beta \geq 4T^2(T-1)c_\rho$ and since $1 + \exp(0.9 c_\rho) \geq 4c_\rho^2$ for any $c_\rho \geq 6$.

Moreover, by Eqs. 40 and 41, we have that

$$
j \in \mathcal{N}, \quad -\ell'_{t=1,j} \geq 1/(1 + \exp(-0.9 c_\beta/4T^2)) := m_{\mathcal{N}}^{t=1} \geq 0.99 \tag{44}
$$
$$
j \in \mathcal{N}, \quad -\ell'_{t=1,j} \leq 1/(1 + \exp(-1.1 c_\beta/4T^2)) := M_{\mathcal{N}}^{t=1} \leq 1 \tag{45}
$$

The notations $M_{\mathcal{C}}^t$ and $m_{\mathcal{C}}^t$ ($M_{\mathcal{N}}^t$ and $m_{\mathcal{N}}^t$) denote the upper and lower bounds, respectively, on the derivative of the loss for clean (noisy) samples at time $t$, and we use them throughout the proof. We remind that $\gamma_{i,\tau}^t = y_i \boldsymbol{v}_t^\top \boldsymbol{x}_{i,\tau}$. Then by Eq. 35, for $i \in \mathcal{C}_k$ we have that

$$
\begin{aligned}
\gamma_{i,1}^{t=1} &\in \frac{\beta}{4T}(1 - 2\eta \pm o_n(1))\rho^2 = \frac{c_\beta}{4T}(1 - 2\eta \pm o_n(1)) \\
\gamma_{i,\tau}^{t=1} &\in \frac{\beta}{2Tn} \cdot d(1 \pm o_n(1)) = \frac{c_\beta}{2T}(1/c_\rho^2 \pm o_n(1)), \forall \tau \in \{2, \dots, T\} \\
\gamma_{i,1}^{t=1} - \gamma_{i,2}^{t=1} &\in \frac{c_\beta}{4T}(1 - 2/c_\rho^2 - 2\eta \pm o_n(1)) .
\end{aligned} \tag{46}
$$

where in the calculation of $\gamma_{i,2}^{t=1}$ we use $\sum_{i \in [n]} y_i y_j \theta_j^{t=1} \sum_{\tau \neq \tau'} \boldsymbol{\xi}_{i,\tau}^\top \boldsymbol{\xi}_{j,\tau'} = o_n(1) \cdot d$, since the training set is good. For $i \in \mathcal{N}_k$, we have that

$$
\begin{aligned}
\gamma_{i,1}^{t=1} &\in -\frac{\beta}{4T}(1 - 2\eta \pm o_n(1))\rho^2 = -\frac{c_\beta}{4T}(1 - 2\eta \pm o_n(1)) \\
\gamma_{i,\tau}^{t=1} &\in \frac{\beta}{2Tn} \cdot d(1 \pm o_n(1)) = \frac{c_\beta}{2T}(1/c_\rho^2 \pm o_n(1)), \forall \tau \in \{2, \dots, T\} \\
\gamma_{i,2}^{t=1} - \gamma_{i,1}^{t=1} &\in \frac{c_\beta}{4T}(1 + 2/c_\rho^2 - 2\eta \pm o_n(1)) .
\end{aligned} \tag{47}
$$

**GD after 2 iterations.**
**Analysis of $\boldsymbol{v}_{t=2}$.**
Observe that

$$
-\beta \nabla_{\boldsymbol{v}} \mathcal{L}(\boldsymbol{v}_1, \boldsymbol{p}_1) = -\frac{\beta}{n} \sum_{i=1}^n \ell'_{1,i} \cdot y_i \boldsymbol{X}_i^\top \mathbb{S}(X_i \boldsymbol{p}_1) = -\frac{\beta}{Tn} \sum_{i=1}^n \ell'_{1,i} \cdot y_i \sum_{\tau=1}^T \boldsymbol{x}_i.
$$

We start by analyzing the first coordinate of $\nabla_v \mathcal{L}(v_1, p_1)$.

$$-\beta \nabla_v \mathcal{L}(v_1, p_1)[1] = \frac{\beta}{Tn} \sum_{i \in \mathcal{C}_1} -\ell'_{1,i} \cdot y_i x_{i,1}[1] + \frac{\beta}{Tn} \sum_{i \in \mathcal{N}_1} -\ell'_{1,i} \cdot y_i x_{i,1}[1]$$

$$= \frac{\beta}{Tn} \sum_{i \in \mathcal{C}_1} -\ell'_{1,i} \cdot \rho - \frac{\beta}{Tn} \sum_{i \in \mathcal{N}_1} -\ell'_{1,i} \cdot \rho$$

$$= \frac{\beta}{Tn} \left( \sum_{i \in \mathcal{C}_1} -\ell'_{1,i} - \sum_{j \in \mathcal{N}_1} -\ell'_{1,j} \right) \cdot \rho . \tag{48}$$

Observe that

$$\sum_{i \in \mathcal{C}_k} -\ell'_{1,i} - \sum_{j \in \mathcal{N}_k} -\ell'_{1,j} \geq \frac{n}{2}(1 - \eta - o_n(1)) \cdot m_{\mathcal{C}} - \frac{n}{2}(\eta + o_n(1)) \cdot M_{\mathcal{N}} \quad \text{good training set}$$

$$> 0 \qquad\qquad\qquad\qquad \text{Eqs. 42 and 45,}$$

where the last inequality holds for small enough $\eta \leq 1/C$, where $C := C(c_\rho, c_\beta)$ (see Assumption 4). Substituting it into Eq. 48, we obtain that

$$-\beta \nabla_v \mathcal{L}(v_1, p_1)[1] > 0.$$

On the other hand, by Eq. 48, we can upper bound the first coordinate of the gradient of $v$ by

$$-\beta \nabla_v \mathcal{L}(v_1, p_1)[1] \leq \frac{\beta}{Tn} \left( \sum_{i \in \mathcal{C}_1} -\ell'_{1,i} \right) \cdot \rho$$

$$\leq \frac{\beta}{17T} \cdot \rho \qquad\qquad\qquad -\ell'_{1,i} < 1/17, \text{Eq. 43.}$$

Similarly, we can estimate the second coordinate of $\nabla_v \mathcal{L}(v_1, p_1)$:

$$0 \geq -\beta \nabla_v \mathcal{L}(v_1, p_1)[2] \geq -\frac{\beta}{17T} \cdot \rho.$$

Write $v_{t=2} = \lambda_1^{t=2} \mu_1 + \lambda_2^{t=2} \mu_2 + \sum_{i=1}^{n} y_i \theta_i^{t=2} \sum_{\tau=1}^{T-1} \xi_i$. Together with Eq. 35, we get that

$$\lambda_1^{t=2} = \lambda_1^{t=1} - \beta \nabla_v \mathcal{L}(v_1, p_1)[1]/\rho \leq \frac{\beta}{4T}(1 + o_n(1)) + \frac{\beta}{17T} \leq \frac{5\beta}{16T} \tag{49}$$

$$\lambda_1^{t=2} \geq \lambda_1^{t=1} \geq \frac{\beta}{4T}(1 - 2\eta - o_n(1)) \tag{50}$$

$$\lambda_2^{t=2} = \lambda_2^{t=1} - \beta \nabla_v \mathcal{L}(v_1, p_1)[2] \geq -\frac{\beta}{4T}(1 + o_n(1)) - \frac{\beta}{17T} \geq -\frac{5\beta}{16} \tag{51}$$

$$\lambda_2^{t=2} \leq \lambda_2^{t=1} \leq -\frac{\beta}{4T}(1 - 2\eta - o_n(1)) . \tag{52}$$

Next, we analyze the rest of the coordinates of $\nabla_v \mathcal{L}(v_1, p_1)$.

$$-\beta \nabla_v \mathcal{L}(v_1, p_1)[3:d] = \frac{\beta}{Tn} \sum_{i \in \mathcal{C}} -\ell'_{1,i} \cdot y_i \sum_{\tau=2}^{T} \xi_i + \frac{\beta}{Tn} \sum_{j \in \mathcal{N}} -\ell'_{1,j} \cdot y_j \sum_{\tau=2}^{T} \xi_j,$$

and use it to analyze the coefficients of the noise (second) tokens in $v_{t=2}$, i.e., $\theta_i^{t=2}$. Indeed, for $i \in \mathcal{C}$ we have that

$$\theta_i^{t=2} = \theta_i^{t=1} - \frac{\beta}{Tn} \ell'_{1,i} = \frac{\beta}{Tn}(-\ell'_{1,i} + 0.5) \qquad\qquad \text{Eq. 35}$$

$$\in \left[ \frac{\beta}{Tn}(m_{\mathcal{C}} + 0.5), \frac{\beta}{Tn}(M_{\mathcal{C}} + 0.5) \right] . \tag{53}$$

For $j \in \mathcal{N}$ we have that

$$\theta_j^{t=2} = \theta_j^{t=1} - \frac{\beta}{Tn} \ell'_{1,j} = \frac{\beta}{Tn}(-\ell'_{1,j} + 0.5) \qquad\qquad \text{Eq. 35}$$

$$\in \left[ \frac{\beta}{Tn}(m_{\mathcal{N}} + 0.5), \frac{\beta}{Tn}(M_{\mathcal{N}} + 0.5) \right] . \tag{54}$$

Now we move to analyze $\boldsymbol{p}_{t=2}$ and show that $\boldsymbol{p}_{t=2}$ focuses on optimal tokens for training samples.

$\boldsymbol{p}_{t=2}$ **focuses on optimal tokens**. Define $\gamma_{i,min} := \min_{\tau \geq 2} \gamma_{i,\tau}$, $\gamma_{i,max} := \max_{\tau \geq 2} \gamma_{i,\tau}$, $\gamma_i := (\gamma_{min} + \gamma_{max})/2$ and $\epsilon_i := (\gamma_{max} - \gamma_{min})/2$. By Eqs. 46 and 47 we have that $\epsilon_i = o_n(1)$ for any $i \in [n]$. Observe that $\boldsymbol{p}_2 = -\beta \nabla_{\boldsymbol{p}} \mathcal{L}(\boldsymbol{v}_1, \boldsymbol{p}_1)$. Therefore, for $j \in \mathcal{C}_k$ and any $\tau \in \{1, 2, \ldots, T\}$ we have:

$$\boldsymbol{p}_2^\top (\boldsymbol{x}_{j,1} - \boldsymbol{x}_{j,\tau})$$

$$= -(\boldsymbol{x}_{j,1} - \boldsymbol{x}_{j,\tau})^\top \beta \nabla_{\boldsymbol{p}} \mathcal{L}(\boldsymbol{v}_t, \boldsymbol{p}_t) = (\boldsymbol{x}_{j,1} - \boldsymbol{x}_{j,\tau})^\top \frac{\beta}{n} \sum_{i=1}^n -\ell'_{1,i} \cdot \boldsymbol{X}_i^\top \mathbb{S}'(X_i \boldsymbol{p}_t) \boldsymbol{\gamma}_i^{t=1}$$

$$= \frac{\beta}{n} \sum_{i=1}^n -\ell'_{1,i} \cdot \boldsymbol{x}_{j,1}^\top \boldsymbol{X}_i^\top \mathbb{S}'(X_i \boldsymbol{p}_t) \boldsymbol{\gamma}_i^{t=1} - \frac{\beta}{n} \sum_{i=1}^n -\ell'_{1,i} \cdot \boldsymbol{x}_{j,\tau}^\top \boldsymbol{X}_i^\top \mathbb{S}'(X_i \boldsymbol{p}_t) \boldsymbol{\gamma}_i^{t=1}. \qquad (55)$$

Write $\boldsymbol{z}_{1,i,j} := \boldsymbol{X}_i \boldsymbol{x}_{j,1}$. Observe that $\boldsymbol{z}_{1,i,j} = (\boldsymbol{x}_i^\top \boldsymbol{x}_j, 0, \ldots, 0)$ for $i \in \mathcal{C}_k \cup \mathcal{N}_k$ and $\boldsymbol{z}_{1,i,j} = \boldsymbol{0}$ otherwise. By Lemma 29, we can lower bound the first term in Eq. 55 as

$$\frac{\beta}{n} \sum_{i=1}^n -\ell'_{1,i} \cdot \boldsymbol{z}_{1,i,j}^\top \mathbb{S}'(X_i \boldsymbol{p}_t) \boldsymbol{\gamma}_i^{t=1} \geq \frac{\beta}{n} \sum_{i \in \mathcal{C}_k} -\ell'_{1,i} \cdot (\gamma_{i,1}^{t=1} - \gamma_i^{t=1})(1 - \alpha_{i,1}^{t=1})\alpha_{i,1}^{t=1}(1 - o_n(1))\boldsymbol{x}_i^\top \boldsymbol{x}_j$$

$$- \frac{\beta}{n} \sum_{i \in \mathcal{N}_k} -\ell'_{1,i} \cdot (\gamma_i^{t=1} - \gamma_{i,1}^{t=1})(1 - \alpha_{i,1}^{t=1})\alpha_{i,1}^{t=1}(1 - o_n(1))\boldsymbol{x}_{i,1}^\top \boldsymbol{x}_{j,1},$$

where the $(1 - o_n(1))$ term is from the second tern in Lemma 29 and since $\epsilon_i = o_n(1)$. Now we move to the second term of Eq. 55. Write $\boldsymbol{z}_{\tau,i,j} := \boldsymbol{X}_i \boldsymbol{x}_{j,\tau}$. Observe that $\boldsymbol{X}_i \boldsymbol{x}_{j,\tau} = (0, \boldsymbol{x}_{i,1}^\top \boldsymbol{x}_{j,\tau}, \ldots, \boldsymbol{x}_{i,\tau}^\top \boldsymbol{x}_{j,\tau}, \ldots, \boldsymbol{x}_{i,T}^\top \boldsymbol{x}_{j,\tau})$. By Lemma 29, we can lower bound the second term in Eq. 55 as

$$-\frac{\beta}{n} \sum_{i=1}^n -\ell'_{1,i} \cdot \boldsymbol{z}_{\tau,i,j}^\top \mathbb{S}'(X_i \boldsymbol{p}_t) \boldsymbol{\gamma}_i^{t=1}$$

$$= \frac{\beta}{(T-1)n} \sum_{i=1}^n -\ell'_{1,i} \cdot (\gamma_i^{t=1} - \gamma_{i,1}^{t=1})(1 - \alpha_{i,1}^{t=1})\alpha_{i,1}^{t=1}(1 - o_n(1)) \sum_{\tau'=1}^T \boldsymbol{x}_{i,\tau'}^\top \boldsymbol{x}_{j,\tau}$$

$$\geq \frac{\beta}{(T-1)n}(-\ell'_{1,j}) \cdot (\gamma_{j,1}^{t=1} - \gamma_j^{t=1})(1 - \alpha_{j,1}^{t=1})\alpha_{j,1}^{t=1}(1 - o_n(1)) \left( \|\boldsymbol{x}_{j,\tau}\|^2 + \sum_{\tau' \neq \tau}^T \boldsymbol{x}_{j,\tau'}^\top \boldsymbol{x}_{j,\tau} \right)$$

$$- \frac{\beta}{(T-1)n} \sum_{i \in n: i \neq j} -\ell'_{1,i} \cdot |\gamma_i^{t=1} - \gamma_{i,1}^{t=1}| \cdot (1 - \alpha_{i,1}^{t=1})\alpha_{i,1}^{t=1}(1 + o_n(1)) \sum_{\tau'=2}^T \boldsymbol{x}_{i,\tau'}^\top \boldsymbol{x}_{j,\tau}$$

Overall,

$$\boldsymbol{p}_2^\top (\boldsymbol{x}_{j,1} - \boldsymbol{x}_{j,\tau})$$

$$\geq \frac{\beta}{n}(-\ell'_{1,j})(\gamma_{j,1}^{t=1} - \gamma_j^{t=1})(1 - \alpha_{j,1})\alpha_{j,1}(1 - o_n(1)) \left( \|\boldsymbol{x}_{j,1}\|^2 + \|\boldsymbol{x}_{j,\tau}\|^2 /(T-1) \right)$$

$$+ \frac{\beta}{n} \sum_{i \in \mathcal{C}_k: i \neq j} -\ell'_{1,i} \cdot (\gamma_{i,1}^{t=1} - \gamma_i^{t=1})(1 - \alpha_{i,1}^{t=1})(1 - o_n(1))\alpha_{i,1}^{t=1}(\boldsymbol{x}_{j,1}^\top \boldsymbol{x}_{i,1})$$

$$- \frac{\beta}{n} \sum_{i \in \mathcal{N}_k: i \neq j} -\ell'_{1,i} \cdot (\gamma_i^{t=1} - \gamma_{i,1}^{t=1})(1 - \alpha_{i,1}^{t=1})\alpha_{i,1}^{t=1}(1 - o_n(1))(\boldsymbol{x}_{j,1}^\top \boldsymbol{x}_{i,1})$$

$$- \frac{\beta}{(T-1)n} \sum_{i \in [n]: i \neq j} \sum_{\tau':} -\ell'_{1,i} \cdot |\gamma_{i,1}^{t=1} - \gamma_{i,2}^{t=1}| \cdot (1 - \alpha_{i,1}^{t=1})\alpha_{i,1}^{t=1}(1 + o_n(1)) \sum_{\tau'=2}^T (\boldsymbol{x}_{j,\tau}^\top \boldsymbol{x}_{i,\tau'}).$$

Observe that $\alpha_{i,1}^{t=1} = 1/T$ and that $(1 - \alpha_{i,1})\alpha_{i,1} = (T-1)/T^2$ for any $i \in [n]$. In Eqs. 46 and 47 we calculate the score (e.g. $\gamma_{i,\tau}^{t=1}$). Overall, we can lower bound the above equation by:

$$
\geq \frac{\beta}{T^2 n} \left( m_{\mathcal{C}} \cdot \frac{c_\beta}{4T}(1 - 2/c_\rho^2 - 2\eta - o_n(1)) \cdot d(1 - o_n(1)) \right)
$$
$$
+ \frac{(T-1)\beta}{T^2 n} \left( (1 - \eta - o_n(1)) \cdot \frac{n}{2} \cdot m_{\mathcal{C}} \frac{c_\beta}{4T}(1 - 2/c_\rho^2 - 2\eta - o_n(1)) \frac{d}{n} c_\rho^2 \right)
$$
$$
- \frac{(T-1)\beta}{T^2 n} \left( (\eta + o_n(1)) \cdot \frac{n}{2} \cdot M_{\mathcal{N}} \frac{c_\beta}{4T}(1 + 2/c_\rho^2 - 2\eta + o_n(1)) \frac{d}{n} c_\rho^2 \right)
$$
$$
- \frac{(T-1)\beta}{T^2 n} \left( n \cdot M_{\mathcal{N}} \frac{c_\beta}{4T}(1 + 2/c_\rho^2 - 2\eta + o_n(1)) \sqrt{d \log(12n^2 T^2/\delta)} \right).
$$

The first term dominates the last term since $d \gg n\sqrt{d \log(12n^2/\delta)}$ (see Assumption 30 (item 2)). The second term dominates the third term for small enough $\eta$ (see Assumption 4). Overall, we obtain that for any $\tau \in \{2, \dots, T\}$ that

$$
\boldsymbol{p}_2^\top (\boldsymbol{x}_{j,1} - \boldsymbol{x}_{j,\tau}) > 0, \tag{56}
$$

which means that for any $i \in \mathcal{C}$ we have:

$$
\alpha_{i,1}^{t=2} = \frac{1}{1 + \sum_{\tau=2}^T \exp(\boldsymbol{p}_2^\top (\boldsymbol{x}_{j,\tau} - \boldsymbol{x}_{j,1}))} > \frac{1}{T}. \tag{57}
$$

Similarly, for any $j \in \mathcal{N}_k$ and $\tau \in \{2, \dots, T\}$,

$$
\boldsymbol{p}_2^\top (\boldsymbol{x}_{j,\tau} - \boldsymbol{x}_{j,1})
$$
$$
\geq \frac{\beta}{n}(-\ell'_{1,j})(\gamma_j^{t=1} - \gamma_{j,1}^{t=1})(1 - \alpha_{j,1})\alpha_{j,1}(1 - o_n(1)) \left( \|\boldsymbol{x}_{j,1}\|^2 + \|\boldsymbol{x}_{j,\tau}\|^2 /(T-1) \right)
$$
$$
- \frac{\beta}{n} \sum_{i \in \mathcal{C}_k : i \neq j} -\ell'_{1,i} \cdot (\gamma_{i,1}^{t=1} - \gamma_i^{t=1})(1 - \alpha_{i,1}^{t=1})(1 + o_n(1))\alpha_{i,1}^{t=1}(\boldsymbol{x}_{j,1}^\top \boldsymbol{x}_{i,1})
$$
$$
+ \frac{\beta}{n} \sum_{i \in \mathcal{N}_k : i \neq j} -\ell'_{1,i} \cdot (\gamma_i^{t=1} - \gamma_{i,1}^{t=1})(1 - \alpha_{i,1}^{t=1})\alpha_{i,1}^{t=1}(1 - o_n(1))(\boldsymbol{x}_{j,1}^\top \boldsymbol{x}_{i,1})
$$
$$
- \frac{\beta}{(T-1)n} \sum_{i \in [n] : i \neq j} \sum_{\tau' :} -\ell'_{1,i} \cdot |\gamma_{i,1}^{t=1} - \gamma_{i,2}^{t=1}| \cdot (1 - \alpha_{i,1}^{t=1})\alpha_{i,1}^{t=1}(1 + o_n(1)) \sum_{\tau'=2}^T (\boldsymbol{x}_{j,\tau}^\top \boldsymbol{x}_{i,\tau'}).
$$

Observe that $\alpha_{i,1}^{t=1} = 1/T$ and that $(1 - \alpha_{i,1})\alpha_{i,1} = (T-1)/T^2$ for any $i \in [n]$. In Eqs. 46 and 47 we calculate the score (e.g. $\gamma_{i,\tau}^{t=1}$). Overall, we can lower bound the above equation by:

$$
\geq \frac{\beta}{T^2 n} \left( m_{\mathcal{N}} \cdot \frac{c_\beta}{4T}(1 + 2/c_\rho^2 - 2\eta - o_n(1)) \cdot d(1 - o_n(1)) \right)
$$
$$
- \frac{(T-1)\beta}{T^2 n} \left( (1 - \eta - o_n(1)) \cdot \frac{n}{2} \cdot M_{\mathcal{C}} \frac{c_\beta}{4T}(1 - 2/c_\rho^2 - 2\eta - o_n(1)) \frac{d}{n} c_\rho^2 \right)
$$
$$
+ \frac{(T-1)\beta}{T^2 n} \left( |\mathcal{N}_k| \cdot m_{\mathcal{N}} \frac{c_\beta}{4T}(1 + 2/c_\rho^2 - 2\eta + o_n(1)) \frac{d}{n} c_\rho^2 \right)
$$
$$
- \frac{(T-1)\beta}{T^2 n} \left( n \cdot M_{\mathcal{N}} \frac{c_\beta}{4T}(1 + 2/c_\rho^2 - 2\eta + o_n(1)) \sqrt{d \log(12n^2 T^2/\delta)} \right).
$$

Observe that the third term is non-negative. Moreover, we argue that the first term is at least twice the sum of the second and last terms. Indeed, enough to show that

$$
\left( m_{\mathcal{N}} \cdot (1 + 2/c_\rho^2 - 2\eta - o_n(1)) \cdot d(1 - o_n(1)) \right) \geq
$$
$$
2(T-1)\left( (1 + o_n(1)) \cdot \frac{1}{2} \cdot M_{\mathcal{C}} \cdot dc_\rho^2 \right) + 2(T-1)\left( n \cdot M_{\mathcal{N}}(1 + 2/c_\rho^2 + o_n(1)) \sqrt{d \log(12n^2 T^2/\delta)} \right),
$$

which indeed holds since $n\sqrt{d\log(12n^2/\delta)} = d \cdot o_n(1)$, and $M_{\mathcal{C}} \cdot c_\rho^2 \leq 0.25/(T-1)$ while $m_{\mathcal{N}} \geq 0.99$ (see Eqs. 44 and 43). Overall, for any $i \in \mathcal{N}$ we have that:

$$\boldsymbol{p}_2^\top (\boldsymbol{x}_{j,\tau} - \boldsymbol{x}_{j,1}) \geq \frac{\beta}{2T^2 n} \left( m_{\mathcal{N}} \cdot \frac{c_\beta}{4T}(1 + 2/c_\rho^2 - 2\eta - o_n(1)) \cdot d(1 - o_n(1)) \right)$$

$$= \frac{c_\beta^2}{8T^3 c_\rho^2} \left( m_{\mathcal{N}} \cdot (1 + 2/c_\rho^2 - 2\eta - o_n(1)) \cdot (1 - o_n(1)) \right)$$

$$\geq 2\log(c_\rho),$$

where that last inequality holds since $c_\beta \geq 4T^2 c_\rho \log(c_\rho)$, which implies that $0.9c_\beta^2/8T^3 c_\rho^2 \geq 2\log(c_\rho) = \log(c_\rho^2)$. We conclude that,

$$\alpha_{i,1}^{t=2} = \frac{1}{1 + \sum_{\tau=2}^T \exp(\boldsymbol{p}_2^\top (\boldsymbol{x}_{j,\tau} - \boldsymbol{x}_{j,1}))} \leq \frac{1}{1 + (T-1)\exp(\log(c_\rho^2))} = \frac{1}{1 + (T-1)c_\rho^2}$$

$$\leq \frac{1}{(T-1)c_\rho^2}. \tag{58}$$

We conclude that for any $j \in \mathcal{N}$ we have that

$$\alpha_{j,1}^{t=2} \leq \frac{1}{(T-1)c_\rho^2}, \quad \sum_{\tau=2}^T \alpha_{j,\tau}^{t=2} \geq 1 - \frac{1}{(T-1)c_\rho^2}. \tag{59}$$

Together with Eq. 57, this proves the first part of the Thm.

**The classifier $\text{sign}(f(\boldsymbol{X}; \boldsymbol{v}_{t=2}, \boldsymbol{p}_{t=2}))$ classifies correctly clean training samples.** Let $(\boldsymbol{X}_j = (\boldsymbol{x}_{j,1}, \ldots, \boldsymbol{x}_{j,T}), y_j)$ for $j \in \mathcal{C}$. We remind that $\boldsymbol{x}_{j,1} = \boldsymbol{\mu}_k$ for $k \in \{1, 2\}$ and $\boldsymbol{x}_{j,\tau} = \boldsymbol{\xi}_{j,\tau}$. we have that,

$$f(\boldsymbol{X}_j; \boldsymbol{v}_{t=2}, \boldsymbol{p}_{t=2}) = \alpha_{j,1}^{t=2} \boldsymbol{v}_2^\top \boldsymbol{x}_{j,1} + \sum_{\tau=2}^T \alpha_{j,\tau}^{t=2} \boldsymbol{v}_2^\top \boldsymbol{x}_{j,\tau},$$

and it suffices to prove that

$$y_j(f(\boldsymbol{X}_j; \boldsymbol{v}_2, \boldsymbol{p}_2)) > 0.$$

Indeed,

$$y_j f(\boldsymbol{X}_j; \boldsymbol{v}, \boldsymbol{p}) = y_j \alpha_{j,1}^{t=2} \boldsymbol{v}_2^\top \boldsymbol{x}_{j,1} + y_j \sum_{\tau=2}^T \alpha_{j,\tau}^{t=2} \boldsymbol{v}_2^\top \boldsymbol{x}_{j,\tau}$$

$$= \alpha_{j,1}^{t=2} |\lambda_k| \, \|\boldsymbol{\mu}_k\|^2 + \sum_{\tau=2}^T \alpha_{j,\tau}^{t=2} \theta_j \, \|\boldsymbol{\xi}_{j,\tau}\|^2 + \sum_{\tau=2}^T \alpha_{j,\tau}^{t=2} y_j \sum_{i\in[n],\tau':i\neq j \vee \tau\neq\tau'} y_i \theta_i^{t=2} \boldsymbol{\xi}_{i,\tau}^\top \boldsymbol{\xi}_{j,\tau'} \quad y_j\lambda_k > 0$$

$$\geq \alpha_{j,1}^{t=2} |\lambda_k| \, \|\boldsymbol{\mu}_k\|^2 - \sum_{\tau=2}^{T-1} \alpha_{j,\tau}^{t=2} n(T-1)^2 \max_i |\theta_i| \sqrt{d\log(12n^2/\delta)}$$

$$\geq \alpha_{j,1}^{t=2} \left( \frac{\beta}{4T+1} \right) \frac{d}{n} c_\rho^2 - \sum_{\tau=2}^{T-1} \alpha_{j,\tau}^{t=2} n(T-1) \frac{\beta}{Tn}(M_{\mathcal{N}} + 0.5)\sqrt{d\log(12n^2T^2/\delta)} \quad \text{Eqs. 54, 50 and 52}$$

$$\geq \frac{1}{T}\left( \frac{\beta}{4T+1} \right) \frac{d}{n} c_\rho^2 - \left( 1 - \frac{1}{T} \right) n(T-1) \frac{\beta}{Tn}(M_{\mathcal{N}} + 0.5)\sqrt{d\log(12n^2T^2/\delta)} \quad \text{Eq. 57}$$

$$> 0, \quad d \gg n\sqrt{d\log(12n^2T^2/\delta)}$$

as required.

**The classifier $\text{sign}(f(\boldsymbol{X}; \boldsymbol{v}_{t=2}, \boldsymbol{p}_{t=2}))$ classifies correctly noisy training samples.** Let $(\boldsymbol{X}_j = (\boldsymbol{x}_{j,1}, \ldots, \boldsymbol{x}_{j,T}), y_j)$ for $j \in \mathcal{N}$. We remind that $\boldsymbol{x}_{j,1} = \boldsymbol{\mu}_k$ for $k \in \{1, 2\}$ and $\boldsymbol{x}_{j,\tau} = \boldsymbol{\xi}_{j,\tau}$. we have that,

$$f(\boldsymbol{X}_j; \boldsymbol{v}_{t=2}, \boldsymbol{p}_{t=2}) = \alpha_{j,1}^{t=2} \boldsymbol{v}_2^\top \boldsymbol{x}_{j,1} + \sum_{\tau=2}^T \alpha_{j,\tau}^{t=2} \boldsymbol{v}_2^\top \boldsymbol{x}_{j,\tau},$$

and it suffices to prove that

$$y_j(f(\boldsymbol{X}_j; \boldsymbol{v}_2, \boldsymbol{p}_2)) > 0.$$

Indeed,

$$y_j f(\boldsymbol{X}_j; \boldsymbol{v}, \boldsymbol{p}) = y_j \alpha_{j,1}^{t=2} \boldsymbol{v}_2^\top \boldsymbol{x}_{j,1} + y_j \sum_{\tau=2}^{T} \alpha_{j,\tau}^{t=2} \boldsymbol{v}_2^\top \boldsymbol{x}_{j,\tau}$$

$$= -\alpha_{j,1}^{t=2} |\lambda_k| \|\boldsymbol{\mu}_k\|^2 + \sum_{\tau=2}^{T} \alpha_{j,\tau}^{t=2} \theta_j \|\boldsymbol{\xi}_{j,\tau}\|^2 + \sum_{\tau=2}^{T} \alpha_{j,\tau}^{t=2} y_j \sum_{i \in [n], \tau': i \neq j \vee \tau \neq \tau'} y_i \theta_i^{t=2} \boldsymbol{\xi}_{i,\tau}^\top \boldsymbol{\xi}_{j,\tau'} \quad y_j \lambda_k < 0$$

$$\geq -\alpha_{j,1}^{t=2} \left( \frac{5\beta}{16T} \right) \frac{d}{n} c_\rho^2 + \sum_{\tau=2}^{T} \alpha_{j,\tau}^{t=2} \frac{\beta}{Tn} (m_\mathcal{N} + 0.5) d(1 - o_n(1))$$

$$- \sum_{\tau=2}^{T-1} \alpha_{j,\tau}^{t=2} n(T-1) \frac{\beta}{Tn} (M_\mathcal{N} + 0.5) \sqrt{d \log(12n^2 T^2/\delta)} \qquad \text{Eqs. 54, 49 and 51}$$

$$\geq \left( \frac{1}{c_\rho^2} \right) \left( \frac{5\beta}{16T} \right) \frac{d}{n} c_\rho^2 + \left( 1 - \frac{1}{c_\rho^2} \right) \frac{\beta}{Tn} (m_\mathcal{N} + 0.5) d(1 - o_n(1)$$

$$- n(T-1) \frac{\beta}{Tn} (M_\mathcal{N} + 0.5) \sqrt{d \log(12n^2 T^2/\delta)} \qquad \text{Eq. 59}$$

$$> 0, \qquad d \gg n\sqrt{d \log(12n^2 T^2/\delta)}$$

as required.

**The classifier** $\text{sign}(f(\boldsymbol{X}; \boldsymbol{v}_{t=2}, \boldsymbol{p}_{t=2}))$ **classifies correctly clean test samples.**

Let $(\boldsymbol{X} = (\boldsymbol{x}_1, \ldots, \boldsymbol{x}_T), y)$ be a fresh clean sample i.e. $(\boldsymbol{X}, y) \sim \mathcal{D}_{\text{clean}}$. Observe that $\boldsymbol{x}_1 = \boldsymbol{\mu}_k$ for some $k \in \{1, 2\}$ and $y = 1$ iff $k = 1$. By Remark 65, with probability at least $1 - 6n \exp(-d/4C_1 n^2)$ for some constant $C_1 = C_1(c_\rho, c_\beta)$ that will be chosen later, we have that $(\boldsymbol{X}, y)$ is a good test sample w.r.t. $C_1$ (Def. 28). We work under the event that $(\boldsymbol{X}, y)$ is a good test sample and show that $y = \text{sign}(f(\boldsymbol{X}; \boldsymbol{v}_{t=2}, \boldsymbol{p}_{t=2})$. Recall that $\boldsymbol{p}_2 = -\beta \nabla_{\boldsymbol{p}} \mathcal{L}(\boldsymbol{v}_1, \boldsymbol{p}_1)$ and therefore (similar to the clean sample case) for any $\tau \in \{2, \ldots, T\}$:

$$\boldsymbol{p}_2^\top (\boldsymbol{x}_{j,1} - \boldsymbol{x}_{j,\tau})$$

$$\geq \frac{\beta}{n} \sum_{i \in \mathcal{C}_k} -\ell_{1,i}' \cdot (\gamma_{i,1}^{t=1} - \gamma_i^{t=1})(1 - \alpha_{i,1}^{t=1}) \alpha_{i,1}^{t=1} (1 - o_n(1))(\boldsymbol{x}_{j,1}^\top \boldsymbol{x}_{i,1})$$

$$- \frac{\beta}{n} \sum_{i \in \mathcal{N}_k} -\ell_{1,i}' \cdot (\gamma_i^{t=1} - \gamma_{i,1}^{t=1})(1 - \alpha_{i,1}^{t=1}) \alpha_{i,1}^{t=1} (1 - o_n(1))(\boldsymbol{x}_{j,1}^\top \boldsymbol{x}_{i,1})$$

$$- \frac{\beta}{(T-1)n} \sum_{i \in [n]} \sum_{\tau=2}^{T} -\ell_{1,i}' \cdot |\gamma_{i,1}^{t=1} - \gamma_i^{t=1}| \cdot (1 - \alpha_{i,1}^{t=1}) \alpha_{i,1}^{t=1} (1 + o_n(1)) \sum_{\tau'=2}^{T} (\boldsymbol{x}_{j,\tau}^\top \boldsymbol{x}_{i,\tau'}) .$$

Observe that $\alpha_{i,1}^{t=1} = 1/T$ and that $(1 - \alpha_{i,1}) \alpha_{i,1} = (T-1)/T^2$ for any $i \in [n]$. In Eqs. 46 and 47 we calculate the score (e.g. $\gamma_{i,\tau}^{t=1}$). Overall, we can lower bound the above equation by:

$$+ \frac{(T-1)\beta}{T^2 n} \left( (1 - \eta - o_n(1)) \cdot \frac{n}{2} \cdot m_\mathcal{C} \frac{c_\beta}{4T} (1 - 2/c_\rho^2 - 2\eta - o_n(1)) \frac{d}{n} c_\rho^2 \right)$$

$$- \frac{(T-1)\beta}{T^2 n} \left( (\eta + o_n(1)) \cdot \frac{n}{2} \cdot M_\mathcal{N} \frac{c_\beta}{4T} (1 + 2/c_\rho^2 - 2\eta + o_n(1)) \frac{d}{n} c_\rho^2 \right)$$

$$- \frac{(T-1)\beta}{T^2 n} \left( n \cdot M_\mathcal{N} \frac{c_\beta}{4T} (1 + 2/c_\rho^2 - 2\eta + o_n(1)) \frac{d}{C_1 n} \right) .$$

Once again, the first term dominates the last two terms when $C_1$ is large enough and when $\eta$ is small enough. Overall, for any $\tau \in \{2, \ldots, T\}$ we have

$$\boldsymbol{p}_2^\top (\boldsymbol{x}_{j,1} - \boldsymbol{x}_{j,\tau}) > 0. \tag{60}$$

This means that the softmax probability of the first token is::

$$\alpha_{i,1}^{t=2} = \frac{1}{1 + \sum_{\tau=2}^{T} \exp(\boldsymbol{p}_2^\top (\boldsymbol{x}_{j,\tau} - \boldsymbol{x}_{j,1}))} > \frac{1}{T} \ . \tag{61}$$

Let $\boldsymbol{x}_{j,1} = \boldsymbol{\mu}_k$ for $k \in \{1, 2\}$ and $\boldsymbol{x}_\tau = \boldsymbol{\xi}_\tau$ for $\tau \in \{2, \dots, T\}$. We have that,

$$f(\boldsymbol{X}; \boldsymbol{v}_{t=2}, \boldsymbol{p}_{t=2}) = \alpha_1^{t=2} \boldsymbol{v}_2^\top \boldsymbol{x}_1 + \sum_{\tau=2}^{T} \alpha_\tau^{t=2} \boldsymbol{v}_2^\top \boldsymbol{x}_\tau,$$

and it suffices to prove that

$$y(f(\boldsymbol{X}; \boldsymbol{v}_2, \boldsymbol{p}_2)) > 0.$$

Indeed,

$$
\begin{aligned}
y f(\boldsymbol{X}_j; \boldsymbol{v}, \boldsymbol{p}) &= y_j \alpha_1^{t=2} \boldsymbol{v}_2^\top \boldsymbol{x}_1 + y \sum_{\tau=2}^{T} \alpha_\tau^{t=2} \boldsymbol{v}_2^\top \boldsymbol{x}_\tau \\
&= \alpha_1^{t=2} |\lambda_k| \, \|\boldsymbol{\mu}_k\|^2 + \sum_{\tau=2}^{T} \alpha_\tau^{t=2} y \sum_{i \in [n], \tau'} y_i \theta_i^{t=2} \boldsymbol{\xi}_{i,\tau}^\top \boldsymbol{\xi}_{\tau'} && y\lambda_k > 0 \\
&\geq \alpha_1^{t=2} |\lambda_k| \, \|\boldsymbol{\mu}_k\|^2 - \sum_{\tau=2}^{T-1} \alpha_\tau^{t=2} n(T-1)^2 \max_i |\theta_i| \frac{d}{C_1 n} \\
&\geq \alpha_1^{t=2} \left( \frac{\beta}{4T+1} \right) \frac{d}{n} c_\rho^2 - \sum_{\tau=2}^{T-1} \alpha_\tau^{t=2} n(T-1) \frac{\beta}{Tn} (M_\mathcal{N} + 0.5) \frac{d}{C_1 n} && \text{Eqs. 54, 50 and 52} \\
&\geq \frac{1}{T} \left( \frac{\beta}{4T+1} \right) \frac{d}{n} c_\rho^2 - \left( 1 - \frac{1}{T} \right) n(T-1) \frac{\beta}{Tn} (M_\mathcal{N} + 0.5) \frac{d}{C_1 n} \\
&> 0,
\end{aligned}
$$

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

(6 n^2 / \delta)} \leq \frac{2 n_2 \sqrt{d \log(6 n^2 / \delta)}}{((1 - \kappa) d - 2 n_2 \sqrt{d \log(6 n^2 / \delta)})^2} = \widetilde{O} \left( \frac{n}{d^{3/2}} \right).$$

The first inequality is from Lemma 64; The second inequality is from Lemma 34. Similarly, for $|I_3|$ we have

$$|I_3| \leq \sum_{i \in \mathcal{C}_1} \max_{i \in \mathcal{C}_1} \breve{\theta}_i^2 \cdot 2 \sqrt{d \log(6 n^2 / \delta)} \leq \frac{2 n_{11} \alpha^2 \sqrt{d \log(6 n^2 / \delta)}}{((1 - \kappa) d - 2 n_{11} \sqrt{d \log(6 n^2 / \delta)})^2} = \widetilde{O} \left( \frac{n}{d^{3/2}} \right).$$

The second inequality is from Lemma 41. Combining the above results, we have

$$\|\boldsymbol{v}''\|_2^2 - \|\boldsymbol{v}\|_2^2 \geq \Theta \left( \frac{n_{11}}{d} \right) - \widetilde{O} \left( \frac{n}{d^{3/2}} \right) \geq \frac{C_3 n (1 - \beta)}{d}.$$

The remaining proof is the same as *Case 2.1* and we omit it for convenience. □

Therefore, we complete the proof for all possible scenarios. □

**Training and Test Error Analysis**

From Proposition 15 we can analyze the properties of both parameters to estimate the training and test error.

In this section, we first get the convergence direction of parameters $p$ and $v$. The main difference between our setting with Ataee Tarzanagh et al. (2023b) is that they only consider the infinite case and their results hold only when $R, r \to \infty$. We extend their results to the finite case. Specifically, given fixed upper bound $R$ and $r$ for $\|p\|$ and $\|v\|$ respectively, we denote the solution of the constrained optimization (2) as $(v_r, p_R)$ in this section for brevity.

Our main theorem in this section estimates the corresponding deviation of $p_R/R$ and $v_r/r$ from their convergence direction $p_{mm}/\|p_{mm}\|$ and $v_{mm}/\|v_{mm}\|$. For a given $p$, it is elementary that the margin induced by $p$ is $\min_{i,t_i \neq \alpha_i}(x_{i\alpha_i} - x_{it_i})^\top p/\|p\|$, thus when $\|p\| = 1$, the margin becomes $\min_{i,t_i \neq \alpha_i}(x_{i\alpha_i} - x_{it_i})^\top p$. And for a given $v$, the label margin induced by $v$ is $\min_i y_i v^\top r_i/\|v\|$. Recall that the label margin induced by $v_{mm}$ is $\Gamma$ and the margin of *p-SVM* induced by $p_{mm}$ is $\Xi$.

First we introduce a lemma to estimate the norm of $\|p_{mm}\|$. This will benefit our proof of the main theorem.

**Lemma 42** (Norm of $p_{mm}$). *Suppose that Assumption 5 holds, recall that the solution of (p-SVM) is $p_{mm}$. With probability at least $1 - \delta$ on the training dataset we have*

$$\frac{1}{\rho^2} + \frac{\eta n}{d} \leq \|p_{mm}\|^2 \leq \frac{8}{\rho^2} + \frac{17\eta n}{d}.$$

*This implies*

$$\|p_{mm}\| = \Theta\left(\sqrt{\frac{1}{\rho^2} + \frac{\eta n}{d}}\right).$$

*Proof of Lemma 42.* First we prove the upper bound. Consider the following possible solution $\widetilde{p}$:

$$\widetilde{p} = \frac{2(\mu_1 + \mu_2)}{\rho^2} + \sum_{i \in \mathcal{N}} 4\frac{\xi_i}{d}. \tag{78}$$

We then proved that $\widetilde{p}$ satisfies (63). For $k \in \mathcal{C}$ we have

$$\widetilde{p}^\top(\mu_k - \xi_k) = 2 - \sum_{i \in \mathcal{N}} 4\frac{\langle \xi_i, \xi_k \rangle}{d} \geq 2 - \frac{4n_2\sqrt{d\log(6n^2/\delta)}}{d} \geq 1.$$

The first inequality is from the definition of $d$ in Lemma 64 and the second inequality is from Assumption 5. And for $k \in \mathcal{N}$, we have

$$\widetilde{p}^\top(\xi_k - \mu_k) = -2 + \sum_{i \in \mathcal{N}} 4\frac{\langle \xi_i, \xi_k \rangle}{d} \geq -2 + 4(1 - \kappa) + \sum_{i \in \mathcal{N}, i \neq k} 4\frac{\langle \xi_i, \xi_k \rangle}{d}$$

$$\geq -2 + 4(1 - \kappa) + \frac{4n_2\sqrt{d\log(6n^2/\delta)}}{d} \geq 1.$$

The first and second inequalities are from Lemma 64; The last inequality is from Assumption 5.

Therefore, the max-margin solution $p_{mm}$ must have no greater norm than $\widetilde{p}$. So we can upper bound $p_{mm}$ as

$$\|p_{mm}\|^2 \leq \|\widetilde{p}\|^2 = \frac{8}{\rho^2} + \frac{16}{d^2}\left(\sum_{i \in \mathcal{N}} \|\

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

$$\left|\,|\mathcal{C}| - n(1-\eta)\right| \leq \sqrt{n \log(\frac{1}{\delta})}; \quad \left|\,|\mathcal{N}| - n\eta\right| \leq \sqrt{n \log(\frac{1}{\delta})};$$

$$\left|\,|\mathcal{C}_i| - \frac{n(1-\eta)}{2}\right| \leq \sqrt{n \log(\frac{1}{\delta})}; \quad \left|\,|\mathcal{N}_i| - \frac{n\eta}{2}\right| \leq \sqrt{n \log(\frac{1}{\delta})}, \quad i = 1, 2.$$

*Proof.* Note that $|\mathcal{C}| \sim \text{Binom}(n, 1 - \eta)$. Applying Hoeffding's inequality, we have

$$\mathbb{P}\left(\left|\,|\mathcal{C}| - (1-\eta)n\right| > t\right) \leq 2 \exp(-\frac{2t^2}{n}).$$

Let $t = \sqrt{n \log(1/\delta)}$. We have that with probability at least $1 - \delta$,

$$\left|\,|\mathcal{C}| - (1-\eta)n\right| \leq \sqrt{n \log(\frac{1}{\delta})}.$$

Similarly, note that $|\mathcal{N}| \sim \text{Binom}(n, \eta), |\mathcal{C}_1| \sim \text{Binom}(n, (1-\eta)/2), |\mathcal{C}_2| \sim \text{Binom}(n, (1-\eta)/2), |\mathcal{N}_1| \sim \text{Binom}(n, \eta/2)$ and $|\mathcal{N}_2| \sim \text{Binom}(n, \eta/2)$, we have that each of the following events holds with probability at least $1 - \delta$:

$$\left|\,|\mathcal{C}| - n(1-\eta)\right| \leq \sqrt{n \log(\frac{1}{\delta})}; \quad \left|\,|\mathcal{N}| - n\eta\right| \leq \sqrt{n \log(\frac{1}{\delta})};$$