# OpenReview forum: "Benign Overfitting in Single-Head Attention"
_ICLR.cc/2025/Conference — Submitted to ICLR 2025_

### Official Review · Reviewer_Ja9x · 2024-11-02

**Soundness:** 3
**Presentation:** 3
**Contribution:** 2
**Rating:** 5
**Confidence:** 4

**Summary:**

The paper investigates benign overfitting within single-head softmax attention models, foundational components of Transformers, by examining when these models can fit noisy training data while still generalizing effectively to test data. The authors provide theoretical conditions under which gradient descent with logistic loss leads to benign overfitting within two iterations, requiring a specific signal-to-noise ratio (SNR) scaled as $\Theta(1/\sqrt{n})$. Additionally, they show that minimum-norm (or maximum-margin) interpolation allows for benign overfitting with a relaxed SNR requirement of $\Omega(1/\sqrt{n})$. Empirical results validate these findings, demonstrating that a sufficiently large SNR and input dimension are essential for benign overfitting, while lower SNR leads to harmful overfitting. This study offers new insights into the benign overfitting behavior of attention mechanisms, highlighting theoretical and empirical conditions for achieving this phenomenon in single-head attention models.

**Strengths:**

- The authors provide a rigorous theoretical analysis, supported by empirical validation. The proofs are well-structured, with assumptions and conditions clearly stated, lending robustness to the theoretical results.

- The paper is clearly written and structured, allowing readers to follow the logical flow of arguments and results.

**Weaknesses:**

- Compared to [1], the data model in this paper is similar; however, the attention network considered here is relatively simple. For example, [1] trains both the query and key matrices, whereas this work only trains a single vector within the softmax function. This simplification reduces the complexity of the analysis significantly but may also limit the applicability of the findings to more complex attention networks.

- For benign overfitting with gradient descent, the requirement for both upper and lower bounds on the Signal-to-Noise Ratio (SNR) restricts the scope of the data model, potentially limiting the range of scenarios where the results apply.

- The data model only considers two tokens, which could constrain the generality of the findings. An analysis including more tokens or varying token arrangements would strengthen the applicability of the conclusions.

- Minor: Big-O notation (line 100-101)



[1] Jiang, Jiarui, Wei Huang, Miao Zhang, Taiji Suzuki, and Liqiang Nie. "Unveil Benign Overfitting for Transformer in Vision: Training Dynamics, Convergence, and Generalization." arXiv preprint arXiv:2409.19345 (2024).

**Questions:**

- For **Assumption 3**, why is an upper bound on the Signal-to-Noise Ratio (SNR) necessary? Additionally, how would the results change if a different step size is used during training, such as a larger or smaller step size?

- For **Assumption 3**, what would be the effect of not using zero initialization? For instance, have you considered using Gaussian initialization, and if so, how would this impact the behavior of benign overfitting?

- In the experiments, have you considered using a heatmap for test accuracy to illustrate the separation condition based on SNR values? Such a visualization could help clarify the relationship between SNR and model generalization in different settings.

---

> ### Author Response · Authors · 2024-11-22
>
> We thank the reviewers for their thoughtful comments and valuable feedback.
>
> **Comparison with related work**
>
> First, we note that Jiang et al. [1] posted their paper on arxiv a few days before the ICLR submission deadline, so it should be considered concurrent work to ours. Moreover, to the best of our knowledge, our work is the first to establish benign overfitting in a softmax attention model with noisy data, in contrast to [1], which considers clean data. Note that our analysis shows that the learning patterns for clean and noisy samples differ significantly, which is a crucial component of our analysis.
>
> **Multi-token settings**
>
> While the setting of [1] involves multiple tokens, they rely on additional assumptions about the data distribution. In their Condition 4.1.(10), they assume that the second noise token is sufficiently larger than that of other noises, hence it essentially plays a similar role to the noise token in our setting.
>
> We conducted some new experiments which show that our results also hold for the multi-token setting (see Fig. 6 in Appendix A.4 of the updated version). In this scenario, the key idea presented in our paper remains consistent: the attention layer focuses on optimal tokens to achieve benign overfitting. The main difference lies in the optimal token selection mechanism. As shown in Figure 6,  the optimal tokens for clean samples are still the signal tokens. However, for noisy samples, the optimal tokens may be a combination of noise tokens. This figure also illustrates that, in the multi-token setting without additional assumptions, the interactions between noise tokens become more complex, making the analysis of their attention probabilities more challenging. Nevertheless, the key concept of "optimal token selection" continues to hold in more complex settings.
>
> For our GD analysis, we believe that we know how to extend the result to multiple tokens, but we still need to verify a few technical details. The proof strategy is very similar and is based on analyzing the coefficients of the tokens in the attention head $v$, showing that the softmax weight $p$ assigns a high softmax probability to the signal token for clean samples, and a low softmax probability to the signal token for noisy samples (instead of showing a high softmax probability for a specific noisy token, as in the case of two tokens).
>
> **Upper Bound on SNR**
>
> Regarding the SNR upper bound, a full answer would require many technical details.
> The bottom line is that proving benign overfitting with a larger SNR requires sharper bounds on the softmax probabilities of optimal tokens, i.e., stronger than 1-c, for some constant c>0, as we have now. If needed, we can provide more technical details about that.
>
> **Different Step-size**
>
> Regarding the step size, with a smaller step size, more than 2 iterations will be required to fit the noisy samples and achieve benign overfitting. Roughly speaking, the loss of noisy samples will be smaller, therefore more iterations will be required to learn the noisy tokens of the noisy samples. We also demonstrate this in the experiments (see figure 2, line 495). With a larger step size, we suspect that the model will not converge, since clean and noisy samples have the same signal token, but with opposite directions, the model might alternate between fitting the noisy samples and fitting the clean samples.
>
> **Zero and Gaussian Initialization**
>
> We prove the result using zero initialization for simplicity, but the same results hold for Gaussian initialization with sufficiently small variance. This is because the model is smooth: the gradient with respect to the softmax weight p is Lipschitz (as shown in Lemma 6 of [1]), and the same holds also for the gradient with respect to the attention head v (f(X;p,v) is linear in v). Consequently, the loss and gradient for any sample under Gaussian initialization will closely resemble the loss and gradient under zero initialization. Using this reasoning, we conclude that the model also exhibits benign overfitting (after two iterations) with Gaussian initialization. To support this claim, we have included additional experiments using Gaussian initialization (See Figure 8 in the additional experiments section in the Appendix). If required, we can formally prove this result in the final version of the paper.
>
> **Heatmap for SNR Threshold**
>
> Thanks for the suggestion! See Figure 7 in the additional experiments section in the Appendix, which confirms the SNR threshold in our theory.
>
> [1] Jiang, Jiarui, Wei Huang, Miao Zhang, Taiji Suzuki, and Liqiang Nie. "Unveil Benign Overfitting for Transformer in Vision: Training Dynamics, Convergence, and Generalization." arXiv preprint arXiv:2409.19345 (2024).
>
> [2] Davoud Ataee Tarzanagh, Yingcong Li, Xuechen Zhang, and Samet Oymak. Max-margin token selection in attention mechanism

---

> ### Comment · Reviewer_Ja9x · 2024-11-29
>
> Thank you for your detailed response.
>
> > Benign overfitting in a softmax attention model with noisy data
>
> Including the analysis with label flipping is interesting. However, I am still concerned about whether this constitutes a significant contribution. I reviewed your response to Reviewer 7cCk and remain curious about the role of attention in the presence of label noise. An experimental simulation/theoretical explanation addressing this question would be appreciated.
>
> > Upper Bound on SNR
>
> Thank you for your explanation; I now understand the difficulties. However, could you please list some potential methods that might address this issue?
>
> > Heatmap for SNR Threshold
>
> Thank you. I am glad to see this new result.

---

> > ### Author Response · Authors · 2024-12-01
> >
> > **Benign overfitting in a softmax attention model with noisy data**
> >
> > We are not sure we fully understand the concern. The novelty of this work lies in exploring how attention mechanisms handle noisy labeled data and when they exhibit benign overfitting. Throughout all of our theorems, we assume noisy labeled data, and all experiments are conducted under this assumption. The paper includes numerous experiments covering a variety of conditions, such as different step sizes, SNR, Gaussian initialization, dimensions, architectures, and more. Several experiments were added after the original submission and can be found in Appendix A.4. We also refer the reviewer to the clarifications in our new response to Reviewer 7cCk.
> >
> > **Upper Bound on SNR**
> >
> > To relax the upper bound on the SNR, we need to establish sharper bounds on the coefficients of the attention head v and the softmax probabilities. We believe this is achievable but we still need to verify some technical details. This would not require a new method or significant changes in our current approach, just a more careful and precise analysis of the existing framework.

---

### Official Review · Reviewer_96r4 · 2024-11-03

**Soundness:** 3
**Presentation:** 3
**Contribution:** 3
**Rating:** 6
**Confidence:** 2

**Summary:**

This paper analyzes benign overfitting in single-head attention depending on the signal-to-noise ratio (SNR) of the data distribution.
GD with the logistic loss shows benign overfitting when SNR is $\Theta(1/\sqrt{n})$. For minimum-norm interpolators, the requirement for the SNR becomes weaker ($\Omega(1/\sqrt{n})$) and this is tight.

**Strengths:**

I couldn't check all the technical details, but surely this paper has its strengths as follows:

- This paper provides a clear characterization of the role of the SNR (i.e., that of $d$ and $n$) for benign overfitting.

**Weaknesses:**

- Some theory appear to lack empirical evidence. Can we confirm Theorem 6,8,10 (in Section 4) with some experiments on the max-margin learning rule? Experiments in Section 6 seem like they are only for Theorem 4. If not, which experiments (figures) correspond to which theorem?
- It would be better to have experiments (or at least a discussion) on a relatively complex model (e.g., multi-head attention, multi-layer Transformer, more than two tokens). For example, a complex model may have similar behavior with the simple one used in Section 6. We may get some intuition from the experiments that a simple structure is enough to exhibit benign overfitting or some complex structures are crucial.
- The single-attention $S(XWq)$ or $S(Xp)$ considered in the paper is linear in $X$ before taking the softmax which is not usual attention as the authors consider the query vector $q$ independent of the input $X$. Is there any justification that we can use such simplification? This is a big limitation of this paper.

**Questions:**

- Can you define the (unusual) notations SNR $\geq \Omega(1/\sqrt{n})$ or SNR $\leq O(1/\sqrt{n})$ explicitly? Is it something like "SNR $\geq f(n)$ and $f(n)=\Omega(1/\sqrt{n})$"?
- Why does $I-\mu_1\mu_1^\top/\rho-\mu_2\mu_2^\top/\rho$ in the covariance make that the noise token is orthogonal to the signal one?
- Can you draw (final or $t=2$ test-accuracy) vs ($n/d$ or $n^2/d$) plot to get a better intuitive explanation?
- Can we somehow neglect the cross term $\sum_{i,j} y_iy_j\theta_j^{t=1}\xi_i^\top \xi_j$ without the assumption item 2, at least experimentally?
- What if we use a normalized noise $\xi \sim N(0, \Sigma/d)$ so that SNR $=\rho$ where $\Sigma=I_d-\mu_1\mu_1^\top/\rho-\mu_2\mu_2^\top/\rho$?

---

> ### Author Response · Authors · 2024-11-22
>
> **Verification of Theorem 6, 8, 10**
>
> To verify Theorems under the max-margin learning rule, we trained using GD with weight-decay (as GD with weight decay corresponds to norm-minimization/margin-maximization) and got similar plots to the ones of GD without weight decay (see Figure 10, page 69 in the updated version).
>
> We also refer to Figure 7 (page 68 in the updated version) which presents a heatmap of the test accuracy, plotted across varying SNR and sample sizes n, which also demonstrates the tight bound on the SNR. These empirical results support Theorem 6,8,10.
>
> **Experiments on more complex model**
>
> We appreciate the question regarding the extension to more general settings and we have conducted additional experiments on the self-attention model, multi-token data model and multi-layer transformer, demonstrating that our analysis can extend to more complex settings.
>
> First, for self-attention model $v^{\top} X^{\top} S(X W x_1)$ where $x_1$ is the first token, the results in Fig.5 (Appendix A.4 in the updated version) confirm the presence of benign overfitting. The softmax probabilities assigned to the signal and noise tokens align with the optimal token selection mechanism outlined in our paper.
>
> Second, for multi-token data, we consider T=5 tokens, where all noise tokens are generated in the same approach. The results in Fig.6 show that the model continues to exhibit benign overfitting. For clean samples, most of the softmax attention is focused on the signal token, which corresponds to the optimal token in our setting. In contrast, for noisy samples, the softmax probabilities are more evenly distributed across all tokens, with significantly lower attention on the signal token.
>
> Finally, for multi-layer attention model, we consider a 4-layer single-head attention and the data model remains the same. The results in Fig.9 indicate that benign overfitting phenomenon still appears and the attention weights of the first layer have the same optimal token selection mechanism.
>
> **Justification of Model Simplification**
>
> Our setting corresponds to self-attention where q is a token concatenated to the raw input X to get $[q, X^{\top}]^{\top}$, and thus our output corresponding to position q is $v^{\top} X^{\top} S(XWX^{\top})$. In practice, this token serves as a learnable parameter that enables the model to adapt flexibly to new tasks. This setup is widely used in applications, such as the [CLS] token in BERT and Vision Transformers (ViT) for classification tasks. Also, the prompt-tuning technique in [2] and [3] leverages similar mechanisms.
>
> [1] Davoud Ataee Tarzanagh, Yingcong Li, Xuechen Zhang, and Samet Oymak. Max-margin token selection in attention mechanism
>
> [2] Xiang Lisa Li and Percy Liang. Prefix-tuning: Optimizing continuous prompts for generation
>
> [3] Brian Lester, Rami Al-Rfou, and Noah Constant. The power of scale for parameter-efficient prompt tuning.
>
> **About SNR Definition**
>
> Yes, the definition you wrote is correct.
>
> **Signal-Noise Orthogonality**
>
> We can understand this expression as removing the variance along the directions of $\mu_1$ and $\mu_2$ by subtracting the terms $\frac{\mu_1 \mu_1^\top}{\rho^2}​$​ and $\frac{\mu_2 \mu_2^\top}{\rho}$​​. So there is no vector component along the direction of $\mu_1$ or $\mu_2$ in noise vectors and the inner product between signal and noise is 0.
>
> **Accuracy vs n/d Plot**
>
> Yes, see Figure 4b in the Appendix A.4 of the updated version.
>
> **Neglect Cross Term**
>
> Assumption item 2 states that $d >> n^2$, and implies that noise tokens from different training samples are nearly orthogonal. This assumption appears in many prior works on benign overfitting (see line 190 for references), and we agree that it is an interesting theoretical challenge to relax this assumption. Experimentally, Figure 3b (page 13) shows results for different values of dimension d when the number of samples n is fixed. We observe that benign overfitting occurs even for $d=1000=2n<<n^2$.
>
> **Normalized Noise**
>
> Then there is only a scale difference in our results and the SNR threshold remains the same, i.e. $\rho = \Omega(1/\sqrt{n})$ to achieve benign overfitting.

---

> > ### Comment · Reviewer_96r4 · 2024-11-26
> >
> > Thank you for the additional explanation, especially some experiments regarding Theorem 6, 8, 10.
> > The setting is limited. GD with weight decay is not exactly the max-margin learning rule (different trajectory).

---

> > > ### Author Response · Authors · 2024-12-01
> > >
> > > The max-margin learning rule roughly corresponds to GD with weight decay (albeit not exactly, as the reviewer pointed out). We refer the reviewers to lines 234-240, which provide additional motivation for this learning rule:
> > >
> > > “We note that learning rules that return min-norm (or max-margin) solutions are considered natural, and hence understanding properties of min-norm interpolators has attracted much interest in recent years, even in settings where the implicit bias of GD does not necessarily lead to a min-norm solution (see, e.g., Savarese et al. (2019); Ongie et al. (2019); Ergen & Pilanci (2021); Hanin (2021); Debarre et al. (2022); Boursier & Flammarion (2023)). More directly related to our work, min-norm interpolation with Transformers has been studied in Ataee Tarzanagh et al. (2023b; a), and benign/tempered overfitting in min-norm univariate neural network interpolators has been studied in Joshi et al. (2023).”

---

### Official Review · Reviewer_7cCk · 2024-11-03

**Soundness:** 3
**Presentation:** 3
**Contribution:** 2
**Rating:** 5
**Confidence:** 4

**Summary:**

This work examines the phenomenon of benign overfitting in a single-head attention model on a binary classification task with label noise. First, they show that two steps of GD, under signal-to-noise ratio (SNR) of $\Theta(1/\sqrt{n})$, exhibit benign overfitting. Next, they examine the max-margin/min-norm interpolator and show that the SNR requirement in that case is only $\Omega(1/\sqrt{n})$. Finally, they show that this is necessary (upto some constant factors), because for lower SNR, the max-margin interpolator exhibits harmful overfitting.

**Strengths:**

This work takes a step towards extending the understanding of benign overfitting from other neural network architectures to attention models.

Overall, the paper is well-written and easy to follow.

**Weaknesses:**

- The main weakness is that the results are under very strong assumptions.
    - First, the authors consider prompt attention, not self-attention, which is the more prevalent attention architecture. The authors also assume that the number of tokens is two, which is far from practice.
    - Second, the authors only analyze two steps of gradient descent (GD), and not the full trajectory. While the experiments show that with large step size, the model converges in two steps of GD, the theoretical results, which are the main contribution, seem rather weak since only two steps of GD are analyzed.
    - Third, the authors assume that the parameters are initialized at zero, instead of random initialization, which is used in practice.

    Given the strong assumptions, it is not clear what insights they offer about training attention models.

- Other minor suggestions:
    - There are some missing references in the related work section on optimization in transformers, such as [1-4].
    - In line 89, overfitting is misspelled.

References:

[1] Puneesh Deora, Rouzbeh Ghaderi, Hossein Taheri, Christos Thrampoulidis. On the Optimization and Generalization of Multi-head Attention, TMLR 2024.

[2] Yuandong Tian, Yiping Wang, Beidi Chen, and Simon Du. Scan and snap: Understanding training dynamics and token composition in 1-layer transformer, NeurIPS 2023.

[3] Yuandong Tian, Yiping Wang, Zhenyu Zhang, Beidi Chen, and Simon Du. Joma: Demystifying multilayer transformers via joint dynamics of mlp and attention, ICLR 2024.

[4] Heejune Sheen, Siyu Chen, Tianhao Wang, and Harrison H. Zhou. Implicit regularization of gradient flow on one-layer softmax attention, 2024.

**Questions:**

- Would the results generalize to a more realistic model, such as self-attention and with more than two tokens?
- Can the authors discuss the challenges for analyzing further steps for GD instead of just two steps?
- Can the authors include some discussion on how the SNR requirement for benign overfitting for self-attention compares with other architectures, particularly CNNs, since similar data models have been considered for benign overfitting in CNNs?

---

> ### Author Response · Authors · 2024-11-22
> **Weaknesses**
>
> We thank the reviewers for their comments and feedback. Below, we address each concern and provide clarifications. We will incorporate the suggested references into the final version of our paper and correct the noted typographical error in line 89.
>
> Regarding the strong assumptions, we emphasize that, to the best of our knowledge, this is the first work to study benign overfitting in transformers with label noise. Therefore, it is natural to begin with simplified assumptions to better understand the behavior of transformers in this context. These assumptions provide a tractable framework for studying softmax attention with label noise, serving as a foundation for future work to explore more general settings. Even with these assumptions, the problem is highly non-trivial to analyze, as evident in our proofs.
>
> We emphasize that Theorems 6-8, which show that the max-margin solution exhibits benign overfitting, can reasonably be viewed as a form of 'fully converged' analysis, correspond to training with weight decay.
>
> **Initialization at zero**
>
> We use zero initialization for simplicity, but the results extend for example to Gaussian initialization with sufficiently small variance. This follows from the model's smoothness, ensuring that the loss and gradient under Gaussian initialization closely match these under zero initialization. Thus, the model exhibits benign overfitting after two iterations also with Gaussian initialization, as confirmed by additional experiments that we conducted – see Fig. 8 in Appendix A.4 in the updated version of the paper. If required, we can formally prove this in the final version.

---

> ### Author Response · Authors · 2024-11-22
> **Question 1 & 2**
>
> **Generalizing to Self-Attention and Multi-Token Settings**
>
> **Self-Attention Models:** While our work focuses on a simplified prompt-attention model, we believe the results can generalize to self-attention settings. For self-attention models, we can consider the model $v^{\top} X^{\top} S(X W x_1)$ where $x_1$ is the first token. Specifically, the optimization behavior may differ due to the additional interactions in self-attention. However, as demonstrated by the numerical results from new experiments that we conducted (see Fig.5, page 67 in the new version), the two core findings from our paper—optimal token selection and the benign overfitting phenomenon—remain consistent in this setting.  Additionally, we have some primary conjectures for theoretical analysis. The analysis of v (in all of the results) can directly follow our analysis. As for the analysis of W, we need to reconsider its convergence direction and from our empirical results we think it will also converge to its max-margin solution (W-SVM), i.e. the minimizer of certain norm under the constraints:
> &nbsp;&nbsp;&nbsp;&nbsp;$(\mu_i - \xi_i) W x_1 \ge 1 (i \in C)$
> &nbsp;&nbsp;&nbsp;&nbsp;$(\xi_i - \mu_i) W x_1 \ge 1 (i \in N)$
>
> **Multi-Token Settings:** regarding the extension to multi-token settings, we conducted new experiments and showed that our results hold also in this case – see Fig. 6 on page 67 of the new version. The key idea presented in our paper remains consistent: the attention layer focuses on optimal tokens to achieve benign overfitting. The main difference lies in the optimal token selection mechanism. As shown in Figure 6(c), the optimal tokens for noisy samples can be a combination of noise tokens in this case. Once the optimal tokens are determined, benign or harmful overfitting follows similarly.
>
> The figure also demonstrates that, in the multiple tokens setting, without additional assumptions the interaction between noise tokens becomes more complex, making it harder to analyze their attention probabilities. Existing works on multiple tokens setting all need additional assumptions. For example, [1] assumes a strict data generation rule (Assumption C) connecting data and the induced max-margin. [2] requires the second noise token to be much larger than others (Condition 4.1.(10)). and [3] fixes the linear prediction head v and assumes identical token scores for all noise tokens. We aim to explore this more general setting in future work.
>
> **Further steps for GD**
>
> In our setup, both clean and noisy samples share the same signal token (the first token), which differs from other works that often assume nearly orthogonal tokens across samples [1,4].
>
> To illustrate the difficulty, consider a scenario where the model correctly classifies clean samples but not noisy samples (which indeed happened after one iteration of GD in our analysis). In this case, the loss of the noisy samples dominates the loss of clean samples, causing the model to prioritize fitting the noisy samples. Since clean and noisy samples have the same signal token, albeit in opposite directions, additional iterations of GD may cause the attention head and softmax weights to reduce the focus on the signal token. This undesirable behavior can ultimately degrade test accuracy.
>
> This, along with other technical difficulties, especially analyzing the gradient with respect to the softmax weight p (with label-noisy data), makes even two steps of GD a challenging task. We hope future work can build upon our techniques to extend the analysis to further iterations.

---

> ### Author Response · Authors · 2024-11-22
> **Question 3 & References**
>
> **SNR- self-attention vs convolutional**
>
> Here we compare our results with Kou et al. [5] which also considers a signal-noise data model with label flipping noise. Their SNR threshold separating benign and harmful overfitting is $||\mu||/\sqrt{d} = \Theta(1/n^{1/2} d^{1/4})$, which is smaller than ours ($\Theta(1/n^{1/2})$). Notably, they only analyze the case where SNR is small ($||\mu||/\sqrt{d} = O(1/n^{1/2})$), while we do not require an upper bound for SNR. The difference arises from the distinct architectures of CNNs and attention models, leading to different mechanisms for overfitting training data. In CNNs, signal and noise patches are learned together because the output is the sum of their contributions. In contrast, due to the optimal token selection mechanism of the attention layer, our model can focus on either the signal or the noise token for each sample, based on which has a higher softmax probability. This enables our analysis to handle cases with larger SNR, which is not possible in Kou et al.'s framework.
>
> [1] Davoud Ataee Tarzanagh, Yingcong Li, Xuechen Zhang, and Samet Oymak. Max-margin token selection in attention mechanism.
>
> [2] Jiarui Jiang, Wei Huang, Miao Zhang, Taiji Suzuki, and Liqiang Nie. Unveil benign overfit-
> ting for transformer in vision.
>
> [3] Heejune Sheen, Siyu Chen, Tianhao Wang, and Harrison H. Zhou. Implicit regularization of gradient flow on one-layer softmax attention.
>
> [4] Bhavya Vasudeva, Puneesh Deora, and Christos Thrampoulidis. Implicit bias and fast convergence rates for self-attention.
>
> [5] Yiwen Kou, Zixiang Chen, Yuanzhou Chen, and Quanquan Gu. Benign overfitting in two-layer relu convolutional neural networks.

---

> > ### Comment · Reviewer_7cCk · 2024-11-28
> >
> > Thank you for the detailed rebuttal. I still have some concerns with the paper.
> >
> > **Regarding the strong assumptions**
> >
> > The reasoning that “it is natural to begin with simplified assumptions to better understand the behavior of transformers in this context” is not satisfactory for two reasons. First, the paper studies a single-head softmax attention model, not a full transformer model, which in itself is a massive simplification. Second, while I agree that some simplified assumptions could be necessary for theoretical results, it’s important that the assumptions are realistic (at least to some extent), and that despite the assumptions, we can get some insights about the behaviour of the model. In this regard, studying two steps of GD seems too restrictive, unless there is reason to believe that further training would not lead to different behaviour eventually (see below).
> >
> >  ****Initialization at zero****
> >
> > Thank you for the clarification, it addresses my concern.
> >
> > ****Generalizing to Self-Attention and Multi-Token Settings****
> >
> > Thank you for the new experimental results in these settings that show similar behavior. The authors have also included a new result that shows that their results hold in the multi-token setting, which addresses this point.
> >
> >  ****Further steps for GD****
> >
> > The authors state that, “Since clean and noisy samples have the same signal token, albeit in opposite directions, additional iterations of GD may cause the attention head and softmax weights to reduce the focus on the signal token. This undesirable behavior can ultimately degrade test accuracy.”
> >
> > This is concerning because if further training can lead to harmful overfitting, it seems that the analysis is specific to two steps of GD, which is very restrictive. This doesn’t just seem to be a technical difficulty, if early stopping seems to be crucial to the behaviour (benign vs harmful overfitting). Can the authors comment on this?
> >
> > **Comparison between single-head attention and CNNs**
> >
> > Thank you for discussing and comparing the results with [5]. I was wondering if such simple settings could give some insight into why ViTs are more robust to label noise compared to CNNs. However, it seems that the SNR requirement for CNNs in this setting is actually lower compared to the attention model. Could this point to other components of the transformer architecture playing a bigger role in their robustness? I realize this is a bit tangential to the paper, but I think it’s relevant in regards to why understanding benign overfitting in attention models is important. I am curious about the authors’ thoughts on this.

---

> > > ### Author Response · Authors · 2024-12-01
> > >
> > > **Further steps for GD:** We believe that the results can, in principle, be extended to further steps:
> > >
> > > - **Comprehensive Experiments:** Our empirical results span a wide range of parameters and consistently demonstrate benign overfitting even after significantly more than two iterations (see Figures 1,2 and 3 on pages 9-10) . Importantly, these experiments show that the model consistently separates optimal tokens from others, providing further evidence that the behavior observed in the first two steps extends throughout training.
> > >
> > > - **Balancing Clean and Noisy Samples:** The statement regarding the potential degradation of test accuracy due to the focus on the signal token was not meant to imply that harmful overfitting is inevitable but rather to highlight the unique challenges in analyzing the interplay between clean and noisy samples. Unlike settings that only consider clean samples, our setup requires balancing the influence of clean and noisy samples, which share the same signal token but in opposite directions. This creates a delicate trade-off: the model must fit noisy samples without excessively diminishing the focus on the signal token. Our proof carefully demonstrates how this balance is achieved within two steps, despite these challenges. We believe that extending the results to further steps will require very careful and technical analysis.
> > >
> > > - **Intuition for Further Steps:**  After the model managed to classify correctly both clean samples and noisy samples (i.e. after two steps for large enough step size), the loss for both clean and noisy samples becomes small. As a result, the gradient updates should have a diminishing effect on the weights direction, meaning that the model’s parameters should stabilize rather than drift significantly. Thus, we expect that extending the analysis to additional steps, while technically involved, would yield similar results.
> > >
> > > - **Minimum-Norm/Maximum Margin Interpolator:**  We also provided the following result, which serves as a kind of "fully converged" learning rule: We proved that the minimum-norm or maximum-margin interpolator also exhibits benign overfitting. This roughly corresponds to gradient descent (GD) with weight decay. We suspect that GD with weight decay tends (after enough iterations) to a solution that minimizes the loss while preserving a small weight norm.
> > >
> > > **Comparison between single-head attention and CNNs**
> > >
> > > It is indeed an interesting question whether other components of the transformer architecture play a larger role in their robustness to label noise. The SNR requirement stems from the token-selection mechanism in our single-head attention model. It is possible that multi-head attention could generalize with a smaller SNR bound, but this question needs further exploration.

---

> > > > ### Comment · Reviewer_7cCk · 2024-12-03
> > > >
> > > > Thank you for the response. I appreciate the authors' efforts in the rebuttal phase, and since many of my concerns have been addressed, I raised my score to 5. However, I still think this paper needs more work not just in motivating and discussing the assumptions and limitations better, but also in further strengthening the results by extending the analysis to further steps of GD. Therefore, I suggest revising and resubmitting the paper.

---

### Official Review · Reviewer_UW8h · 2024-11-07

**Soundness:** 2
**Presentation:** 2
**Contribution:** 2
**Rating:** 5
**Confidence:** 3

**Summary:**

This paper studies benign overfitting in a single-head attention model, proving that under certain signal-to-noise ratio (SNR) conditions, the model achieves benign overfitting after two gradient descent steps or through a max-margin solution.

**Strengths:**

The paper provides the theoretical analysis of benign overfitting in attention mechanisms, expanding understanding beyond linear and CNN models. It also establishes conditions for benign overfitting.

**Weaknesses:**

This paper relies on overly simplified settings and assumptions for its theoretical analysis. Additionally, its approach heavily depends on prior research, offering limited novelty and inheriting weaknesses from existing studies.

**Questions:**

It seems that this paper primarily combines ideas from two prior works, "Benign Overfitting in Two-layer Convolutional Neural Networks" (Cao et al., NeurIPS 2024) and "Max-Margin Token Selection in Attention Mechanism" (Ataee Tarzanagh et al., NeurIPS 2024). By relying heavily on these works, it lacks novelty and inherits their weaknesses without addressing them.
Additionally, it seems that the weaknesses in these individual studies interact in a way that undermines each other's strengths.

Q1. Can the analysis extend to multiple tokens? Also, is it possible to relax the strong signal-noise separability assumptions that have been identified as limitations in existing studies?
In Cao et al.'s CNN work, only two patches are considered, which has been criticized in OpenReview for oversimplifying the signal and noise separation. Similarly, this paper assumes only two tokens and retains assumptions of orthogonality and separability between signal and noise, as in Cao et al.'s paper. However, in an attention model, especially one focused on token selection, analyzing only two tokens is even more problematic. For applications like Vision Transformers and sequence modeling, the ability to select among multiple tokens is fundamental to the attention mechanism, and overlooking this critical functionality limits the paper’s scope. While Ataee Tarzanagh et al. analyze max-margin solutions for multi-token scenarios, this paper fails to leverage those benefits by retaining the two-token simplification.

Q2. What is the essential difference between this work and benign overfitting in a linear model with input by linearly interpolating signal and noize vectors?
Due to the highly simplified setup, the analysis in this paper appears comparable to a straightforward two-vector linear interpolation. With two tokens—one signal and one noise—the role of attention is reduced to a linear interpolation between the two vectors. Since the signal is fixed and noise orthogonal, analyzing the behavior of the attention weights boils down to tracking a single-dimensional parameter representing this linear combination like lambda*singal+(1-lambda)*noize. As a result, the model resembles a simple two-class linear classification task using signal-noise interpolation. Thus, is this analysis merely a study of benign overfitting in a linear model? Given the extensive research on benign overfitting in linear models, it is unclear what essential differences this work offers from simpler linear models.

Q3. Is it feasible to analyze benign overfitting conditions under fully converged solutions?
Although Cao et al.'s CNN paper contributes by analyzing gradient descent dynamics, this paper only examines two-step gradient descent without exploring convergence behaviors in depth. While it does provide max-margin solution analysis, it lacks a general guarantee of convergence to this solution using gradient descent. This limitation stems from a similar issue in Ataee Tarzanagh et al.'s study, where convergence guarantees hold only for initializations close to the solution—a known weakness that seems inherited

---

> ### Author Response · Authors · 2024-11-22
> **Weaknesses**
>
> We thank the reviewers for their comments and feedback.
>
> First, the prior works that the reviewer mentioned (Tarzanagh et al. and Cao et al.) are conceptually very different than our work. Indeed, Tarzanagh et al. studied the implicit bias of a single-head attention model, and did not consider generalization or benign overfitting. Cao et al. studied convolutional networks, and thus did not consider the effect of the attention mechanism which is central in our work.
>
> The novelty of this work lies in exploring how attention mechanisms handle noisy training data and when it exhibits benign overfitting. Specifically, we investigate how attention mechanisms can effectively attend to noise tokens in noisy-labeled examples while maintaining strong generalization, a structural behavior absent in CNNs and prior studies on benign overfitting. This analysis provides fresh insights into the interplay between attention mechanisms and label noise, advancing the theoretical understanding of the overfitting behavior in attention-based models.
>
> At the technical level, almost all of our proofs required techniques that are significantly different from prior work. Since we analyzed softmax attention-based model, there is virtually no technical similarity to Cao et al. which studied CNNs (interestingly, some assumptions repeat in several works on benign overfitting in different settings, but it does not indicate a technical similarity). Regarding Tarzanagh et al. [2], which studied implicit bias in attention mechanisms, we emphasize that our proof is significantly different from theirs. Here we highlight several specific distinctions:
>
> 1. **The Proof of Theorem 4:**
>
> This proof is novel and analyzes the gradient of a softmax model with noisy labels, optimizing both the attention head v and softmax weights p, unlike [2] which assumes a fixed v in its GD results.
>
> 2. **The Proof of Theorem 6:**
>
> While inspired by Tarzanagh et al. (specifically, Theorem 5 on page 7 of their arXiv version), our proof diverges significantly. Tarzanagh et al.'s proof relies on **Assumption C** (page 7 in their arXiv version), which appears somewhat unnatural. Our work extends their results in three key aspects:
>
> - **Specific Natural Distributions:** We show a specific natural distribution (the one studied in Cao et al.) that satisfies Assumption C with high probability. See Proposition 15 (Section 5, Line 419) and its very non-trivial proof (pages 26–42), which is analog to Assumption C from Tarzanagh et al.
>
> - **Non-Asymptotic Results:** We generalize Theorem 5 from Tarzanagh et al. to include non-asymptotic results. See Theorem 16 (Line 427), which provides explicit rates for the margin as a function of the norm constraints (r and R).
>
> - **Implicit Bias implies Benign Overfitting:** We demonstrate that the implicit bias of attention leads to benign overfitting.
>
> 3. **Theorem 8:**
>
> We analyze the max-margin solution under joint norm minimization of v and p, corresponding to GD with weight decay. This approach is distinct from [2] and other works, which impose separate norm constraints.
>
> Regarding the reviewer’s comment on the simplified settings and assumptions. All existing works on benign overfitting consider very simple models (even analyzing benign overfitting in linear models is already challenging) and strong assumptions on the data. Also, all existing works on implicit bias in attention-based models require very strong assumptions. Our assumptions are not stronger than prior works, and analyzing benign overfitting in our setting is already very challenging. We believe that exploring simplified settings provides valuable insights into the overfitting behavior in deep learning.

---

> ### Author Response · Authors · 2024-11-22
> **Questions**
>
> We appreciate the question regarding the extension to more general data models.
>
> **Multiple tokens setting**
>
> We conducted some new experiments which appear in Fig. 6, Page 67 in the updated pdf. We empirically showed that our results hold also in this case. The key idea presented in our paper remains consistent: the attention layer focuses on optimal tokens to achieve benign overfitting.
>
> Regarding the extension of the proof of the GD, we think it should be possible, but we still need to verify a few technical details. The proof strategy is very similar and is based on analyzing the coefficients of the tokens in the attention head $v$, showing that the softmax weight $p$ assigns a high softmax probability to the signal token for clean samples, and a low softmax probability to the signal token for noisy samples (instead of showing a high softmax probability for a specific noisy token, as in the case of two tokens).
>
> Regarding the max-margin part, the situation is more complicated. The main difference lies in the optimal token selection mechanism: as shown in Figure 6(c), the optimal tokens for noisy samples can be a combination of noise tokens in this case. Therefore, in the multiple tokens setting, without additional assumptions, the interaction between noise tokens becomes more complex, making it harder to analyze their attention probabilities. Existing works on multiple tokens setting all need additional assumptions. For example, [1] assumes a strict data generation rule (Assumption C) connecting data and the induced max-margin. [2] requires the second noise token to be much larger than others (Condition 4.1.(10)). and [3] fixes the linear prediction head v and assumes identical token scores for all noise tokens. We aim to explore this more general setting in future work.  We note that once the optimal tokens are determined, benign or harmful overfitting follows similarly to our current proof.
>
> **Near orthogonality of data points**
>
> This assumption is used crucially in a number of works on benign overfitting ([4], [5], [6]). But we agree that this condition might be relaxable and this should be investigated further in the future.
>
> **Comparison to linear model**
>
> The essential difference lies in the **token selection mechanism** of the softmax layer, which induces significantly different dynamics and implicit bias. In the attention model, the softmax layer learns from the data and assigns probabilities to the first-layer output. While the optimization problem for the output layer resembles linear interpolation in the form, the softmax layer will automatically focus most of the attention on the signal token for clean samples ($\lambda \rightarrow1$) and the noise token for non-clean samples ($\lambda \rightarrow 0$). In contrast, linear interpolation lacks this token selection mechanism, leading to simultaneous learning of signal and noise tokens.
>
> **Fully Converged Solutions**
>
> Indeed, it is reasonable to think of theorem 6-8, which state that the max margin solution exhibits benign overfitting, as a kind of "fully converged" learning rule (which corresponds to training with weight decay). Regarding convergence of GD, this is indeed an intriguing question for future research. Additionally, we refer to the response to Q2 from Reviewer 7cCk, which offers some intuition on the challenges of analyzing further steps of gradient descent.
>
> [1] Davoud Ataee Tarzanagh, Yingcong Li, Xuechen Zhang, and Samet Oymak. Max-margin token selection in attention mechanism
>
> [2] Jiarui Jiang, Wei Huang, Miao Zhang, Taiji Suzuki, and Liqiang Nie. Unveil benign overfit-
> ting for transformer in vision
>
> [3] Heejune Sheen, Siyu Chen, Tianhao Wang, and Harrison H. Zhou. Implicit regularization of gradient flow on one-layer softmax attention, 2024.
>
> [4] Yuan Cao, Zixiang Chen, Misha Belkin, and Quanquan Gu. Benign overfitting in two-layer convolutional neural networks
>
> [5] Yiwen Kou, Zixiang Chen, Yuanzhou Chen, and Quanquan Gu. Benign overfitting in two-layer relu convolutional neural networks
>
> [6] Xuran Meng, Difan Zou, and Yuan Cao. Benign overfitting in two-layer relu convolutional neural networks for xor data

---

> > ### Comment · Reviewer_UW8h · 2024-11-30
> >
> > Thank you for your response to the review. It took some time to thoroughly examine the response.
> > I raised the score. However, addressing the questions raised seems to require significant revisions to the paper.
> > Therefore, it would be appropriate to relax some assumptions, make the problem setting more realistic, and resubmit the paper.

---

> > > ### Author Response · Authors · 2024-12-01
> > >
> > > Thank you for your reply. We would also like to highlight that we have extended the gradient descent result (Theorem 4) to the multiple-tokens setting, addressing one of your concerns. For more details, please refer to the top comment addressed to all reviewers.

---

### Author Response · Authors · 2024-11-28
**Multiple tokens setting**

We extend the gradient descent (GD) result (i.e., Theorem 4) to the multiple-token setting. The updated version is available in the Supplementary Material, specifically in the appendix section, where all section headlines added after the original submission are highlighted in red.
- Section A.1.4 (page 23 of the Supplementary Material) introduces the new data setting for the multiple-token case and presents the main result (i.e., Theorem 26), which demonstrates benign overfitting after two iterations of GD.
- Section A.1.5 adapts the definitions and lemmas from Section A.1.2 to accommodate the multiple-token case.
- Section A.1.6 contains the complete proof of Theorem 26.

Key points regarding the extension:
- $\textbf{Proof strategy}$:
The strategy is similar to the two-token case. After the first iteration, the model correctly classifies only the clean training samples. After the second iteration, the model interpolates all the training data.
- $\textbf{Higher softmax probability for optimal tokens}$:
After two iterations, the softmax probability of the signal token for a clean sample is at least $1/T$​ (instead of $1/2$​) and at most $1/(T-1)c_{\rho}^2$ for a noisy sample, where $T$ is the number of tokens (see the first item in Theorem 26).
- $\textbf{Constant number of tokens}$:
We assume the number of tokens is constant (see Assumption 25, item 7, page 23 of the Supplementary Material).

---

### Meta-Review · Area_Chair_GuRF · 2024-12-22

**Metareview:**

This paper theoretically investigates benign overfitting for a single head attention model.

Although benign overfitting of transformers is an important research topic, this paper has several drawbacks:
(i) The biggest concern of this paper is its novelty. First, the analysis heavily depends on exiting two papers. Unfortunately, this paper does not resolve their weakness. Moreover, the assumptions is quite strong (as described below), and thus is not strong as some other existing work.
(ii) This paper considers a prompt-attention not self-attention and the token length is limited to 2, which restrict its applicability significantly. Hence, the theoretical implication to practice is quite limited.

Unfortunately, these issues cannot be resolved by merely re-editing the paper, but It requires substantial revision. Hence, I think this paper cannot be accepted.

**Additional Comments On Reviewer Discussion:**

After extensive discussions between reviewers and authors, it appeared that the paper still has substantial drawbacks. The paper is still difficult to accept.

---

### Decision · Program_Chairs · 2025-01-22

Reject